# ADBench: Anomaly Detection Benchmark

**Songqiao Han**[1,*]**, Xiyang Hu**[2,*]**, Hailiang Huang**[1,*†]**, Minqi Jiang**[1,*]**, Yue Zhao**[2,*†]
[1] Shanghai University of Finance and Economics [2] Carnegie Mellon University
{han.songqiao,hlhuang}@shufe.edu.cn, {2020310191}@live.sufe.edu.cn,
{xiyanghu,zhaoy}@cmu.edu

## Abstract

Given a long list of anomaly detection algorithms developed in the last few decades, how do they perform with regard to (*i*) varying levels of supervision, (*ii*) different types of anomalies, and (*iii*) noisy and corrupted data? In this work, we answer these key questions by conducting (to our best knowledge) the most comprehensive anomaly detection benchmark with 30 algorithms on 57 benchmark datasets, named ADBench. Our extensive experiments (98,436 in total) identify meaningful insights into the role of supervision and anomaly types, and unlock future directions for researchers in algorithm selection and design. With ADBench, researchers can efficiently conduct comprehensive and fair evaluations for newly proposed methods on the datasets (including our contributed ones from natural language and computer vision domains) against the existing baselines. To foster accessibility and reproducibility, we fully open-source ADBench and the corresponding results.

## 1 Introduction

Anomaly detection (AD), which is also known as outlier detection, is a key machine learning (ML) task with numerous applications, including anti-money laundering [94], rare disease detection [196], social media analysis [186, 193], and intrusion detection [88]. AD algorithms aim to identify data instances that deviate significantly from the majority of data objects [59, 139, 146, 160], and numerous methods have been developed in the last few decades [3, 85, 102, 103, 129, 156, 172, 198]. Among them, the majority are designed for tabular data (i.e., no time dependency and graph structure). Thus, we focus on the *tabular* AD algorithms and datasets in this work.

Although there are already some benchmark and evaluation works for tabular AD [25, 38, 42, 53, 166], they generally have the limitations as follows: (*i*) primary emphasis on unsupervised methods only without including emerging (semi-)supervised AD methods; (*ii*) limited analysis of the algorithm performance concerning anomaly types (e.g., local vs. global); *(iii)* the lack of analysis on model robustness (e.g., noisy labels and irrelevant features); *(iv)* the absence of using statistical tests for algorithm comparison; and *(v)* no coverage of more complex CV and NLP datasets, which have attracted extensive attention nowadays.

To address these limitations, we design (to our best knowledge) the most comprehensive tabular anomaly detection benchmark called ADBench. By analyzing both research needs and deployment requirements in the industry, we design the experiments with three major angles in anomaly detection (see §3.3): (*i*) the availability of supervision (e.g., ground truth labels) by including 14 unsupervised, 7 semi-supervised, and 9 supervised methods; (*ii*) algorithm performance under different types of anomalies by simulating the environments with four types of anomalies; and (*iii*) algorithm robustness and stability under three settings of data corruptions. Fig. 1 provides an overview of ADBench.

**Key takeaways**: Through extensive experiments, we find (*i*) surprisingly none of the benchmarked unsupervised algorithms is statistically better than others, emphasizing the importance of algorithm

---

[*]All authors contribute equally. Names are listed in alphabetical ordering by the last name.
[†]Corresponding authors. Direct technical questions to Minqi Jiang and Yue Zhao.

36th Conference on Neural Information Processing Systems (NeurIPS 2022) Track on Datasets and Benchmarks.

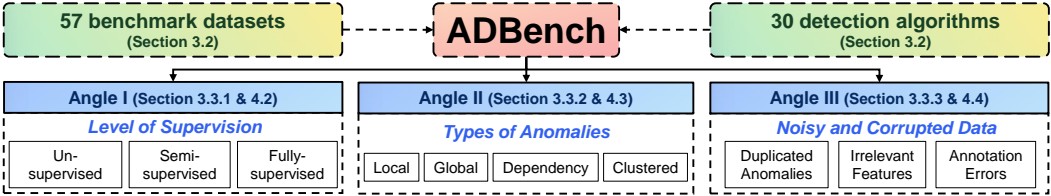

Figure 1: The design of the proposed ADBench is driven by research and application needs.

selection; (*ii*) with merely 1% labeled anomalies, most semi-supervised methods can outperform the best unsupervised method, justifying the importance of supervision; (*iii*) in controlled environments, we observe that the best unsupervised methods for specific types of anomalies are even better than semi- and fully-supervised methods, revealing the necessity of understanding data characteristics; (*iv*) semi-supervised methods show potential in achieving robustness in noisy and corrupted data, possibly due to their efficiency in using labels and feature selection. See §4 for additional results and insights.

We summarize the primary contributions of ADBench as below:

1. **The most comprehensive AD benchmark**. ADBench examines 30 detection algorithms' performance on 57 benchmark datasets (of which 47 are existing ones and we create 10).
2. **Research and application-driven benchmark angles**. By analyzing the needs of research and real-world applications, we focus on three critical comparison angles: availability of supervision, anomaly types, and algorithm robustness under noise and data corruption.
3. **Insights and future directions for researchers and practitioners**. With extensive results, we show the necessity of algorithm selection, and the value of supervision and prior knowledge.
4. **Fair and accessible AD evaluation**. We open-source ADBench with BSD-2 License at `https://github.com/Minqi824/ADBench`, for benchmarking newly proposed methods.

## 2 Related Work

### 2.1 Anomaly Detection Algorithms

**Unsupervised Methods by Assuming Anomaly Data Distributions**. *Unsupervised AD methods are proposed with different assumptions of data distribution* [3], e.g., anomalies located in low-density regions, and their performance depends on the agreement between the input data and the algorithm assumption(s). Many unsupervised methods have been proposed in the last few decades [3, 15, 129, 150, 198], which can be roughly categorized into shallow and deep (neural network) methods. The former often carries better interpretability, while the latter handles large, high-dimensional data better. Please see Appx. §A.1, recent book [3], and surveys [129, 150] for additional information.

**Supervised Methods by Treating Anomaly Detection as Binary Classification**. *With the accessibility of full ground truth labels (which is rare), supervised classifiers may identify known anomalies at the risk of missing unknown anomalies*. Arguably, there are no specialized supervised anomaly detection algorithms, and people often use existing classifiers for this purpose [3, 170] such as Random Forest [21] and neural networks [89]. One known risk of supervised methods is that ground truth labels are not necessarily sufficient to capture all types of anomalies during annotation. These methods are therefore limited to detecting unknown types of anomalies [3]. Recent machine learning books [4, 54] and scikit-learn [133] may serve as good sources of supervised ML methods.

**Semi-supervised Methods with Efficient Use of Labels.** *Semi-supervised AD algorithms can capitalize the supervision from partial labels, while keeping the ability to detect unseen types of anomalies.* To this end, some recent studies investigate using partially labeled data for improving detection performance and leveraging unlabeled data to facilitate representation learning. For instance, some semi-supervised models are trained only on normal samples, and detect anomalies that deviate from the normal representations learned in the training process [7, 8, 188]. In ADBench, semi-supervision mostly refers to *incomplete label learning* in weak-supervision (see [206]). More discussions on semi-supervised AD are deferred to Appx. §A.3.

### 2.2 Existing Datasets and Benchmarks for Tabular AD

**AD Datasets in Literature**. Existing benchmarks mainly evaluate a part of the datasets derived from the ODDS Library [145], DAMI Repository [25], ADRepository [129], and Anomaly Detection

Table 1: Comparison among ADBench and existing benchmarks, where ADBench comprehensively includes the most datasets and algorithms, uses both benchmark and synthetic datasets, covers both shallow and deep learning (DL) algorithms, and considers multiple comparison angles.

| Benchmark | Coverage (§3.2) | | Data Source | | Algorithm Type | | Comparison Angle (§3.3) | | |
|---|---|---|---|---|---|---|---|---|---|
| | # datasets | # algo. | Real-world | Synthetic | Shallow | DL | Supervision | Types | Robustness |
| Ruff et al. [150] | 3 | 9 | ✓ | ✓ | ✓ | ✓ | ✗ | ✓ | ✗ |
| Goldstein et al. [53] | 10 | 19 | ✓ | ✗ | ✓ | ✗ | ✗ | ✓ | ✗ |
| Domingues et al. [38] | 15 | 14 | ✓ | ✗ | ✓ | ✗ | ✗ | ✗ | ✓ |
| Soenen et al. [164] | 16 | 6 | ✓ | ✗ | ✓ | ✗ | ✗ | ✗ | ✗ |
| Steinbuss et al. [166] | 19 | 4 | ✗ | ✓ | ✓ | ✗ | ✗ | ✓ | ✗ |
| Emmott et al. [42] | 19 | 8 | ✓ | ✓ | ✓ | ✗ | ✗ | ✓ | ✓ |
| Campos et al. [25] | 23 | 12 | ✓ | ✗ | ✓ | ✗ | ✗ | ✗ | ✗ |
| **ADBench (ours)** | 57 | 30 | ✓ | ✓ | ✓ | ✓ | ✓ | ✓ | ✓ |

Meta-Analysis Benchmarks [42]. In ADBench, we include almost all publicly available datasets, and add larger datasets adapted from CV and NLP domains, for a more holistic view. See details in §3.2.

**Existing Benchmarks**. There are some notable works that take effort to benchmark AD methods on tabular data, e.g., [25, 38, 42, 150, 166] (see Appx. A.4). How does ADBench differ from them?

First, previous studies mainly focus on benchmarking the shallow unsupervised AD methods. Considering the rapid advancement of ensemble learning and deep learning methods, we argue that a comprehensive benchmark should also consider them. Second, most existing works only evaluate public benchmark datasets and/or some fully synthetic datasets; we organically incorporate both of them to unlock deeper insights. More importantly, existing benchmarks primarily focus on direct performance comparisons, while the settings may not be sufficiently complex to understand AD algorithm characteristics. We strive to address the above issues in ADBench, and illustrate the main differences between the proposed ADBench and existing AD benchmarks in Table 1.

Also, "anomaly detection" is an overloaded term; there are AD benchmarks for time-series [85, 87, 132], graph [101], CV [6, 27, 203] and NLP [143], but they are different from tabular AD in nature.

### 2.3 Connections with Related Fields and Other Opportunities

While ADBench focuses on the AD tasks, we note that there are some closely related problems, including out-of-distribution (OOD) detection [182, 183], novelty detection [116, 137], and open-set recognition (OSR) [51, 112]. Uniquely, AD usually does not assume the train set is anomaly-free, while other related tasks may do. Some methods designed for these related fields, e.g., OCSVM [157], can be used for AD as well; future benchmark can consider including: (*i*) OOD methods: MSP [65], energy-based EBO [104], and Mahalanobis distance-based MDS [92]; (*ii*) novelty detection methods: OCGAN [135] and Adversarial One-Class Classifier [154]; and (*iii*) OSR methods: OpenGAN [79] and PROSER [204]. See [155] for deeper connections and differences between AD and these fields.

We consider saliency detection (SD) [44, 46] and camouflage detection (CD) [45] as good inspirations and applications of AD tasks. Saliency detection identifies important regions in the images, where explainable AD algorithms [123], e.g., FCDD [106], may help the task. Camouflage detection finds concealed objects in the background, e.g., camouflaged anomalies blurred with normal objects [110], where camouflage-resistant AD methods [40] help detect concealed objects (that look normal but are abnormal). Future work can explore the explainability of detected objects in AD.

## 3 ADBench: AD Benchmark Driven by Research and Application Needs

### 3.1 Preliminaries and Problem Definition

**Unsupervised AD** often presents a collection of $n$ samples $\mathbf{X} = \{\boldsymbol{x}_1, \ldots, \boldsymbol{x}_n\} \in \mathbb{R}^{n \times d}$, where each sample has $d$ features. Given the inductive setting, the goal is to train an AD model $M$ to output anomaly score $\mathbf{O} := M(\mathbf{X}) \in \mathbb{R}^{n \times 1}$, where higher scores denote for more outlyingness. In the inductive setting, we need to predict on $\mathbf{X}_{\text{test}} \in \mathbb{R}^{m \times d}$, so to return $\mathbf{O}_{\text{test}} := M(\mathbf{X}_{\text{test}}) \in \mathbb{R}^{m \times 1}$.

**Supervised AD** also has the (binary) ground truth labels of $\mathbf{X}$, i.e., $\mathbf{y} \in \mathbb{R}^{n \times 1}$. A supervised AD model $M$ is first trained on $\{\mathbf{X}, \mathbf{y}\}$, and then returns anomaly scores for the $\mathbf{O}_{\text{test}} := M(\mathbf{X}_{\text{test}})$.

**Semi-supervised AD** only has the partial label information $\mathbf{y}^l \in \mathbf{y}$. The AD model $M$ is trained on the entire feature space $\mathbf{X}$ with the partial label $\mathbf{y}^l$, i.e., $\{\mathbf{X}, \mathbf{y}^l\}$, and then outputs $\mathbf{O}_{\text{test}} := M(\mathbf{X}_{\text{test}})$.

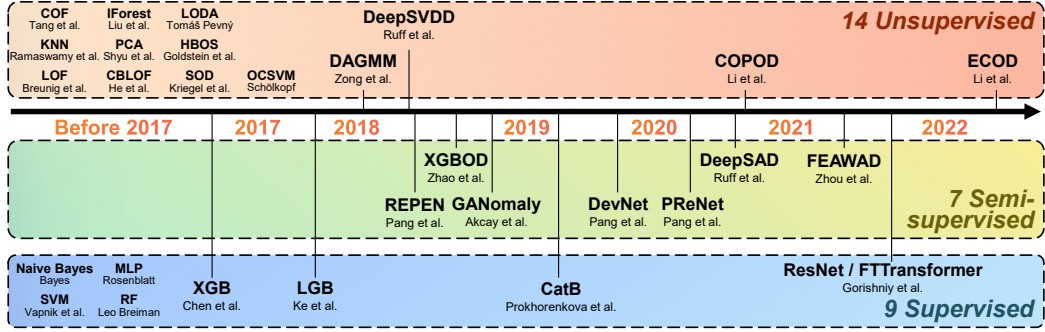

Figure 2: ADBench covers a wide range of AD algorithms. See Appx. B.1 for more details.

**Remark**. Irrespective of the types of underlying AD algorithms, the goal of ADBench is to understand AD algorithms' performance under the inductive setting. Collectively, we refer semi-supervised and supervised AD methods as "label-informed" methods. Refer to §4.1 for specific experiment settings.

## 3.2 The Largest AD Benchmark with 30 Algorithms and 57 Datasets

**Algorithms**. Compared to the previous benchmarks, we have a larger algorithm collection with (*i*) the latest unsupervised AD algorithms like DeepSVDD [151] and ECOD [97]; (*ii*) SOTA semi-supervised algorithms, including DeepSAD [152] and DevNet [131]; (*iii*) latest network architectures like ResNet [62] in computer vision (CV) and Transformer [171] in the natural language processing (NLP) domain—we adapt ResNet and FTTransformer models [56] for tabular AD in the proposed ADBench; and (*iv*) ensemble learning methods like LightGBM [74], XGBoost [29], and CatBoost [138] that have shown effectiveness in AD tasks [170]. Fig. 2 shows the 30 algorithms (14 unsupervised, 7 semi-supervised, and 9 supervised algorithms) evaluated in ADBench, where we provide more information about them in Appx. B.1.

**Algorithm Implementation**. Most unsupervised algorithms are readily available in our early work Python Outlier Detection (PyOD) [198], and some supervised methods are available in scikit-learn [133] and corresponding libraries. Supervised ResNet and FTTransformer tailored for tabular data have been open-sourced in their original paper [56]. We implement the semi-supervised methods and release them along with ADBench.

**Public AD Datasets**. In ADBench, we gather more than 40 benchmark datasets [25, 42, 129, 145], for model evaluation, as shown in Appx. Table B1. These datasets cover many application domains, including healthcare (e.g., disease diagnosis), audio and language processing (e.g., speech recognition), image processing (e.g., object identification), finance (e.g., financial fraud detection), etc. For due diligence, we keep the datasets where the anomaly ratio is below 40% (Appx. Fig. B1).

**Newly-added Datasets in ADBench**. Since most of these datasets are relatively small, we introduce 10 more complex datasets from CV and NLP domains with more samples and richer features in ADBench (highlighted in Appx. Table B1). Pretrained models are applied to extract data embedding from CV and NLP datasets to access more complex representations, which has been widely used in AD literature [33, 115, 152] and shown better results than using the raw features. For NLP datasets, we use BERT [75] pretrained on the BookCorpus and English Wikipedia to extract the embedding of the [CLS] token. For CV datasets, we use ResNet18 [62] pretrained on the ImageNet [35] to extract the embedding after the last average pooling layer. Following previous works [151, 152], we set one of the multi-classes as normal, downsample the remaining classes to 5% of the total instances as anomalies, and report the average results over all the respective classes. Including these originally non-tabular datasets helps to see whether tabular AD methods can work on CV/NLP data after necessary preprocessing. See Appx. B.2 for more details on datasets.

## 3.3 Benchmark Angles in ADBench

### 3.3.1 Angle I: Availability of Ground Truth Labels (Supervision)

**Motivation**. As shown in Table 1, existing benchmarks only focus on the unsupervised setting, i.e., none of the labeled anomalies is available. Despite, in addition to unlabeled samples, one may have access to a limited number of labeled anomalies in real-world applications, e.g., a few anomalies identified by domain experts or human-in-the-loop techniques like active learning [5, 7, 78, 189].

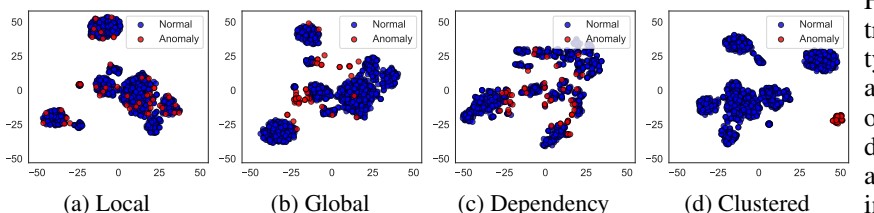

Figure 3: Illustration of four types of synthetic anomalies shown on Lymphography dataset. See the additional demo in Appx. Fig. B2.

(a) Local     (b) Global     (c) Dependency     (d) Clustered

Notably, there is a group of semi-supervised AD algorithms [127, 128, 130, 131, 152, 168, 205] that have not been covered by existing benchmarks.

**Our design**: We first benchmark existing unsupervised anomaly detection methods, and then evaluate both semi-supervised and fully-supervised methods with varying levels of supervision following the settings in [127, 131, 205] to provide a fair comparison. For example, labeled anomalies $\gamma_l = 10\%$ means that $10\%$ anomalies in the train set are known while other samples remain unlabeled. The complete experiment results of un-, semi-, and full-supervised algorithms are presented in §4.2.

### 3.3.2   Angle II: Types of Anomalies

**Motivation**. While extensive public datasets can be used for benchmarking, they often consist of a mixture of different types of anomalies, making it challenging to understand the pros and cons of AD algorithms regarding specific types of anomalies [55, 166]. In real-world applications, one may know specific types of anomalies of interest. To better understand the impact of anomaly types, we create synthetic datasets based on public datasets by injecting specific types of anomalies to analyze the response of AD algorithms.

**Our design**: In ADBench, we create *realistic* synthetic datasets from benchmark datasets by injecting specific types of anomalies. Some existing works, such as PyOD [198], generate fully synthetic anomalies by assuming their data distribution, which fails to create complex anomalies. We follow and enrich the approach in [166] to generate "realistic" synthetic data; ours supports more types of anomaly generation. The core idea is to build a generative model (e.g., Gaussian mixture model GMM used in [166], Sparx [191], and ADBench) using the normal samples from a benchmark dataset and discard its original anomalies as we do not know their types. Then, We could generate normal samples and different types of anomalies based on their definitions by tweaking the generative model. The generation of normal samples is the same in all settings if not noted, and we provide the generation process of four types of anomalies below (also see our codebase for details).

**Definition and Generation Process of Four Types of Common Anomalies Used in ADBench**:

- **Local anomalies** refer to the anomalies that are deviant from their local neighborhoods [22]. We follow the GMM procedure [118, 166] to generate synthetic normal samples, and then scale the covariance matrix $\hat{\Sigma} = \alpha \hat{\Sigma}$ by a scaling parameter $\alpha = 5$ to generate local anomalies.
- **Global anomalies** are more different from the normal data [68], generated from a uniform distribution $\text{Unif}\left(\alpha \cdot min\left(\mathbf{X}^{\mathbf{k}}\right), \alpha \cdot max\left(\mathbf{X}^{\mathbf{k}}\right)\right)$, where the boundaries are defined as the *min* and *max* of an input feature, e.g., $k$-th feature $\mathbf{X}^{\mathbf{k}}$, and $\alpha = 1.1$ controls the outlyingness of anomalies.
- **Dependency anomalies** refer to the samples that do not follow the dependency structure which normal data follow [117], i.e., the input features of dependency anomalies are assumed to be independent of each other. Vine Copula [1] method is applied to model the dependency structure of original data, where the probability density function of generated anomalies is set to complete independence by removing the modeled dependency (see [117]). We use Kernel Density Estimation (KDE) [61] to estimate the probability density function of features and generate normal samples.
- **Clustered anomalies**, also known as group anomalies [93], exhibit similar characteristics [42, 99]. We scale the mean feature vector of normal samples by $\alpha = 5$, i.e., $\hat{\mu} = \alpha \hat{\mu}$, where $\alpha$ controls the distance between anomaly clusters and the normal, and use the scaled GMM to generate anomalies.

Fig. 3 shows 2-d t-SNE [169] visualization of the four types of synthetic outliers generated from Lymphography dataset, where they generally satisfy the expected characteristics. Local anomalies (Fig. 3a) are well overlapped with the normal samples. Global anomalies (Fig. 3b) are more deviated from the normal samples and on the edges of normal clusters. The other two types of anomalies are as expected, with no clear dependency structure in Fig. 3c and having anomaly cluster(s) in Fig. 3d. In ADBench, we analyze the algorithm performances under all four types of anomalies above (§4.3).

### 3.3.3 Angle III: Model Robustness with Noisy and Corrupted Data

**Motivation**. Model robustness has been an important aspect of anomaly detection and adversarial machine learning [24, 41, 47, 76, 177]. Meanwhile, the input data likely suffers from noise and corruption to some extent in real-world applications [42, 55, 60, 124]. However, this important view has not been well studied in existing benchmarks, and we try to understand this by evaluating AD algorithms under three noisy and corruption settings (see results in §4.4):

- **Duplicated Anomalies**. In many applications, certain anomalies likely repeat multiple times in the data for reasons such as recording errors [83]. The presence of duplicated anomalies is also called the "anomaly masking" [55, 60, 100], posing challenges to many AD algorithms [25], e.g., the density-based KNN [11, 144]. Besides, the change of anomaly frequency would also affect the behavior of detection methods [42]. Therefore, we simulate this setting by splitting the data into train and test set, then duplicating the anomalies (both features and labels) up to 6 times in both sets, and observing how AD algorithms change.
- **Irrelevant Features**. Tabular data may contain irrelevant features caused by measurement noise or inconsistent measuring units [28, 55], where these noisy dimensions could hide the characteristics of anomaly data and thus make the detection process more difficult [128, 150]. We add irrelevant features up to $50\%$ of the total input features (i.e., $d$ in the problem definition) by generating uniform noise features from $\text{Unif}\left(\min\left(\mathbf{X}^{\mathbf{k}}\right), \max\left(\mathbf{X}^{\mathbf{k}}\right)\right)$ of randomly selected $k$-th input feature $\mathbf{X}^{\mathbf{k}}$ while the labels stay correct, and summarize the algorithm performance changes.
- **Annotation Errors**. While existing studies [131, 152] explored anomaly contamination in the unlabeled samples, we further discuss the more generalized impact of label contamination on the algorithm performance, where the label flips [122, 202] between the normal samples and anomalies are considered (up to $50\%$ of total labels). Note this setting does not affect unsupervised methods as they do not use any labels. Discussion of annotation errors is meaningful since manual annotation or some automatic labeling techniques are always noisy while being treated as perfect.

## 4 Experiment Results and Analyses

We conduct 98,436 experiments (Appx. C) to answer **Q1** (§4.2): How do AD algorithms perform with varying levels of supervision? **Q2** (§4.3): How do AD algorithms respond to different types of anomalies? **Q3** (§4.4): How robust are AD algorithms with noisy and corrupted data? In each subsection, we first present the key results and analyses (please refer to the additional points in Appx. D), and then propose a few open questions and future research directions.

### 4.1 Experiment Setting

**Datasets, Train/test Data Split, and Independent Trials**. As described in §3.2 and Appx. Table B1, ADBench includes 57 existing and freshly proposed datasets, which cover different fields including healthcare, security, and more. Although unsupervised AD algorithms are primarily designed for the transductive setting (i.e., outputting the anomaly scores on the input data only other than making predictions on the newcoming data), we adapt all the algorithms for the inductive setting to predict the newcoming data, which is helpful in applications and also common in popular AD library PyOD [198], TODS [84], and PyGOD [102]. Thus, we use $70\%$ data for training and the remaining $30\%$ as the test set. We use stratified sampling to keep the anomaly ratio consistent. We repeat each experiment 3 times and report the average. Detailed settings are described in Appx. C.

**Hyperparameter Settings**. For all the algorithms in ADBench, we use their default hyperparameter (HP) settings in the original paper for a fair comparison. Refer to the Appx. C for more information.

**Evaluation Metrics and Statistical Tests**. We evaluate different AD methods by two widely used metrics: AUCROC (Area Under Receiver Operating Characteristic Curve) and AUCPR (Area Under Precision-Recall Curve) value[1]. Besides, the critical difference diagram (CD diagram) [34, 70] based on the Wilcoxon-Holm method is used for comparing groups of AD methods statistically ($p \leq 0.05$).

### 4.2 Overall Model Performance on Datasets with Varying Degrees of Supervision

As introduced in §3.3.1, we first present the results of unsupervised methods on 57 datasets in Fig. 4a, and then compare label-informed semi- and fully-supervised methods under varying degrees of supervision, i.e., different label ratios of $\gamma_l$ (from $1\%$ to $100\%$ full labeled anomalies) in Fig. 4b.

---

[1]We present the results based on AUCROC and observe similar results for AUCPR; See Appx. D for all.

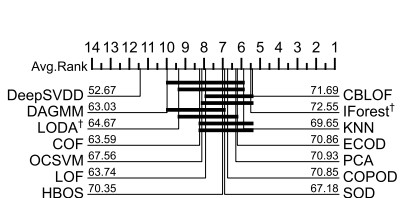

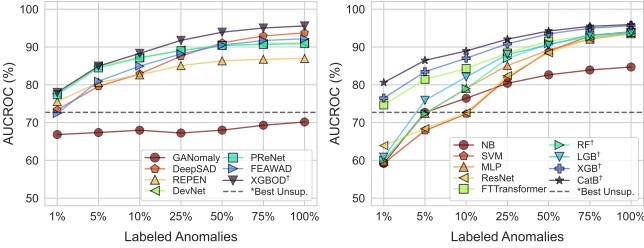

(a) Avg. rank (lower the better) and avg. AUCROC (on each line) of unsupervised methods; groups of algorithms not statistically different are connected horizontally.

(b) Avg. AUCROC (on 57 datasets) vs. % of labeled anomalies (x-axis); semi-supervised (left) and fully-supervised (right). Most label-informed algorithms outperform the best *unsupervised* algorithm CBLOF (denoted as the dashed line) with 10% labeled anomalies.

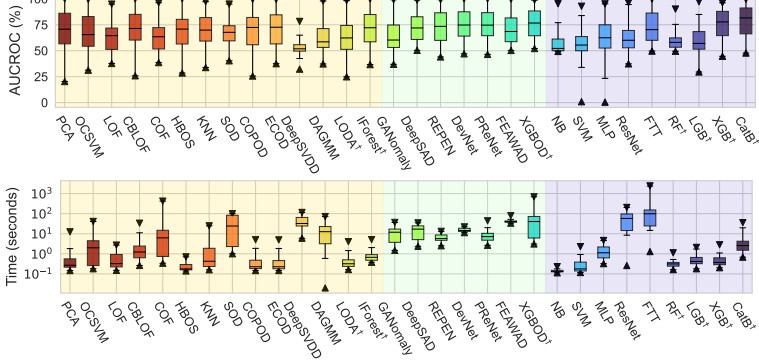

(c) Boxplot of AUCROC (@1% labeled anomalies) on 57 datasets; we denote un-, semi-, and fully supervised methods in light yellow, green, and purple.

(d) Boxplot of train time (see inf. time in Appx. Fig. D6) on 57 datasets; we denote un-, semi-, and fully supervised methods in light yellow, green, and purple.

Figure 4: Average AD model performance across 57 benchmark datasets. (a) shows that no unsupervised algorithm statistically outperforms the rest. (b) shows that semi-supervised methods leverage the labels more efficiently than fully-supervised methods with a small labeled anomaly ratio $\gamma_l$. (c) and (d) present the boxplots of AUCROC and runtime. Ensemble methods are marked with "†".

**None of the unsupervised methods is statistically better than the others**, as shown in the critical difference diagram of Fig. 4a (where most algorithms are horizontally connected without statistical significance). We also note that some DL-based unsupervised methods like DeepSVDD and DAGMM are surprisingly worse than shallow methods. Without the guidance of label information, DL-based unsupervised algorithms are harder to train (due to more hyperparameters) and more difficult to tune hyperparameters, leading to unsatisfactory performance.

**Semi-supervised methods outperform supervised methods when limited label information is available**. For $\gamma_l \leq 5\%$, i.e., only less than 5% labeled anomalies are available during training, the detection performance of semi-supervised methods (median AUCROC= 75.56% for $\gamma_l = 1\%$ and AUCROC= 80.95% for $\gamma_l = 5\%$) are generally better than that of fully-supervised algorithms (median AUCROC= 60.84% for $\gamma_l = 1\%$ and AUCROC= 72.69% for $\gamma_l = 5\%$). For most semi-supervised methods, merely 1% labeled anomalies are sufficient to surpass the best unsupervised method (shown as the dashed line in Fig. 4b), while most supervised methods need 10% labeled anomalies to achieve so. We also show the improvement of algorithm performances about the increasing $\gamma_l$, and notice that with a large number of labeled anomalies, both semi-supervised and supervised methods have comparable performance. Putting these together, we verify the assumed advantage of semi-supervised methods in leveraging limited label information more efficiently.

**Latest network architectures like Transformer and emerging ensemble methods yield competitive performance in AD**. Fig. 4b shows FTTransformer and ensemble methods like XGB(oost) and CatB(oost) provide satisfying detection performance among all the label-informed algorithms, even these methods are not specifically proposed for the anomaly detection tasks. For $\gamma_l = 1\%$, the AUCROC of FTTransformer and the median AUCROC of ensemble methods are 74.68% and 76.47%, respectively, outperforming the median AUCROC of all label-informed methods 72.91%. The great performance of tree-based ensembles (in tabular AD) is consistent with the findings in literature [20, 58, 170], which may be credited to their capacity to handle imbalanced AD datasets via aggregation. Future research may focus on understanding the cause and other merits of ensemble trees in tabular AD, e.g., better model efficiency.

**Runtime Analysis.** We present the train and inference time in Fig. 4d and Appx. Fig. D6. Runtime analysis finds that HBOS, COPOD, ECOD, and NB are the fastest as they treat each feature independently. In contrast, more complex representation learning methods like XGBOD, ResNet, and FTTansformer are computationally heavy. This should be factored in for algorithm selection.

*Future Direction 1: Unsupervised Algorithm Evaluation, Selection, and Design*. For unsupervised AD, the results suggest that future algorithms should be evaluated on large testbeds like ADBench for statistical tests (such as via critical difference diagrams). Meanwhile, the no-free-lunch theorem [175] suggests there is no universal winner for all tasks, and more focus should be spent on understanding the suitability of each AD algorithm. Notably, algorithm selection and hyperparameter optimization are important in unsupervised AD, but limited works [13, 109, 194, 199, 200] have studied them. We may consider self-supervision [140, 158, 161, 179] and transfer learning [33] to improve tabular AD as well. Thus, we call for attention to large-scale evaluation, task-driven algorithm selection, and data augmentation/transfer for unsupervised AD.

*Future Direction 2: Semi-supervised Learning*. By observing the success of using limited labels in AD, we would call for more attention to semi-supervised AD methods which can leverage both the guidance from labels efficiently and the exploration of the unlabeled data. Regarding backbones, the latest network architectures like Transformer and ensembling show their superiority in AD tasks.

## 4.3 Algorithm Performance under Different Types of Anomalies

Under four types of anomalies introduced in §3.3.2), we show the performances of unsupervised methods in Fig. 5, and then compare both semi- and fully-supervised methods in Fig. 6.

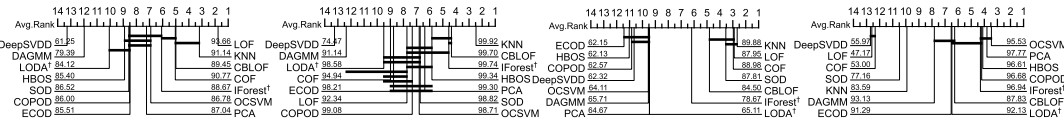

|     (a) Local anomalies     |     (b) Global anomalies     |     (c) Dependency anomalies     |     (d) Clustered anomalies     |

Figure 5: Avg. rank (lower the better) of unsupervised methods on different types of anomalies. Groups of algorithms not significantly different are connected horizontally in the CD diagrams. The unsupervised methods perform well when their assumptions conform to the underlying anomaly type.

**Performance of unsupervised algorithms highly depends on the alignment of its assumptions and the underlying anomaly type**. As expected, *local* anomaly factor (LOF) is statistically better than other unsupervised methods for the local anomalies (Fig. 5a), and KNN, which uses $k$-th (*global*) nearest neighbor's distance as anomaly scores, is the statistically best detector for global anomalies (Fig. 5b). Again, there is no algorithm performing well on all types of anomalies; LOF achieves the best AUCROC on local anomalies (Fig. 5a) and the second best AUCROC rank on dependency anomalies (Fig. 5c), but performs poorly on clustered anomalies (Fig. 5d). Practitioners should select algorithms based on the characteristics of the underlying task, and consider the algorithm which may cover more high-interest anomaly types [93].

**The "power" of prior knowledge on anomaly types may overweigh the usage of partial labels**. For the local, global, and dependency anomalies, most label-informed methods perform worse than the best unsupervised methods of each type (corresponding to LOF, KNN, and KNN). For example, the detection performance of XGBOD for the local anomalies is inferior to the best unsupervised method LOF when $\gamma_l \leq 50\%$, while other methods perform worse than LOF in all cases (See Fig. 6a). Why could not label-informed algorithms beat unsupervised methods in this setting? We believe that partially labeled anomalies cannot well capture all characteristics of specific types of anomalies, and learning such decision boundaries is challenging. For instance, different local anomalies often exhibit various behaviors, as shown in Fig. 3a, which may be easier to identify by a generic definition of "locality" in unsupervised methods other than specific labels. Thus, incomplete label information may bias the learning process of these label-informed methods, which explains their relatively inferior performances compared to the best unsupervised methods. This conclusion is further verified by the results of clustered anomalies (See Fig. 6d), where label-informed (especially semi-supervised) methods outperform the best unsupervised method OCSVM, as few labeled anomalies can already represent similar behaviors in the clustered anomalies (Fig. 3d).

*Future Direction 3: Leveraging Anomaly Types as Valuable Prior Knowledge*. The above results emphasize the importance of knowing anomaly types in achieving high detection performance even without labels, and call for attention to designing anomaly-type-aware detection algorithms. In an

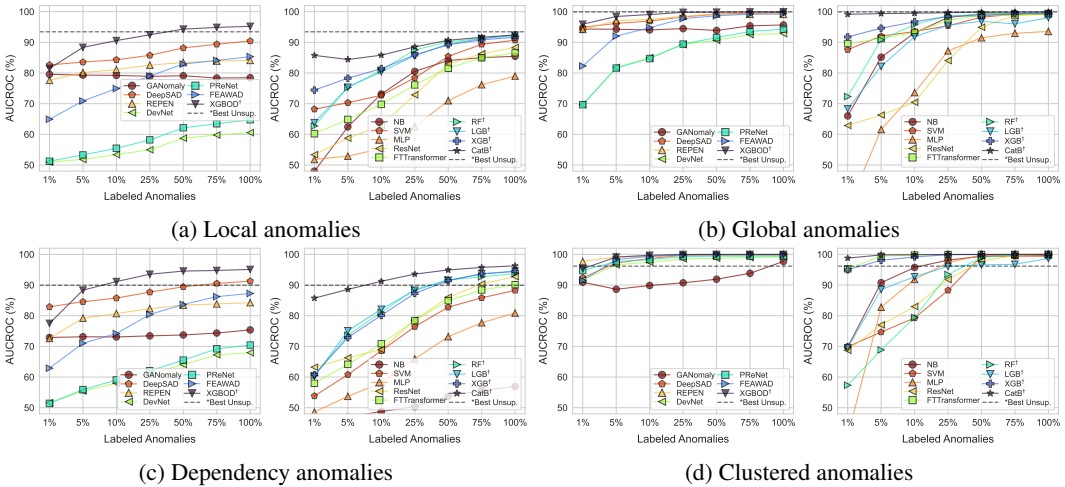

(a) Local anomalies

(b) Global anomalies

(c) Dependency anomalies

(d) Clustered anomalies

Figure 6: Semi- (left of each subfigure) and supervised (right) algorithms' performance on different types of anomalies with varying levels of labeled anomalies. Surprisingly, these label-informed algorithms are *inferior* to the best unsupervised method except for the clustered anomalies.

ideal world, one may combine multiple AD algorithms based on the composition of anomaly types, via frameworks like dynamic model selection and combination [197]. To our knowledge, the latest advancement in this end [71] provides an equivalence criterion for measuring to what degree two anomaly detection algorithms detect the same kind of anomalies. Furthermore, future research may also consider designing semi-supervised AD methods capable of detecting different types of unknown anomalies while effectively improving performance by the partially available labeled data. Another interesting direction is to train an offline AD model using synthetically generated anomalies and then adapt it for online prediction on real-world datasets with likely similar anomaly types. Unsupervised domain adaption and transfer learning for AD [33, 185] may serve as useful references.

### 4.4 Algorithm Robustness under Noisy and Corrupted Data

In this section, we investigate the algorithm robustness (i.e., $\Delta$performance; see absolute performance plot in Appx. D9) of different AD algorithms under noisy and data corruption described in §3.3.3. The default $\gamma_l$ is set to $100\%$ since we only care about the relative change of model performance. Fig. 7 demonstrates the results.

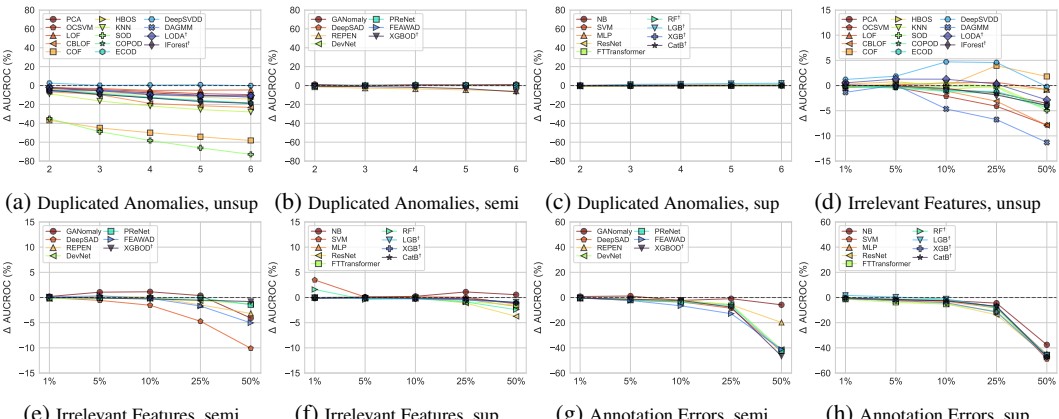

(a) Duplicated Anomalies, unsup  (b) Duplicated Anomalies, semi  (c) Duplicated Anomalies, sup  (d) Irrelevant Features, unsup

(e) Irrelevant Features, semi  (f) Irrelevant Features, sup  (g) Annotation Errors, semi  (h) Annotation Errors, sup

Figure 7: Algorithm performance change under noisy and corrupted data (i.e., duplicated anomalies for (a)-(c), irrelevant features for (d)-(f), and annotation errors for (g) and (h)). X-axis denotes either the duplicated times or the noise ratio. Y-axis denotes the % of performance change ($\Delta$AUCROC), and its range remains consistent across different algorithms. The results reveal unsupervised methods' susceptibility to duplicated anomalies and the usage of label information in defending irrelevant features. Un-, semi-, and fully-supervised methods are denoted as *unsup*, *semi*, and *sup*, respectively.

**Unsupervised methods are more susceptible to duplicated anomalies**. As shown in Fig. 7a, almost all unsupervised methods are severely impacted by duplicated anomalies. Their AUCROC

deteriorates proportionally with the increase in duplication. When anomalies are duplicated by 6 times, the median $\Delta$AUCROC of unsupervised methods is $-16.43\%$, compared to that of semi-supervised methods $-0.05\%$ (Fig. 7b) and supervised methods $0.13\%$ (Fig. 7c). One explanation is that unsupervised methods often assume the underlying data is imbalanced with only a smaller percentage of anomalies—they rely on this assumption to detect anomalies. With more duplicated anomalies, the underlying data becomes more balanced, and the minority assumption of anomalies is violated, causing the degradation of unsupervised methods. Differently, more balanced datasets do not affect the performance of semi- and fully-supervised methods remarkably, with the help of labels.

**Irrelevant features cause little impact on supervised methods due to feature selection**. Compared to the unsupervised and most semi-supervised methods, the training process of supervised methods is fully guided by the data labels ($\mathbf{y}$), therefore performing robustly to the irrelevant features (i.e., corrupted $\mathbf{X}$) due to the direct (or indirect) feature selection process. For instance, ensemble trees like XGBoost can filter irrelevant features. As shown in Fig. 7f, even the worst performing supervised algorithm (say ResNet) in this setting yields $\leq 5\%$ degradation when 50% of the input features are corrupted by the uniform noises, while the un- and semi-supervised methods could face up to $10\%$ degradation. Besides, the robust performances of supervised methods (and some semi-supervised methods like DevNet) indicate that the label information can be beneficial for feature selection. Also, Fig. 7f shows that minor irrelevant features (e.g., 1%) help supervised methods as regularization to generalize better.

**Both semi- and fully-supervised methods show great resilience to minor annotation errors**. Although the detection performance of these methods is significantly downgraded when the annotation errors are severe (as shown in Fig. 7g and 7h), their degradation with regard to minor annotation errors is acceptable. The median $\Delta$AUCROC of semi- and fully-supervised methods for $5\%$ annotation errors is $-1.52\%$ and $-1.91\%$, respectively. That being said, label-informed methods are still acceptable in practice as the annotation error should be relatively small [95, 181].

*Future Direction 4: Noise-resilient AD Algorithms*. Our results indicate there is an improvement space for robust unsupervised AD algorithms. One immediate remedy is to incorporate unsupervised feature selection [30, 125, 126] to combat irrelevant features. Moreover, label information could serve as effective guidance for model training against data noise, and it helps semi- and fully-supervised methods to be more robust. Given the difficulty of acquiring full labels, we suggest using semi-supervised methods as the backbone for designing more robust AD algorithms. Also, recent works on leveraging multiple sets of noisy labels collectively for learning AD models are also relevant [201].

## 5    Conclusions and Future Work

In this paper, we introduce ADBench, the most comprehensive tabular anomaly detection benchmark with 30 algorithms and 57 benchmark datasets. Based on the analyses of multiple comparison angles, we unlock insights into the role of supervision, the importance of prior knowledge of anomaly types, and the principles of designing robust detection algorithms. On top of them, we summarize a few promising future research directions for anomaly detection, along with the fully released benchmark suite for evaluating new algorithms.

ADBench can extend to understand the algorithm performance with (*i*) mixed types of anomalies; (*ii*) different levels of (intrinsic) anomaly ratio; and (*iii*) more data modalities. Also, future benchmarks can consider the latest algorithms [28, 99, 161], and curate datasets from emerging fields like drug discovery [69], molecule optimization [49, 50], interpretability and explainability [123, 180], and bias and fairness [32, 67, 123, 159, 165, 190].

## Aknowledgement

We briefly describe the authors' contributions. *Problem scoping*: M.J., Y.Z., S.H., X.H., and H.H.; *Experiment and Implementation*: M.J. and Y.Z.; *Result Analysis*: M.J., Y.Z., and X.H.; *Paper Drafting*: M.J., Y.Z., S.H., X.H., and H.H.; *Paper Revision*: M.J., Y.Z., S.H., and X.H.

We thank anonymous reviewers for their insightful feedback and comments. We appreciate the suggestions of Xueying Ding, Kwei-Herng (Henry) Lai, Meng-Chieh Lee, Ninghao Liu, Yuwen Yang, Allen Zhu, Chaochuan Hou, and Xu Yao. Y.Z. is partly supported by the Norton Graduate Fellowship. H.H., S.H., and M.J. are supported by the National Natural Science Foundation of China under Grant No. 72271151, 92146004, and the National Key Research and Development Program of China under Grant No. 2022YFC3303301. H.H., S.H., and M.J. thank the financial support provided by FlagInfo-SHUFE Joint Laboratory.

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
