# Supplementary Material for *ADBench: Anomaly Detection Benchmark*

*Additional information on related works, algorithms, datasets, and additional experiment settings and results*

## A    Related Works with More Details

We provide more details on existing AD algorithms and benchmarks, and the primary content discussed in §2.

### A.1    Unsupervised Methods

**Unsupervised Methods by Assuming Anomaly Data Distributions**. Unsupervised AD methods are proposed with different assumptions of data distribution [3], e.g., anomalies located in low-density regions, and their performance often depends on the agreement between the input data and the algorithm assumption(s). Many unsupervised methods have been proposed in the last few decades [3, 15, 129, 150, 198], which can be roughly categorized into shallow and deep (neural network) methods. The former often carry better interpretability, while the latter handles large, high-dimensional data better. Recent book [3] and surveys [129, 150] provide great details on these algorithms, while we further elaborate on a few representative unsupervised methods. More algorithm details and hyperparameter settings are illustrated in Appx. §B.1

**Representative Shallow Methods**. Some representative shallow methods include: (*i*) Isolation Forest (IForest) [100] builds an ensemble of trees to isolate the data points and defines the anomaly score as the distance of an individual instance to the root; (*ii*) One-class SVM (OCSVM) [157] maximizes the margin between origin and the normal samples, where the decision boundary is the hyper-plane that determines the margin; and (*iii*) Empirical-Cumulative-distribution-based Outlier Detection (ECOD) [97] computes the empirical cumulative distribution per dimension of the input data, and then aggregates the tail probabilities per dimension for calculating the anomaly score.

**Representative Deep Methods**. Deep (neural network) methods have gained more attention recently, and we briefly review some representative ones in this section. Deep Autoencoding Gaussian Mixture Model (DAGMM) [206] jointly optimizes the deep autoencoder and the Gaussian mixture model in an end-to-end neural network fashion. The joint optimization balances autoencoding reconstruction, density estimation of latent representation, and regularization and helps the autoencoder escape from less attractive local optima and further reduce reconstruction errors, avoiding pre-training. Deep Support Vector Data Description (DeepSVDD) [151] trains a neural network to learn a transformation that minimizes the volume of a data-enclosing hypersphere in the output space, and calculates the anomaly score as the distance of transformed embedding to the center of the hypersphere.

### A.2    Supervised Methods

Due to the difficulty and cost of collecting large-scale labeled data, fully-supervised anomaly detection is often impractical as it assumes the availability of labeled training data with both normal and anomaly samples [129]. Although some loss functions (e.g., focal loss [98]) are devised to address the class imbalance problem, they are often not specific for AD tasks. There also exist a few works [57, 73] discussing the relationship between fully-supervised and semi-supervised AD methods, and argue that semi-supervised AD needs to be ground on the unsupervised learning paradigm instead of the supervised one for detecting both known and unknown anomalies. We implement several representative supervised classification algorithms in ADBench (as shown in Appx. §B.1), and recommend interesting readers to recent machine learning books [4, 54] and scikit-learn [133] for more details about recent supervised methods designed for the classification tasks.

### A.3    Semi-supervised Methods

Semi-supervised AD algorithms are designed to capitalize the supervision from partial labels, while keeping the ability to detect unseen types of anomalies. To this end, some recent studies investigate efficiently using partially labeled data for improving detection performance, and leverage the unlabeled data to facilitate representation learning. We further provide some technical details on

representative semi-supervised AD methods here. Please see Appx. §B.1 for more algorithm details and hyperparameter settings in ADBench.

**Representative Methods**. Extreme Gradient Boosting Outlier Detection (XGBOD) [195] uses multiple unsupervised AD algorithms to extract useful representations from the underlying data that augment the predictive capabilities of an embedded supervised classifier on an improved feature space. Deep Semi-supervised Anomaly Detection (DeepSAD) [152] is an end-to-end methodology considered the state-of-the-art method in semi-supervised anomaly detection. DeepSAD improves the DeepSVDD [151] model by the inverse loss function for the labeled anomalies. REPresentations for a random nEarest Neighbor distance-based method (REPEN) [127] proposes a ranking model-based framework, which unifies representation learning and anomaly detection to learn low-dimensional representations tailored for random distance-based detectors. Deviation Networks (DevNet) [131] constructs an end-to-end neural network for learning anomaly scores, which forces the network to produce statistically higher anomaly scores for identified anomalies than that of unlabeled data. **P**airwise **R**elation prediction-based ordinal regression **Net**work (PReNet) [130] formulates the anomaly detection problem as a pairwise relation prediction task, which defines a two-stream ordinal regression neural network to learn the relation of randomly sampled instance pairs. Feature Encoding with AutoEncoders for Weakly-supervised Anomaly Detection (FEAWAD) [204] leverages an autoencoder to encode the input data and utilize hidden representation, reconstruction residual vector and reconstruction error as the new representations for improving the DevNet [131] and DAGMM [206].

## A.4 Existing AD Benchmarks

As we show in Table 1, there is a line of existing AD benchmarks. [150] discusses a unifying review of both the shallow and deep anomaly detection methods, but they mainly focus on the theoretical perspective and thus lack results from the experimental views. [25] benchmarks 19 different unsupervised methods on 10 datasets, and analyzes the characteristics of density-based and clustering-based algorithms. [38] tests 14 unsupervised anomaly detection methods on 15 public datasets, and analyzes the scalability, memory consumption, and robustness of different methods. [166] proposes a generic process for the generation of realistic synthetic data. The synthetic normal instances are reconstructed from existing real-world benchmark data, while synthetic anomalies are in line with a characterizable deviation from the modeling of synthetic normal data. [42] evaluates 8 unsupervised methods on 19 public datasets, and produces a large corpus of synthetic anomaly detection datasets that vary in their construction across several dimensions that are important to real-world applications. [25] compares 12 unsupervised anomaly detection approaches on 23 datasets, providing a characterization of benchmark datasets and their suitability as anomaly detection benchmark sets.

All these existing works lay the foundation of AD algorithm design, and we further improve the foundation by considering more datasets, algorithms, and comparison aspects.

# B  More Details on ADBench

## B.1  ADBench Algorithm List

We organize all the algorithms in ADBench into the following three categories and report their hyperparameter settings which mainly refer to the settings of their original papers or repositories (e.g., PyOD[1] and scikit-learn[2]).

*(i) 14 unsupervised algorithms*:

1. **Principal Component Analysis (PCA)** [162]. PCA is a linear dimensionality reduction using singular value decomposition of the data to project it to a lower dimensional space. When used for AD, it projects the data to the lower dimensional space and then uses the reconstruction errors as the anomaly scores. If not specified, the default hyperparameters in PyOD are used for the PCA (and the other unsupervised algorithms deployed by PyOD).

---

[1] https://pyod.readthedocs.io/en/latest/pyod.html
[2] https://scikit-learn.org/stable/

2. **One-class SVM (OCSVM)** [157]. OCSVM maximizes the margin between the origin and the normal samples, and defines the decision boundary as the hyperplane that determines the margin.
3. **Local Outlier Factor (LOF)** [22]. LOF measures the local deviation of the density of a given sample with respect to its neighbors.
4. **Clustering Based Local Outlier Factor (CBLOF)** [64]. CBLOF calculates the anomaly score by first assigning samples to clusters, and then using the distance among clusters as anomaly scores.
5. **Connectivity-Based Outlier Factor (COF)** [167]. COF uses the ratio of the average chaining distance of data points and the average chaining distance of $k$-th nearest neighbor of the data point, as the anomaly score for observations.
6. **Histogram- based outlier detection (HBOS)** [52]. HBOS assumes feature independence and calculates the degree of outlyingness by building histograms.
7. **K-Nearest Neighbors (KNN)** [144]. KNN views the anomaly score of the input instance as the distance to its $k$-th nearest neighbor.
8. **Subspace Outlier Detection (SOD)** [80]. SOD aims to detect outliers in varying subspaces of high-dimensional feature space.
9. **Copula Based Outlier Detector (COPOD)** [96]. COPOD is a hyperparameter-free, highly interpretable outlier detection algorithm based on empirical copula models.
10. **Empirical-Cumulative-distribution-based Outlier Detection (ECOD)** [97]. ECOD is a hyperparameter-free, highly interpretable outlier detection algorithm based on empirical CDF functions. Basically, it uses ECDF to estimate the density of each feature independently, and assumes that outliers locate the tails of the distribution.
11. **Deep Support Vector Data Description (DeepSVDD)** [151]. DeepSVDD trains a neural network while minimizing the volume of a hypersphere that encloses the network representations of the data, forcing the network to extract the common factors of variation.
12. **Deep Autoencoding Gaussian Mixture Model (DAGMM)** [206]. DAGMM utilizes a deep autoencoder to generate a low-dimensional representation and reconstruction error for each input data point, which is further fed into a Gaussian Mixture Model (GMM). We train the DAGMM for 200 epochs with 256 batch size, where the patience of early stopping is set to 50. The learning rate of Adam [77] optimizer is 0.0001 and is decayed once the number of epochs reaches 50. The latent dimension of DAGMM is set to 1 and the number of Gaussian mixture components is set to 4. The $\lambda_1$ and $\lambda_2$ for energy and covariance in the objective function are set to 0.1 and 0.005, respectively.
13. **Lightweight on-line detector of anomalies (LODA)** [136]. LODA is an ensemble method and is particularly useful in domains where a large number of samples need to be processed in real-time or in domains where the data stream is subject to concept drift and the detector needs to be updated online.
14. **Isolation Forest (IForest)** [100]. IForest isolates observations by randomly selecting a feature and then randomly selecting a split value between the maximum and minimum values of the selected feature.

*(ii) 7 semi-supervised algorithms*:

1. **Semi-Supervised Anomaly Detection via Adversarial Training (GANomaly)** [7]. A GAN-based method that defines the reconstruction error of the input instance as the anomaly score. We replace the convolutional layer in the original GANomaly with the dense layer with tanh activation function for evaluating it on the tabular data, where the hidden size of the encoder-decoder-encoder structure of GANomaly is set to half of the input dimension. We train the GANomaly for 50 epochs with 64 batch size, where the SGD [149] optimizer with 0.01 learning rate and 0.7 momentum is applied for both the generator and the discriminator.
2. **Deep Semi-supervised Anomaly Detection (DeepSAD)** [152]. A deep one-class method that improves the unsupervised DeepSVDD [151] by penalizing the inverse of the distances of anomaly representation such that anomalies must be mapped further away from the hypersphere center. The hyperparameter $\eta$ in the loss function is set to 1.0, where DeepSAD is trained for 50 epochs with 128 batch size. Adam optimizer with 0.001 learning rate and $10^{-6}$ weight decay is applied for updating the network parameters. DeepSAD additionally employs an autoencoder for calculating the initial center of the hypersphere, where the autoencoder is trained for 100 epochs with 128 batch size, and optimized by Adam optimizer with learning rate 0.001 and $10^{-6}$ weight decay.
3. **REPresentations for a random nEarest Neighbor distance-based method (REPEN)** [127]. A neural network-based model that leverages transformed low-dimensional representation for

random distance-based detectors. The hidden size of REPEN is set to 20, and the margin of triplet loss is set to 1000. REPEN is trained for 30 epochs with 256 batch size, where the total number of steps (batches of samples) is set to 50. Adadelta [187] optimizer with 0.001 learning rate and 0.95 $\rho$ is applied to update network parameters.

4. **Deviation Networks (DevNet)** [131]. A neural network-based model uses a prior probability to enforce a statistical deviation score of input instances. The margin hyperparameter $a$ in the deviation loss is set to 5. DevNet is trained for 50 epochs with 512 batch size, where the total number of steps is set to 20. RMSprop [149] optimizer with 0.001 learning rate and 0.95 $\rho$ is applied to update network parameters.

5. **Pairwise Relation prediction-based ordinal regression Network (PReNet)** [130]. A neural network-based model that defines a two-stream ordinal regression to learn the relation of instance pairs. The score targets of {unlabeled, unlabeled}, {labeled, unlabeled} and {labeled, labeled} sample pairs are set to 0, 4 and 8, respectively. PReNet is trained for 50 epochs with 512 batch size, where the total number of steps is set to 20. RMSprop optimizer with a learning rate of 0.001 and 0.01 weight decay is applied to update network parameters.

6. **Feature Encoding With Autoencoders for Weakly Supervised Anomaly Detection (FEAWAD)** [204]. A neural network-based model that incorporates the network architecture of DAGMM [206] with the deviation loss of DevNet [131]. FEAWAD is trained for 30 epochs with 512 batch size, where the total number of steps is set to 20. Adam optimizer with 0.0001 learning rate is applied to update network parameters.

7. **Extreme Gradient Boosting Outlier Detection (XGBOD)** [195]. XGBOD first uses the passed-in unsupervised outlier detectors to extract richer representations of the data and then concatenates the newly generated features to the original feature for constructing the augmented feature space. An XGBoost classifier is then applied to this augmented feature space. We use the default hyperparameters in PyOD.

*(iii) 9 supervised algorithms*:

1. **Naive Bayes (NB)** [14]. NB methods are based on applying Bayes' theorem with the "naive" assumption of conditional independence between every pair of features given the value of the class variable. We use the Gaussian NB in ADBench.

2. **Support Vector Machine (SVM)** [31]. SVM is effective in high-dimensional spaces and could still be effective in cases where the number of dimensions is greater than the number of samples. We use the default hyperparameters in scikit-learn for SVM (and for the following MLP and RF).

3. **Multi-layer Perceptron (MLP)** [148]. MLP uses the binary cross entropy loss to update network parameters.

4. **Random Forest (RF)** [21]. RF is a meta estimator that fits several decision tree classifiers on various sub-samples of the dataset and uses averaging to improve the predictive accuracy and control over-fitting.

5. **eXtreme Gradient Boosting (XGBoost)** [29]. XGBoost is an optimized distributed gradient boosting method designed to be highly efficient, flexible, and portable. We use the default hyperparameter settings in the XGBoost official repository[1].

6. **Highly Efficient Gradient Boosting Decision Tree (LightGBM)** [74]. LightGBM is a gradient boosting framework that uses tree-based learning algorithms with faster training speed, higher efficiency, lower memory usage, and better accuracy. The default hyperparameter settings in the LightGBM official repository[2] are used.

7. **Categorical Boosting (CatBoost)** [138]. CatBoost is a fast, scalable, high-performance gradient boosting on decision trees. CatBoost uses the default hyperparameter settings in its official repository[3].

8. **Residual Nets (ResNet)** [56]. This method introduces a ResNet-like architecture [62] for tabular based data. ResNet is trained for 100 epochs with 64 batch size. AdamW [108] optimizer with 0.001 learning rate is applied to update network parameters.

9. **Feature Tokenizer + Transformer (FTTransformer)** [56]. FTTransformer is an effective adaptation of the Transformer architecture [171] for tabular data. FTTransformer is trained for 100 epochs with 64 batch size. AdamW optimizer with 0.0001 learning rate and $10^{-5}$ weight decay is applied to update network parameters.

---

[1]https://xgboost.readthedocs.io/en/stable/parameter.html

[2]https://lightgbm.readthedocs.io/en/latest/Parameters.html

[3]https://catboost.ai/en/docs/references/training-parameters/

## B.2 ADBench Dataset List

**Overview**. As described in §3.2, ADBench is the largest AD benchmark with 57 datasets. More specifically, Table B1 shows the datasets used in ADBench, covering many application domains, including healthcare (e.g., disease diagnosis), audio and language processing (e.g., speech recognition), image processing (e.g., object identification), finance (e.g., financial fraud detection), and more, where we show this information in the last column. We resample the sample size to 1,000 for those datasets smaller than 1,000, and use the subsets of 10,000 for those datasets greater than 10,000 due to the computational cost. Fig. B1 provides the anomaly ratio distribution of the datasets, where the median is equal to 5%. We release all the datasets and their raw version(s) when possible at https://github.com/Minqi824/ADBench/tree/main/datasets.

**Newly-added Datasets in ADBench**. Since most of the public datasets are relatively small and simple, we introduce 10 more complex datasets from CV and NLP domains with more samples and richer features in ADBench (highlighted in Table B1 in blue).

*Reasoning of Using CV/NLP Datasets*. It is often challenging to directly run large CV and NLP datasets on selected shallow methods, e.g., OCSVM [157] and kNN [144] with high time complexity, we follow DeepSAD [152], ADIB [33], and DATE [115] to extract representations of CV and NLP datasets by neural networks for downstream detection tasks. More specifically, ADIB [33] shows that "transferring features from semantic tasks can provide powerful and generic representations for various AD problems", which is true even when the pre-trained task is only loosely related to downstream AD tasks. Similarly, DeepSAD [152] uses pre-trained autoencoder to extract features for training classical AD detectors like OCSVM [157] and IForest [100]. For NLP datasets, DATE [115] uses fastText [72] and Glove [134] embeddings for evaluating classical AD methods (e.g., OCSVM [157] and IForest [100]) against proposed methods in NLP datasets.

We want to elaborate further on the reasons for adapting CV and NLP datasets for tabular AD. First, some shallow models, such as OCSVM [157], cannot directly run on (large, high-dimensional) CV datasets. Second, it is interesting to see whether tabular AD methods can work on CV/NLP data representations, which carry values in real-world applications where deep models are infeasible to run. Moreover, the extracted representations often lead to better downstream detection results [33]. Thus, we extract features from CV and NLP datasets by deep models to create "tabular" versions of them. Although not perfect, this may provide insights into shallow methods' performance on (originally infeasible) CV and NLP datasets.

*CV Datasets*: For MNIST-C, we set original MNIST images to be normal and corrupted images in MNIST-C to be abnormal, like in the recent work [91]. For MVTec-10, we test different AD algorithms on the 15 image sets, where anomalies correspond to various manufacturing defects. For CIFAR10, FashionMNIST, and SVHN, we follow previous works [151, 152] and set one of the multi-classes as normal and downsample the remaining classes to 5% of the total instances as anomalies by default, and report the average results over all the respective classes.

*NLP Datasets*: For Amazon and Imdb, we regard the original negative class as the anomaly class. For Yelp, we regard the reviews of 0 and 1 stars as the anomaly class, and the reviews of 3 and 4 stars as the normal class. For 20newsgroups dataset, like in DATE [115] and CVDD [153], we only take the articles from the six top-level classes: *computer*, *recreation*, *science*, *miscellaneous*, *politics*, *religion*. Similarly, for the multi-classes datasets 20newsgroups and Agnews, we set one of the classes as normal and downsample the remaining classes to 5% of the total instances as anomalies.

*Backbone Choices of Feature Extraction*. Pretrained models are applied to extract data embedding from CV and NLP datasets to access these more complex representations. For CV datasets, following [16] and [147], we use ResNet18[1] [62] pretrained on the ImageNet [35] to extract meaningful embedding after the last average pooling layer. We also provide the embedding version that are extracted by the ImageNet-pretrained ViT[2] [39]. For NLP datasets, instead of using traditional embedding methods like fastText [19, 72] or Glove [134], we apply BERT[3] [75] pretrained on the BookCorpus and English Wikipedia to extract more enriching embedding of the [CLS] token. In addition, we provide the embedding version that are extracted by the pretrained RoBERTa[4] [105]

---

[1]https://pytorch.org/hub/pytorch_vision_resnet/

[2]https://github.com/lukemelas/PyTorch-Pretrained-ViT

[3]https://huggingface.co/bert-base-uncased

[4]https://huggingface.co/roberta-base

in our codebase[1]. Although we release all the generated datasets for completeness, we analyze the results based on the datasets generated by BERT and ResNet18. Future work may consider analyzing the impact of backbones on detection performance.

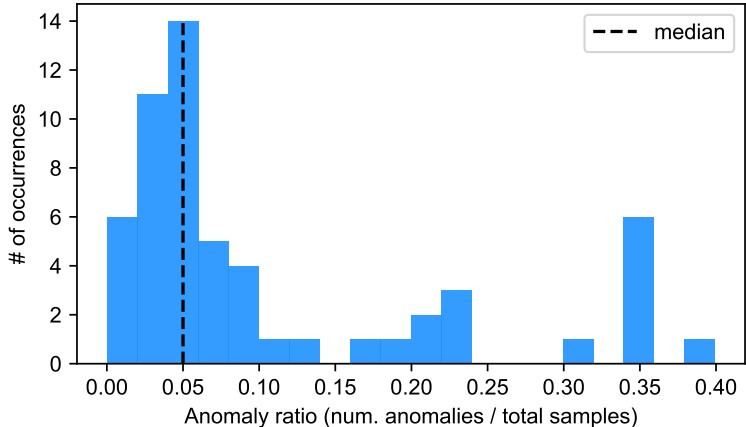

Figure B1: Distribution of anomaly ratios in 57 datasets in ADBench, where 40 datasets' anomaly ratio is below 10% (median=5%).

---
[1]https://github.com/Minqi824/ADBench/tree/main/datasets

Table B1: Data description of the 57 datasets included in ADBench; 10 newly added datasets from CV and NLP domain are highlighted in blue at the bottom of the table.

| Data | # Samples | # Features | # Anomaly | % Anomaly | Category | Reference |
|---|---|---|---|---|---|---|
| ALOI | 49534 | 27 | 1508 | 3.04 | Image | [42] |
| annthyroid | 7200 | 6 | 534 | 7.42 | Healthcare | [141] |
| backdoor | 95329 | 196 | 2329 | 2.44 | Network | [119] |
| breastw | 683 | 9 | 239 | 34.99 | Healthcare | [173] |
| campaign | 41188 | 62 | 4640 | 11.27 | Finance | [131] |
| cardio | 1831 | 21 | 176 | 9.61 | Healthcare | [12] |
| Cardiotocography | 2114 | 21 | 466 | 22.04 | Healthcare | [12] |
| celeba | 202599 | 39 | 4547 | 2.24 | Image | [131] |
| census | 299285 | 500 | 18568 | 6.20 | Sociology | [131] |
| cover | 286048 | 10 | 2747 | 0.96 | Botany | [18] |
| donors | 619326 | 10 | 36710 | 5.93 | Sociology | [131] |
| fault | 1941 | 27 | 673 | 34.67 | Physical | [42] |
| fraud | 284807 | 29 | 492 | 0.17 | Finance | [131] |
| glass | 214 | 7 | 9 | 4.21 | Forensic | [43] |
| Hepatitis | 80 | 19 | 13 | 16.25 | Healthcare | [36] |
| http | 567498 | 3 | 2211 | 0.39 | Web | [145] |
| InternetAds | 1966 | 1555 | 368 | 18.72 | Image | [25] |
| Ionosphere | 351 | 33 | 126 | 35.90 | Oryctognosy | [163] |
| landsat | 6435 | 36 | 1333 | 20.71 | Astronautics | [42] |
| letter | 1600 | 32 | 100 | 6.25 | Image | [48] |
| Lymphography | 148 | 18 | 6 | 4.05 | Healthcare | [26] |
| magic.gamma | 19020 | 10 | 6688 | 35.16 | Physical | [42] |
| mammography | 11183 | 6 | 260 | 2.32 | Healthcare | [176] |
| mnist | 7603 | 100 | 700 | 9.21 | Image | [90] |
| musk | 3062 | 166 | 97 | 3.17 | Chemistry | [37] |
| optdigits | 5216 | 64 | 150 | 2.88 | Image | [10] |
| PageBlocks | 5393 | 10 | 510 | 9.46 | Document | [113] |
| pendigits | 6870 | 16 | 156 | 2.27 | Image | [9] |
| Pima | 768 | 8 | 268 | 34.90 | Healthcare | [145] |
| satellite | 6435 | 36 | 2036 | 31.64 | Astronautics | [145] |
| satimage-2 | 5803 | 36 | 71 | 1.22 | Astronautics | [145] |
| shuttle | 49097 | 9 | 3511 | 7.15 | Astronautics | [145] |
| skin | 245057 | 3 | 50859 | 20.75 | Image | [42] |
| smtp | 95156 | 3 | 30 | 0.03 | Web | [145] |
| SpamBase | 4207 | 57 | 1679 | 39.91 | Document | [25] |
| speech | 3686 | 400 | 61 | 1.65 | Linguistics | [23] |
| Stamps | 340 | 9 | 31 | 9.12 | Document | [25] |
| thyroid | 3772 | 6 | 93 | 2.47 | Healthcare | [142] |
| vertebral | 240 | 6 | 30 | 12.50 | Biology | [17] |
| vowels | 1456 | 12 | 50 | 3.43 | Linguistics | [82] |
| Waveform | 3443 | 21 | 100 | 2.90 | Physics | [107] |
| WBC | 223 | 9 | 10 | 4.48 | Healthcare | [114] |
| WDBC | 367 | 30 | 10 | 2.72 | Healthcare | [114] |
| Wilt | 4819 | 5 | 257 | 5.33 | Botany | [25] |
| wine | 129 | 13 | 10 | 7.75 | Chemistry | [2] |
| WPBC | 198 | 33 | 47 | 23.74 | Healthcare | [114] |
| yeast | 1484 | 8 | 507 | 34.16 | Biology | [66] |
| CIFAR10 | 5263 | 512 | 263 | 5.00 | Image | [81] |
| FashionMNIST | 6315 | 512 | 315 | 5.00 | Image | [178] |
| MNIST-C | 10000 | 512 | 500 | 5.00 | Image | [120] |
| MVTec-AD | | See Table B2. | | | Image | [16] |
| SVHN | 5208 | 512 | 260 | 5.00 | Image | [121] |
| Agnews | 10000 | 768 | 500 | 5.00 | NLP | [192] |
| Amazon | 10000 | 768 | 500 | 5.00 | NLP | [63] |
| Imdb | 10000 | 768 | 500 | 5.00 | NLP | [111] |
| Yelp | 10000 | 768 | 500 | 5.00 | NLP | [192] |
| 20newsgroups | | See Table B3. | | | NLP | [86] |

Table B2: Detailed description of the MVTec-AD dataset; see the full dataset list in Table B1. For MVTec-AD dataset, we evaluate 30 algorithms on each class and report the average performance of all classes.

| Class | # Samples | # Features | # Anomaly | % Anomaly |
|---|---|---|---|---|
| Carpet | 397 | 512 | 89 | 22.42 |
| Grid | 342 | 512 | 57 | 16.67 |
| Leather | 369 | 512 | 92 | 24.93 |
| Tile | 347 | 512 | 84 | 24.21 |
| Wood | 326 | 512 | 60 | 18.40 |
| Bottle | 292 | 512 | 63 | 21.58 |
| Cable | 374 | 512 | 92 | 24.60 |
| Capsule | 351 | 512 | 109 | 31.05 |
| Hazelnut | 501 | 512 | 70 | 13.97 |
| Metal Nut | 335 | 512 | 93 | 27.76 |
| Pill | 434 | 512 | 141 | 32.49 |
| Screw | 480 | 512 | 119 | 24.79 |
| Toothbrush | 102 | 512 | 30 | 29.41 |
| Transistor | 313 | 512 | 40 | 12.78 |
| Zipper | 391 | 512 | 119 | 30.43 |
| Total | 5354 | 512 | 1258 | 23.50 |

Table B3: Detailed description of the 20newsgroups dataset; see the full dataset list in Table B1. For 20newsgroups dataset, we evaluate 30 algorithms on each class and report the average performance of all classes.

| Class | # Samples | # Features | # Anomaly | % Anomaly |
|---|---|---|---|---|
| Computer | 3090 | 768 | 154 | 4.98 |
| Recreation | 2514 | 768 | 125 | 4.97 |
| Science | 2497 | 768 | 124 | 4.97 |
| Miscellaneous | 615 | 768 | 30 | 4.88 |
| Politics | 1657 | 768 | 82 | 4.95 |
| Religion | 1532 | 768 | 76 | 4.96 |
| Total | 11905 | 768 | 591 | 4.96 |

## B.3 Additional Demonstration of Synthetic Anomalies for §3.3.2

In addition to Fig. 3 that demonstrates the synthetic anomalies on Lymphography dataset in §3.3.2, we provide another example here for Ionosphere data.

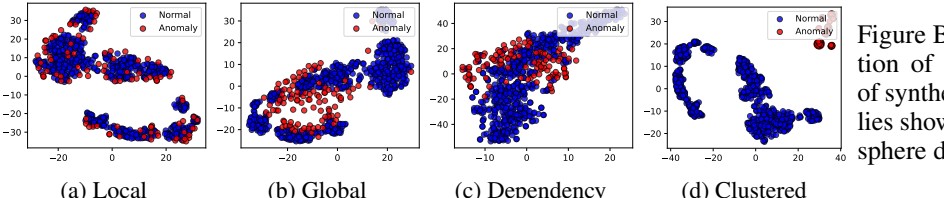

| (a) Local | (b) Global | (c) Dependency | (d) Clustered |

Figure B2: Illustration of four types of synthetic anomalies shown on Ionosphere dataset.

## B.4 Open-source Release

As mentioned before, the full experiment code, datasets, and examples of benchmarking new algorithms are available at `https://github.com/Minqi824/ADBench`. We specify the key environment setting of using ADBench, e.g., `scikit-learn==0.20.3`, `pyod==0.9.8`, etc. With our interactive example in Jupyter notebooks, one may compare a newly proposed AD algorithm easily.

# C   Details on Experiment Setting

We provide additional details on experiment setting to §4.1 in this section.

**General Experimental Settings**. Although unsupervised AD algorithms are primarily designed for the transductive setting (i.e., outputting the anomaly scores on the input data only other than making predictions on newcoming data), we adapt all the algorithms for the inductive setting to predict the newcoming data, which is helpful in applications and also common in popular AD library PyOD [198], TODS [84, 85], and PyGOD [102]. Thus, we use $70\%$ data for training and the remaining $30\%$ as a testing set. We use stratified sampling to keep the anomaly ratio consistent. We repeat each experiment 3 times and report the average. The 10 complex CV and NLP datasets are mainly considered for evaluating algorithm performance on the public datasets and are not included in the experiments of different types of anomalies and algorithm robustness, since such high-dimensional data could make it hard to generate synthetic anomalies (e.g., the Vine Copula is computationally expensive for fitting such high-dimensional data), or introduce too much noise in input data (e.g., the noise ratio of irrelevant features $50\%$ would lead to 384 noise features in the 768 input dimensions of NLP data). Future works may resort to the help of the latest generative methods like diffusion models [184].

**Hyperparameter Settings**. For all the algorithms in ADBench, we use their default hyperparameter (HP) settings in the original paper for a fair comparison. Specific values can be found in Appx.B.1 and our codebase[1]. It is also acknowledged that it is possible to use a small hold-out data for hyperparameter tuning for semi- and fully-supervised methods [164], while we do not consider this setting in this work.

**Extensive Experiments**. In total ADBench conducts 98,436 experiments, where each denotes one algorithm's result on a dataset under a specific setting. More specifically, we have 27,090 experiments in §4.2. For 47 classical datasets:

- Unsupervised methods on benchmark ~~real-world~~ datasets {14 algorithms, 47 datasets, 3 repeat times} leads to 1,974 experiments.
- Semi- and fully-supervised on real-world datasets {16 algorithms, 47 datasets, 3 repeat times, 7 settings of labeled anomalies} leads to 15,792 experiments.

As we described in Appx. B.2, we totally have 74 subclasses for the 10 CV and NLP datasets, thus generating:

- Unsupervised methods on benchmark datasets {14 algorithms, 74 subclasses, 1 repeat times} leads to 1,036 experiments.
- Semi- and fully-supervised on real-world datasets {16 algorithms, 74 subclasses, 1 repeat times, 7 settings of labeled anomalies} leads to 8,288 experiments.

Additionally, we have 17,766 experiments for understanding the algorithm performances under four types of anomalies in §4.3:

- Unsupervised methods on benchmark ~~real-world~~ datasets {14 algorithms, 47 datasets, 3 repeat times} leads to 1,974 experiments.
- Semi- and fully-supervised on benchmark datasets {16 algorithms, 47 datasets, 3 repeat times, 7 settings of labeled anomalies} leads to 15,792 experiments.

Finally, we have 53,580 experiments for evaluating the algorithm robustness under three settings of data noises and corruptions in §4.4:

- For duplicated anomalies and irrelevant features {30 algorithms, 47 datasets, 3 repeat times, 5 settings of data noises, 2 scenarios} leads to 42,300 experiments.
- For annotation errors {16 algorithms, 47 datasets, 3 repeat times, 5 settings of data noises} leads to 11,280 experiments.

**Computational Resources**. Classical anomaly detection models are run on an Intel i7-8700 @3.20 GHz, 16GB RAM, 12-core workstation. For deep learning models (especially for ResNet and FTTransformer), we run experiments on an NVIDIA Tesla V100 GPU accelerator. The model runtime on benchmark datasets is reported in Appx. §D.1.

---

[1]ADBench repo: `https://github.com/Minqi824/ADBench`

# D Additional Experiment Results

## D.1 Additional Results for Overall Model Performance on Benchmark Datasets in §4.2

In addition to the AUCROC results presented in §4.2, we also show the AUCPR results of model performance on 57 benchmark datasets in Fig. D3, where the corresponding conclusions are similar to that of AUCROC results. There is still no statistically superior solution for unsupervised methods regarding AUCPR. Semi-supervised methods perform better than supervised methods when only limited label data is available, say the labeled anomalies $\gamma_l$ is less than $5\%$. Besides, we show that the semi-supervised GANomaly, which learns an intermediate representation of the normal data, performs worse than those anomaly-informed models leveraging labeled anomalies (see Fig. D3(b)). This conclusion verifies that merely capturing the normal behaviors is not enough for detecting the underlying anomalies, where the lack of knowledge about the true anomalies would lead to high false positives/negatives [128, 130, 131].

Fig. D4 and D5 show the boxplots of AUCROC and AUCPR of 30 algorithms on the 57 benchmark datasets. These results validate the no-free-lunch theorem, where no model is both the best and the most stable performer. For example, DeepSVDD and RF are the most stable detectors among un- and fully-supervised methods, respectively, but they are inferior to most of the other algorithms. Besides, IForest and CatB(oost) can be regarded as two very competitive methods among un- and fully-supervised methods, respectively, but their variances of model performance are relatively large compared to the other methods.

Additionally, we also present the full results in tables in §D.4.

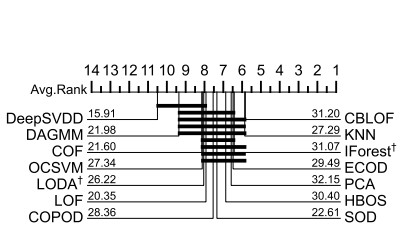

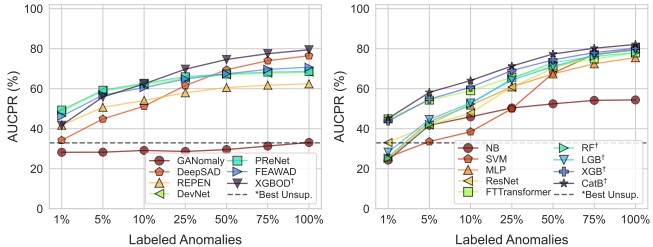

(a) Avg. rank (lower the better) and avg. AUCPR (on each line) of unsupervised methods; groups of algorithms not statistically different are connected horizontally.

(b) Avg. AUCPR (on 57 datasets) vs. % of labeled anomalies (x-axis); semi-supervised (left) and fully-supervised (right). Most label-informed algorithms outperform the best *unsupervised* algorithm CBLOF (denoted as the dashed line) with 10% labeled anomalies.

Figure D3: AD model's AUCPR on 57 benchmark datasets. Generally, the AUCPR results are consistent with the AUCROC results in §4.2. (a) shows that no unsupervised algorithm can statistically outperform. (b) shows the AUCPR of semi- and supervised methods under varying ratios of labeled anomalies $\gamma_l$. The semi-supervised methods leverage the labels more efficiently w/ small $\gamma_l$.

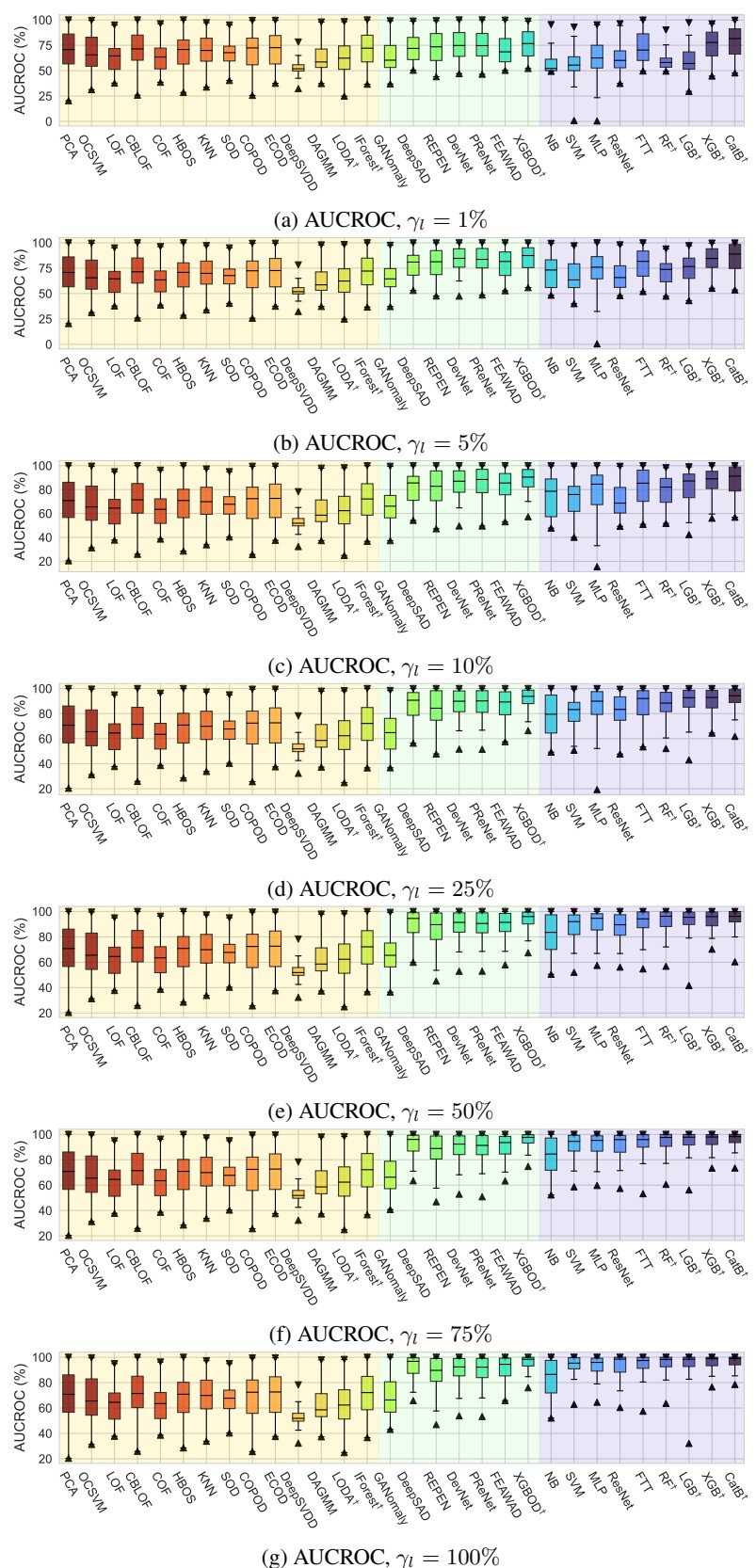

(a) AUCROC, $\gamma_l = 1\%$

(b) AUCROC, $\gamma_l = 5\%$

(c) AUCROC, $\gamma_l = 10\%$

(d) AUCROC, $\gamma_l = 25\%$

(e) AUCROC, $\gamma_l = 50\%$

(f) AUCROC, $\gamma_l = 75\%$

(g) AUCROC, $\gamma_l = 100\%$

Figure D4: Boxplot of AUCROC. We denote unsupervised methods in ▨ (light yellow), semi-supervised methods in ▨ (light green), and supervised methods in ▨ (light purple). Consistent with the CD diagrams, we notice that none of the unsupervised methods visually outperform.

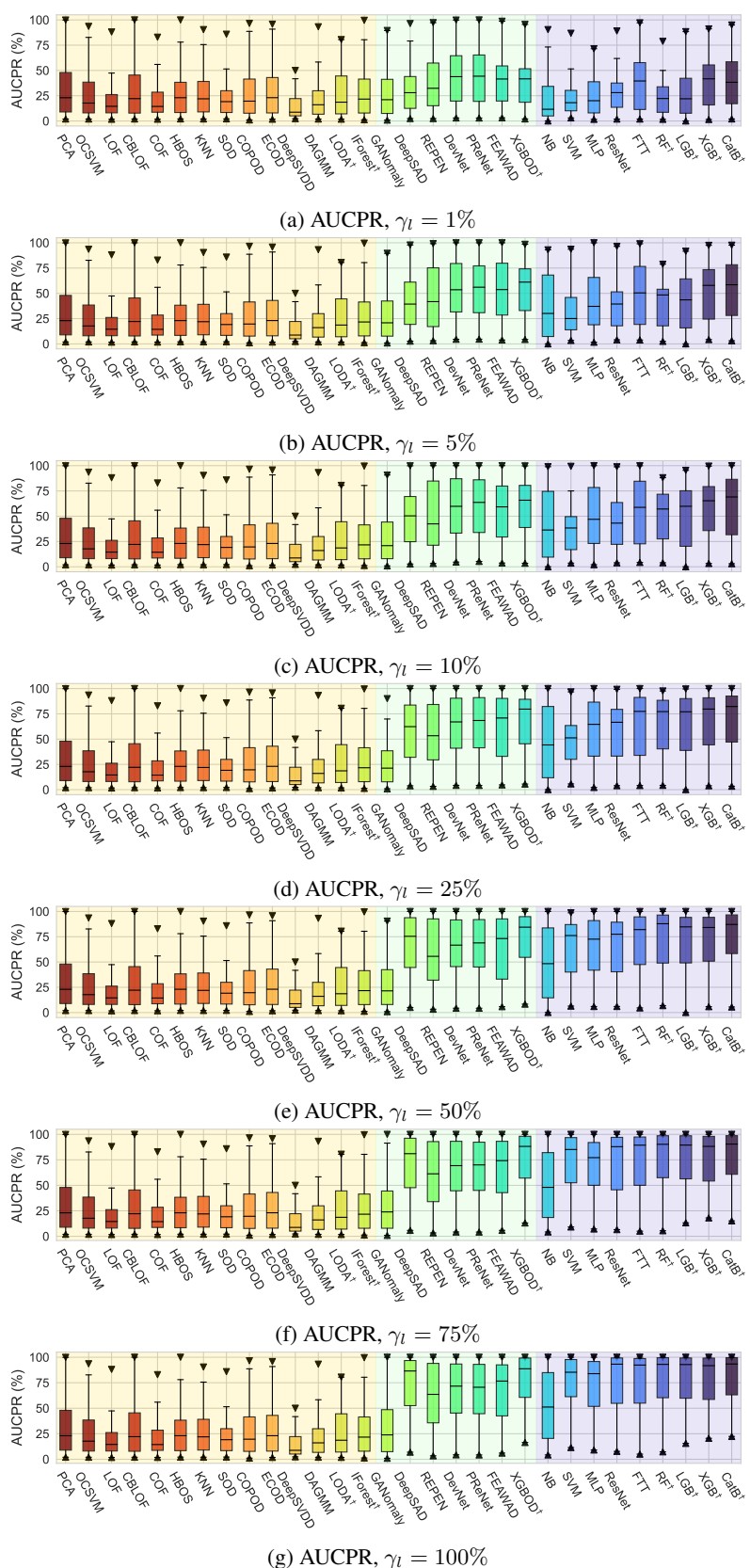

(a) AUCPR, $\gamma_l = 1\%$

(b) AUCPR, $\gamma_l = 5\%$

(c) AUCPR, $\gamma_l = 10\%$

(d) AUCPR, $\gamma_l = 25\%$

(e) AUCPR, $\gamma_l = 50\%$

(f) AUCPR, $\gamma_l = 75\%$

(g) AUCPR, $\gamma_l = 100\%$

Figure D5: Boxplot of AUCPR. We denote unsupervised methods in ▢ (light yellow), semi-supervised methods in ▢ (light green), and supervised methods in ▢ (light purple). Consistent with the CD diagrams, we notice that none of the unsupervised methods visually outperform.

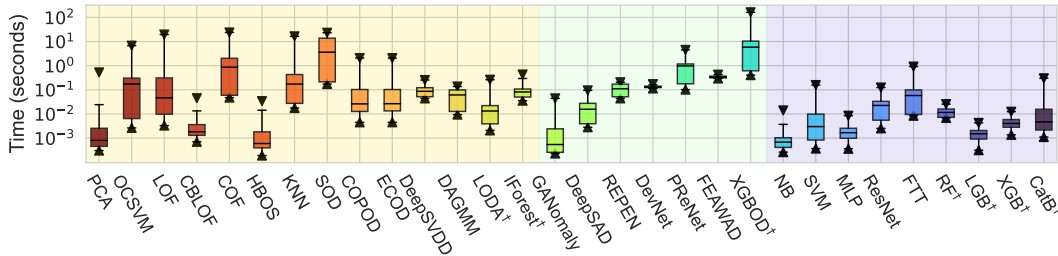

Figure D6: Inference time of included algorithms. We denote unsupervised methods in ▮ (light yellow), semi-supervised methods in ▮ (light green), and supervised methods in ▮ (light purple). Consistent with the train time in Fig. 4d, PCA, HBOS, GANomaly and NB take the least inference time on test datasets, while more complex feature representation methods like SOD and XGBOD spend more time due to the search of the feature subspace.

## D.2 Additional Results for Different Types of Anomalies §4.3

We additionally show the AUCPR results for model performance on different types of anomalies in Fig. D7 and Fig. D8, which are consistent with the conclusions drawn in §4.3, i.e., the unsupervised methods are significantly better if their model assumptions conform to the underlying anomaly types. Moreover, the prior knowledge of anomaly types can be more important than that of label information, where those label-informed algorithms generally underperform the best unsupervised methods for local, global, and dependency anomalies.

We want to note that XGBOD can be regarded as an exception to the above observations, which is comparable to or even outperforms the best unsupervised model when more labeled anomalies are available. Recall that XGBOD employs the stacking ensemble method [174], where heterogeneous unsupervised methods are integrated with the supervised model XGBoost, therefore XGBOD is more adaptable to different data assumptions while effectively leveraging the label information. This validates the conclusion that such ensemble learning techniques should be considered in future research directions.

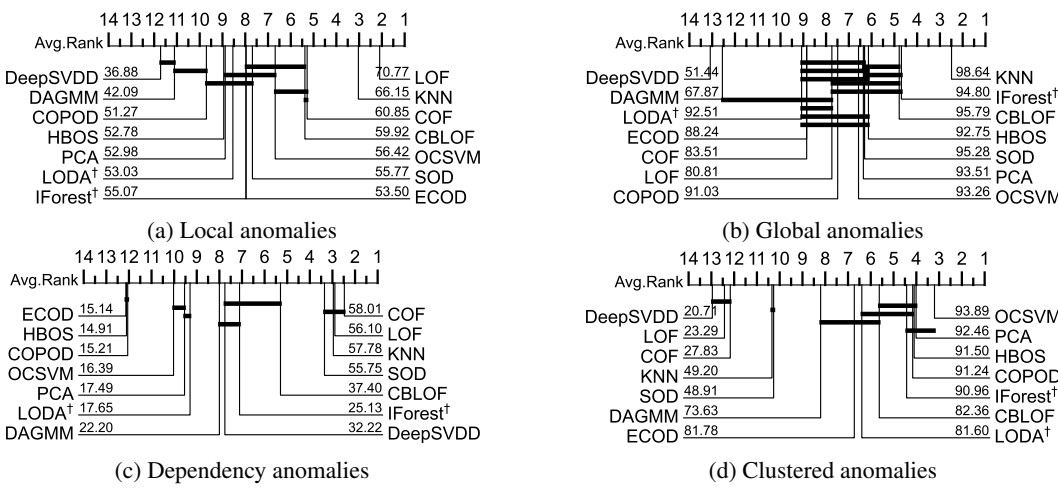

Figure D7: AUCPR CD Diagram of unsupervised methods on different types of anomalies. The unsupervised methods perform well when their assumptions conform to the anomaly types.

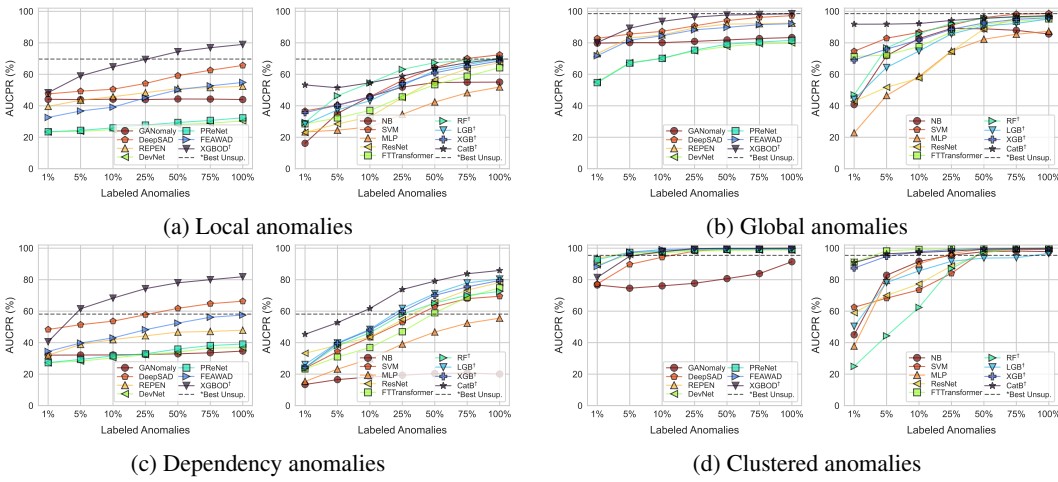

Figure D8: Semi- (left of each subfigure) and supervised (right) algorithms' performance on different types of anomalies with varying levels of labeled anomalies for AUCPR performance. Surprisingly, these label-informed algorithms are *inferior* to the best unsupervised method except for the clustered anomalies.

## D.3 Additional Results for Algorithm Robustness in §4.4

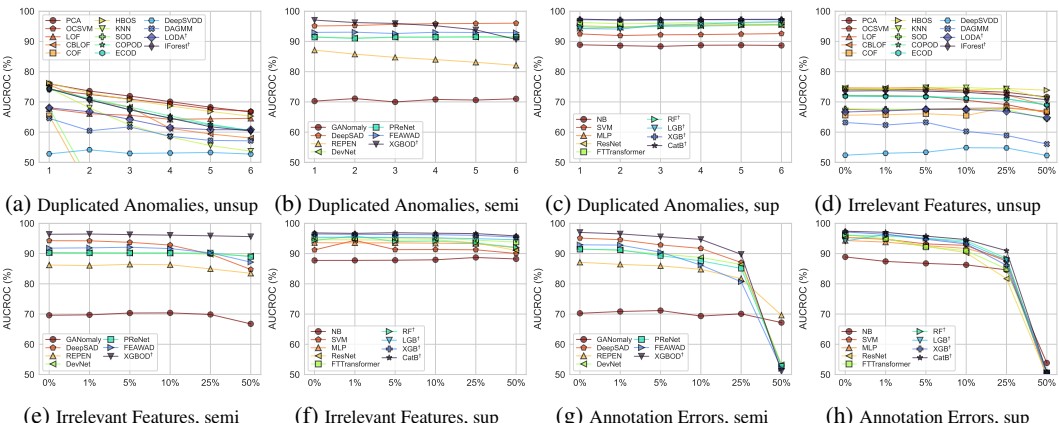

(a) Duplicated Anomalies, unsup    (b) Duplicated Anomalies, semi    (c) Duplicated Anomalies, sup    (d) Irrelevant Features, unsup

(e) Irrelevant Features, semi    (f) Irrelevant Features, sup    (g) Annotation Errors, semi    (h) Annotation Errors, sup

Figure D9: Algorithm performance under noisy and corrupted data (i.e., duplicated anomalies for (a)-(c), irrelevant features for (d)-(f), and annotation errors for (g) and (h)). X-axis denotes either the duplicated times or the noise ratio. Y-axis denotes the AUCROC performance and its range remains consistent across different algorithms. The results reveal unsupervised methods' susceptibility to duplicated anomalies and the usage of label information in defending irrelevant features. Un-, semi-, and fully-supervised methods are denoted as *unsup*, *semi*, and *sup*, respectively. The results are mostly consistent with the observations in Fig. 7 (§4.4) showing the relative performance change.

In Fig. D9, we provide the performance of the AD algorithms under noisy and corrupted data. Along with the relative performance changes shown in Fig. 7, the analysis in 4.4 still stands.

In addition to the primary results shown in §4.4, we provide the AUCPR results for algorithm robustness in Fig. D10 and D11. The AUCPR results confirm the robustness of supervised methods for irrelevant features. Besides, both semi- and fully-supervised methods are robust to minor annotation errors, say the annotation errors are less than $10\%$.

One thing to note is we observe AUCPR performance improves under the setting of duplicated anomalies (see Fig. D10 (a)-(c)). This is expected as AUCPR emphasizes the positive classes (i.e., anomalies), and more duplicated anomalies favor this metric. Since this observation is consistently true for both unsupervised and label-informed methods, it would not largely impact our selection of algorithms. However, if we care about both anomaly and normal classes equally, the results on AUCROC in §4.4 still stand —unsupervised methods are more susceptible to duplicate anomalies.

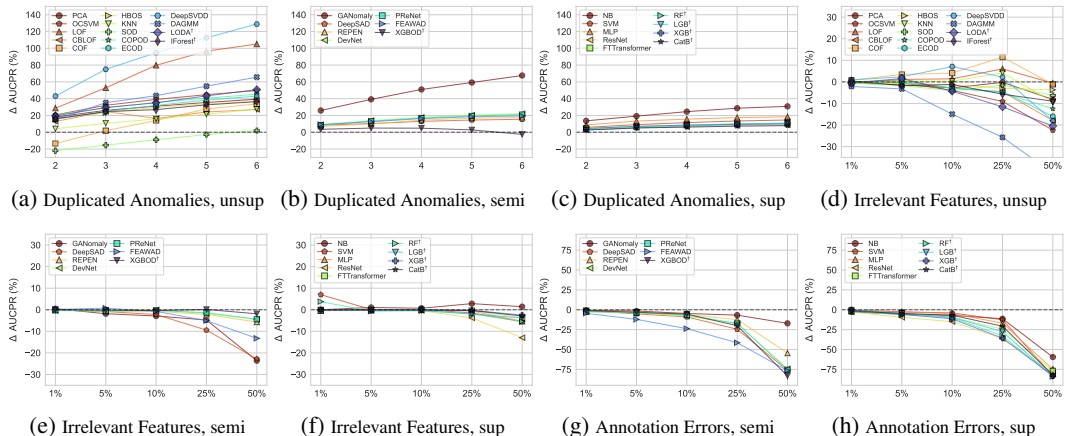

Figure D10: Algorithm performance change under noisy and corrupted data (i.e., duplicated anomalies for (a)-(c), irrelevant features for (d)-(f), and annotation errors for (g) and (h)). y-axis denotes the % of performance change (ΔAUCPR) and its range remains consistent across different algorithms. The results reveal the usage of label information in defending irrelevant features, and the robustness of label-informed methods to the minor annotation errors. Un-, semi-, and fully-supervised methods are denoted as *unsup*, *semi*, and *sup*, respectively. The results are mostly consistent with the observations in Fig. 7 (§4.4) showing the AUCROC.

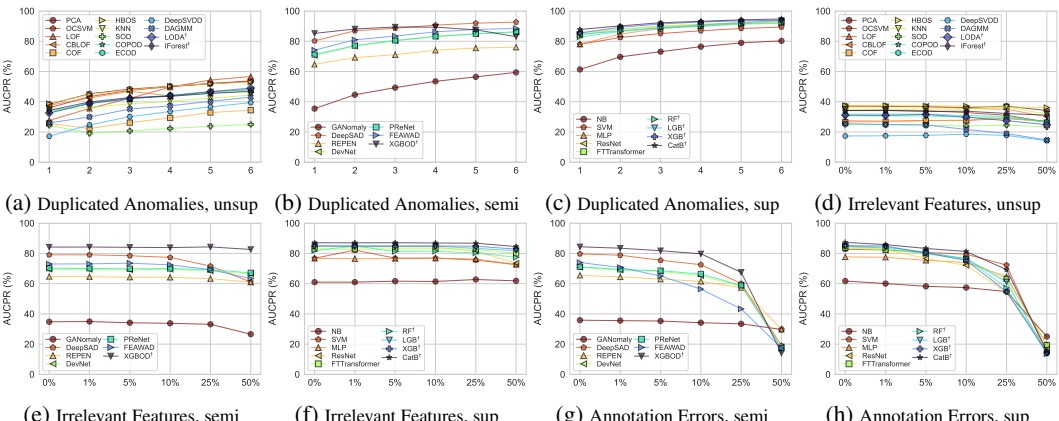

Figure D11: Algorithm performance under noisy and corrupted data (i.e., duplicated anomalies for (a)-(c), irrelevant features for (d)-(f), and annotation errors for (g) and (h)). X-axis denotes either the duplicated times or the noise ratio. Y-axis denotes the AUCPR performance and its range remains consistent across different algorithms. The results reveal unsupervised methods' susceptibility to duplicated anomalies and the usage of label information in defending irrelevant features. Un-, semi-, and fully-supervised methods are denoted as *unsup*, *semi*, and *sup*, respectively.

## D.4 Full Performance Tables on Benchmark Datasets (in addition to §4.2 and Appendix D.1)

In the following tables, we first present the AUCROC and AUCPR for all unsupervised methods, and then show the label-informed methods' performance at different levels of labeled anomaly ratio (i.e., $\gamma_l = \{1\%, ..., 100\%\}$). We would expect these results are useful in constructing unsupervised anomaly detection model selection methods like MetaOD [199], where the historical algorithm performance table serves as a great source for building strong meta-learning methods.

Table D4: AUCROC of 14 unsupervised algorithms on 57 benchmark datasets. We show the performance rank in parenthesis (the lower, the better), and mark the best performing method(s) in **bold**.

| Datasets | PCA | OCSVM | LOF | CBLOF | COF | HBOS | KNN | SOD | COPOD | ECOD | Deep SVDD | DA GMM | LODA | IForest |
|---|---|---|---|---|---|---|---|---|---|---|---|---|---|---|
| ALOI | 56.65(6) | 55.85(8) | **66.63(1)** | 55.22(9) | 64.68(2) | 52.63(11) | 61.47(3) | 61.09(4) | 53.75(10) | 56.60(7) | 50.29(14) | 51.96(12) | 51.33(13) | 56.66(5) |
| annthyroid | 66.25(8) | 57.23(12) | 70.20(7) | 62.26(10) | 65.92(9) | 60.15(11) | 71.69(6) | 77.38(3) | 76.80(4) | 78.03(2) | 76.62(5) | 56.53(13) | 41.02(14) | **82.01(1)** |
| backdoor | 80.13(7) | 86.20(2) | 85.68(3) | 81.16(4) | 73.03(8) | 71.43(10) | 80.82(6) | 69.54(11) | 80.97(5) | **86.33(1)** | 55.16(14) | 56.26(13) | 69.22(12) | 72.15(9) |
| breastw | 95.13(8) | 80.30(10) | 40.61(12) | 96.81(7) | 38.84(13) | 98.94(3) | 97.01(6) | 93.97(9) | **99.68(1)** | 99.17(2) | 65.66(11) | N/A(N/A) | 98.49(4) | 98.32(5) |
| campaign | 72.78(4) | 65.52(9) | 58.85(10) | 66.61(8) | 57.26(11) | **78.61(1)** | 72.10(5) | 69.04(7) | 77.69(2) | 76.78(3) | 48.70(14) | 56.08(12) | 51.43(13) | 71.71(6) |
| cardio | **95.55(1)** | 93.91(3) | 66.33(13) | 89.93(7) | 71.41(12) | 84.67(8) | 76.64(9) | 73.25(11) | 92.35(5) | 94.44(2) | 58.96(14) | 75.01(10) | 90.34(6) | 93.19(4) |
| Cardiotocography | 74.67(2) | **77.86(1)** | 59.51(10) | 64.54(7) | 53.77(12) | 60.86(9) | 56.23(11) | 51.69(14) | 67.02(6) | 68.92(4) | 53.53(13) | 62.01(8) | 73.65(3) | 67.57(5) |
| celeba | **79.38(1)** | 70.70(6) | 38.55(14) | 73.99(4) | 38.58(13) | 76.18(2) | 59.63(9) | 47.85(11) | 75.68(3) | 72.82(5) | 50.36(10) | 44.74(12) | 60.11(8) | 70.41(7) |
| census | 68.74(2) | 54.58(10) | 47.19(12) | 59.41(8) | 41.35(13) | 64.94(5) | 66.75(4) | 62.31(6) | 68.44(3) | 51.07(11) | 59.29(9) | 36.86(14) | 59.52(7) | 59.52(7) |
| cover | **93.73(1)** | 92.62(3) | 84.58(10) | 89.30(6) | 76.91(12) | 80.24(11) | 85.97(9) | 74.46(13) | 88.64(7) | 93.42(2) | 46.20(14) | 89.89(5) | 92.34(4) | 86.74(8) |
| donors | **83.15(1)** | 71.93(7) | 55.49(11) | 60.44(10) | 70.54(9) | 78.23(4) | 81.09(3) | 55.21(12) | 81.76(2) | 74.45(6) | 50.27(13) | 70.57(8) | 24.86(14) | 77.68(5) |
| fault | 46.02(10) | 47.69(9) | 58.93(5) | 64.06(3) | 62.10(4) | 51.28(8) | **72.98(1)** | 68.11(2) | 43.88(12) | 43.41(13) | 51.67(7) | 45.86(11) | 41.71(14) | 57.02(6) |
| fraud | 90.35(8) | 90.62(6) | 94.92(2) | 91.70(5) | 93.05(4) | 90.29(9) | 93.56(3) | 88.32(13) | 89.85(10) | 64.98(14) | 89.53(11) | 88.99(12) | 90.38(7) | 90.38(7) |
| glass | 66.29(12) | 35.36(14) | 69.20(11) | **82.94(1)** | 72.24(10) | 77.23(3) | 82.29(2) | 73.36(7) | 72.43(9) | 75.70(6) | 47.49(13) | 76.09(5) | 73.13(8) | 77.13(4) |
| Hepatitis | 75.95(4) | 67.75(7) | 38.02(14) | 66.40(8) | 41.45(13) | 79.85(2) | 52.76(11) | 68.17(6) | **82.05(1)** | 79.67(3) | 50.96(12) | 54.80(10) | 64.87(9) | 69.75(5) |
| http | 99.72(2) | 99.59(4) | 27.46(11) | 99.60(3) | 88.78(8) | 99.53(5) | 3.37(13) | 78.04(9) | 99.29(6) | 98.10(7) | 69.05(10) | N/A(N/A) | 12.48(12) | **99.96(1)** |
| InternetAds | 61.67(11) | 68.28(4) | 65.83(8) | **70.58(1)** | 63.79(9) | 68.03(5) | 69.99(2) | 61.85(10) | 67.05(7) | 67.10(6) | 60.20(12) | N/A(N/A) | 55.38(13) | 69.01(3) |
| Ionosphere | 79.19(8) | 75.92(10) | 90.59(2) | **90.72(1)** | 86.76(4) | 62.49(13) | 88.26(3) | 86.41(5) | 79.34(7) | 75.59(11) | 50.89(14) | 73.41(12) | 78.42(9) | 84.50(6) |
| landsat | 35.76(14) | 36.15(13) | 53.90(7) | 63.55(2) | 53.50(8) | 55.14(6) | 57.95(4) | 59.54(3) | 41.55(11) | 56.61(5) | **63.61(1)** | 43.92(10) | 38.17(12) | 47.64(9) |
| letter | 50.29(12) | 46.18(14) | 84.49(2) | 75.62(5) | 80.03(4) | 59.74(7) | **86.19(1)** | 84.09(3) | 54.32(9) | 50.76(10) | 56.64(8) | 50.42(11) | 50.24(13) | 61.07(6) |
| Lymphography | 99.82(2) | 99.54(4) | 89.86(9) | **99.83(1)** | 90.85(8) | 99.49(6) | 55.91(13) | 72.49(11) | 99.48(7) | 99.52(5) | 32.29(14) | 72.11(12) | 85.55(10) | 99.81(3) |
| magic.gamma | 67.22(9) | 60.65(12) | 68.51(6) | 75.13(3) | 66.64(10) | 70.86(5) | **82.38(1)** | 75.40(2) | 68.33(7) | 64.36(11) | 60.26(13) | 58.58(14) | 68.02(8) | 73.25(4) |
| mammography | 88.72(3) | 84.95(6) | 74.39(12) | 83.74(9) | 77.53(11) | 86.27(5) | 84.53(7) | 81.51(10) | 90.69(2) | **90.75(1)** | 56.98(13) | N/A(N/A) | 83.91(8) | 86.39(4) |
| mnist | **85.29(1)** | 82.95(3) | 67.13(11) | 79.45(6) | 70.78(9) | 60.42(12) | 80.58(5) | 60.10(13) | 77.74(7) | 84.60(2) | 53.40(14) | 67.23(10) | 72.27(8) | 80.98(4) |
| musk | 100.00(1) | 80.58(8) | 41.18(13) | **100.00(1)** | 38.69(14) | **100.00(1)** | 69.89(11) | 74.09(10) | 94.20(7) | 95.11(6) | 43.52(12) | 76.85(9) | 91.15(5) | 99.99(4) |
| optdigits | 51.65(11) | 54.00(10) | 56.10(9) | **87.51(1)** | 49.15(12) | 81.63(2) | 41.73(13) | 58.92(8) | 68.71(4) | 61.04(7) | 38.89(14) | 62.57(5) | 61.74(6) | 70.92(3) |
| PageBlocks | 90.64(2) | 88.76(5) | 75.90(12) | 85.04(7) | 72.65(13) | 80.58(10) | 81.94(9) | 77.75(11) | 88.05(6) | **90.92(1)** | 57.77(14) | 89.61(3) | 83.34(8) | 89.57(4) |
| pendigits | 93.73(3) | 93.75(2) | 47.99(12) | 90.40(7) | 45.07(13) | 93.04(4) | 72.95(9) | 66.29(10) | 90.68(6) | 91.22(5) | 39.92(14) | 64.22(11) | 89.10(8) | **94.76(1)** |
| Pima | 70.77(5) | 66.92(7) | 65.71(9) | 71.42(3) | 61.05(11) | 71.07(4) | 73.43(1) | 61.25(10) | 69.10(6) | 51.54(13) | 51.03(14) | 55.92(12) | 65.93(8) | 72.87(2) |
| satellite | 59.62(10) | 59.02(11) | 55.88(12) | 71.32(3) | 54.74(14) | 74.80(2) | 65.18(5) | 63.96(6) | 63.20(7) | **75.06(1)** | 55.30(13) | 62.33(8) | 61.98(9) | 70.43(4) |
| satimage-2 | 97.62(4) | 97.35(6) | 47.36(14) | **99.84(1)** | 56.70(12) | 97.65(3) | 92.60(10) | 83.08(11) | 97.21(7) | 97.11(8) | 53.14(13) | 96.29(9) | 97.56(5) | 99.16(2) |
| shuttle | 98.62(5) | 97.40(7) | 57.11(12) | 83.48(8) | 51.72(14) | 98.63(4) | 69.64(9) | 69.51(10) | 99.35(3) | 99.40(2) | 52.05(13) | 97.92(6) | 60.95(11) | **99.56(1)** |
| skin | 45.26(10) | 49.45(6) | 46.47(8) | 69.49(2) | 41.66(12) | 60.15(5) | 71.46(1) | 60.35(4) | 47.55(7) | 39.09(13) | 44.05(11) | N/A(N/A) | 45.75(9) | 68.21(3) |
| smtp | 88.41(3) | 80.70(4) | 71.84(10) | 79.68(5) | 79.60(6) | 70.52(12) | 89.62(2) | 59.85(14) | 79.09(7) | 71.86(9) | 78.24(8) | 71.32(11) | 67.43(13) | **89.73(1)** |
| SpamBase | 54.66(6) | 52.47(9) | 43.33(11) | 54.97(5) | 40.96(13) | 54.33(8) | 53.35(8) | 52.35(10) | 70.09(1) | 66.89(2) | 53.55(7) | N/A(N/A) | 41.99(12) | 64.76(3) |
| speech | 50.79(9) | 50.19(13) | 52.48(6) | 50.58(12) | **55.97(1)** | 50.59(11) | 51.03(8) | 55.86(2) | 52.89(4) | 51.58(7) | 53.43(3) | 52.75(5) | 49.84(14) | 50.74(10) |
| Stamps | 91.47(2) | 83.86(8) | 51.26(14) | 68.18(11) | 53.81(13) | 90.73(5) | 68.61(10) | 73.26(9) | **93.40(1)** | 91.41(3) | 55.84(12) | 88.88(6) | 87.18(7) | 91.21(4) |
| thyroid | 96.34(3) | 87.92(10) | 86.86(11) | 94.73(6) | 90.87(9) | 95.62(5) | 95.93(4) | 92.81(8) | 94.30(7) | 97.78(2) | 49.64(14) | 79.75(12) | 74.30(13) | **98.30(1)** |
| vertebral | 37.06(8) | 37.99(6) | 49.29(2) | 41.41(4) | 48.71(3) | 28.56(13) | 33.79(11) | 40.32(5) | 25.64(14) | 37.51(7) | 36.67(9) | **53.20(1)** | 30.57(12) | 36.66(10) |
| vowels | 65.29(9) | 61.59(10) | 93.12(3) | 89.92(5) | 94.04(2) | 72.21(7) | **97.26(1)** | 92.65(4) | 53.15(12) | 45.81(14) | 52.49(13) | 60.58(11) | 70.36(8) | 73.94(6) |
| Waveform | 65.48(10) | 56.29(12) | 73.32(3) | 72.42(6) | 72.56(5) | 68.77(8) | 73.78(2) | 68.57(9) | **75.03(1)** | 73.25(4) | 54.47(13) | 49.35(14) | 60.13(11) | 71.47(7) |
| WBC | 98.20(7) | 99.03(4) | 54.17(13) | **99.46(1)** | 60.90(11) | 98.72(6) | 90.56(10) | 94.60(9) | 99.11(2) | 99.11(2) | 55.50(12) | N/A(N/A) | 96.91(8) | 99.01(5) |
| WDBC | 99.05(4) | 98.86(6) | 89.00(12) | 99.32(3) | 96.26(9) | **99.50(1)** | 91.72(11) | 91.90(10) | 99.42(2) | 97.20(8) | 65.69(14) | 76.67(13) | 98.26(7) | 98.95(5) |
| Wilt | 20.39(14) | 31.28(12) | 50.65(2) | 32.54(10) | 49.66(3) | 32.49(11) | 48.42(4) | **53.25(1)** | 33.40(9) | 39.43(7) | 46.08(5) | 37.29(8) | 26.42(13) | 41.94(6) |
| wine | 84.37(4) | 73.07(6) | 37.74(13) | 25.86(14) | 44.44(12) | 91.36(1) | 44.98(11) | 46.11(10) | 88.65(3) | 71.34(7) | 59.52(9) | 61.70(8) | 90.12(2) | 80.37(5) |
| WPBC | 46.01(10) | 45.35(12) | 41.41(14) | 44.77(13) | 45.88(11) | **51.24(1)** | 46.59(9) | 51.14(2) | 49.34(4) | 46.83(7) | 49.79(3) | 47.80(6) | 49.31(5) | 46.63(8) |
| yeast | 41.15(7) | 41.00(9) | 45.31(2) | 44.85(3) | 44.48(5) | 39.64(10) | 39.06(12) | 42.46(6) | 36.99(14) | 39.61(11) | **47.92(1)** | 41.11(8) | 44.58(4) | 37.76(13) |
| CIFAR10 | 63.87(6) | 63.76(7) | **68.57(1)** | 64.23(4) | 64.70(3) | 57.50(13) | 64.75(2) | 64.22(5) | 58.64(11) | 61.04(10) | 56.04(14) | 58.08(12) | 62.34(8) | 61.28(9) |
| FashionMNIST | 86.09(3) | 85.24(4) | 67.57(12) | **88.17(1)** | 71.44(11) | 78.68(10) | 86.60(2) | 81.73(7) | 81.07(8) | 83.63(6) | 63.32(14) | 67.29(13) | 80.28(9) | 84.89(5) |
| MNIST-C | 73.75(5) | 72.21(8) | 68.27(12) | 80.86(2) | 69.81(11) | 70.82(10) | **81.26(1)** | 74.00(4) | 71.26(9) | 72.64(7) | 51.85(14) | 58.56(13) | 74.37(3) | 73.74(6) |
| MVTec-AD | 72.42(8) | 69.84(10) | 74.19(2) | **75.98(1)** | 69.70(11) | 73.36(4) | 72.96(6) | 71.57(9) | 72.91(7) | 73.46(3) | 57.10(14) | 66.47(13) | 68.51(12) | 73.19(5) |
| SVHN | 60.53(6) | 60.73(5) | **64.51(1)** | 60.30(7) | 63.47(2) | 56.08(13) | 62.63(3) | 61.09(4) | 56.75(12) | 58.27(9) | 53.47(14) | 57.22(11) | 58.26(10) | 58.62(8) |
| Agnews | 54.70(8) | 54.34(9) | **71.80(1)** | 60.02(5) | 68.97(2) | 53.87(10) | 62.81(4) | 52.98(12) | 53.04(11) | 42.51(14) | 52.02(13) | 55.47(7) | 56.74(6) | 56.74(6) |
| Amazon | 55.06(10) | 54.14(12) | 56.11(9) | 57.36(3) | 56.96(4) | 56.52(7) | 60.03(2) | **60.05(1)** | 56.94(5) | 56.79(6) | 39.08(14) | 53.58(13) | 54.20(11) | 56.13(8) |
| Imdb | 47.06(12) | 46.07(14) | 48.71(9) | 49.35(6) | 49.64(5) | 49.10(7) | 47.83(10) | 49.86(4) | 50.68(3) | **50.73(2)** | 47.67(11) | 46.43(13) | 49.09(8) | 49.09(8) |
| Yelp | 60.71(11) | 60.28(12) | 67.09(3) | 64.90(5) | 66.11(4) | 61.85(9) | **69.84(1)** | 67.74(2) | 62.36(7) | 62.15(8) | 54.62(14) | 56.28(13) | 61.36(10) | 62.53(6) |
| 20news | 56.66(7) | 56.45(8) | **62.14(1)** | 57.59(5) | 61.80(2) | 56.28(9) | 59.33(3) | 58.56(4) | 55.79(11) | 56.00(10) | 50.24(14) | 54.17(13) | 55.53(12) | 56.90(6) |

Table D5: AUCPR of 14 unsupervised algorithms on 57 benchmark datasets. We show the performance rank in parenthesis (lower the better), and mark the best performing method(s) in **bold**.

| Datasets | PCA | OCSVM | LOF | CBLOF | COF | HBOS | KNN | SOD | COPOD | ECOD | Deep SVDD | DA GMM | LODA | IForest |
|---|---|---|---|---|---|---|---|---|---|---|---|---|---|---|
| ALOI | 4.17(9) | 5.02(5) | **8.08(1)** | 4.46(7) | 6.85(2) | 3.69(13) | 6.02(3) | 5.97(4) | 3.62(14) | 3.90(11) | 4.01(10) | 4.33(8) | 4.53(6) | 3.90(12) |
| annthyroid | 16.12(8) | 10.37(12) | 15.71(9) | 13.69(11) | 14.39(10) | 16.99(5) | 16.74(6) | 18.84(4) | 16.58(7) | 24.65(2) | 21.95(3) | 9.64(13) | 7.06(14) | **30.47(1)** |
| backdoor | 31.29(3) | 9.69(9) | 26.14(4) | 6.96(11) | 24.68(5) | 4.91(13) | **45.22(1)** | 39.41(2) | 7.69(10) | 11.25(8) | 12.85(7) | 6.50(12) | 14.51(6) | 4.75(14) |
| breastw | 95.11(6) | 82.70(10) | 28.55(12) | 91.54(8) | 27.60(13) | 97.71(3) | 92.19(7) | 84.88(9) | **99.40(1)** | 98.54(2) | 50.92(11) | N/A(N/A) | 97.04(4) | 96.04(5) |
| campaign | 27.90(6) | 29.22(5) | 14.51(11) | 23.99(8) | 13.01(13) | 37.99(2) | 27.18(7) | 18.88(9) | **38.58(1)** | 37.40(3) | 11.60(14) | 14.62(10) | 13.47(12) | 32.26(4) |
| cardio | 66.06(2) | 62.89(3) | 23.79(13) | 61.95(4) | 28.67(11) | 52.10(8) | 40.72(9) | 28.54(12) | 60.42(5) | **68.59(1)** | 22.50(14) | 28.92(10) | 53.41(7) | 59.95(6) |
| Cardiotocography | 47.95(3) | **52.61(1)** | 30.66(11) | 45.44(4) | 28.21(13) | 38.28(8) | 34.79(9) | 27.99(14) | 40.46(7) | 43.57(5) | 34.03(10) | 30.61(12) | 48.00(2) | 41.47(6) |
| celeba | **15.89(1)** | 10.73(6) | 1.71(14) | 11.33(5) | 1.77(13) | 13.82(2) | 3.14(9) | 2.66(10) | 13.69(3) | 12.37(4) | 2.34(11) | 1.95(12) | 4.04(8) | 8.96(7) |
| census | **10.02(1)** | 6.76(11) | 5.45(12) | 7.44(9) | 4.88(14) | 8.69(6) | 9.00(4) | 8.52(7) | 9.92(2) | 9.72(3) | 6.87(10) | 8.71(5) | 5.01(13) | 7.78(8) |
| cover | 9.80(6) | 11.41(4) | 8.12(8) | 5.83(12) | 4.00(13) | 6.83(10) | 6.16(11) | 3.88(14) | 11.37(5) | 15.63(2) | 8.12(9) | **27.59(1)** | 13.06(3) | 8.85(7) |
| donors | 17.90(3) | 9.86(8) | 7.88(11) | 6.89(12) | 8.80(10) | **23.36(1)** | 14.75(4) | 9.69(9) | 21.58(2) | 14.17(5) | 6.38(13) | 10.53(7) | 3.78(14) | 12.74(6) |
| fault | 32.76(11) | 38.44(7) | 38.38(8) | 43.98(3) | 41.56(4) | 36.47(9) | **54.45(1)** | 48.01(2) | 30.54(14) | 30.82(13) | 39.15(6) | 33.48(10) | 31.03(12) | 41.09(5) |
| fraud | 22.91(10) | **47.58(1)** | 47.40(3) | 47.52(2) | 22.86(11) | 25.89(9) | 47.30(4) | 31.37(8) | 42.82(7) | 42.99(6) | 8.97(14) | 21.32(13) | 46.37(5) | 21.67(12) |
| glass | 10.05(11) | 8.02(14) | 20.11(3) | 13.84(6) | 11.81(9) | 11.82(8) | 20.26(2) | 18.73(4) | 9.78(12) | 18.43(5) | 8.72(13) | **24.58(1)** | 13.37(7) | 10.99(10) |
| Hepatitis | 36.65(4) | 29.44(7) | 13.69(14) | 31.54(5) | 14.39(13) | 37.73(3) | 21.95(12) | 24.89(9) | **41.50(1)** | 37.82(2) | 22.17(11) | 22.96(10) | 30.90(6) | 26.25(8) |
| http | 56.43(2) | 46.86(4) | 3.82(11) | 47.53(3) | 9.57(9) | 44.79(5) | 0.70(12) | 8.32(10) | 35.19(6) | 16.61(8) | 29.30(7) | N/A(N/A) | 0.67(13) | **90.83(1)** |
| InternetAds | 32.55(10) | 54.68(2) | 40.49(8) | 58.13(1) | 38.67(9) | 53.97(3) | 43.23(7) | 27.69(12) | 50.97(5) | 51.07(4) | 27.91(11) | N/A(N/A) | 23.89(13) | 48.60(6) |
| Ionosphere | 73.92(8) | 74.54(7) | 88.06(3) | 90.27(2) | 82.91(5) | 41.78(14) | **90.41(1)** | 85.87(4) | 69.89(10) | 65.99(11) | 41.79(13) | 64.98(12) | 73.04(9) | 80.41(6) |
| landsat | 16.18(14) | 16.21(13) | 24.69(6) | 30.97(2) | 24.95(5) | 22.03(9) | 24.65(7) | 26.38(3) | 17.48(12) | 25.17(4) | **38.83(1)** | 24.48(8) | 18.86(11) | 19.81(10) |
| letter | 6.86(12) | 6.10(14) | **34.02(1)** | 14.80(5) | 21.43(4) | 8.38(9) | 30.00(2) | 28.63(3) | 6.77(13) | 6.94(10) | 9.29(7) | 11.68(6) | 6.87(11) | 8.49(8) |
| Lymphography | 97.02(3) | 93.59(4) | 23.08(11) | **97.62(1)** | 36.68(10) | 91.83(5) | 38.69(9) | 22.65(12) | 88.68(7) | 90.87(6) | 4.58(14) | 19.52(13) | 44.54(8) | 97.31(2) |
| magic.gamma | 59.27(6) | 51.43(12) | 54.76(9) | 68.85(2) | 54.12(11) | 62.41(5) | **75.63(1)** | 67.89(3) | 59.18(7) | 54.38(10) | 49.17(13) | 46.92(14) | 58.49(8) | 64.72(4) |
| mammography | 19.25(5) | 12.94(9) | 9.80(12) | 11.14(10) | 11.14(11) | 21.31(3) | 15.91(6) | 13.41(8) | 40.67(2) | **41.28(1)** | 6.26(13) | N/A(N/A) | 14.75(7) | 20.67(4) |
| mnist | **39.93(1)** | 33.20(3) | 20.90(11) | 28.82(5) | 25.51(8) | 12.51(14) | 39.15(2) | 19.15(13) | 21.35(10) | 31.93(4) | 19.72(12) | 23.75(9) | 25.86(7) | 27.71(6) |
| musk | 99.89(3) | 10.61(9) | 2.82(13) | 100.00(2) | 2.61(14) | **100.00(1)** | 9.65(10) | 7.59(11) | 34.79(7) | 34.95(6) | 5.39(12) | 32.75(8) | 47.60(5) | 99.61(4) |
| optdigits | 2.76(13) | 2.92(12) | 6.06(3) | **10.08(1)** | 4.42(6) | 10.03(2) | 3.06(11) | 4.39(7) | 4.36(8) | 3.43(10) | 2.50(14) | 5.59(4) | 3.95(9) | 5.09(5) |
| PageBlocks | 51.71(2) | 49.14(6) | 39.64(10) | 49.65(4) | 41.02(9) | 33.32(13) | 45.39(8) | 37.83(11) | 37.65(12) | 49.30(5) | 31.45(14) | **53.25(1)** | 51.29(3) | 46.04(7) |
| pendigits | 23.65(3) | 23.52(4) | 3.78(12) | 17.27(8) | 2.89(13) | **29.27(1)** | 6.50(9) | 4.46(11) | 21.22(6) | 23.07(5) | 2.45(14) | 4.67(10) | 18.71(7) | 26.05(2) |
| Pima | 54.03(5) | 50.00(7) | 47.18(9) | 53.19(6) | 44.70(10) | **56.61(1)** | 55.14(4) | 48.24(8) | 55.19(3) | 37.30(13) | 35.87(14) | 41.55(12) | 44.09(11) | 55.82(2) |
| satellite | 59.64(6) | 57.61(8) | 37.68(14) | 61.48(5) | 39.70(13) | **67.25(1)** | 50.01(10) | 47.23(11) | 56.58(9) | 65.94(2) | 40.11(12) | 58.33(7) | 61.94(4) | 65.92(3) |
| satimage-2 | 85.69(3) | 82.71(4) | 4.29(13) | **97.09(1)** | 8.80(12) | 78.04(6) | 39.14(9) | 26.11(10) | 76.55(7) | 63.25(8) | 3.08(14) | 22.07(11) | 80.52(5) | 93.45(2) |
| shuttle | 92.35(6) | 85.29(7) | 13.76(13) | 60.98(8) | 12.17(14) | 96.40(3) | 20.38(10) | 20.27(11) | 96.56(2) | 95.76(4) | 15.86(12) | 93.20(5) | 48.75(9) | **97.62(1)** |
| skin | 17.40(11) | 19.03(6) | 18.25(9) | **29.82(1)** | 16.38(12) | 23.70(5) | 28.72(2) | 24.61(4) | 17.99(10) | 15.96(13) | 18.48(7) | N/A(N/A) | 18.44(8) | 26.08(3) |
| smtp | 66.70(2) | 18.90(12) | 26.68(10) | 61.13(3) | 35.20(8) | 35.20(8) | **66.70(1)** | 33.36(9) | 1.08(14) | 50.01(6) | 50.02(5) | 50.03(4) | 35.77(7) | 1.24(13) |
| SpamBase | 41.57(6) | 40.12(9) | 35.16(12) | 41.18(8) | 34.73(13) | 50.03(4) | 41.42(7) | 40.03(10) | **56.68(1)** | 53.95(2) | 42.23(5) | 49.16(3) | 35.88(11) | 51.75(3) |
| speech | 1.97(10) | 1.96(11) | 2.52(2) | 1.99(9) | 2.25(4) | 2.09(6) | 2.02(8) | 2.13(5) | 1.94(12) | 1.77(14) | **5.12(1)** | 2.03(7) | 1.79(13) | 2.31(3) |
| Stamps | 41.09(3) | 31.39(8) | 21.29(11) | 23.66(9) | 16.50(13) | 35.24(6) | 23.53(10) | 20.28(12) | 43.10(2) | 38.17(5) | 11.40(14) | **43.72(1)** | 34.60(7) | 39.49(4) |
| thyroid | 44.34(4) | 21.23(9) | 20.81(10) | 29.95(6) | 28.50(7) | 50.98(3) | 34.98(5) | 23.56(8) | 19.64(11) | 54.05(2) | 2.50(14) | 16.06(12) | 14.68(13) | **63.11(1)** |
| vertebral | 10.49(10) | 10.94(7) | 14.24(2) | 11.58(5) | 13.85(3) | 9.23(13) | 10.57(8) | 11.79(4) | 8.89(14) | 11.24(6) | 10.49(9) | **15.24(1)** | 9.68(12) | 10.46(11) |
| vowels | 8.92(10) | 8.24(11) | 34.42(4) | 22.12(5) | 55.96(2) | 13.41(8) | **63.41(1)** | 38.88(3) | 4.14(13) | 3.92(14) | 4.99(12) | 12.22(9) | 13.82(7) | 15.12(6) |
| Waveform | 5.79(10) | 4.37(13) | 11.33(4) | **18.98(1)** | 14.11(2) | 5.86(9) | 13.04(3) | 9.66(5) | 6.90(6) | 6.86(7) | 4.83(11) | 3.11(14) | 4.71(12) | 6.24(8) |
| WBC | 82.29(6) | 89.87(3) | 5.57(13) | **92.27(1)** | 9.73(11) | 73.56(8) | 66.55(9) | 54.00(10) | 86.19(4) | 86.19(4) | 6.38(12) | N/A(N/A) | 78.67(7) | 90.49(2) |
| WDBC | 75.46(5) | 71.88(6) | 14.93(13) | 79.62(3) | 50.52(9) | **88.98(1)** | 43.72(10) | 35.60(11) | 84.78(2) | 57.91(8) | 6.57(14) | 18.48(12) | 66.11(7) | 78.53(4) |
| Wilt | 3.13(14) | 3.62(12) | 5.05(2) | 3.64(11) | 4.98(3) | 3.84(9) | 4.73(4) | **5.53(1)** | 3.24(14) | 4.14(7) | 4.67(5) | 4.00(8) | 3.36(13) | 4.23(6) |
| wine | 30.87(4) | 21.56(6) | 7.77(13) | 5.83(14) | 8.45(10) | 43.08(3) | 8.43(11) | 7.95(12) | 45.71(2) | 18.37(8) | 21.14(7) | 17.51(9) | **48.82(1)** | 25.96(5) |
| WPBC | 23.01(5) | 22.93(6) | 20.29(14) | 21.32(12) | 21.30(13) | 23.04(4) | 21.49(10) | 25.37(3) | 22.81(7) | 21.38(11) | **26.24(1)** | 22.49(8) | 25.58(2) | 22.42(9) |
| yeast | 29.90(11) | 29.84(12) | 31.64(4) | 30.93(7) | 31.27(6) | 32.75(3) | 29.33(14) | 29.96(9) | 30.71(8) | 31.36(5) | 33.03(2) | 29.92(10) | **33.29(1)** | 29.80(13) |
| CIFAR10 | 10.59(6) | 10.19(7) | **13.02(1)** | 10.61(5) | 11.61(2) | 8.38(12) | 11.13(3) | 11.06(4) | 8.77(11) | 9.29(9) | 8.05(13) | 7.73(14) | 9.72(8) | 8.97(10) |
| FashionMNIST | 31.42(6) | 31.97(5) | 16.85(13) | **38.90(1)** | 20.73(11) | 29.43(8) | 33.87(2) | 28.72(9) | 30.32(7) | 32.53(3) | 17.43(12) | 14.44(14) | 27.32(10) | 32.35(4) |
| MNIST-C | 16.88(7) | 17.72(6) | 13.84(12) | **27.62(1)** | 14.53(11) | 15.46(10) | 22.98(2) | 15.68(9) | 15.90(8) | 18.24(4) | 8.34(14) | 11.37(13) | 18.63(3) | 17.99(5) |
| MVTec-AD | 54.06(8) | 51.44(10) | 54.90(6) | **58.52(1)** | 46.59(12) | 55.22(5) | 55.55(3) | 51.48(9) | 54.64(7) | 55.44(4) | 36.50(14) | 45.66(13) | 49.73(11) | 56.04(2) |
| SVHN | 8.66(5) | 8.65(6) | 9.24(2) | 8.58(7) | 8.97(3) | 7.45(12) | **9.46(1)** | 8.52(8) | 7.61(11) | 7.82(10) | 6.99(14) | 7.29(13) | 8.70(4) | 8.10(9) |
| Agnews | 5.74(8) | 5.69(9) | **14.35(1)** | 7.02(5) | 12.21(2) | 5.58(10) | 8.61(3) | 8.40(4) | 5.43(12) | 5.43(11) | 4.45(14) | 5.41(13) | 5.93(7) | 6.04(6) |
| Amazon | 5.85(9) | 5.64(13) | 5.72(11) | 6.07(4) | 5.74(10) | 5.98(6) | 6.23(2) | **6.40(1)** | 6.08(3) | 6.06(5) | 3.84(14) | 5.65(12) | 5.92(8) | 5.95(7) |
| Imdb | 4.55(12) | 4.44(14) | 4.83(5) | 4.75(6) | **5.16(1)** | 4.74(8) | 4.49(13) | 4.70(9) | 4.90(3) | 4.90(4) | 5.06(2) | 4.65(10) | 4.59(11) | 4.74(7) |
| Yelp | 7.62(12) | 7.75(9) | 8.52(4) | 7.68(10) | 8.68(3) | 7.81(8) | **9.85(1)** | 9.20(2) | 8.01(5) | 7.98(6) | 6.39(14) | 6.72(13) | 7.65(11) | 7.88(7) |
| 20news | 7.97(5) | 7.53(11) | **9.13(1)** | 7.81(9) | 9.02(2) | 7.80(10) | 8.54(3) | 8.19(4) | 7.95(6) | 7.92(7) | 7.82(8) | 6.68(14) | 7.37(12) | 7.29(13) |

Table D6: AUCROC of 16 label-informed algorithms on 57 benchmark datasets, with labeled anomaly ratio $\gamma_l = 1\%$. We show the performance rank in parenthesis (lower the better), and mark the best performing method(s) in **bold**.

| Datasets | GANomaly | DeepSAD | REPEN | DevNet | PReNet | FEAWAD | XGBOD | NB | SVM | MLP | ResNet | FTTransformer | RF | LGB | XGB | CatB |
|---|---|---|---|---|---|---|---|---|---|---|---|---|---|---|---|---|
| ALOI | 55.53(3) | 59.13(2) | 54.53(5) | 47.03(15) | 46.47(16) | 55.06(4) | 60.53(1) | 49.31(13) | 52.42(8) | 48.50(14) | 49.89(12) | 51.74(9) | 51.04(10) | 50.60(11) | 53.08(7) | 53.22(6) |
| annthyroid | 55.67(9) | 76.82(7) | 72.20(11) | 74.78(10) | 75.95(8) | 77.66(5) | 92.89(2) | 80.62(4) | 59.25(15) | 62.84(12) | 52.04(16) | 76.97(6) | 60.93(14) | 62.62(13) | 89.91(3) | **95.58(1)** |
| backdoor | 82.28(9) | 91.98(3) | 89.44(5) | 92.91(2) | **94.23(1)** | 82.60(7) | 85.67(6) | 63.10(13) | 80.03(10) | 89.69(4) | 71.15(12) | 62.89(14) | 58.05(15) | 37.26(16) | 82.55(8) | 76.47(11) |
| breastw | 91.99(3) | 88.36(5) | 86.03(7) | 74.61(10) | 69.24(12) | 78.39(9) | 86.57(6) | 52.11(16) | 71.27(11) | 52.59(15) | 53.60(14) | 95.70(2) | 65.01(13) | 90.20(4) | 84.81(8) | **97.30(1)** |
| campaign | 56.58(12) | 64.32(7) | 57.57(11) | 66.20(6) | 67.62(5) | 58.45(10) | 74.21(2) | 51.87(14) | 47.20(16) | 60.69(9) | 49.37(15) | 61.16(8) | 56.28(13) | 68.27(4) | 73.81(3) | **81.61(1)** |
| cardio | 82.03(6) | 69.21(12) | 83.07(5) | 89.74(2) | 87.54(3) | 79.57(8) | 83.94(4) | 58.89(15) | 79.91(7) | 61.55(14) | N/A(N/A) | 75.38(9) | 63.98(13) | 73.56(11) | 74.67(10) | **94.01(1)** |
| Cardiotocography | 53.99(15) | 64.94(12) | 81.22(3) | 81.91(2) | 79.33(4) | 70.67(10) | 71.54(9) | 76.00(7) | 54.85(14) | 48.18(16) | 65.16(11) | 76.31(6) | 60.95(13) | 75.31(8) | 78.81(5) | **85.89(1)** |
| celeba | 81.23(2) | 54.65(14) | 54.85(13) | 73.84(5) | 71.65(6) | 71.59(7) | 74.86(4) | 50.71(15) | 55.00(11) | 62.62(8) | 59.78(9) | 55.63(10) | 54.91(12) | 29.53(16) | 78.00(3) | **81.50(1)** |
| census | 58.91(12) | 60.87(11) | 68.18(8) | 73.80(5) | 66.57(10) | 67.16(9) | 77.20(3) | 50.66(16) | 54.87(14) | 76.76(4) | 51.06(15) | 70.22(7) | 58.28(13) | 71.26(6) | 79.67(2) | **84.53(1)** |
| cover | 42.98(16) | 87.11(9) | 98.60(4) | 99.04(2) | 98.63(3) | 86.37(10) | 92.92(8) | 50.00(14) | 80.88(11) | 93.84(7) | 62.00(13) | **99.80(1)** | 62.41(12) | 43.18(15) | 96.09(6) | 97.90(5) |
| donors | 49.17(15) | 97.54(4) | 82.72(10) | 99.71(2) | **99.89(1)** | 96.82(6) | 95.52(8) | 95.34(9) | 0.89(16) | 70.69(12) | 60.51(13) | 97.89(3) | 60.49(14) | 77.58(11) | 96.18(7) | 96.98(5) |
| fault | 63.88(2) | **67.84(1)** | 63.79(3) | 61.70(4) | 61.19(6) | 56.16(8) | 54.62(10) | 61.26(5) | 39.48(16) | 47.40(14) | 47.10(15) | 55.25(9) | 53.37(13) | 53.89(12) | 53.98(11) | 60.88(7) |
| fraud | 90.52(5) | 87.66(9) | 92.18(2) | 89.77(7) | 91.91(3) | 81.61(12) | 89.84(6) | 50.00(15) | **92.44(1)** | 85.73(10) | 50.99(14) | 75.38(13) | 84.92(11) | 44.99(16) | 87.90(8) | 90.86(4) |
| glass | 67.58(15) | 71.95(13) | 85.97(6) | 87.54(4) | 90.77(2) | 74.71(12) | 81.83(11) | 55.90(16) | 83.87(9) | 82.49(10) | 84.57(8) | 86.30(5) | 67.63(14) | 84.94(7) | 88.63(3) | **91.09(1)** |
| Hepatitis | 56.31(13) | 62.67(11) | 70.29(7) | 68.49(9) | 69.21(8) | 50.98(15) | 76.56(4) | 53.62(14) | 50.00(16) | 74.39(6) | 67.50(10) | 74.76(5) | 56.31(12) | 82.29(2) | 80.78(3) | **83.14(1)** |
| http | 99.80(9) | 99.88(8) | 99.98(7) | **100.00(1)** | 99.99(6) | 99.78(11) | 99.78(11) | 81.67(14) | 83.31(13) | 0.10(16) | **100.00(1)** | 64.41(11) | 83.33(12) | 42.47(15) | **100.00(1)** | 99.79(10) |
| InternetAds | 67.89(4) | 71.41(2) | 62.48(6) | 51.66(13) | 53.10(11) | 60.17(7) | 63.57(5) | 50.28(14) | 57.56(8) | 42.13(15) | 51.93(12) | N/A(N/A) | 54.25(10) | 55.15(9) | 69.52(3) | **77.18(1)** |
| Ionosphere | **91.98(1)** | 73.84(9) | 77.56(3) | 55.28(13) | 54.48(14) | 73.67(4) | 75.87(5) | 50.75(15) | 74.22(7) | 74.09(8) | 64.41(11) | 63.63(10) | 59.44(12) | 74.91(6) | 77.37(4) | 77.27(2) |
| landsat | 45.19(16) | 74.50(3) | 57.12(14) | 70.89(7) | 73.67(4) | 64.90(11) | 76.46(2) | 73.59(5) | 56.48(15) | 69.12(8) | 59.53(13) | 71.78(6) | 60.09(12) | 68.47(9) | 67.12(10) | **80.98(1)** |
| letter | 69.52(4) | 72.18(2) | 59.98(6) | 50.85(12) | 50.54(13) | 54.87(9) | 71.41(3) | 50.00(14) | 48.39(15) | 41.45(16) | **74.71(1)** | 51.19(11) | 54.16(10) | 57.03(8) | 58.71(7) | 63.54(5) |
| Lymphography | 96.80(4) | 80.40(10) | 93.63(7) | 80.56(9) | 78.14(11) | 82.39(8) | 99.73(2) | 66.78(14) | 50.00(16) | 62.50(15) | 77.79(12) | 94.88(6) | 72.04(13) | 97.25(3) | 95.89(5) | **99.65(1)** |
| magic.gamma | 52.62(14) | 76.58(6) | 73.47(7) | **80.89(1)** | 80.46(2) | 70.00(11) | 76.70(5) | 77.04(4) | 50.99(16) | 53.69(13) | 51.56(15) | 77.36(3) | 59.53(12) | 72.14(9) | 70.33(10) | 72.51(8) |
| mammography | 77.28(8) | 84.40(6) | 88.92(3) | 88.57(4) | 84.47(5) | 89.16(2) | 78.78(7) | 73.34(10) | 60.78(13) | 19.48(16) | 61.43(12) | 77.23(9) | 58.85(14) | 39.41(15) | 66.95(11) | **89.83(1)** |
| mnist | 69.68(11) | 75.32(8) | 79.96(5) | 78.71(7) | 79.73(6) | 71.04(10) | 88.35(2) | 64.07(14) | 69.29(12) | 85.93(3) | 67.36(13) | 74.27(9) | 64.05(15) | 61.14(16) | 80.02(4) | **91.19(1)** |
| musk | 99.12(3) | 86.41(11) | 84.71(12) | 88.19(10) | **99.93(1)** | 88.26(9) | 89.38(8) | 50.00(15) | 95.66(6) | 96.23(5) | 97.39(4) | 63.78(13) | 55.51(14) | 49.42(7) | 99.44(2) | 99.99(1) |
| optdigits | 45.68(15) | 84.32(10) | 99.23(4) | **99.93(1)** | 99.71(2) | 99.39(3) | 94.15(8) | 50.74(14) | 87.83(9) | 82.84(11) | 67.12(13) | 98.55(5) | 75.39(12) | 43.88(16) | 94.92(7) | 96.46(6) |
| PageBlocks | 74.79(6) | 84.76(4) | 88.50(2) | 70.47(8) | 73.14(7) | 61.84(12) | 83.79(5) | 64.43(11) | 53.97(14) | 36.07(16) | 37.37(15) | 70.32(9) | 55.59(13) | 65.55(10) | 86.54(3) | **93.81(1)** |
| pendigits | 55.69(12) | 82.47(7) | 86.56(4) | 87.20(3) | 82.95(6) | 68.54(10) | 84.71(5) | 55.56(13) | 49.41(14) | 74.99(9) | 45.87(15) | 87.79(2) | 60.46(11) | 43.88(16) | 77.76(8) | **89.96(1)** |
| Pima | 59.71(8) | 58.30(10) | 67.54(2) | 59.88(7) | 58.30(11) | 58.78(9) | 64.07(3) | 56.31(12) | 48.99(15) | 23.38(16) | 54.05(14) | 63.06(4) | 54.29(13) | 62.21(5) | 61.70(6) | **71.28(1)** |
| satellite | 68.69(13) | 78.70(6) | 75.58(8) | **82.60(1)** | 81.44(2) | 77.99(7) | 80.29(4) | 78.99(5) | 55.48(16) | 75.44(9) | 59.20(15) | 75.09(10) | 64.68(14) | 73.73(11) | 70.75(12) | 80.40(3) |
| satimage-2 | 96.55(7) | 95.66(9) | 96.40(8) | 98.37(3) | 98.44(2) | 97.99(4) | 97.86(5) | 50.00(14) | 47.99(15) | 79.10(12) | 94.53(11) | 96.78(6) | 69.75(13) | 40.98(16) | 94.64(10) | **98.77(1)** |
| shuttle | 73.14(14) | 98.80(2) | **98.92(1)** | 97.62(8) | 97.80(6) | 98.13(5) | 98.76(4) | 93.12(10) | 82.87(11) | 48.11(16) | 92.43(12) | 97.74(7) | 57.76(15) | 74.26(13) | 96.18(9) | 98.77(3) |
| skin | 52.14(16) | 96.98(5) | 92.43(12) | 95.38(7) | 95.03(9) | 96.42(6) | 91.98(13) | 93.22(10) | **99.74(1)** | 92.72(11) | 97.14(4) | 97.84(3) | 57.37(15) | 76.50(14) | 95.34(8) | 99.02(2) |
| smtp | 51.69(14) | 82.74(3) | 76.47(8) | 75.99(9) | 75.24(10) | 64.06(11) | 79.87(5) | 50.00(15) | 62.98(12) | **86.40(1)** | 52.83(13) | 79.07(6) | 83.32(2) | 43.01(16) | 81.12(4) | 78.85(7) |
| SpamBase | 53.46(14) | 58.01(13) | 75.55(8) | 83.90(2) | 80.46(4) | 62.61(11) | 77.84(5) | 67.52(10) | 36.27(16) | 69.28(9) | 47.11(15) | 81.04(3) | 58.96(12) | 75.70(6) | 75.64(7) | **84.57(1)** |
| speech | 47.46(14) | 52.35(7) | 52.03(8) | 56.06(3) | **58.16(1)** | 53.12(5) | 54.63(4) | 50.00(12) | 58.15(2) | 46.78(15) | 50.35(10) | 53.02(6) | 49.52(13) | 50.30(11) | 44.69(16) | 51.54(9) |
| Stamps | 71.87(9) | 63.28(13) | 88.76(2) | 84.81(5) | 84.88(4) | 67.60(10) | 81.48(6) | 51.19(16) | 67.00(12) | 54.56(15) | 72.83(8) | **89.04(1)** | 55.40(14) | 67.26(11) | 80.95(7) | 86.75(3) |
| thyroid | 92.36(6) | 91.81(7) | 98.53(2) | 96.75(4) | 97.78(3) | 89.02(10) | 91.16(9) | 50.00(15) | 54.16(14) | 71.35(11) | 56.02(13) | 91.64(8) | 59.84(12) | 42.36(16) | 94.80(5) | **98.57(1)** |
| vertebral | 36.95(16) | 53.66(11) | 45.66(15) | 64.09(6) | 66.15(3) | **70.46(1)** | 60.27(8) | 51.66(13) | 64.57(5) | 66.93(2) | 55.40(10) | 58.59(9) | 50.48(14) | 60.27(7) | 64.76(4) | 53.56(12) |
| vowels | 78.31(9) | 79.02(6) | 78.68(7) | 78.79(7) | **86.68(1)** | 84.77(3) | 86.46(2) | 52.22(15) | 63.66(14) | 13.49(16) | 81.67(5) | 73.06(11) | 64.13(13) | 64.87(12) | 73.85(10) | 83.24(4) |
| Waveform | 53.14(11) | 63.29(7) | 58.57(9) | **73.37(1)** | 70.59(2) | 51.49(13) | 67.01(5) | 50.00(16) | 58.74(8) | 57.57(10) | 69.15(3) | 67.55(4) | 52.23(13) | 52.81(12) | 51.77(14) | 66.29(6) |
| WBC | 93.28(8) | 93.20(9) | 97.59(5) | **98.77(1)** | 98.25(4) | 87.79(10) | 96.44(7) | 54.52(16) | 58.43(15) | 63.64(14) | 80.39(11) | 98.65(2) | 65.35(13) | 75.76(12) | 97.11(6) | 98.63(3) |
| WDBC | 97.52(7) | 90.44(11) | 98.81(5) | **99.91(1)** | 99.82(3) | 99.76(4) | 95.24(10) | 56.02(15) | 61.79(14) | 0.45(16) | 82.49(13) | 99.87(2) | 90.15(12) | 95.27(9) | 95.46(8) | 98.49(6) |
| Wilt | 42.80(16) | 63.86(9) | 44.02(15) | 65.31(7) | 66.26(5) | 59.60(10) | 59.39(11) | 66.89(4) | **72.34(1)** | 65.67(6) | 70.32(2) | 58.73(12) | 53.51(13) | 52.71(14) | 66.94(3) | 65.08(8) |
| wine | 65.53(11) | 64.33(12) | 95.95(5) | 99.98(2) | **100.00(1)** | 96.11(4) | 90.23(8) | 56.18(14) | 50.00(15) | 2.79(16) | 80.56(9) | 98.92(3) | 58.21(13) | 79.38(10) | 93.41(6) | 91.60(7) |
| WPBC | 47.57(15) | 50.38(11) | 49.88(13) | 60.50(2) | **60.73(1)** | 56.13(3) | 52.14(8) | 53.41(6) | 50.00(12) | 46.33(16) | 54.05(4) | 53.46(5) | 52.25(7) | 51.51(10) | 51.82(9) | 47.78(14) |
| yeast | 49.88(14) | 50.62(12) | 44.59(15) | 59.06(4) | 58.75(5) | 54.13(7) | 55.95(6) | 59.50(3) | **61.90(1)** | 53.99(8) | 44.38(16) | 60.48(2) | 52.26(11) | 52.77(9) | 50.26(13) | 52.55(10) |
| CIFAR10 | 60.30(9) | 61.10(7) | 62.48(4) | 59.91(11) | 57.78(13) | 60.95(8) | 64.46(3) | 50.00(16) | 61.20(6) | 69.27(12) | 51.75(9) | 53.52(15) | 55.50(14) | 50.61(15) | 64.58(2) | **67.54(1)** |
| FashionMNIST | 80.12(5) | 83.18(3) | 76.44(7) | 72.02(8) | 71.75(9) | 77.27(6) | 84.64(2) | 50.00(16) | 57.06(12) | 69.27(10) | 53.52(14) | 65.71(11) | 53.60(13) | 50.61(15) | 82.65(4) | **87.70(1)** |
| MNIST-C | 74.58(11) | 78.48(9) | 80.96(6) | **88.04(1)** | 84.59(4) | 75.28(10) | 85.34(2) | 50.24(15) | 61.88(13) | 79.90(7) | 47.11(16) | 78.58(8) | 63.60(12) | 58.72(14) | 83.45(5) | 85.24(3) |
| MVTec-AD | 74.00(2) | 64.46(4) | 72.87(3) | 63.16(6) | 63.40(5) | 60.73(12) | 63.34(7) | 50.90(16) | 51.27(15) | 60.85(11) | 55.02(14) | 61.29(9) | 58.09(13) | 61.21(10) | 63.40(6) | **76.12(1)** |
| SVHN | 57.82(8) | 57.72(9) | 60.05(2) | 59.96(3) | 55.45(10) | 58.74(7) | 59.45(5) | 50.05(16) | 50.52(15) | 55.36(11) | 55.21(12) | 59.45(4) | 51.25(14) | 54.89(13) | 59.20(6) | **62.12(1)** |
| Agnews | 59.31(12) | 63.27(10) | 72.66(3) | 75.06(2) | **75.96(1)** | 64.79(9) | 68.87(5) | 53.39(15) | 68.81(6) | 61.65(11) | 69.53(4) | 57.97(13) | 50.77(16) | 55.93(14) | 68.80(7) | 68.28(8) |
| Amazon | 58.47(12) | 59.22(11) | 62.62(7) | 74.64(2) | **74.67(1)** | 66.65(3) | 64.47(6) | 50.85(14) | 33.95(16) | 58.21(13) | 65.01(5) | 60.04(8) | 50.67(15) | 59.30(10) | 65.94(4) | 59.76(9) |
| Imdb | 49.34(16) | 53.36(11) | 57.98(8) | **66.07(1)** | 64.51(2) | 58.60(6) | 59.85(5) | 53.55(10) | 58.49(7) | 52.57(12) | 63.24(4) | 52.09(14) | 49.99(15) | 54.11(9) | 63.44(3) | 52.45(13) |
| Yelp | 67.92(8) | 60.66(12) | 69.83(5) | **77.80(1)** | 73.99(2) | 71.38(3) | 69.12(7) | 57.20(13) | 45.28(16) | 63.45(10) | 67.11(9) | 61.20(11) | 51.06(15) | 56.82(14) | 69.65(6) | 70.12(4) |
| 20news | 57.29(4) | 57.86(3) | 56.65(6) | **60.43(1)** | 59.18(2) | 54.79(10) | 56.13(7) | 50.04(15) | 53.99(13) | 56.97(5) | 55.55(9) | 49.65(16) | 54.50(11) | 53.89(14) | 54.48(12) | 55.69(8) |

Table D7: AUCPR of 16 label-informed algorithms on 57 benchmark datasets, with labeled anomaly ratio $\gamma_l = 1\%$. We show the performance rank in parenthesis (lower the better), and mark the best performing method(s) in **bold**.

| Datasets | GANomaly | DeepSAD | REPEN | DevNet | PReNet | FEAWAD | XGBOD | NB | SVM | MLP | ResNet | FTTransformer | RF | LGB | XGB | CatB |
|---|---|---|---|---|---|---|---|---|---|---|---|---|---|---|---|---|
| ALOI | 3.83(14) | 6.10(2) | 4.51(6) | 3.76(15) | 3.86(13) | 4.56(4) | **6.68(1)** | 3.29(16) | 4.99(3) | 4.08(11) | 4.33(7) | 3.95(12) | 4.25(8) | 4.12(9) | 4.08(10) | 4.53(5) |
| annthyroid | 34.35(5) | 29.25(9) | 27.38(11) | 31.79(8) | 33.92(6) | 32.05(7) | **65.84(1)** | 43.64(4) | 13.79(15) | 12.99(16) | 15.88(14) | 29.12(10) | 22.34(12) | 21.96(13) | 54.84(3) | 61.79(2) |
| backdoor | 6.95(16) | 38.49(7) | 28.84(10) | **87.83(1)** | 86.20(2) | 68.50(4) | 25.79(12) | 28.06(11) | 51.54(5) | 71.48(3) | 34.99(8) | 33.83(9) | 16.55(14) | 10.42(15) | 42.00(6) | 19.24(13) |
| breastw | 85.14(3) | 79.62(9) | 81.19(7) | 77.74(10) | 72.83(12) | 80.86(8) | 83.82(5) | 54.37(14) | 76.04(11) | 56.04(13) | 51.70(16) | 90.82(2) | 54.04(15) | 85.12(4) | 81.93(6) | **93.23(1)** |
| campaign | 23.13(7) | 19.87(9) | 15.09(13) | 23.18(6) | 23.62(5) | 20.35(8) | 32.98(2) | 11.83(16) | 13.13(15) | 19.21(10) | 13.51(14) | 18.71(11) | 16.55(12) | 25.70(4) | 28.63(3) | **36.63(1)** |
| cardio | 38.52(10) | 30.03(12) | 54.24(4) | 66.86(2) | 63.47(3) | 52.19(5) | 50.41(6) | 20.48(14) | 41.60(9) | 14.11(15) | N/A(N/A) | 48.60(7) | 29.30(13) | 42.16(8) | 36.18(11) | **70.42(1)** |
| Cardiotocography | 33.18(15) | 40.99(12) | 58.09(5) | 61.74(2) | 58.84(4) | 52.60(9) | 46.13(10) | 53.30(7) | 37.51(13) | 21.50(16) | 46.12(11) | 59.03(3) | 34.71(14) | 52.95(8) | 55.45(6) | **65.09(1)** |
| celeba | 11.72(6) | 3.79(13) | 2.94(15) | **15.19(1)** | 14.62(2) | 14.47(3) | 11.64(7) | 2.64(16) | 7.71(9) | 7.51(10) | 7.81(8) | 6.87(11) | 5.46(12) | 3.10(14) | 14.31(4) | 12.62(5) |
| census | 7.47(15) | 9.57(13) | 9.88(12) | 19.70(6) | 18.33(9) | 19.76(5) | 27.07(3) | 6.71(16) | 13.22(10) | 19.66(7) | 8.87(14) | 19.01(8) | 12.47(11) | 21.98(4) | **30.88(1)** | 30.59(2) |
| cover | 0.89(16) | 24.22(10) | 58.34(6) | 68.57(2) | 65.28(3) | 51.36(7) | 58.61(5) | 1.02(15) | 18.12(11) | 35.14(9) | 15.24(13) | **81.57(1)** | 15.73(12) | 1.04(14) | 62.00(4) | 38.30(8) |
| donors | 9.68(15) | 79.14(6) | 17.03(14) | 90.73(3) | **98.08(1)** | 94.62(2) | 60.93(9) | 84.66(5) | 3.07(16) | 20.07(13) | 36.26(11) | 85.13(4) | 23.75(12) | 38.36(10) | 68.71(8) | 72.67(7) |
| fault | 48.92(3) | **49.92(1)** | 49.84(2) | 48.46(5) | 48.91(4) | 44.25(9) | 43.21(10) | 45.17(7) | 29.76(16) | 35.17(15) | 36.23(14) | 44.88(8) | 38.52(13) | 43.06(11) | 42.53(12) | 47.30(6) |
| fraud | 42.43(6) | 28.00(13) | 42.35(7) | 52.80(3) | **55.13(1)** | 54.48(2) | 38.57(10) | 0.14(15) | 21.23(14) | 41.61(8) | 31.53(12) | 48.90(4) | 48.38(5) | 0.14(15) | 36.75(11) | 39.79(9) |
| glass | 13.34(16) | 15.19(15) | 23.20(11) | 18.25(13) | 31.20(8) | **49.47(1)** | 36.70(3) | 15.35(14) | 44.54(2) | 21.84(12) | 29.46(9) | 23.18(10) | 32.33(6) | 32.00(7) | 35.87(4) | 35.84(5) |
| Hepatitis | 21.59(14) | 28.50(12) | 43.55(4) | 43.85(7) | 44.05(6) | 35.85(10) | 52.70(2) | 21.43(15) | 16.00(16) | 34.40(11) | 42.58(9) | 45.40(5) | 26.75(13) | 50.91(3) | **54.55(1)** | 49.15(4) |
| http | 73.48(11) | 74.48(10) | 93.99(7) | **100.00(1)** | **100.00(1)** | 97.78(6) | 80.39(8) | 63.50(13) | 61.84(14) | 0.37(16) | **100.00(1)** | 99.70(5) | 66.21(12) | 0.38(15) | **100.00(1)** | 77.95(9) |
| InternetAds | 46.05(3) | 42.68(4) | 41.92(5) | 33.91(10) | 34.32(9) | 39.10(6) | 36.29(7) | 18.82(14) | 34.55(8) | 17.30(15) | 30.48(11) | N/A(N/A) | 25.24(13) | 30.27(12) | 47.94(2) | **55.26(1)** |
| Ionosphere | **89.55(1)** | 69.44(3) | 64.46(6) | 50.93(13) | 53.97(12) | 49.57(14) | 65.38(6) | 46.42(16) | 63.89(9) | 68.45(4) | 57.52(11) | 57.75(10) | 48.79(15) | 65.26(7) | 64.24(5) | 82.86(2) |
| landsat | 18.28(16) | 45.05(4) | 28.74(15) | 37.63(9) | 37.40(11) | 39.89(7) | 50.99(2) | 39.34(8) | 32.44(12) | 45.06(3) | 31.81(13) | 37.61(10) | 30.74(14) | 40.65(6) | 44.72(5) | **53.27(1)** |
| letter | 15.94(14) | 18.33(13) | 13.33(9) | 14.25(7) | 14.87(5) | 14.36(6) | 18.55(2) | 6.25(16) | 12.38(10) | 7.45(15) | **28.06(1)** | 8.07(13) | 7.87(14) | 13.46(8) | 11.80(11) | 11.44(12) |
| Lymphography | 65.66(6) | 43.96(14) | 62.57(8) | 60.68(10) | 57.53(11) | 63.50(7) | 83.69(2) | 36.34(15) | 4.11(16) | 49.48(12) | 61.85(9) | 80.75(4) | 45.60(13) | 80.81(3) | 79.46(5) | **94.13(1)** |
| magic.gamma | 39.96(16) | 64.64(5) | 66.45(4) | 71.97(2) | **72.46(1)** | 62.42(8) | 63.84(7) | 70.82(3) | 42.53(14) | 48.51(12) | 64.16(6) | 49.74(13) | 58.22(9) | 56.69(11) | 57.72(10) | 61.79(8) |
| mammography | 15.75(13) | 32.24(7) | 40.20(4) | 40.49(3) | 36.69(6) | **53.56(1)** | 23.52(10) | 29.90(8) | 16.01(14) | 1.37(16) | 22.86(11) | 42.24(2) | 12.30(15) | 2.44(15) | 28.94(9) | 37.72(5) |
| mnist | 17.88(16) | 32.84(10) | 48.06(7) | 56.31(2) | 55.93(3) | 50.16(6) | 51.69(5) | 22.77(15) | 28.89(12) | 52.97(4) | 32.16(11) | 39.59(9) | 25.39(13) | 24.23(14) | 41.73(8) | **56.31(1)** |
| musk | 89.49(2) | 46.21(12) | 39.11(13) | 78.41(6) | **94.05(1)** | 78.76(5) | 55.84(8) | 3.16(16) | 20.90(15) | 55.74(9) | 89.07(3) | 74.64(7) | 29.36(14) | 48.96(11) | 54.93(10) | 88.03(4) |
| optdigits | 2.83(16) | 27.71(13) | 96.95(4) | **99.02(1)** | 98.73(2) | 98.69(3) | 48.64(8) | 4.31(14) | 35.64(10) | 38.83(9) | 29.25(12) | 93.10(5) | 33.03(11) | 3.15(15) | 57.85(6) | 57.02(7) |
| PageBlocks | 41.22(7) | 51.76(3) | 57.34(2) | 40.91(8) | 45.31(6) | 33.40(12) | 47.12(5) | 34.27(11) | 26.74(13) | 20.62(15) | 20.93(14) | 39.79(9) | 17.00(16) | 34.30(10) | 49.13(4) | **67.14(1)** |
| pendigits | 2.80(15) | 27.59(8) | 46.66(4) | 53.88(2) | **57.07(1)** | 41.55(5) | 34.75(6) | 13.37(13) | 24.20(10) | 9.47(14) | 27.38(9) | 48.91(3) | 19.12(12) | 2.25(16) | 36.37(7) | 53.30(1) |
| Pima | 43.77(11) | 44.26(10) | 50.96(2) | 45.76(7) | 44.44(9) | 42.87(12) | 50.03(3) | 44.93(8) | 37.15(15) | 23.61(16) | 41.13(13) | 47.15(6) | 38.80(14) | 47.18(5) | 47.40(4) | **53.50(1)** |
| satellite | 65.05(10) | 67.43(9) | 71.23(5) | **79.41(1)** | 77.14(2) | 75.32(3) | 70.55(6) | 73.20(4) | 44.82(16) | 60.43(12) | 50.63(14) | 70.54(7) | 48.47(15) | 60.63(11) | 58.41(13) | 70.24(8) |
| satimage-2 | 43.15(10) | 40.06(11) | 85.70(7) | 89.99(3) | 92.20(2) | **94.21(1)** | 46.18(9) | 1.21(16) | 30.29(13) | 2.87(15) | 78.32(8) | 88.59(5) | 33.82(12) | 9.10(14) | 85.77(6) | 88.73(4) |
| shuttle | 46.19(14) | 96.40(4) | **97.27(1)** | 96.72(3) | 96.50(3) | 96.39(5) | 95.52(7) | 90.40(10) | 86.80(12) | 25.53(15) | 88.44(11) | 96.06(6) | 20.75(16) | 54.32(13) | 91.09(9) | 94.95(8) |
| skin | 21.98(16) | 83.45(8) | 58.19(14) | 68.55(10) | 66.43(11) | 87.83(5) | 77.24(9) | 84.91(6) | **98.57(1)** | 60.08(13) | 94.57(3) | 88.78(4) | 31.08(15) | 61.98(12) | 83.77(7) | 95.74(2) |
| smtp | 21.46(12) | 61.13(2) | 41.68(7) | 50.02(4) | **66.68(1)** | 25.12(11) | 50.02(3) | 0.04(15) | 17.73(13) | 38.92(8) | 11.71(14) | 33.94(9) | 50.01(5) | 50.01(5) | 28.13(10) | 59.75(6) |
| SpamBase | 40.69(15) | 45.89(13) | 63.07(8) | 75.16(2) | 72.64(3) | 60.04(9) | 68.17(5) | 53.63(11) | 34.12(16) | 56.96(10) | 43.22(14) | 71.31(4) | 48.87(12) | 66.16(7) | 67.01(6) | **76.82(1)** |
| speech | 1.61(15) | 2.44(5) | 2.41(6) | 3.00(3) | 2.97(4) | 3.15(2) | 2.06(7) | 1.63(14) | **3.48(1)** | 1.81(11) | 1.89(10) | 1.93(9) | 1.65(13) | 1.69(12) | 1.50(16) | 2.04(8) |
| Stamps | 29.94(9) | 15.90(15) | **53.59(1)** | 48.27(4) | 49.94(3) | 38.55(6) | 27.26(10) | 11.60(16) | 39.29(5) | 18.26(13) | 26.67(11) | 52.80(2) | 19.95(14) | 24.17(12) | 30.14(8) | 34.72(7) |
| thyroid | 53.25(8) | 38.94(10) | 79.14(2) | 74.23(3) | **79.44(1)** | 62.31(5) | 45.13(9) | 2.47(15) | 12.08(14) | 17.01(12) | 21.29(11) | 57.06(7) | 13.43(13) | 2.47(15) | 73.72(4) | 58.56(6) |
| vertebral | 10.60(16) | 16.44(13) | 20.03(11) | 31.27(2) | 30.73(3) | **38.38(1)** | 20.29(10) | 15.98(14) | 21.77(8) | 18.80(12) | 24.05(6) | 20.58(9) | 15.98(14) | 26.33(5) | 28.37(4) | 23.74(7) |
| vowels | 24.96(10) | 18.93(14) | 29.27(8) | 34.85(6) | 38.61(2) | **42.34(1)** | 34.97(5) | 7.72(15) | 20.88(13) | 2.02(16) | 37.55(4) | 24.53(11) | 23.54(12) | 26.69(9) | 29.55(7) | 38.45(3) |
| Waveform | 4.44(10) | 12.50(2) | 4.50(9) | 9.00(3) | 6.65(7) | 3.73(14) | 7.21(4) | 2.90(16) | 5.54(8) | 4.17(13) | **13.56(1)** | 6.82(5) | 3.50(15) | 4.38(11) | 4.20(12) | 6.74(6) |
| WBC | 42.31(12) | 56.30(10) | 73.49(6) | 83.17(2) | 81.48(4) | 68.44(7) | 65.07(8) | 12.94(16) | 13.72(15) | 34.75(13) | 46.09(11) | 81.78(3) | 28.14(14) | 60.18(9) | **86.33(1)** | 76.80(5) |
| WDBC | 59.54(11) | 39.38(13) | 91.29(5) | **98.21(1)** | 96.59(3) | 94.21(4) | 68.19(10) | 14.48(14) | 10.09(15) | 1.64(16) | 44.39(12) | 97.13(2) | 78.80(7) | 88.39(6) | 74.97(9) | 77.12(8) |
| Wilt | 4.60(16) | 9.71(8) | 5.07(15) | 8.33(10) | 8.23(11) | 10.67(7) | 12.08(5) | 11.15(6) | 12.45(4) | 8.00(12) | **22.71(1)** | 7.12(14) | 8.56(9) | 7.61(13) | 15.87(3) | 17.13(2) |
| wine | 14.75(14) | 18.73(13) | 83.97(5) | 99.85(2) | **100.00(1)** | 92.74(3) | 59.74(7) | 19.47(12) | 8.11(15) | 4.72(16) | 45.56(10) | 91.62(4) | 23.51(11) | 59.17(8) | 77.45(6) | 53.41(9) |
| WPBC | 22.57(15) | 27.27(13) | 29.49(7) | 33.66(3) | 34.91(2) | 32.97(4) | 29.67(6) | 29.23(9) | 23.00(14) | 22.31(16) | **37.53(1)** | 31.41(5) | 28.75(11) | 29.39(8) | 29.14(10) | 27.87(12) |
| yeast | 33.70(14) | 33.95(13) | 32.36(16) | 41.43(5) | 40.74(6) | 38.78(7) | 41.68(3) | 41.84(2) | **43.43(1)** | 34.25(12) | 32.88(15) | 41.49(4) | 35.62(11) | 36.79(9) | 36.05(10) | 36.73(8) |
| CIFAR10 | 9.20(12) | 9.16(13) | 11.62(8) | 13.02(3) | 12.66(7) | **13.67(1)** | 13.31(2) | 5.00(16) | 12.80(6) | 10.29(10) | 9.82(11) | 10.54(9) | 5.71(15) | 8.04(14) | 12.80(5) | 12.91(4) |
| FashionMNIST | 21.07(11) | 32.45(7) | 26.71(10) | 39.41(4) | 39.84(3) | **45.76(1)** | 40.81(2) | 5.01(16) | 15.81(13) | 31.85(8) | 17.53(12) | 27.08(9) | 9.33(15) | 11.47(14) | 38.78(5) | 36.64(6) |
| MNIST-C | 22.44(12) | 29.20(10) | 38.69(9) | **64.55(1)** | 60.66(2) | 48.14(5) | 50.35(3) | 5.39(16) | 23.36(11) | 48.01(6) | 10.85(15) | 42.30(8) | 22.23(13) | 12.95(14) | 48.65(4) | 46.00(7) |
| MVTec-AD | 57.38(2) | 45.33(5) | 55.41(3) | 44.41(7) | 44.31(8) | 44.85(6) | 42.43(10) | 25.76(16) | 28.76(15) | 43.84(9) | 35.13(14) | 47.87(4) | 35.63(13) | 42.25(11) | 41.85(12) | **60.55(1)** |
| SVHN | 8.12(11) | 7.70(12) | 10.36(5) | **12.08(1)** | 10.45(4) | 11.13(2) | 10.31(6) | 5.07(16) | 7.16(14) | 8.67(9) | 8.50(10) | 11.12(3) | 5.55(15) | 7.69(13) | 10.25(7) | 10.22(8) |
| Agnews | 6.74(14) | 10.92(9) | 16.93(4) | 25.45(2) | **26.51(1)** | 20.22(3) | 13.70(7) | 5.78(15) | 16.41(5) | 9.48(11) | 15.02(6) | 7.93(13) | 5.31(16) | 9.30(12) | 13.26(8) | 9.54(10) |
| Amazon | 6.19(13) | 7.32(10) | 9.62(6) | 16.14(2) | **17.30(1)** | 13.57(5) | 9.50(7) | 5.11(14) | 7.12(11) | 3.57(16) | 13.17(4) | 8.67(8) | 5.22(15) | 8.36(9) | 9.97(5) | 6.95(12) |
| Imdb | 4.88(16) | 6.09(10) | 7.73(8) | **9.40(1)** | 8.95(3) | 7.80(7) | 8.08(6) | 5.66(13) | 9.00(2) | 5.79(11) | 8.89(4) | 5.75(12) | 5.09(15) | 6.17(9) | 8.77(5) | 5.59(14) |
| Yelp | 9.03(12) | 9.04(11) | 17.12(4) | **20.32(1)** | 19.42(2) | 17.30(3) | 13.40(6) | 6.49(15) | 8.33(14) | 11.38(10) | 14.27(5) | 11.38(9) | 5.37(16) | 8.50(13) | 12.83(7) | 12.21(8) |
| 20news | 7.20(14) | **10.52(1)** | 7.95(9) | 10.30(2) | 9.85(3) | 7.97(7) | 7.55(12) | 4.83(16) | 7.99(6) | 7.85(10) | 7.30(13) | 6.11(15) | 8.60(5) | 9.35(4) | 7.96(8) | 7.84(11) |

Table D8: AUCROC of 16 label-informed algorithms on 57 benchmark datasets, with labeled anomaly ratio $\gamma_l = 5\%$. We show the performance rank in parenthesis (lower the better), and mark the best performing method(s) in **bold**.

| Datasets | GANomaly | DeepSAD | REPEN | DevNet | PReNet | FEAWAD | XGBOD | NB | SVM | MLP | ResNet | FTTransformer | RF | LGB | XGB | CatB |
|---|---|---|---|---|---|---|---|---|---|---|---|---|---|---|---|---|
| ALOI | 56.71(4) | 57.88(2) | 52.13(9) | 47.46(16) | 48.51(13) | 52.77(8) | **65.99(1)** | 48.37(14) | 49.86(11) | 48.95(12) | 47.91(15) | 51.97(10) | 56.68(5) | 55.28(6) | 57.22(3) | 55.05(7) |
| annthyroid | 76.22(14) | 79.92(11) | 78.91(12) | 81.64(10) | 82.42(7) | 89.05(6) | 97.24(2) | 81.99(9) | 49.80(16) | 71.31(15) | 77.35(13) | 95.13(4) | 82.25(8) | 93.14(5) | 95.24(3) | **98.51(1)** |
| backdoor | 81.81(12) | 95.13(3) | 89.33(10) | 95.58(2) | 94.37(4) | **96.80(1)** | 91.07(8) | 79.06(13) | 86.80(11) | 93.12(7) | 66.31(16) | 89.84(9) | 76.68(14) | 76.63(15) | 93.34(6) | 94.34(5) |
| breastw | 92.28(11) | 90.67(12) | 97.61(4) | 99.18(2) | 92.79(9) | 92.67(10) | 96.72(6) | **99.73(1)** | 72.44(15) | 95.42(7) | 49.28(16) | 88.59(13) | 85.59(14) | 97.53(5) | 95.12(8) | 99.10(3) |
| campaign | 64.43(14) | 69.71(10) | 77.88(16) | 81.03(5) | 80.65(6) | 71.01(8) | 88.41(2) | 69.02(11) | 68.58(12) | 70.18(9) | 59.17(15) | 67.08(13) | 72.78(7) | 84.86(4) | 86.48(3) | **88.89(1)** |
| cardio | 82.46(13) | 79.71(14) | 95.56(5) | **97.12(1)** | 96.36(3) | 87.96(10) | 95.73(4) | 71.04(15) | 83.30(12) | 90.85(7) | N/A(N/A) | 92.65(6) | 85.07(11) | 88.03(9) | 89.79(8) | 96.82(2) |
| Cardiotocography | 54.77(16) | 81.24(12) | 89.88(3) | **91.17(1)** | 90.73(2) | 81.89(10) | 83.73(8) | 86.77(5) | 80.19(13) | 84.67(6) | 60.64(15) | 81.73(11) | 79.49(14) | 83.03(9) | 84.55(7) | 89.12(4) |
| celeba | 68.79(12) | 75.40(10) | 56.47(16) | **91.01(1)** | 90.64(3) | 86.63(6) | 87.45(5) | 74.33(11) | 56.62(15) | 90.93(2) | 58.14(14) | 89.93(4) | 63.12(13) | 83.72(9) | 83.90(8) | 84.69(7) |
| census | 59.66(14) | 69.05(13) | 69.20(12) | 81.03(5) | 75.34(10) | 76.32(8) | 87.48(2) | 57.28(15) | 79.56(6) | 75.94(9) | 56.06(16) | 78.81(7) | 70.47(11) | 83.37(4) | **88.04(1)** | 86.81(3) |
| cover | 44.46(16) | 94.43(9) | 99.69(2) | 99.47(4) | 99.47(3) | 91.01(11) | 97.18(6) | 96.92(7) | 89.40(12) | 96.38(8) | 68.67(14) | **99.92(1)** | 84.32(13) | 57.48(15) | 94.08(10) | 99.24(5) |
| donors | 68.62(15) | 99.92(4) | 82.81(13) | **99.99(1)** | 99.95(3) | 99.99(2) | 99.59(7) | 99.42(8) | 64.53(16) | 99.62(6) | 95.70(11) | 99.90(5) | 77.22(14) | 95.13(12) | 97.90(10) | 99.29(9) |
| fault | 64.26(12) | 68.76(4) | 68.70(5) | 67.54(7) | 66.67(9) | 62.94(14) | 69.30(3) | 63.65(13) | 62.96(15) | 66.07(10) | 50.92(16) | 66.99(8) | 65.00(11) | 69.68(2) | 68.15(6) | **72.02(1)** |
| fraud | 90.52(5) | 87.66(9) | 92.18(2) | 89.77(7) | 91.91(3) | 81.61(12) | 89.84(6) | 50.00(15) | **92.44(1)** | 85.73(10) | 50.99(14) | 75.38(13) | 84.92(11) | 44.99(16) | 87.90(8) | 90.86(4) |
| glass | 67.84(16) | 84.34(11) | 82.27(13) | 91.37(9) | 94.46(4) | 93.72(5) | 95.73(3) | 92.73(6) | 80.63(14) | 82.59(12) | 89.96(10) | 91.48(8) | 78.85(15) | 92.17(7) | 95.97(2) | **98.37(1)** |
| Hepatitis | 56.94(15) | 74.83(12) | 88.10(4) | 86.62(5) | 83.71(8) | 72.08(13) | 85.44(6) | 77.79(11) | 50.00(16) | 85.11(7) | 80.48(10) | 82.99(9) | 69.71(14) | 89.29(3) | 89.66(2) | **91.72(1)** |
| http | 99.80(11) | 99.90(10) | 99.98(8) | **100.00(1)** | 99.99(7) | 99.99(6) | 98.33(12) | 83.31(14) | 0.14(16) | **100.00(1)** | **100.00(1)** | **100.00(1)** | **100.00(1)** | 39.87(15) | **100.00(1)** | 99.97(9) |
| InternetAds | 67.94(8) | 72.98(4) | 79.79(2) | 61.68(11) | 57.82(12) | 63.57(10) | 75.77(3) | 51.49(14) | 69.28(5) | 57.14(13) | 50.82(15) | N/A(N/A) | 68.01(7) | 68.68(6) | 67.87(9) | **81.94(1)** |
| Ionosphere | 92.09(4) | 81.01(9) | **94.21(1)** | 68.67(13) | 67.54(14) | 63.46(15) | 93.43(3) | 85.42(7) | 83.55(8) | 71.41(12) | 50.49(16) | 76.83(10) | 74.95(11) | 88.77(5) | 87.23(6) | 93.99(2) |
| landsat | 46.33(16) | 85.54(4) | 59.84(15) | 77.78(12) | 78.32(11) | 81.22(7) | 87.34(2) | 71.02(13) | 79.27(10) | 80.25(8) | 66.81(14) | 86.98(3) | 79.52(9) | 84.70(5) | 82.67(6) | **88.44(1)** |
| letter | 69.64(4) | 73.09(3) | 66.43(5) | 65.01(7) | 63.18(11) | 64.18(8) | **83.13(1)** | 48.83(14) | 40.02(16) | 42.09(15) | 76.96(2) | 59.82(13) | 62.33(12) | 65.92(6) | 63.80(9) | 63.50(10) |
| Lymphography | 96.84(2) | 86.90(8) | 92.94(4) | 80.25(9) | 79.68(10) | 93.47(3) | 77.40(13) | 73.19(14) | 50.00(16) | 70.06(15) | 87.01(7) | 91.87(5) | 78.22(12) | 91.31(6) | 78.66(11) | **99.71(1)** |
| magic.gamma | 61.44(15) | 82.08(6) | 78.24(10) | 82.76(5) | 83.38(4) | 84.04(3) | 84.58(2) | 76.10(13) | 49.30(16) | **85.08(1)** | 71.05(14) | 81.91(7) | 78.06(12) | 79.11(9) | 78.22(11) | 81.85(8) |
| mammography | 75.46(13) | 90.90(6) | 91.90(3) | 92.90(2) | **93.17(1)** | 91.28(4) | 87.22(10) | 87.26(9) | 69.49(16) | 89.65(8) | 77.88(12) | 90.80(7) | 72.62(14) | 71.32(15) | 86.44(11) | 91.15(5) |
| mnist | 69.09(14) | 84.99(12) | 92.30(7) | 95.39(3) | 93.37(5) | 84.77(13) | 96.28(2) | 57.22(16) | 87.89(10) | 89.03(9) | 67.59(15) | 92.84(6) | 86.55(11) | 92.07(8) | 94.45(4) | **97.02(1)** |
| musk | 97.34(9) | 96.46(10) | 93.97(11) | **100.00(1)** | **100.00(1)** | **100.00(1)** | 99.90(5) | 83.33(13) | 54.07(16) | **100.00(1)** | 78.75(15) | 99.89(6) | 81.60(14) | 83.97(12) | 99.66(8) | 99.78(7) |
| optdigits | 46.44(16) | 94.98(10) | 99.67(4) | **99.98(1)** | 99.98(2) | 99.94(3) | 98.23(8) | 80.46(13) | 90.94(12) | 99.48(5) | 52.71(15) | 98.84(7) | 93.91(11) | 78.07(14) | 96.75(9) | 99.04(6) |
| PageBlocks | 72.69(15) | 93.01(4) | 91.22(6) | 86.42(12) | 90.38(7) | 89.67(9) | 94.73(2) | 88.46(10) | 58.16(16) | 90.15(8) | 82.94(13) | 88.03(11) | 76.84(14) | 92.74(5) | 94.28(3) | **96.80(1)** |
| pendigits | 56.82(16) | 97.41(11) | 99.75(2) | **99.76(1)** | 99.69(3) | 99.59(9) | 99.66(4) | 99.10(5) | 84.95(12) | 79.23(13) | 70.59(14) | 98.27(9) | 84.29(13) | 64.95(15) | 98.33(7) | 99.03(6) |
| Pima | 60.11(13) | 63.23(12) | 73.18(4) | **78.02(1)** | 76.43(2) | 63.55(11) | 69.92(8) | 71.13(6) | 47.99(15) | 32.37(16) | 51.59(14) | 68.69(9) | 67.36(10) | 71.80(5) | 70.80(7) | 73.80(3) |
| satellite | 69.49(15) | 88.65(3) | 81.31(12) | 84.34(10) | 84.41(9) | 85.32(7) | 89.34(2) | 79.37(13) | 61.94(16) | 87.79(5) | 75.45(14) | 87.82(4) | 82.71(11) | 86.74(6) | 85.20(8) | **90.22(1)** |
| satimage-2 | 96.60(10) | 98.11(3) | 97.79(7) | 98.03(4) | 97.82(6) | 98.55(2) | 97.83(5) | 91.43(12) | 65.61(15) | 96.65(9) | 85.08(14) | 96.99(8) | 87.97(13) | 50.02(16) | 93.66(11) | **98.93(1)** |
| shuttle | 77.41(16) | 98.71(4) | 98.83(3) | 97.57(8) | 97.69(7) | 97.29(11) | **99.73(1)** | 94.63(14) | 97.04(13) | 97.49(9) | 97.97(6) | 97.44(10) | 78.01(15) | 97.16(12) | 98.00(5) | 99.22(2) |
| skin | 52.78(16) | 99.41(4) | 90.56(14) | 95.72(11) | 95.24(12) | 98.56(8) | 99.22(5) | 94.89(13) | 99.60(3) | 97.23(10) | **99.83(1)** | 99.06(6) | 78.23(15) | 99.00(7) | 98.93(9) | 99.72(2) |
| smtp | 51.69(14) | 82.74(3) | 76.47(8) | 75.99(9) | 75.24(10) | 74.57(11) | 79.87(5) | 50.00(15) | 62.98(12) | **86.40(1)** | 52.83(13) | 79.07(6) | 83.32(2) | 43.01(16) | 81.12(4) | 78.85(7) |
| SpamBase | 53.66(16) | 70.72(13) | 83.55(8) | 90.34(3) | 90.87(2) | 79.54(12) | 89.24(4) | 79.84(10) | 68.44(14) | 87.16(6) | 54.44(15) | 87.75(5) | 78.85(12) | 84.84(7) | 82.86(9) | **91.87(1)** |
| speech | 47.49(15) | 53.11(11) | 53.13(10) | 62.36(3) | **63.60(1)** | 54.61(7) | 58.20(4) | 49.88(14) | 63.40(2) | 50.39(13) | 53.45(9) | 56.61(6) | 47.39(16) | 51.71(12) | 56.82(5) | 53.58(8) |
| Stamps | 72.13(14) | 74.61(12) | 95.30(3) | 94.14(5) | 96.07(2) | 79.40(10) | 95.09(4) | 83.22(9) | 54.66(16) | 75.90(11) | 71.39(15) | 91.02(6) | 73.95(13) | 84.90(8) | 87.03(7) | **96.07(1)** |
| thyroid | 92.65(11) | 95.21(10) | **99.55(1)** | 93.50(9) | 99.50(2) | 99.42(4) | 98.89(7) | 96.53(8) | 71.37(14) | 85.18(12) | 51.29(16) | 99.35(5) | 78.61(13) | 71.20(15) | 95.76(9) | 98.99(6) |
| vertebral | 37.14(16) | 61.49(13) | 55.86(15) | 77.65(4) | 77.24(5) | 77.20(7) | 69.29(11) | 81.21(3) | 71.90(9) | 67.05(12) | 69.77(10) | **81.88(1)** | 56.29(14) | 73.50(8) | 81.65(2) | 77.22(6) |
| vowels | 78.54(9) | 80.31(7) | 86.47(6) | 89.12(2) | **92.49(1)** | 86.69(5) | 87.82(3) | 73.29(12) | 55.90(15) | 14.35(16) | 80.28(8) | 73.35(11) | 66.06(14) | 73.15(13) | 73.74(10) | 87.31(4) |
| Waveform | 52.99(16) | 68.05(10) | 80.58(5) | **84.84(1)** | 82.30(3) | 64.09(13) | 74.68(7) | 77.42(6) | 60.62(15) | 61.03(14) | 72.60(8) | 80.85(4) | 64.74(12) | 64.81(11) | 71.76(9) | 84.82(2) |
| WBC | 92.78(7) | 94.96(6) | 97.88(5) | 98.38(2) | 98.28(3) | 68.09(14) | 74.32(12) | 65.00(15) | 58.91(16) | 76.82(10) | 72.86(13) | 97.98(4) | 81.68(8) | 81.23(9) | 76.14(11) | **98.61(1)** |
| WDBC | 97.52(7) | 90.44(11) | 98.81(5) | **99.91(1)** | 99.82(3) | 99.73(4) | 95.24(10) | 56.02(15) | 61.79(14) | 0.45(16) | 82.49(13) | 99.87(2) | 90.15(12) | 95.27(9) | 95.46(8) | 98.49(6) |
| Wilt | 38.88(16) | 81.89(4) | 49.93(15) | 67.91(12) | 68.39(10) | 80.51(6) | 83.57(3) | 74.33(9) | 79.01(8) | 65.79(13) | 87.83(2) | 64.93(14) | 68.20(11) | 80.75(5) | 80.35(7) | **89.99(1)** |
| wine | 65.76(14) | 82.24(12) | 99.91(3) | **100.00(1)** | **100.00(1)** | 99.82(6) | 99.82(5) | 95.82(7) | 50.00(15) | 3.90(16) | 88.46(10) | 99.82(4) | 73.73(13) | 89.63(9) | 94.27(8) | 99.20(6) |
| WPBC | 47.66(15) | 55.28(13) | 57.32(10) | 68.74(2) | **69.92(1)** | 57.00(11) | 61.95(6) | 55.92(12) | 44.53(16) | 54.06(14) | 64.60(3) | 60.63(7) | 60.11(8) | 57.48(9) | 62.26(5) | 64.39(4) |
| yeast | 48.22(15) | 54.78(12) | 47.53(16) | 65.99(2) | **66.13(1)** | 59.70(7) | 55.83(10) | 63.75(3) | 62.62(4) | 53.76(13) | 50.93(14) | 60.65(6) | 56.89(9) | 57.22(8) | 55.18(11) | 62.04(5) |
| CIFAR10 | 60.37(13) | 64.91(9) | 71.88(4) | 71.06(5) | 65.98(8) | 67.83(7) | 75.50(2) | 50.54(16) | 61.36(11) | 64.51(10) | 55.79(14) | 61.16(12) | 54.31(15) | 70.83(6) | **75.67(1)** | 73.73(3) |
| FashionMNIST | 79.87(10) | 86.83(6) | 87.79(5) | 88.59(4) | 86.04(7) | 81.73(9) | **91.92(1)** | 52.53(16) | 68.13(13) | 74.74(12) | 62.50(14) | 77.93(11) | 61.54(15) | 83.07(8) | 90.37(2) | 90.15(3) |
| MNIST-C | 74.58(14) | 85.83(7) | 82.41(12) | **93.29(1)** | 91.68(2) | 87.72(6) | 90.97(3) | 60.20(16) | 83.56(10) | 84.27(8) | 60.69(15) | 84.12(9) | 82.86(11) | 82.09(13) | 90.65(4) | 88.24(5) |
| MVTec-AD | 74.14(5) | 68.81(11) | 72.72(6) | 71.58(7) | 71.29(8) | 62.91(13) | 76.56(4) | 53.39(16) | 56.56(14) | 64.59(12) | 54.82(15) | 69.49(10) | 70.60(9) | 76.56(3) | 77.29(2) | **82.41(1)** |
| SVHN | 57.94(12) | 59.56(10) | 64.82(4) | **67.36(1)** | 62.21(9) | 63.69(6) | 67.23(2) | 49.84(16) | 62.54(8) | 58.40(11) | 57.44(13) | 56.13(14) | 52.59(15) | 63.56(7) | 66.56(3) | 63.93(5) |
| Agnews | 58.78(14) | 61.53(12) | 69.54(8) | 82.86(2) | **83.35(1)** | 72.33(6) | 74.70(5) | 51.41(15) | 72.21(7) | 78.45(3) | 59.85(13) | 66.80(11) | 51.11(16) | 68.07(10) | 76.52(4) | 69.50(9) |
| Amazon | 49.59(16) | 59.52(13) | 72.56(5) | **77.35(1)** | 77.09(2) | 72.52(6) | 69.03(9) | 52.76(14) | 72.80(4) | 73.43(3) | 65.74(10) | 60.16(12) | 51.58(15) | 63.62(11) | 69.74(8) | 71.08(7) |
| Imdb | 67.87(12) | 64.79(13) | 79.86(7) | **86.80(1)** | 86.73(2) | 82.74(4) | 80.12(6) | 58.21(15) | 76.40(8) | 86.15(3) | 63.24(14) | 70.17(11) | 53.62(16) | 75.66(9) | 80.88(5) | 74.69(10) |
| Yelp | 67.87(12) | 64.79(13) | 79.86(7) | **86.80(1)** | 86.73(2) | 82.74(4) | 80.12(6) | 58.21(15) | 76.40(8) | 86.15(3) | 63.24(14) | 70.17(11) | 53.62(16) | 75.66(9) | 80.88(5) | 74.69(10) |
| 20news | 57.42(12) | 64.54(6) | 56.56(14) | **66.66(1)** | 65.49(2) | 61.30(8) | 64.95(4) | 56.48(15) | 51.87(16) | 58.33(11) | 60.13(9) | 56.89(13) | 58.58(10) | 64.40(7) | 65.12(3) | 64.81(5) |

Table D9: AUCPR of 16 label-informed algorithms on 57 benchmark datasets, with labeled anomaly ratio $\gamma_l = 5\%$. We show the performance rank in parenthesis (lower the better), and mark the best performing method(s) in **bold**.

| Datasets | GANomaly | DeepSAD | REPEN | DevNet | PReNet | FEAWAD | XGBOD | NB | SVM | MLP | ResNet | FTTransformer | RF | LGB | XGB | CatB |
|---|---|---|---|---|---|---|---|---|---|---|---|---|---|---|---|---|
| ALOI | 3.92(11) | 5.85(4) | 4.35(9) | 4.55(8) | 4.67(7) | 3.83(14) | **7.59(1)** | 3.49(15) | 3.39(16) | 3.88(12) | 4.02(10) | 3.85(13) | 5.88(3) | 4.77(6) | 6.00(2) | 5.45(5) |
| annthyroid | 36.68(12) | 37.97(11) | 36.09(14) | 43.73(8) | 43.73(9) | 51.83(6) | 77.54(2) | 42.73(10) | 11.84(16) | 25.83(15) | 36.36(13) | 76.62(3) | 49.66(7) | 64.75(5) | 72.25(4) | **78.01(1)** |
| backdoor | 7.05(16) | 62.79(7) | 28.48(15) | 88.48(3) | 88.58(2) | **88.99(1)** | 61.12(8) | 30.12(14) | 51.30(9) | 85.62(4) | 42.59(13) | 74.80(5) | 43.47(12) | 47.18(11) | 63.30(6) | 50.67(10) |
| breastw | 85.59(12) | 83.65(13) | 95.32(4) | 98.65(2) | 92.91(8) | 93.09(7) | 91.96(9) | **99.49(1)** | 75.47(15) | 94.88(6) | 56.83(16) | 85.66(11) | 79.61(14) | 95.09(5) | 91.30(10) | 98.04(3) |
| campaign | 23.50(14) | 26.68(10) | 15.33(16) | 39.15(6) | 39.32(5) | 29.98(8) | **47.53(1)** | 24.15(12) | 24.90(11) | 27.35(9) | 20.49(15) | 23.76(13) | 31.65(7) | 43.55(4) | 44.55(3) | 46.68(2) |
| cardio | 39.02(15) | 45.81(13) | 82.12(4) | **85.56(1)** | 82.34(3) | 77.24(6) | 79.07(5) | 73.90(4) | 55.11(14) | 67.55(8) | N/A(N/A) | 64.82(9) | 57.80(12) | 63.75(10) | 83.54(2) | 70.42(5) |
| Cardiotocography | 33.71(16) | 60.87(12) | 75.22(3) | **76.46(1)** | 75.97(2) | 64.45(8) | 62.98(10) | 73.90(4) | 55.11(14) | 65.29(6) | 45.15(15) | 63.22(9) | 58.32(13) | 61.57(11) | 64.77(7) | 70.42(5) |
| celeba | 10.97(11) | 9.34(12) | 3.12(16) | 24.67(2) | **26.54(1)** | 22.52(3) | 18.27(6) | 12.02(10) | 7.09(14) | 22.35(4) | 5.76(15) | 19.25(5) | 8.64(13) | 15.78(8) | 16.37(7) | 12.24(9) |
| census | 7.65(16) | 15.02(12) | 10.07(14) | 29.48(5) | 25.59(6) | 25.55(7) | 34.55(2) | 9.40(15) | 22.77(8) | 20.63(10) | 13.12(13) | 19.57(11) | 22.59(9) | 34.12(3) | **36.65(1)** | 33.00(4) |
| cover | 0.91(16) | 71.12(9) | **93.61(1)** | 90.55(3) | 89.60(4) | 80.01(7) | 79.02(8) | 68.07(10) | 45.91(13) | 44.76(14) | 50.32(11) | 92.30(2) | 50.22(12) | 13.43(15) | 83.02(5) | 81.41(6) |
| donors | 15.61(16) | 97.85(5) | 17.11(15) | **99.87(1)** | 99.15(3) | 99.86(2) | 92.83(7) | 88.51(10) | 65.57(13) | 94.42(6) | 91.83(8) | 99.02(4) | 52.91(14) | 70.40(12) | 80.89(11) | 89.69(9) |
| fault | 49.56(12) | 50.97(9) | 53.82(4) | 53.54(5) | 54.77(2) | 50.95(10) | 53.32(6) | 47.65(14) | 46.80(15) | 52.37(8) | 42.40(16) | 50.00(11) | 48.39(13) | 53.93(3) | 53.23(7) | **56.50(1)** |
| fraud | 42.43(6) | 28.00(13) | 42.35(7) | 52.80(3) | **55.13(1)** | 54.48(2) | 38.57(10) | 0.14(15) | 21.23(14) | 41.61(8) | 31.53(12) | 48.90(4) | 48.58(5) | 0.14(15) | 36.75(11) | 39.79(9) |
| glass | 13.43(16) | 23.23(14) | 24.01(13) | 36.61(12) | 48.55(10) | 79.66(2) | 70.33(5) | 68.90(6) | 57.76(8) | 21.35(15) | 77.46(3) | 43.89(11) | 67.65(9) | 68.54(7) | 74.45(4) | **81.23(1)** |
| Hepatitis | 22.05(15) | 44.29(14) | 74.73(4) | 70.90(6) | 67.16(8) | 62.61(11) | 74.33(5) | 55.45(12) | 16.00(16) | 70.66(7) | 65.85(9) | 63.69(10) | 49.16(13) | **79.25(1)** | 77.57(2) | 75.35(3) |
| http | 73.48(13) | 77.70(12) | 93.99(10) | **100.00(1)** | **100.00(1)** | 99.78(7) | 96.68(8) | 61.84(14) | 0.37(16) | **100.00(1)** | **100.00(1)** | **100.00(1)** | 99.44(6) | 0.38(15) | **100.00(1)** | 93.81(11) |
| InternetAds | 46.23(7) | 46.77(5) | 54.16(2) | 46.09(8) | 41.80(10) | 46.66(6) | 54.11(3) | 19.63(15) | 45.08(9) | 27.35(14) | 28.45(13) | N/A(N/A) | 41.40(11) | 38.46(12) | 52.74(4) | **67.24(1)** |
| Ionosphere | 89.61(3) | 75.93(9) | 91.77(2) | 73.16(10) | 72.43(11) | 64.91(15) | 89.44(4) | 78.95(7) | 78.91(8) | 72.40(12) | 52.41(16) | 69.89(13) | 68.82(14) | 85.65(5) | 84.23(6) | **91.91(1)** |
| landsat | 18.65(16) | 61.79(4) | 32.54(14) | 42.02(13) | 44.52(11) | 54.25(7) | 66.48(2) | 28.70(15) | 49.87(10) | 49.94(9) | 42.49(12) | 63.58(3) | 52.28(8) | 58.99(5) | 57.87(6) | **67.36(1)** |
| letter | 16.05(8) | 19.29(6) | 15.50(10) | 21.63(3) | 20.08(5) | 17.75(7) | 32.96(2) | 7.12(15) | 6.41(16) | 7.50(14) | **38.10(1)** | 13.73(13) | 12.94(13) | 20.11(4) | 14.59(11) | 15.86(9) |
| Lymphography | 65.56(10) | 48.29(15) | 77.07(6) | 79.51(3) | 73.13(7) | 79.81(2) | 62.57(12) | 48.16(14) | 41.11(16) | 59.45(13) | 72.61(8) | 77.96(4) | 58.25(13) | 77.90(5) | 63.53(11) | **94.69(1)** |
| magic.gamma | 46.45(15) | 70.93(8) | 71.70(6) | 71.94(5) | 75.37(3) | 75.45(2) | 74.24(4) | 68.48(10) | 43.90(16) | **77.42(1)** | 57.12(14) | 69.98(9) | 64.44(12) | 67.02(11) | 64.42(13) | 71.10(7) |
| mammography | 14.95(16) | 49.12(8) | 56.56(3) | **62.12(1)** | 58.66(2) | 55.70(4) | 40.30(10) | 54.35(5) | 16.23(15) | 51.74(6) | 36.86(11) | 50.87(7) | 31.49(14) | 31.77(13) | 34.62(12) | 48.38(9) |
| mnist | 17.15(15) | 51.61(13) | 73.58(5) | **79.51(1)** | 75.34(4) | 63.18(9) | 75.45(3) | 11.02(16) | 52.33(12) | 61.71(10) | 38.39(14) | 67.50(7) | 58.05(11) | 63.50(8) | 68.15(6) | 79.49(2) |
| musk | 72.41(11) | 78.34(10) | 80.43(9) | **100.00(1)** | **100.00(1)** | **100.00(1)** | 97.87(6) | 67.72(12) | 45.98(12) | **100.00(1)** | 56.84(15) | 98.27(5) | 64.32(13) | 64.29(14) | 97.48(7) | 94.29(8) |
| optdigits | 3.24(16) | 58.85(11) | 98.15(4) | **99.53(1)** | 99.49(2) | 99.12(3) | 74.22(8) | 45.46(13) | 45.98(12) | 36.93(14) | 27.84(15) | 86.67(6) | 76.94(7) | 44.94(14) | 47.27(9) | 77.46(7) |
| PageBlocks | 40.76(15) | 71.06(3) | 69.40(7) | 64.88(10) | 70.08(4) | 73.71(2) | 69.26(8) | 49.36(13) | 36.93(16) | 69.71(6) | 51.49(12) | 64.04(11) | 46.43(14) | 65.42(9) | 69.99(5) | **77.23(1)** |
| pendigits | 2.86(16) | 75.95(10) | **96.17(1)** | 93.70(3) | 93.22(4) | 95.16(2) | 71.88(11) | 88.54(5) | 48.55(13) | 78.34(9) | 37.95(14) | 87.36(6) | 49.91(12) | 28.83(15) | 79.30(7) | 78.50(8) |
| Pima | 44.07(13) | 48.95(11) | 54.73(8) | **64.16(1)** | 61.63(2) | 48.35(12) | 56.16(7) | 59.80(3) | 36.39(15) | 27.21(16) | 41.58(14) | 52.97(9) | 52.55(10) | 58.51(4) | 57.16(6) | 58.49(5) |
| satellite | 64.72(15) | 79.24(9) | 80.04(8) | 82.20(3) | 82.04(4) | 80.13(7) | 82.43(2) | 75.43(11) | 51.19(16) | **82.44(1)** | 69.33(14) | 81.12(6) | 71.14(13) | 75.62(10) | 73.46(12) | 81.83(5) |
| satimage-2 | 43.73(14) | 82.89(7) | 90.44(4) | 91.71(2) | 91.23(3) | **93.64(1)** | 81.62(8) | 79.87(10) | 36.72(15) | 78.51(11) | 66.25(12) | 90.03(5) | 66.05(13) | 20.00(16) | 80.72(9) | 85.76(6) |
| shuttle | 48.60(16) | 97.27(4) | 97.39(3) | 96.44(6) | 96.79(5) | 96.16(8) | **98.51(1)** | 93.02(13) | 93.64(12) | 95.66(10) | 96.90(7) | 95.14(11) | 91.64(14) | 95.04(11) | 95.80(9) | 98.61(2) |
| skin | 22.33(16) | 94.25(7) | 52.32(15) | 70.09(12) | 67.14(13) | 88.12(9) | 96.46(4) | 87.93(10) | 97.48(3) | 80.51(11) | **99.06(1)** | 94.93(6) | 62.52(14) | 95.94(5) | 94.23(8) | 98.61(2) |
| smtp | 21.46(11) | 61.13(2) | 41.68(7) | 50.02(4) | **66.68(1)** | 17.07(13) | 50.02(3) | 0.04(15) | 17.73(12) | 38.92(8) | 11.71(14) | 33.94(9) | 50.01(5) | 0.04(15) | 50.01(5) | 28.13(10) |
| SpamBase | 40.85(16) | 59.54(14) | 76.52(9) | 85.00(3) | 86.26(2) | 78.65(7) | 84.15(4) | 64.79(12) | 61.85(13) | 83.16(5) | 52.00(15) | 82.53(6) | 71.29(11) | 77.88(8) | 75.78(10) | **88.53(1)** |
| speech | 1.61(16) | 2.71(12) | 6.16(4) | 6.61(3) | 6.72(2) | 5.87(5) | 4.35(9) | 1.63(15) | **6.86(1)** | 5.66(6) | 5.32(7) | 4.73(8) | 2.49(14) | 2.62(13) | 4.24(10) | 3.10(11) |
| Stamps | 30.08(14) | 23.23(16) | 72.09(4) | 70.21(6) | 75.41(2) | 57.38(9) | 68.45(7) | **75.44(1)** | 23.40(15) | 37.12(13) | 49.19(11) | 72.13(3) | 40.39(12) | 53.46(10) | 64.78(8) | 71.24(5) |
| thyroid | 53.94(11) | 55.16(10) | 89.64(3) | **90.84(1)** | 89.90(2) | 87.55(4) | 82.37(7) | 83.65(6) | 28.10(15) | 28.82(14) | 17.88(16) | 83.70(5) | 43.07(12) | 36.51(13) | 80.88(8) | 78.27(9) |
| vertebral | 10.63(16) | 20.53(14) | 26.66(13) | 34.40(8) | 32.69(11) | 42.67(4) | 34.07(10) | 40.97(6) | 34.32(9) | 18.91(15) | 38.86(7) | 43.70(2) | 26.74(12) | 42.64(5) | **49.73(1)** | 43.25(3) |
| vowels | 25.13(11) | 21.09(15) | 43.19(5) | 48.91(2) | **53.44(1)** | 42.84(6) | 36.08(7) | 24.46(12) | 21.71(14) | 2.04(16) | 43.68(4) | 35.62(8) | 26.08(10) | 29.38(9) | 24.13(13) | 43.80(3) |
| Waveform | 4.45(16) | 17.60(2) | 12.85(8) | 14.68(5) | 12.30(9) | 10.25(12) | 15.30(4) | 12.95(7) | 10.25(11) | 4.64(15) | 11.93(10) | **18.62(1)** | 8.02(13) | 7.72(14) | 14.03(6) | 16.83(3) |
| WBC | 41.59(14) | 57.93(9) | 78.08(5) | 81.25(2) | 80.25(4) | 63.23(6) | 46.53(13) | 33.00(15) | 15.80(16) | 47.04(12) | 54.21(11) | 81.20(3) | 59.58(7) | 58.91(8) | 56.07(10) | **84.41(1)** |
| WDBC | 59.54(11) | 39.38(13) | 91.29(5) | **98.21(1)** | 96.59(3) | 93.68(4) | 68.19(10) | 14.48(14) | 10.09(15) | 1.64(16) | 44.39(12) | 97.13(2) | 78.80(7) | 88.39(6) | 74.97(9) | 77.12(8) |
| Wilt | 4.32(16) | 19.38(9) | 8.15(12) | 8.60(11) | 32.59(5) | 30.99(6) | 23.12(8) | 25.08(7) | 8.04(13) | 4.76(15) | 80.39(9) | 98.23(5) | 51.74(12) | 78.46(10) | 36.56(3) | **33.56(1)** |
| wine | 14.98(14) | 37.92(13) | 99.15(3) | **100.00(1)** | **100.00(1)** | 98.44(4) | 78.93(5) | 88.68(7) | 8.11(15) | 4.76(16) | 80.39(9) | 98.23(5) | 51.74(12) | 78.46(10) | 82.05(8) | 93.48(6) |
| WPBC | 22.58(15) | 29.86(13) | 30.72(12) | 40.76(8) | 45.11(4) | 37.69(10) | 45.79(3) | 30.99(11) | 22.58(16) | 25.35(14) | **51.48(1)** | 42.56(7) | 39.91(9) | 45.02(5) | 47.26(2) | 44.02(6) |
| yeast | 33.03(16) | 36.31(13) | 34.39(14) | **46.72(1)** | 46.60(3) | 45.30(5) | 40.33(10) | 46.69(2) | 45.58(4) | 34.25(15) | 39.43(12) | 42.77(7) | 40.52(9) | 41.03(8) | 39.80(11) | 45.11(6) |
| CIFAR10 | 9.20(14) | 12.26(12) | 19.48(6) | **21.16(1)** | 20.79(3) | 20.18(5) | 20.99(2) | 5.20(16) | 13.12(10) | 15.32(9) | 12.47(11) | 9.65(13) | 7.28(15) | 15.73(8) | 20.75(4) | 18.09(7) |
| FashionMNIST | 20.85(14) | 43.56(8) | 52.54(7) | **63.79(1)** | 62.12(2) | 53.70(6) | 58.53(3) | 7.30(16) | 26.71(13) | 39.01(10) | 29.10(12) | 32.29(11) | 17.85(15) | 41.03(9) | 55.06(4) | 53.51(5) |
| MNIST-C | 23.82(14) | 50.40(10) | 46.81(7) | **79.16(1)** | 77.18(2) | 68.81(3) | 68.12(4) | 16.48(16) | 43.62(12) | 53.99(7) | 22.67(15) | 50.44(9) | 36.51(13) | 49.38(11) | 67.07(5) | 66.16(6) |
| MVTec-AD | 57.54(5) | 52.50(11) | 54.55(8) | 56.29(6) | 56.07(7) | 52.41(12) | 61.21(3) | 28.19(16) | 36.03(15) | 50.02(13) | 41.82(14) | 53.25(10) | 54.04(9) | 60.76(4) | 62.34(2) | **68.54(1)** |
| SVHN | 8.06(13) | 9.01(12) | 14.41(3) | **16.42(1)** | 14.37(4) | 13.21(6) | 5.04(16) | 12.47(7) | 9.40(11) | 10.32(10) | 7.89(14) | 6.26(15) | 12.36(8) | 13.58(5) | 14.75(2) | 12.22(9) |
| Agnews | 6.57(14) | 18.33(12) | 31.90(4) | 38.88(2) | **40.66(1)** | 31.13(5) | 25.59(6) | 5.33(16) | 22.31(8) | 38.12(3) | 18.92(11) | 16.69(13) | 6.02(15) | 20.91(10) | 24.50(7) | 21.28(9) |
| Amazon | 6.21(14) | 10.13(13) | 14.18(8) | 24.33(2) | **25.09(1)** | 20.95(4) | 15.10(6) | 5.16(16) | 16.34(5) | 21.22(3) | 10.40(11) | 10.35(12) | 5.21(15) | 11.86(10) | 14.95(7) | 13.77(9) |
| Imdb | 4.90(16) | 8.34(12) | 14.59(5) | **19.02(1)** | 18.50(2) | 17.09(3) | 12.45(9) | 5.30(15) | 13.91(7) | 14.56(6) | 12.68(8) | 7.37(13) | 5.42(14) | 9.44(11) | 12.38(10) | 15.03(4) |
| Yelp | 8.93(14) | 9.80(13) | 24.41(5) | 31.59(2) | 30.85(3) | 28.52(4) | 20.23(7) | 6.16(16) | 19.88(8) | **31.77(1)** | 15.36(9) | 15.28(10) | 6.44(15) | 15.04(11) | 21.52(6) | 15.03(12) |
| 20news | 7.17(13) | 12.76(9) | 6.79(15) | **17.88(1)** | 17.64(2) | 12.65(10) | 15.69(4) | 6.05(16) | 10.87(12) | 12.97(8) | 13.19(6) | 6.88(14) | 11.40(11) | 13.10(7) | 17.06(3) | 13.32(5) |

Table D10: AUCROC of 16 label-informed algorithms on 57 benchmark datasets, with labeled anomaly ratio $\gamma_l = 10\%$. We show the performance rank in parenthesis (lower the better), and mark the best performing method(s) in **bold**.

| Datasets | GANomaly | DeepSAD | REPEN | DevNet | PReNet | FEAWAD | XGBOD | NB | SVM | MLP | ResNet | FTTransformer | RF | LGB | XGB | CatB |
|---|---|---|---|---|---|---|---|---|---|---|---|---|---|---|---|---|
| ALOI | 56.59(7) | 56.63(6) | 54.54(8) | 49.55(13) | 49.50(14) | 53.11(9) | 72.10(1) | 47.78(15) | 47.05(16) | 51.34(10) | 49.79(12) | 50.89(11) | 60.67(2) | 57.98(4) | 58.95(3) | 56.85(5) |
| annthyroid | 82.83(11) | 87.41(9) | 82.05(14) | 83.31(10) | 82.78(12) | 93.07(7) | 98.21(3) | 82.54(13) | 40.43(16) | 80.53(15) | 90.88(8) | 98.67(2) | 93.72(6) | 97.39(5) | 97.56(4) | 98.85(1) |
| backdoor | 83.69(14) | 96.47(5) | 89.40(12) | 97.84(1) | 96.39(6) | 96.73(3) | 96.17(7) | 79.46(15) | 94.03(9) | 95.82(8) | 58.20(16) | 93.93(10) | 87.71(13) | 93.93(11) | 97.22(2) | 96.69(4) |
| breastw | 92.59(13) | 94.61(11) | 97.92(8) | 99.49(3) | 98.79(5) | 88.12(14) | 97.88(9) | 99.75(1) | 79.54(15) | 99.54(2) | 60.44(16) | 98.18(7) | 93.69(12) | 99.01(4) | 97.43(10) | 98.78(6) |
| campaign | 73.02(10) | 71.02(13) | 57.92(16) | 83.85(5) | 83.84(6) | 72.49(12) | 90.77(1) | 75.62(9) | 68.88(14) | 72.51(11) | 62.26(15) | 79.61(8) | 80.15(7) | 88.25(4) | 88.72(3) | 90.08(2) |
| cardio | 82.87(15) | 86.51(13) | 98.09(4) | 98.35(2) | 97.97(5) | 96.57(7) | 97.42(6) | 90.04(12) | 86.19(14) | 94.32(8) | N/A(N/A) | 98.27(3) | 91.53(11) | 93.39(10) | 94.30(9) | 98.46(1) |
| Cardiotocography | 55.12(16) | 86.65(10) | 91.80(4) | 93.15(2) | 93.33(1) | 85.38(13) | 89.77(7) | 90.85(6) | 83.19(14) | 91.02(5) | 69.36(15) | 86.43(11) | 85.64(12) | 87.23(8) | 87.03(9) | 91.85(3) |
| celeba | 72.16(13) | 80.00(11) | 57.33(16) | 93.69(2) | 93.65(3) | 88.66(9) | 92.90(5) | 88.89(8) | 79.76(12) | 94.03(1) | 62.02(15) | 93.03(4) | 71.10(14) | 90.03(6) | 89.17(7) | 87.29(10) |
| census | 60.45(16) | 71.90(12) | 69.07(13) | 84.88(5) | 79.66(6) | 79.20(7) | 88.53(1) | 62.38(14) | 79.12(9) | 79.02(10) | 61.42(15) | 76.73(11) | 79.14(8) | 85.87(4) | 86.07(3) | 87.91(2) |
| cover | 45.42(16) | 96.71(10) | 99.92(2) | 99.88(3) | 99.87(4) | 95.43(11) | 97.76(9) | 98.72(6) | 92.73(12) | 97.86(8) | 73.39(14) | 99.92(1) | 92.37(13) | 61.22(15) | 98.25(7) | 99.50(5) |
| donors | 69.68(16) | 99.99(3) | 82.79(15) | 99.99(1) | 99.95(5) | 99.98(4) | 99.83(7) | 99.42(11) | 99.72(8) | 99.84(6) | 99.43(10) | 99.99(2) | 90.19(14) | 99.01(13) | 99.28(12) | 99.72(9) |
| fault | 64.59(14) | 72.45(5) | 72.43(6) | 72.10(7) | 72.04(8) | 68.44(12) | 73.43(3) | 67.35(13) | 63.85(15) | 73.92(2) | 55.75(16) | 71.13(9) | 70.86(10) | 73.16(4) | 70.80(11) | 75.79(1) |
| fraud | 90.51(3) | 87.57(7) | 92.22(1) | 87.01(8) | 88.45(5) | 80.95(13) | 84.86(11) | 50.00(15) | 88.04(6) | 85.69(9) | 59.19(14) | 90.27(4) | 84.91(10) | 42.32(16) | 84.41(12) | 90.98(2) |
| glass | 67.84(16) | 87.31(12) | 88.12(10) | 90.86(8) | 95.77(3) | 89.62(9) | 97.44(2) | 87.85(11) | 86.33(13) | 82.77(14) | 93.27(7) | 95.06(4) | 79.60(15) | 94.50(5) | 94.49(6) | 99.15(1) |
| Hepatitis | 56.34(15) | 85.24(13) | 93.02(8) | 95.68(3) | 93.86(6) | 88.61(12) | 94.25(5) | 91.89(10) | 50.00(16) | 92.91(9) | 91.54(11) | 93.12(7) | 85.17(14) | 96.29(2) | 95.32(4) | 96.35(1) |
| http | 99.80(11) | 99.93(9) | 99.98(7) | 100.00(1) | 100.00(1) | 99.84(10) | 98.33(12) | 98.33(12) | 83.31(14) | 0.17(16) | 100.00(1) | 100.00(1) | 100.00(6) | 44.39(15) | 100.00(1) | 99.97(8) |
| InternetAds | 68.04(11) | 75.03(7) | 82.43(2) | 70.71(9) | 66.90(12) | 70.04(10) | 76.69(3) | 53.78(15) | 75.03(8) | 66.53(13) | 57.26(14) | N/A(N/A) | 75.62(6) | 76.23(5) | 76.41(4) | 84.26(1) |
| Ionosphere | 92.29(5) | 85.56(8) | 95.65(1) | 78.94(11) | 78.19(12) | 76.77(13) | 94.53(3) | 87.70(7) | 82.17(10) | 75.95(14) | 60.35(16) | 70.87(15) | 83.89(9) | 91.81(6) | 92.67(4) | 94.62(2) |
| landsat | 47.40(16) | 90.09(3) | 57.21(15) | 78.71(12) | 79.95(11) | 82.93(9) | 90.50(2) | 71.45(14) | 82.73(10) | 88.41(5) | 77.68(13) | 89.73(4) | 87.15(7) | 87.86(6) | 86.80(8) | 91.93(1) |
| letter | 69.72(10) | 74.65(3) | 73.25(4) | 71.95(7) | 72.77(6) | 68.32(12) | 87.32(1) | 57.29(13) | 56.00(14) | 49.01(16) | 77.99(2) | 52.18(15) | 69.54(11) | 71.15(9) | 71.56(8) | 73.23(5) |
| Lymphography | 96.93(6) | 88.16(13) | 99.60(2) | 94.67(7) | 97.94(4) | 93.02(9) | 89.37(12) | 80.08(15) | 50.00(16) | 90.60(11) | 90.96(10) | 97.13(5) | 85.98(14) | 98.69(3) | 93.40(8) | 99.72(1) |
| magic.gamma | 62.86(15) | 85.00(4) | 79.34(12) | 82.80(9) | 83.35(7) | 83.11(8) | 87.10(1) | 75.87(14) | 55.23(16) | 86.90(2) | 76.90(13) | 85.37(3) | 83.72(6) | 82.77(10) | 80.87(11) | 84.84(5) |
| mammography | 72.50(15) | 91.46(5) | 92.42(3) | 93.44(1) | 93.07(2) | 91.69(4) | 87.59(10) | 90.43(7) | 67.10(16) | 91.19(6) | 78.25(12) | 88.80(9) | 77.68(13) | 72.52(14) | 84.90(11) | 89.16(8) |
| mnist | 69.06(14) | 90.78(12) | 96.72(5) | 97.63(3) | 97.47(4) | 93.65(10) | 98.17(1) | 51.71(16) | 89.94(13) | 92.08(11) | 66.94(15) | 95.62(8) | 94.35(9) | 95.93(7) | 96.66(6) | 98.09(2) |
| musk | 99.47(9) | 98.38(11) | 99.28(10) | 100.00(1) | 100.00(1) | 100.00(1) | 99.99(6) | 93.10(12) | 57.33(16) | 100.00(1) | 80.53(15) | 99.95(8) | 90.79(13) | 88.83(14) | 99.98(7) | 100.00(1) |
| optdigits | 46.08(16) | 98.73(8) | 99.94(4) | 99.98(1) | 99.98(2) | 99.94(3) | 99.07(7) | 91.99(13) | 92.18(12) | 99.63(5) | 72.94(15) | 98.13(11) | 98.27(10) | 87.20(14) | 98.52(9) | 99.51(6) |
| PageBlocks | 80.20(15) | 94.45(5) | 93.81(6) | 86.77(13) | 89.75(10) | 93.45(7) | 96.07(4) | 90.08(9) | 72.80(16) | 92.57(8) | 83.56(14) | 89.57(11) | 89.30(12) | 96.47(2) | 96.42(3) | 97.77(1) |
| pendigits | 57.10(16) | 99.38(9) | 99.78(4) | 99.86(3) | 99.70(6) | 99.87(2) | 98.97(7) | 99.54(7) | 89.42(14) | 99.30(10) | 90.72(13) | 99.91(1) | 90.75(12) | 85.90(15) | 99.47(8) | 99.73(5) |
| Pima | 60.29(13) | 66.07(12) | 69.69(9) | 76.24(2) | 77.35(1) | 71.49(8) | 73.83(6) | 74.76(5) | 48.36(16) | 58.11(14) | 49.26(15) | 69.65(10) | 69.61(11) | 75.29(4) | 73.70(7) | 75.53(3) |
| satellite | 71.53(16) | 91.07(4) | 80.24(13) | 84.48(10) | 83.94(11) | 86.78(9) | 91.41(3) | 79.02(14) | 76.90(15) | 88.21(8) | 81.79(12) | 92.62(1) | 89.68(5) | 89.62(6) | 88.98(7) | 92.46(2) |
| satimage-2 | 96.54(10) | 98.00(7) | 99.46(1) | 98.33(6) | 98.59(4) | 96.75(9) | 98.40(5) | 94.00(11) | 71.95(15) | 96.89(8) | 82.68(14) | 98.88(3) | 93.53(12) | 61.69(16) | 89.33(13) | 99.11(2) |
| shuttle | 68.73(16) | 99.01(3) | 98.72(5) | 97.55(10) | 97.69(8) | 98.30(7) | 99.88(1) | 93.94(13) | 87.58(15) | 97.47(11) | 97.64(9) | 97.27(12) | 90.47(14) | 98.67(6) | 98.94(4) | 99.51(2) |
| skin | 52.26(16) | 99.64(3) | 90.29(14) | 95.70(11) | 95.30(12) | 98.22(10) | 99.58(4) | 94.81(13) | 99.53(5) | 99.14(8) | 99.78(2) | 99.24(6) | 89.54(15) | 99.16(7) | 99.13(9) | 99.84(1) |
| smtp | 51.69(14) | 82.74(3) | 76.47(8) | 75.99(9) | 75.24(11) | 75.69(10) | 79.87(5) | 50.00(15) | 62.98(12) | 86.40(1) | 52.83(13) | 79.07(6) | 83.32(2) | 43.01(16) | 81.12(4) | 78.85(7) |
| SpamBase | 54.01(16) | 80.28(13) | 87.14(9) | 91.35(6) | 92.96(3) | 83.93(11) | 94.64(1) | 83.50(12) | 69.45(14) | 91.92(4) | 63.20(15) | 91.84(5) | 84.55(10) | 91.16(7) | 88.85(8) | 94.32(2) |
| speech | 47.49(16) | 54.23(11) | 57.89(7) | 68.71(2) | 70.28(1) | 59.51(5) | 57.27(9) | 58.63(4) | 57.79(8) | 52.67(12) | 52.58(13) | 68.65(3) | 51.69(15) | 52.34(14) | 56.02(10) | 63.20(4) |
| Stamps | 72.38(15) | 85.97(11) | 96.92(2) | 95.37(8) | 96.40(5) | 87.25(9) | 96.81(3) | 84.57(12) | 87.03(10) | 83.58(13) | 60.44(16) | 96.29(6) | 76.48(14) | 95.72(7) | 96.50(4) | 97.55(1) |
| thyroid | 92.77(13) | 97.10(10) | 99.70(2) | 99.73(1) | 99.64(3) | 99.46(6) | 99.84(8) | 99.34(7) | 64.11(16) | 90.95(14) | 74.21(15) | 99.61(4) | 92.97(12) | 95.65(11) | 97.97(9) | 99.50(5) |
| vertebral | 37.26(16) | 68.06(11) | 56.21(15) | 76.86(9) | 78.26(7) | 77.24(8) | 85.83(2) | 83.34(4) | 75.85(10) | 67.41(12) | 62.55(14) | 80.82(6) | 65.78(13) | 82.36(5) | 87.31(1) | 84.73(3) |
| vowels | 78.52(15) | 82.64(12) | 92.68(5) | 95.69(3) | 96.99(1) | 93.00(4) | 95.74(2) | 88.52(9) | 79.63(14) | 15.44(16) | 81.05(13) | 84.07(11) | 84.80(10) | 88.89(8) | 91.02(7) | 92.56(6) |
| Waveform | 52.95(16) | 74.37(13) | 83.65(5) | 88.17(3) | 88.84(2) | 77.71(12) | 83.77(4) | 81.92(7) | 80.17(10) | 62.68(15) | 78.54(11) | 80.45(9) | 70.38(14) | 81.63(8) | 82.52(6) | 99.82(1) |
| WBC | 93.57(9) | 95.88(7) | 98.28(4) | 98.91(3) | 99.03(2) | 97.37(6) | 87.60(12) | 86.84(13) | 58.91(16) | 90.09(11) | 70.28(15) | 97.64(5) | 86.29(14) | 95.80(8) | 92.67(10) | 99.37(1) |
| WDBC | 97.56(8) | 95.51(12) | 100.00(1) | 100.00(1) | 100.00(1) | 99.82(6) | 97.50(9) | 80.99(14) | 61.79(15) | 33.04(16) | 96.76(10) | 99.92(5) | 96.15(11) | 98.40(7) | 94.81(13) | 99.94(4) |
| Wilt | 47.50(16) | 92.05(5) | 52.98(15) | 68.14(13) | 69.06(12) | 87.92(7) | 90.42(5) | 78.61(10) | 79.70(9) | 65.92(14) | 92.41(2) | 70.15(11) | 82.05(8) | 89.13(6) | 90.59(4) | 95.66(1) |
| wine | 66.14(15) | 89.42(12) | 99.87(5) | 100.00(1) | 100.00(1) | 100.00(1) | 87.18(13) | 99.89(4) | 45.77(16) | 99.20(8) | 94.20(9) | 99.63(7) | 77.52(14) | 92.98(10) | 91.78(11) | 99.69(6) |
| WPBC | 47.62(15) | 64.81(12) | 63.93(13) | 77.76(1) | 77.43(3) | 73.90(8) | 75.00(6) | 62.06(14) | 40.07(16) | 75.14(5) | 72.46(9) | 66.25(11) | 69.37(10) | 74.17(7) | 77.52(2) | 77.39(4) |
| yeast | 48.58(15) | 59.53(11) | 47.25(16) | 64.68(4) | 65.55(3) | 63.30(6) | 59.91(8) | 65.71(2) | 53.51(14) | 53.81(13) | 59.68(9) | 64.61(5) | 57.64(12) | 59.54(10) | 66.23(1) | 66.23(1) |
| CIFAR10 | 60.39(13) | 66.50(11) | 73.80(6) | 74.61(4) | 69.87(8) | 72.96(7) | 76.66(1) | 52.15(16) | 69.04(9) | 66.56(10) | 58.00(15) | 62.54(12) | 58.46(14) | 74.22(5) | 76.08(3) | 76.16(2) |
| FashionMNIST | 79.83(10) | 88.62(9) | 91.59(2) | 91.22(5) | 90.17(6) | 89.01(8) | 92.78(1) | 53.77(16) | 77.62(12) | 77.94(11) | 62.03(15) | 76.32(13) | 69.20(14) | 90.00(7) | 91.42(3) | 91.26(4) |
| MNIST-C | 75.07(14) | 88.25(8) | 82.62(13) | 94.00(1) | 92.98(2) | 91.41(5) | 92.46(4) | 66.12(16) | 86.50(12) | 87.15(10) | 68.68(15) | 87.05(11) | 88.19(9) | 89.83(7) | 92.55(3) | 90.57(6) |
| MVTec-AD | 74.25(8) | 73.68(9) | 73.33(10) | 79.21(5) | 76.77(6) | 70.24(12) | 82.61(2) | 56.36(16) | 68.41(14) | 71.60(11) | 61.40(15) | 70.14(13) | 74.67(7) | 80.55(4) | 82.28(3) | 83.92(1) |
| SVHN | 57.61(14) | 63.11(11) | 70.74(2) | 72.18(1) | 67.64(7) | 70.35(4) | 70.22(5) | 50.09(16) | 63.73(10) | 62.61(12) | 58.15(13) | 63.78(9) | 54.21(15) | 68.17(6) | 70.65(3) | 67.64(8) |
| Agnews | 58.77(14) | 74.98(12) | 84.09(3) | 88.18(2) | 87.59(4) | 83.22(7) | 84.99(5) | 58.23(15) | 81.78(8) | 89.63(1) | 65.01(13) | 76.00(11) | 79.86(10) | 84.41(6) | 81.34(9) | |
| Amazon | 57.80(14) | 59.22(13) | 77.98(5) | 84.97(1) | 84.73(2) | 80.43(4) | 76.04(7) | 50.19(16) | 74.17(9) | 84.54(3) | 63.87(12) | 65.55(11) | 54.48(15) | 73.29(10) | 77.02(6) | 74.21(8) |
| Imdb | 49.39(16) | 63.87(12) | 74.92(7) | 80.80(2) | 80.00(3) | 76.59(5) | 74.83(8) | 55.83(14) | 77.11(4) | 81.21(1) | 65.74(11) | 63.29(13) | 51.96(15) | 71.31(10) | 74.20(9) | 74.95(6) |
| Yelp | 66.76(13) | 68.39(12) | 87.34(4) | 89.48(2) | 89.03(3) | 85.52(5) | 82.23(7) | 53.76(16) | 81.09(8) | 90.32(1) | 66.67(14) | 74.45(11) | 55.79(15) | 77.55(10) | 82.63(6) | 80.41(9) |
| 20news | 57.18(14) | 64.91(9) | 57.15(15) | 72.45(1) | 69.96(5) | 61.60(11) | 70.05(4) | 56.95(16) | 65.03(8) | 64.79(10) | 58.24(13) | 65.84(7) | 61.39(12) | 72.18(2) | 71.70(3) | 67.63(6) |

Table D11: AUCPR of 16 label-informed algorithms on 57 benchmark datasets, with labeled anomaly ratio $\gamma_l = 10\%$. We show the performance rank in parenthesis (lower the better), and mark the best performing method(s) in **bold**.

| Datasets | GANomaly | DeepSAD | REPEN | DevNet | PReNet | FEAWAD | XGBOD | NB | SVM | MLP | ResNet | FTTransformer | RF | LGB | XGB | CatB |
|---|---|---|---|---|---|---|---|---|---|---|---|---|---|---|---|---|
| ALOI | 3.92(12) | 6.36(5) | 5.12(9) | 4.65(10) | 5.30(7) | 3.70(14) | 8.15(1) | 3.61(16) | 3.64(15) | 5.19(8) | 3.91(13) | 4.27(11) | 7.50(2) | 7.46(3) | 7.03(4) | 6.01(6) |
| annthyroid | 45.28(11) | 49.95(9) | 40.81(14) | 46.43(10) | 44.91(12) | 63.56(7) | 80.54(2) | 43.62(13) | 8.28(16) | 40.68(15) | 54.35(8) | 82.40(1) | 69.19(6) | 78.78(5) | 79.32(4) | 79.71(3) |
| backdoor | 18.34(15) | 63.57(10) | 29.10(14) | 90.30(1) | 89.60(2) | 77.61(4) | 65.85(8) | 11.04(16) | 64.49(9) | 79.78(3) | 33.28(13) | 73.24(6) | 62.71(11) | 66.54(7) | 74.83(5) | 57.75(12) |
| breastw | 86.00(14) | 88.30(13) | 96.01(7) | 99.11(2) | 98.54(4) | 90.05(11) | 94.64(9) | 99.52(1) | 80.51(15) | 98.93(3) | 66.65(16) | 94.55(10) | 89.89(12) | 97.84(5) | 95.06(8) | 96.69(6) |
| campaign | 30.76(10) | 28.61(13) | 15.30(16) | 42.04(6) | 44.12(5) | 29.45(12) | 51.43(1) | 33.02(9) | 27.11(14) | 29.75(11) | 23.69(15) | 33.84(8) | 39.61(7) | 47.00(4) | 47.80(3) | 50.86(2) |
| cardio | 39.46(15) | 55.53(13) | 86.47(6) | 88.58(1) | 87.31(5) | 88.78(2) | 67.56(13) | 76.96(5) | 61.68(14) | 75.61(6) | N/A(N/A) | 88.49(4) | 70.36(10) | 69.15(10) | 84.63(6) | 76.75(6) |
| Cardiotocography | 34.00(16) | 69.51(9) | 81.18(2) | 80.91(3) | 82.19(1) | 67.56(13) | 75.08(7) | 76.96(5) | 61.68(14) | 78.52(4) | 51.64(15) | 68.10(12) | 68.36(11) | 69.15(10) | 80.41(13) | 76.75(6) |
| celeba | 5.18(15) | 11.74(12) | 3.21(16) | 27.64(2) | 29.25(1) | 22.76(4) | 21.97(6) | 15.36(9) | 13.79(11) | 24.15(3) | 7.80(14) | 22.71(5) | 10.53(13) | 19.68(7) | 18.11(8) | 15.00(10) |
| census | 8.05(16) | 18.49(12) | 10.07(14) | 33.30(5) | 30.90(6) | 25.57(8) | 38.83(1) | 9.88(15) | 22.68(10) | 23.04(9) | 16.25(13) | 18.85(11) | 28.87(7) | 33.92(4) | 35.81(3) | 36.95(2) |
| cover | 0.91(16) | 78.76(9) | 95.40(1) | 94.46(3) | 94.54(2) | 86.61(5) | 77.94(10) | 85.05(6) | 47.46(14) | 61.23(12) | 49.11(13) | 89.62(4) | 68.20(11) | 26.45(15) | 82.39(8) | 82.77(7) |
| donors | 11.66(16) | 99.87(3) | 17.07(15) | 99.93(1) | 99.80(8) | 99.44(4) | 96.53(9) | 88.56(13) | 99.36(5) | 99.08(6) | 98.90(7) | 99.91(2) | 77.45(14) | 89.43(12) | 90.26(11) | 94.96(10) |
| fault | 49.99(14) | 55.76(11) | 60.23(2) | 57.99(7) | 60.11(4) | 56.01(9) | 58.07(6) | 51.01(13) | 49.61(15) | 60.22(3) | 47.40(16) | 55.75(12) | 55.95(10) | 58.97(5) | 56.39(8) | 62.80(1) |
| fraud | 42.43(10) | 36.13(13) | 42.44(9) | 52.18(5) | 54.51(3) | 54.04(4) | 44.21(8) | 0.14(15) | 36.84(12) | 41.70(11) | 34.30(14) | 54.71(2) | 51.50(6) | 0.14(15) | 41.77(7) | 55.80(1) |
| glass | 13.43(16) | 26.48(14) | 28.90(13) | 34.40(12) | 48.87(11) | 75.56(5) | 80.48(2) | 59.5(9) | 65.98(8) | 21.40(15) | 79.36(3) | 70.07(6) | 86.44(1) | 66.27(7) | 67.87(6) | 86.36(4) |
| Hepatitis | 22.06(15) | 60.03(14) | 86.51(9) | 87.10(6) | 86.04(10) | 84.08(11) | 89.19(4) | 78.23(12) | 16.00(16) | 87.57(5) | 87.07(7) | 86.68(8) | 75.54(13) | 90.66(2) | 89.44(3) | 90.87(1) |
| http | 73.48(13) | 84.95(11) | 93.99(9) | 100.00(1) | 100.00(1) | 83.33(12) | 96.68(7) | 61.84(14) | 0.37(16) | 100.00(1) | 100.00(1) | 100.00(1) | 99.44(6) | 0.38(15) | 100.00(1) | 93.81(10) |
| InternetAds | 46.53(10) | 49.14(8) | 59.70(2) | 54.43(5) | 50.85(7) | 46.45(11) | 55.75(4) | 20.92(15) | 52.42(6) | 35.09(14) | 35.18(13) | N/A(N/A) | 41.36(12) | 48.29(9) | 56.21(3) | 74.41(1) |
| Ionosphere | 89.90(5) | 81.12(8) | 93.27(1) | 81.04(9) | 80.04(11) | 76.02(13) | 91.88(2) | 81.79(7) | 75.02(14) | 76.17(12) | 64.80(16) | 70.61(15) | 80.05(10) | 88.68(6) | 90.76(4) | 90.99(3) |
| landsat | 19.21(16) | 72.80(2) | 33.37(14) | 46.38(13) | 50.90(12) | 57.88(9) | 72.01(4) | 29.36(15) | 55.46(10) | 67.48(5) | 54.03(11) | 72.56(3) | 65.28(8) | 65.33(6) | 65.32(7) | 74.43(1) |
| letter | 16.19(11) | 19.15(3) | 18.23(8) | 18.87(5) | 18.93(4) | 17.07(10) | 32.69(1) | 10.75(14) | 11.53(13) | 8.42(16) | 28.21(2) | 9.12(15) | 16.49(12) | 18.74(6) | 18.68(7) | 18.66(7) |
| Lymphography | 65.60(13) | 63.73(14) | 95.68(1) | 90.91(4) | 91.06(3) | 87.65(6) | 78.56(9) | 54.94(15) | 4.11(16) | 78.42(10) | 73.53(11) | 90.14(5) | 73.24(12) | 86.82(7) | 84.07(8) | 94.88(2) |
| magic.gamma | 46.82(16) | 76.40(3) | 74.06(7) | 72.03(9) | 75.31(6) | 72.94(8) | 78.93(2) | 68.31(12) | 48.44(15) | 81.29(1) | 64.17(14) | 75.71(5) | 72.02(10) | 72.02(11) | 67.83(13) | 76.31(4) |
| mammography | 14.82(16) | 56.62(4) | 57.08(3) | 62.03(1) | 57.98(2) | 56.32(5) | 49.72(9) | 51.30(8) | 20.60(15) | 55.84(7) | 41.35(11) | 56.14(6) | 41.05(12) | 30.67(14) | 35.08(13) | 47.59(10) |
| mnist | 17.30(15) | 65.21(12) | 84.81(3) | 84.03(4) | 82.38(5) | 73.36(10) | 85.18(2) | 9.54(16) | 59.76(13) | 70.51(11) | 38.11(14) | 75.45(8) | 74.66(9) | 77.40(7) | 78.47(6) | 86.68(1) |
| musk | 90.72(10) | 86.97(11) | 95.74(9) | 100.00(1) | 100.00(1) | 100.00(1) | 99.78(6) | 86.64(12) | 43.36(16) | 100.00(1) | 45.40(5) | 98.95(8) | 81.82(13) | 75.66(14) | 99.57(7) | 100.00(1) |
| optdigits | 3.30(16) | 86.34(8) | 98.79(3) | 99.53(1) | 99.35(2) | 98.66(4) | 84.17(9) | 67.98(12) | 49.02(14) | 95.40(5) | 36.11(15) | 89.70(6) | 83.02(10) | 60.67(13) | 80.34(11) | 49.47(7) |
| PageBlocks | 46.01(16) | 76.02(4) | 75.26(5) | 63.51(12) | 67.46(9) | 74.33(8) | 74.75(6) | 53.66(14) | 48.77(15) | 74.44(7) | 57.44(13) | 64.71(11) | 65.96(10) | 76.45(2) | 76.24(3) | 83.24(1) |
| pendigits | 3.01(16) | 89.92(9) | 97.13(1) | 94.03(4) | 91.18(6) | 96.90(2) | 79.01(11) | 90.68(8) | 54.94(15) | 91.69(5) | 75.74(12) | 95.09(3) | 68.81(13) | 59.83(14) | 85.18(10) | 90.96(7) |
| Pima | 44.26(13) | 51.28(12) | 51.98(10) | 59.80(7) | 61.98(3) | 59.09(8) | 61.14(5) | 63.14(1) | 38.38(16) | 43.45(14) | 43.17(15) | 51.49(11) | 58.52(9) | 62.96(2) | 60.78(6) | 61.53(4) |
| satellite | 66.00(15) | 83.83(5) | 78.79(11) | 82.42(6) | 82.19(7) | 77.50(12) | 84.83(3) | 75.40(13) | 64.60(16) | 83.93(4) | 72.68(14) | 86.85(1) | 81.40(8) | 80.76(9) | 78.63(12) | 86.06(2) |
| satimage-2 | 44.03(14) | 86.42(6) | 92.29(2) | 91.58(3) | 88.70(4) | 92.40(1) | 81.92(8) | 81.70(9) | 43.77(15) | 79.01(10) | 63.28(13) | 84.63(7) | 78.55(11) | 34.28(16) | 74.28(12) | 86.50(5) |
| shuttle | 44.74(16) | 97.91(3) | 97.06(4) | 95.68(9) | 96.56(7) | 96.76(5) | 98.95(1) | 92.84(13) | 52.96(15) | 95.83(11) | 96.46(8) | 93.91(12) | 92.81(14) | 97.26(6) | 96.68(6) | 98.99(1) |
| skin | 22.12(16) | 96.30(7) | 51.54(15) | 70.15(13) | 67.49(14) | 82.06(11) | 98.57(2) | 88.10(10) | 96.70(5) | 93.85(9) | 98.19(3) | 95.43(8) | 80.40(12) | 97.26(4) | 96.68(6) | 98.99(1) |
| smtp | 21.46(11) | 61.13(2) | 41.68(7) | 50.02(4) | 66.68(1) | 16.99(13) | 50.02(3) | 0.04(15) | 17.73(12) | 38.92(8) | 11.71(14) | 33.94(9) | 50.01(5) | 0.04(15) | 50.01(5) | 28.13(10) |
| SpamBase | 41.10(16) | 70.26(12) | 81.13(10) | 86.07(7) | 88.58(4) | 82.31(9) | 92.34(1) | 68.50(13) | 61.94(14) | 88.77(3) | 60.42(15) | 87.87(5) | 79.60(11) | 86.29(6) | 83.82(8) | 91.77(2) |
| speech | 1.61(16) | 2.76(15) | 9.01(6) | 12.45(1) | 11.36(2) | 9.48(4) | 3.50(11) | 7.26(8) | 10.30(3) | 4.15(9) | 9.32(5) | 7.58(7) | 3.95(10) | 2.76(14) | 3.23(12) | 3.03(13) |
| Stamps | 30.56(16) | 36.56(15) | 79.44(3) | 72.94(9) | 78.47(4) | 75.58(7) | 78.11(5) | 74.59(8) | 56.74(11) | 57.50(12) | 41.58(14) | 80.52(2) | 50.16(13) | 71.26(10) | 75.95(6) | 82.50(1) |
| thyroid | 54.98(13) | 64.25(12) | 91.72(2) | 82.71(9) | 91.21(3) | 89.96(5) | 87.71(8) | 88.14(7) | 35.73(16) | 54.65(14) | 42.59(15) | 89.31(6) | 71.75(10) | 51.59(10) | 90.51(4) | 84.66(9) |
| vertebral | 10.64(16) | 26.56(13) | 25.42(14) | 31.74(12) | 33.93(11) | 42.06(7) | 61.20(2) | 39.68(9) | 42.70(6) | 19.15(15) | 39.21(10) | 48.90(5) | 40.44(8) | 58.40(3) | 62.91(1) | 58.26(4) |
| vowels | 25.76(14) | 24.81(15) | 54.26(8) | 73.18(2) | 82.43(1) | 55.35(7) | 63.65(3) | 36.38(13) | 39.77(12) | 2.06(16) | 56.06(6) | 58.06(5) | 44.87(11) | 46.55(10) | 52.18(9) | 59.25(4) |
| Waveform | 4.41(16) | 21.36(3) | 12.78(12) | 15.41(11) | 16.90(8) | 15.51(10) | 20.23(5) | 12.05(13) | 16.97(7) | 5.16(15) | 20.67(4) | 21.79(2) | 9.44(14) | 16.87(9) | 17.85(6) | 28.47(1) |
| WBC | 42.29(15) | 61.20(13) | 81.81(5) | 84.41(3) | 85.07(2) | 80.89(6) | 83.06(12) | 68.11(10) | 15.80(16) | 63.69(11) | 56.27(14) | 83.56(4) | 73.11(9) | 78.36(7) | 74.73(8) | 91.59(1) |
| WDBC | 59.89(12) | 50.37(14) | 100.00(1) | 100.00(1) | 100.00(1) | 96.10(6) | 86.90(9) | 57.52(13) | 10.09(16) | 22.66(15) | 89.63(7) | 98.06(5) | 88.09(8) | 86.37(10) | 83.37(11) | 98.40(4) |
| Wilt | 5.13(16) | 38.75(8) | 6.18(15) | 8.20(13) | 8.53(12) | 41.27(6) | 51.14(5) | 33.14(9) | 28.18(10) | 8.07(14) | 35.42(7) | 88.8(1) | 39.92(7) | 51.92(4) | 53.55(3) | 69.07(1) |
| wine | 15.25(15) | 50.17(14) | 98.70(5) | 100.00(1) | 100.00(1) | 100.00(1) | 83.10(11) | 98.99(4) | 12.20(16) | 93.48(9) | 89.10(9) | 98.45(6) | 57.15(13) | 82.75(12) | 85.88(10) | 97.51(7) |
| WPBC | 22.58(15) | 36.16(12) | 34.87(13) | 53.26(8) | 52.31(10) | 54.67(7) | 62.57(5) | 33.95(14) | 21.45(16) | 46.96(11) | 63.57(3) | 52.49(9) | 56.33(6) | 62.91(4) | 64.87(2) | 66.47(1) |
| yeast | 33.14(16) | 40.27(12) | 35.44(14) | 44.13(8) | 45.60(5) | 46.60(3) | 45.33(6) | 47.21(2) | 38.38(13) | 34.17(15) | 45.07(7) | 46.35(4) | 42.91(11) | 43.07(10) | 43.59(9) | 49.30(1) |
| CIFAR10 | 9.18(15) | 14.16(11) | 21.74(6) | 23.84(2) | 22.31(5) | 24.50(1) | 22.40(4) | 5.43(16) | 15.83(10) | 16.41(9) | 12.87(13) | 13.76(12) | 9.95(14) | 20.25(8) | 22.49(3) | 20.37(7) |
| FashionMNIST | 20.90(15) | 51.76(9) | 64.56(5) | 71.07(1) | 68.93(2) | 65.01(4) | 65.03(3) | 7.28(16) | 39.02(11) | 42.65(10) | 27.52(14) | 37.51(12) | 27.57(13) | 57.48(6) | 60.52(7) | 63.93(6) |
| MNIST-C | 25.93(15) | 58.52(11) | 42.26(13) | 82.22(1) | 80.70(2) | 76.31(3) | 74.22(5) | 21.01(16) | 52.52(12) | 62.14(9) | 32.58(14) | 61.33(10) | 63.21(8) | 66.40(7) | 74.58(4) | 72.81(6) |
| MVTec-AD | 57.66(12) | 59.33(8) | 55.40(13) | 65.28(5) | 63.69(6) | 59.27(9) | 69.78(2) | 31.48(16) | 47.98(15) | 59.24(10) | 50.16(14) | 58.78(11) | 60.67(7) | 68.09(4) | 69.61(3) | 72.27(1) |
| SVHN | 8.00(14) | 10.49(13) | 17.94(4) | 20.22(1) | 18.04(3) | 18.69(2) | 16.02(6) | 5.09(16) | 13.43(9) | 12.13(10) | 10.97(12) | 11.23(11) | 6.64(15) | 14.80(7) | 17.08(5) | 14.08(8) |
| Agnews | 6.51(15) | 26.27(10) | 42.93(4) | 46.31(3) | 46.49(2) | 39.46(5) | 33.28(6) | 6.46(16) | 32.96(7) | 53.54(1) | 22.19(13) | 24.50(12) | 7.65(14) | 24.50(11) | 31.99(8) | 31.72(9) |
| Amazon | 6.11(15) | 9.25(13) | 21.31(5) | 26.26(3) | 26.42(2) | 25.49(4) | 15.06(8) | 5.02(16) | 15.85(7) | 26.63(1) | 14.13(11) | 11.45(12) | 6.27(14) | 14.39(10) | 16.19(6) | 14.93(9) |
| Imdb | 4.95(16) | 11.65(12) | 18.69(6) | 23.21(2) | 20.68(4) | 23.16(3) | 17.45(8) | 5.69(14) | 18.63(7) | 24.11(1) | 14.50(10) | 7.82(13) | 5.64(15) | 13.75(11) | 16.20(9) | 18.89(5) |
| Yelp | 8.60(14) | 13.11(13) | 33.06(5) | 37.57(2) | 37.33(3) | 34.94(4) | 23.36(8) | 5.54(16) | 26.55(6) | 43.95(1) | 19.56(10) | 16.36(12) | 6.66(15) | 18.76(11) | 25.52(7) | 19.68(9) |
| 20news | 7.16(14) | 12.48(11) | 6.90(15) | 20.58(2) | 21.50(1) | 14.49(8) | 15.29(6) | 6.06(16) | 17.00(4) | 19.03(3) | 14.73(7) | 10.85(13) | 11.73(12) | 13.42(10) | 15.35(5) | 14.12(9) |

Table D12: AUCROC of 16 label-informed algorithms on 57 benchmark datasets, with labeled anomaly ratio $\gamma_l = 25\%$. We show the performance rank in parenthesis (lower the better), and mark the best performing method(s) in **bold**.

| Datasets | GANomaly | DeepSAD | REPEN | DevNet | PReNet | FEAWAD | XGBOD | NB | SVM | MLP | ResNet | FTTransformer | RF | LGB | XGB | CatB |
|---|---|---|---|---|---|---|---|---|---|---|---|---|---|---|---|---|
| ALOI | 55.25(8) | 59.29(6) | 53.09(11) | 51.65(13) | 51.44(14) | 57.54(7) | **75.96(1)** | 49.27(15) | 53.86(9) | 52.16(12) | 47.74(16) | 53.45(10) | 71.83(2) | 65.23(3) | 64.77(4) | 62.78(5) |
| annthyroid | 80.96(15) | 93.06(9) | 82.78(12) | 82.39(13) | 82.11(14) | 96.95(7) | 98.78(4) | 82.95(11) | 50.73(16) | 91.29(10) | 96.08(8) | **99.05(1)** | 98.58(6) | 98.70(5) | 98.83(3) | 98.97(2) |
| backdoor | 82.96(15) | 97.07(8) | 89.49(13) | 97.49(6) | 96.45(9) | 98.00(4) | 97.20(7) | 76.77(16) | 94.51(12) | 95.98(10) | 86.10(14) | 94.98(11) | 97.92(5) | 98.40(2) | 98.39(3) | **98.92(1)** |
| breastw | 93.13(15) | 97.44(12) | 98.72(8) | 99.67(2) | 99.27(5) | 93.83(14) | 99.14(6) | **99.75(1)** | 94.57(13) | 99.42(3) | 80.40(16) | 98.45(10) | 98.23(11) | 99.30(4) | 98.72(9) | 99.11(7) |
| campaign | 66.57(15) | 73.12(14) | 57.92(16) | 85.83(7) | 85.95(6) | 76.42(11) | **92.20(1)** | 79.15(9) | 73.59(13) | 77.29(10) | 75.45(12) | 84.62(8) | 87.85(5) | 90.41(3) | 89.16(4) | 91.23(2) |
| cardio | 83.68(15) | 93.98(12) | 98.65(2) | 98.02(7) | 98.26(5) | 97.11(8) | 98.46(3) | 93.74(13) | 91.48(14) | 95.21(11) | N/A(N/A) | 98.24(6) | **98.73(1)** | 95.73(10) | 96.02(9) | 98.37(4) |
| Cardiotocography | 56.41(16) | 92.52(12) | 94.43(6) | 94.91(3) | 94.92(2) | 90.40(13) | 93.81(7) | 93.20(10) | 87.28(14) | 94.80(4) | 82.53(15) | 94.48(5) | 93.34(9) | 93.50(8) | 92.89(11) | **95.13(1)** |
| celeba | 76.18(15) | 89.65(11) | 57.39(16) | **95.41(1)** | 95.17(3) | 90.65(9) | 95.15(4) | 89.68(10) | 85.43(12) | 94.46(6) | 77.31(14) | 95.20(2) | 81.65(13) | 94.51(5) | 91.86(8) | 92.40(7) |
| census | 46.38(16) | 78.77(12) | 69.41(14) | 90.33(3) | 88.07(6) | 82.45(10) | **91.07(1)** | 70.09(13) | 83.98(9) | 84.61(8) | 61.69(15) | 82.04(11) | 87.47(7) | 89.95(4) | 89.72(5) | 90.85(2) |
| cover | 42.98(16) | 99.86(5) | **99.94(1)** | 99.92(3) | 99.94(2) | 99.83(7) | 99.68(12) | 99.75(11) | 98.97(13) | 99.77(9) | 96.56(14) | 99.81(8) | 99.85(6) | 90.19(15) | 99.76(10) | 99.91(4) |
| donors | 51.09(16) | **100.00(1)** | 82.84(15) | 99.99(2) | 99.94(4) | 99.98(3) | 99.93(6) | 99.42(14) | 99.57(12) | 99.89(8) | 99.43(13) | 99.92(7) | 99.58(11) | 99.88(9) | 99.82(10) | 99.93(5) |
| fault | 65.21(16) | 75.58(8) | 74.74(10) | 74.10(11) | 76.06(7) | 71.88(12) | 77.71(3) | 71.15(13) | 68.87(14) | 77.91(2) | 68.79(15) | 75.57(9) | 77.62(4) | 76.40(5) | 76.18(6) | **80.69(1)** |
| fraud | 91.06(3) | 90.21(6) | **92.38(1)** | 89.91(7) | 90.47(4) | 79.07(13) | 90.36(5) | 78.35(14) | 77.80(15) | 85.91(9) | 81.48(12) | 83.22(11) | 88.20(8) | 44.72(16) | 85.24(10) | 91.20(2) |
| glass | 68.06(16) | 91.67(12) | 86.42(14) | 89.49(13) | 92.13(11) | 94.24(10) | 98.41(5) | 96.01(8) | 95.21(9) | 84.32(15) | 97.33(7) | 99.22(3) | 97.91(6) | 98.93(4) | 99.22(2) | **99.62(1)** |
| Hepatitis | 58.17(16) | 95.32(14) | 98.57(6) | 98.11(7) | 97.06(9) | 96.81(11) | 98.75(4) | 96.54(12) | 64.77(15) | 98.79(3) | 97.63(8) | 96.16(13) | 97.03(10) | 98.64(5) | 99.22(2) | **99.48(1)** |
| http | 99.80(11) | 99.99(7) | 99.98(8) | **100.00(1)** | **100.00(1)** | 99.96(10) | 98.33(12) | 98.33(12) | 83.31(14) | 0.30(16) | 99.99(6) | **100.00(1)** | 100.00(5) | 43.69(15) | **100.00(1)** | 99.97(9) |
| InternetAds | 68.43(14) | 81.07(9) | 90.28(2) | 84.65(7) | 82.06(8) | 80.87(10) | 86.62(6) | 65.24(15) | 80.36(11) | 79.98(12) | 74.64(13) | N/A(N/A) | 87.18(3) | 86.81(5) | 86.87(4) | **92.63(1)** |
| Ionosphere | 92.52(8) | 93.60(7) | 97.60(4) | 86.74(15) | 86.76(14) | 91.69(10) | **98.75(1)** | 89.69(13) | 89.94(12) | 90.19(11) | 76.30(16) | 92.36(9) | 94.90(6) | 98.11(2) | 97.50(5) | 97.67(3) |
| landsat | 50.88(16) | 92.87(3) | 64.20(15) | 79.93(12) | 79.65(13) | 82.58(11) | 92.89(2) | 71.95(14) | 85.00(10) | 89.93(8) | 85.93(9) | 92.31(5) | 91.88(6) | 92.35(4) | 91.73(7) | **94.27(1)** |
| letter | 69.93(15) | 78.65(10) | 82.72(6) | 79.81(9) | 81.41(7) | 73.01(14) | 90.33(2) | 61.36(16) | 73.85(13) | 75.00(12) | 86.46(3) | 77.73(11) | 85.75(4) | 85.20(5) | 80.78(8) | **90.38(1)** |
| Lymphography | 97.21(11) | 97.27(10) | 99.80(2) | 99.37(6) | 99.72(4) | 94.91(12) | 92.01(15) | **99.85(1)** | 77.53(16) | 99.34(7) | 93.39(13) | 99.58(5) | 93.18(14) | 99.64(8) | 98.61(9) | 99.79(3) |
| magic.gamma | 50.32(16) | 88.02(5) | 81.14(13) | 82.70(11) | 82.91(10) | 84.14(9) | 89.24(2) | 78.10(14) | 65.59(15) | **89.45(1)** | 81.61(12) | 86.72(7) | 88.40(4) | 87.64(6) | 84.57(8) | 88.85(3) |
| mammography | 75.96(15) | **94.46(1)** | 92.81(5) | 92.78(6) | 92.65(7) | 94.14(3) | 91.55(9) | 91.28(10) | 60.48(16) | 92.42(8) | 85.48(13) | 94.35(2) | 86.77(11) | 76.75(14) | 86.33(12) | 93.57(4) |
| mnist | 68.76(16) | 96.65(12) | 98.89(4) | 98.77(5) | 98.93(3) | 97.42(10) | 99.12(2) | 79.55(15) | 95.00(13) | 97.13(11) | 83.36(14) | 97.86(9) | 98.48(7) | 98.66(6) | 98.46(8) | **99.20(1)** |
| musk | 98.46(13) | 100.00(10) | **100.00(1)** | **100.00(1)** | **100.00(1)** | **100.00(1)** | **100.00(1)** | 94.83(14) | 79.66(16) | **100.00(1)** | 93.32(15) | **100.00(1)** | 99.85(12) | 99.98(11) | **100.00(1)** | **100.00(1)** |
| optdigits | 46.49(16) | 99.87(8) | 99.99(2) | 99.99(3) | **99.99(1)** | 99.97(4) | 99.77(9) | 97.64(13) | 96.62(14) | 99.94(5) | 90.00(15) | 99.93(6) | 99.72(10) | 99.62(12) | 99.72(11) | 99.89(7) |
| PageBlocks | 64.80(16) | 96.18(5) | 94.42(10) | 88.59(14) | 90.15(13) | 95.31(7) | 97.65(3) | 91.16(12) | 85.81(15) | 93.45(11) | 94.69(9) | 96.09(6) | 95.05(8) | 97.60(4) | 97.77(2) | **98.36(1)** |
| pendigits | 60.12(16) | **99.93(1)** | 99.52(10) | 99.82(5) | 99.82(6) | 99.21(12) | 99.77(7) | 99.66(8) | 98.49(15) | 99.58(9) | 98.72(14) | 99.86(3) | 99.46(11) | 98.99(13) | 99.82(6) | 99.91(2) |
| Pima | 61.46(16) | 73.27(12) | 71.75(13) | 81.52(4) | 81.06(5) | 76.66(11) | 82.55(3) | 80.49(8) | 63.21(15) | 77.00(10) | 68.15(14) | 77.36(9) | 80.87(6) | **83.38(1)** | 80.78(7) | 82.56(2) |
| satellite | 74.94(16) | 93.57(5) | 79.92(14) | 84.67(11) | 84.17(12) | 85.89(9) | 93.91(2) | 78.73(15) | 80.17(13) | 90.88(8) | 85.44(10) | 93.81(3) | 93.64(4) | 93.09(6) | 92.74(7) | **94.37(1)** |
| satimage-2 | 96.63(13) | 99.38(5) | 99.53(2) | **99.60(1)** | 99.51(3) | 97.81(10) | 99.42(4) | 98.23(7) | 88.83(16) | 97.75(11) | 94.44(14) | 98.05(8) | 96.75(12) | 90.12(15) | 98.00(9) | 99.90(6) |
| shuttle | 79.04(16) | 98.85(6) | 98.94(5) | 97.57(12) | 97.63(11) | 98.68(7) | **99.90(1)** | 97.26(14) | 88.54(15) | 97.57(13) | 97.66(10) | 98.35(8) | 98.15(9) | 99.77(2) | 99.54(3) | 99.54(4) |
| skin | 52.75(16) | 99.88(2) | 89.70(15) | 95.99(12) | 95.37(13) | 98.92(10) | 99.84(4) | 93.96(14) | 99.66(8) | 99.70(6) | 99.77(5) | 99.84(3) | 98.04(11) | 99.66(9) | 99.69(7) | **99.91(1)** |
| smtp | 51.69(14) | 82.74(3) | 76.47(8) | 75.99(9) | 75.24(11) | 75.41(10) | 79.87(5) | 50.00(15) | 62.98(12) | **86.40(1)** | 52.83(13) | 79.07(6) | 83.32(2) | 43.01(16) | 81.12(4) | 78.85(7) |
| SpamBase | 55.09(16) | 90.66(10) | 89.77(11) | 91.69(9) | 93.51(5) | 89.51(12) | 96.80(2) | 82.32(14) | 79.28(15) | 95.30(3) | 84.45(13) | 93.80(6) | 92.01(8) | 95.26(4) | 93.97(5) | **96.85(1)** |
| speech | 47.58(16) | 56.35(14) | 63.59(8) | 71.53(2) | **72.80(1)** | 61.59(9) | 67.92(5) | 63.13(10) | 69.73(4) | 56.81(12) | 56.77(13) | 70.47(3) | 52.11(15) | 64.18(7) | 64.45(6) | 61.82(11) |
| Stamps | 72.82(16) | 90.68(13) | 98.27(8) | 98.26(9) | 98.74(3) | 98.35(6) | 98.21(10) | 89.03(14) | 93.55(12) | 98.28(7) | 87.39(15) | 98.59(5) | 94.96(11) | 98.92(2) | 98.68(4) | **99.00(1)** |
| thyroid | 92.88(16) | 98.52(11) | 99.48(6) | **99.82(1)** | 99.75(3) | 98.59(10) | 99.53(5) | 99.58(4) | 94.05(15) | 99.19(7) | 96.49(14) | 99.15(8) | 97.34(13) | 98.39(12) | 99.13(9) | 99.76(2) |
| vertebral | 38.43(16) | 79.25(13) | 78.61(14) | 80.04(12) | 81.04(11) | 88.98(6) | 93.81(2) | 86.29(8) | 83.22(10) | 68.14(15) | 85.33(9) | 92.07(5) | 86.46(7) | 92.69(4) | 92.99(3) | **94.47(1)** |
| vowels | 78.64(15) | 90.26(11) | 96.92(4) | 97.65(2) | **98.33(1)** | 96.20(7) | 95.90(9) | 95.90(8) | 88.96(12) | 19.32(16) | 87.91(14) | 96.87(5) | 88.04(13) | 96.82(6) | 96.87(6) | 97.04(3) |
| Waveform | 53.30(16) | 80.11(14) | **91.65(1)** | 91.35(2) | 91.04(3) | 84.40(13) | 90.07(5) | 87.06(9) | 86.50(10) | 84.98(12) | 75.47(15) | 85.19(11) | 87.23(8) | 89.35(6) | 88.01(7) | 90.53(4) |
| WBC | 94.03(12) | 94.85(10) | 97.74(5) | 98.66(3) | 99.03(2) | 86.66(15) | 93.10(13) | 98.14(4) | 95.53(9) | 91.41(14) | 77.28(16) | 96.51(7) | 94.32(11) | 96.70(6) | 96.40(8) | **99.35(1)** |
| WDBC | 97.53(13) | 98.77(11) | **100.00(1)** | **100.00(1)** | **100.00(1)** | 94.82(14) | 98.28(12) | 93.55(15) | 93.71(14) | 99.92(7) | 92.99(16) | 99.97(6) | 99.89(8) | 99.72(9) | 99.96(10) | **100.00(1)** |
| Wilt | 36.73(16) | 96.83(2) | 62.64(15) | 67.97(13) | 68.73(12) | 88.56(8) | 94.96(5) | 80.31(10) | 83.22(9) | 66.33(14) | 95.63(3) | 74.33(11) | 93.84(6) | 93.66(7) | 94.98(4) | **97.59(1)** |
| wine | 66.97(15) | 97.51(13) | **100.00(1)** | **100.00(1)** | **100.00(1)** | 94.24(10) | 99.39(9) | 96.01(8) | 80.23(15) | 99.87(8) | 98.59(11) | **100.00(1)** | 94.77(14) | 98.71(10) | 98.47(12) | 99.99(7) |
| WPBC | 47.93(16) | 80.12(10) | 73.53(14) | 79.68(11) | 80.45(8) | 78.50(12) | 87.98(3) | 67.98(15) | 77.10(13) | 80.32(9) | **90.04(1)** | 84.27(7) | 85.18(6) | 86.86(5) | 87.90(4) | 89.28(2) |
| yeast | 48.40(15) | 65.31(10) | 47.77(16) | 66.29(8) | 66.76(4) | 67.70(3) | 66.39(6) | 64.52(12) | 55.68(14) | 64.68(11) | 66.33(7) | 70.07(2) | 63.25(13) | 66.78(5) | 65.98(9) | **72.44(1)** |
| CIFAR10 | 60.91(14) | 70.82(11) | 80.37(6) | 82.32(2) | 76.80(8) | 77.06(7) | 81.58(3) | 53.08(16) | 76.43(9) | 73.54(10) | 59.59(15) | 64.30(12) | 62.01(13) | 80.67(5) | 81.27(4) | **83.26(1)** |
| FashionMNIST | 80.05(14) | 92.31(9) | 95.42(2) | **95.72(1)** | 94.21(7) | 93.89(8) | 95.23(3) | 61.06(16) | 85.88(12) | 86.95(11) | 73.16(15) | 88.19(10) | 81.58(13) | 94.93(5) | 95.07(4) | 94.29(6) |
| MNIST-C | 74.64(15) | 91.50(11) | 82.62(13) | **95.31(1)** | 94.88(2) | 94.03(6) | 94.48(3) | 70.72(16) | 91.67(10) | 92.31(8) | 79.21(14) | 89.13(12) | 92.28(9) | 94.23(4) | 94.16(5) | 93.72(7) |
| MVTec-AD | 74.51(14) | 83.54(9) | 73.42(15) | 86.17(6) | 83.93(7) | 80.52(10) | 89.12(2) | 58.26(16) | 79.11(12) | 79.73(11) | 74.68(13) | 83.71(8) | 86.29(5) | 88.26(4) | 88.94(3) | **89.35(1)** |
| SVHN | 57.55(15) | 68.91(11) | 78.02(2) | **79.40(1)** | 76.37(3) | 74.18(8) | 75.96(5) | 51.11(16) | 73.14(9) | 69.50(10) | 62.83(13) | 64.13(12) | 60.57(14) | 74.70(7) | 76.21(4) | 75.12(6) |
| Agnews | 61.44(16) | 86.32(11) | 91.00(4) | 91.20(2) | 91.16(3) | 89.20(8) | 89.81(5) | 68.43(14) | 88.88(9) | **93.41(1)** | 82.48(12) | 82.25(13) | 64.12(15) | 88.11(10) | 89.48(6) | 89.72(6) |
| Amazon | 57.16(14) | 69.07(13) | 84.30(4) | 87.13(2) | 86.48(3) | 83.44(5) | 82.27(7) | 53.06(16) | 80.02(9) | **89.52(1)** | 70.41(12) | 72.21(11) | 57.03(15) | 79.07(10) | 82.33(6) | 81.85(8) |
| Imdb | 49.25(16) | 70.46(13) | 79.86(9) | 82.26(3) | 81.45(4) | 82.36(2) | 81.41(5) | 55.29(15) | 80.89(6) | **85.52(1)** | 77.27(11) | 71.30(12) | 57.68(14) | 78.09(10) | 80.52(8) | 80.63(7) |
| Yelp | 68.20(14) | 74.28(13) | 90.23(2) | 90.19(3) | 90.07(4) | 89.33(5) | 87.39(6) | 56.13(16) | 84.87(10) | **93.15(1)** | 76.85(12) | 83.94(11) | 62.07(15) | 85.29(9) | 86.93(7) | 86.58(8) |
| 20news | 57.80(15) | 70.58(9) | 58.48(14) | 77.92(2) | 77.59(3) | 70.10(10) | 73.52(7) | 51.95(16) | 71.51(8) | **79.34(1)** | 65.87(12) | 64.62(13) | 68.89(11) | 74.86(6) | 75.83(4) | 75.01(5) |

Table D13: AUCPR of 16 label-informed algorithms on 57 benchmark datasets, with labeled anomaly ratio $\gamma_l = 25\%$. We show the performance rank in parenthesis (lower the better), and mark the best performing method(s) in **bold**.

| Datasets | GANomaly | DeepSAD | REPEN | DevNet | PReNet | FEAWAD | XGBOD | NB | SVM | MLP | ResNet | FTTransformer | RF | LGB | XGB | CatB |
|---|---|---|---|---|---|---|---|---|---|---|---|---|---|---|---|---|
| ALOI | 3.83(16) | 6.13(6) | 4.54(9) | 4.27(11) | 4.19(12) | 4.71(8) | 10.55(2) | 3.97(14) | 5.67(7) | 4.30(10) | 3.92(15) | 4.07(13) | **13.19(1)** | 9.32(3) | 7.79(5) | 8.97(4) |
| annthyroid | 41.61(15) | 62.35(9) | 42.15(14) | 44.72(12) | 45.01(11) | 69.16(8) | 80.46(6) | 44.24(13) | 16.63(16) | 58.00(10) | 70.09(7) | **83.26(1)** | 81.17(5) | 82.87(2) | 82.02(4) | 82.03(3) |
| backdoor | 8.12(15) | 68.46(13) | 29.46(14) | **90.47(1)** | 90.23(2) | 74.19(10) | 71.48(12) | 6.55(16) | 73.80(11) | 76.47(8) | 74.24(9) | 78.55(7) | 87.34(4) | 82.65(6) | 85.95(5) | 87.85(3) |
| breastw | 86.88(15) | 93.23(13) | 97.22(8) | 99.38(2) | 98.96(4) | 94.72(12) | 97.90(7) | **99.53(1)** | 92.66(14) | 99.00(3) | 82.92(16) | 95.91(11) | 96.94(10) | 98.46(5) | 96.98(9) | 97.96(6) |
| campaign | 26.41(15) | 32.04(14) | 15.31(16) | 46.25(6) | 47.11(5) | 32.96(12) | **55.33(1)** | 36.83(9) | 32.30(13) | 34.32(10) | 33.18(11) | 38.30(8) | 48.60(4) | 49.83(3) | 45.65(7) | 52.73(2) |
| cardio | 41.03(15) | 75.75(13) | **93.38(1)** | 87.72(7) | 88.74(4) | 88.74(3) | 88.19(6) | 77.51(12) | 58.40(14) | 82.56(9) | N/A(N/A) | 88.20(5) | 87.29(8) | 80.24(11) | 81.58(10) | 84.49(4) |
| Cardiotocography | 35.04(16) | 80.55(10) | 84.14(5) | 84.58(2) | 84.51(3) | 75.01(13) | 82.53(6) | 80.64(8) | 64.33(15) | **85.69(1)** | 64.76(14) | 78.34(12) | 81.11(7) | 80.61(9) | 79.73(11) | 84.49(4) |
| celeba | 12.94(14) | 22.37(10) | 3.27(16) | **32.54(1)** | 31.44(2) | 24.20(7) | 26.80(5) | 12.71(15) | 19.71(11) | 24.90(6) | 14.40(13) | 28.00(3) | 17.38(12) | 27.35(4) | 23.36(9) | 23.57(8) |
| census | 5.95(16) | 27.78(11) | 10.21(15) | 42.01(4) | 40.89(6) | 32.77(8) | **48.18(1)** | 11.78(14) | 30.02(10) | 30.51(9) | 19.24(13) | 24.46(12) | 40.35(7) | 41.22(5) | 42.39(3) | 47.11(2) |
| cover | 0.88(16) | 95.03(4) | 96.10(2) | **96.43(1)** | 96.02(3) | 92.97(5) | 88.39(10) | 89.60(7) | 69.50(14) | 87.63(12) | 73.81(13) | 88.34(11) | 88.86(9) | 52.38(15) | 90.87(6) | 88.93(8) |
| donors | 7.76(16) | **99.98(1)** | 17.10(15) | 99.92(2) | 99.06(7) | 99.48(5) | 98.69(9) | 88.56(14) | 96.75(13) | 99.70(4) | 99.25(6) | 99.72(3) | 97.08(11) | 97.95(10) | 96.94(12) | 98.98(8) |
| fault | 51.09(16) | 59.72(9) | 61.67(8) | 58.73(11) | 63.50(5) | 57.74(12) | 64.20(2) | 53.90(15) | 56.45(13) | 63.77(3) | 56.14(14) | 58.76(10) | 63.48(6) | 63.00(7) | 63.73(4) | **68.90(1)** |
| fraud | 42.37(12) | 42.47(10) | 42.30(13) | 57.37(4) | 56.27(5) | 52.16(5) | 43.86(9) | 19.42(15) | 27.54(14) | 42.39(11) | 45.09(8) | 57.69(2) | **60.68(1)** | 0.14(16) | 48.75(7) | 50.16(6) |
| glass | 16.11(16) | 33.13(12) | 32.36(13) | 25.37(15) | 41.41(11) | 85.77(7) | 79.67(8) | 58.78(10) | 62.17(9) | 27.07(14) | 90.75(3) | **93.46(1)** | 86.17(6) | 90.07(5) | 90.09(4) | 93.40(2) |
| Hepatitis | 23.13(16) | 86.83(13) | 96.59(6) | 93.43(10) | 91.14(11) | 92.59(12) | 98.15(3) | 85.19(14) | 38.73(15) | 97.25(5) | 95.97(7) | **100.00(1)** | 94.34(9) | 95.10(8) | 98.12(4) | 98.64(2) |
| http | 73.48(13) | 97.78(6) | 93.99(11) | **100.00(1)** | **100.00(1)** | 92.59(12) | 96.68(8) | 96.68(6) | 61.84(14) | 0.37(16) | 97.78(6) | **100.00(1)** | 99.44(5) | 0.38(15) | **100.00(1)** | 94.12(10) |
| InternetAds | 47.46(14) | 62.73(10) | 38.38(2) | 73.76(5) | 70.30(7) | 65.19(9) | 74.36(4) | 31.65(15) | 67.22(8) | 56.00(12) | 55.94(13) | N/A(N/A) | 58.90(11) | 71.58(6) | 75.66(3) | **86.38(1)** |
| Ionosphere | 90.04(10) | 90.60(8) | 96.36(4) | 87.54(13) | 87.62(12) | 92.65(7) | **98.44(1)** | 85.56(14) | 85.01(15) | 89.59(11) | 81.10(16) | 90.19(9) | 93.59(6) | 97.25(2) | 96.76(3) | 96.31(5) |
| landsat | 21.30(16) | 79.94(2) | 42.87(14) | 55.40(11) | 54.06(12) | 53.20(13) | 78.71(3) | 29.27(15) | 61.42(10) | 73.23(8) | 64.93(9) | 77.96(4) | 77.32(5) | 77.05(6) | 74.79(7) | **82.24(1)** |
| letter | 16.29(15) | 23.38(14) | 31.84(8) | 25.85(12) | 30.43(9) | 27.08(11) | 41.96(3) | 10.55(16) | 25.61(13) | 27.57(10) | **49.91(1)** | 31.91(7) | 38.81(5) | 38.82(4) | 47.06(2) | 38.82(6) |
| Lymphography | 69.89(15) | 84.89(13) | **97.12(1)** | 94.33(7) | 96.34(3) | 90.60(10) | 83.08(14) | 70.51(15) | 59.75(16) | 65.52(12) | 85.55(12) | 94.90(6) | 86.09(11) | 92.24(9) | 93.20(8) | 96.58(2) |
| magic.gamma | 38.47(16) | 80.56(4) | 74.45(10) | 71.68(12) | 73.66(11) | 75.85(8) | 82.82(2) | 70.57(13) | 59.77(15) | **84.83(1)** | 69.09(14) | 79.23(7) | 80.40(5) | 79.57(6) | 74.86(9) | 82.59(3) |
| mammography | 16.04(16) | 63.05(2) | 60.85(3) | 59.72(7) | 58.75(8) | 57.10(10) | 60.09(5) | 51.35(12) | 24.97(15) | 59.89(6) | 55.05(11) | **65.96(1)** | 58.15(9) | 39.60(14) | 49.29(13) | 60.61(4) |
| mnist | 17.10(16) | 85.35(10) | 92.54(9) | 89.17(8) | 91.05(4) | 83.11(12) | 92.81(2) | 23.97(15) | 73.26(13) | 85.35(9) | 63.21(14) | 84.26(11) | 89.24(7) | 90.14(5) | 89.81(6) | **93.53(1)** |
| musk | 87.42(14) | 99.88(10) | **100.00(1)** | **100.00(1)** | **100.00(1)** | **100.00(1)** | **100.00(1)** | 89.98(13) | 59.57(16) | **100.00(1)** | 79.45(15) | **100.00(1)** | 97.62(12) | 94.50(13) | 92.87(12) | 97.10(8) |
| optdigits | 3.37(16) | 97.96(7) | **99.73(1)** | 99.66(3) | 99.71(2) | 99.48(4) | 93.88(9) | 60.59(15) | 66.11(14) | 98.54(5) | 71.06(13) | 98.31(6) | 94.50(8) | 92.87(10) | 93.35(11) | 97.10(8) |
| PageBlocks | 36.36(16) | 80.81(6) | 75.58(11) | 67.03(13) | 68.65(12) | 77.69(9) | 81.84(4) | 53.40(15) | 62.54(14) | 77.26(10) | 79.62(7) | 81.53(5) | 78.99(8) | 82.44(3) | 82.89(2) | **86.14(1)** |
| pendigits | 3.45(16) | **98.39(1)** | 96.77(3) | 94.12(11) | 94.48(10) | 96.71(4) | 96.39(5) | 94.79(9) | 67.08(15) | 94.92(8) | 92.40(13) | 95.84(6) | 93.38(12) | 91.64(14) | 95.36(7) | 98.15(2) |
| Pima | 45.05(16) | 55.40(13) | 54.60(14) | 68.77(6) | 68.44(8) | 62.43(11) | 73.06(3) | 68.73(7) | 51.25(15) | 62.82(9) | 61.11(12) | 62.66(10) | 72.25(4) | **75.44(1)** | 71.59(5) | 73.17(2) |
| satellite | 66.01(16) | 87.57(4) | 77.65(12) | 82.63(9) | 82.35(10) | 78.68(11) | **89.43(1)** | 75.52(13) | 71.57(15) | 86.60(7) | 73.74(14) | 87.17(5) | 88.38(3) | 86.97(6) | 85.78(8) | 88.97(2) |
| satimage-2 | 44.59(16) | 94.84(3) | 94.91(2) | **95.37(1)** | 93.85(5) | 94.40(4) | 90.17(9) | 91.72(8) | 63.46(15) | 86.06(12) | 75.74(13) | 93.77(6) | 88.86(10) | 75.14(14) | 92.05(7) | 86.79(11) |
| shuttle | 37.64(16) | 97.96(5) | 97.26(6) | 96.41(11) | 96.30(11) | 98.11(4) | 99.43(2) | 95.41(14) | 65.10(15) | 95.96(12) | 95.71(13) | 96.47(8) | 96.46(10) | 96.68(4) | 98.62(5) | **99.50(1)** |
| skin | 22.00(16) | 98.61(6) | 50.32(15) | 72.12(13) | 71.95(14) | 90.38(11) | 99.43(2) | 81.67(12) | 97.29(9) | 98.24(8) | 98.39(7) | 98.76(3) | 95.13(10) | 98.62(4) | 98.62(5) | **99.50(1)** |
| smtp | 21.46(11) | 61.13(2) | 41.68(7) | 50.02(4) | **66.68(1)** | 17.05(13) | 50.02(3) | 0.04(15) | 17.73(12) | 38.92(8) | 11.71(14) | 33.94(9) | 50.01(5) | 0.04(15) | 50.01(5) | 28.13(10) |
| SpamBase | 41.92(16) | 83.84(12) | 84.01(11) | 86.35(10) | 89.22(7) | 86.42(9) | 66.33(15) | 66.33(15) | 70.79(14) | 92.43(3) | 78.70(13) | 88.82(8) | 89.38(6) | 92.40(4) | 90.44(5) | **95.50(1)** |
| speech | 1.62(16) | 3.81(14) | **20.75(1)** | 12.56(4) | 12.46(5) | 8.67(7) | 5.53(9) | 10.63(6) | 17.59(2) | 4.96(10) | 14.79(3) | 7.09(8) | 4.25(12) | 4.91(11) | 3.85(13) | 3.39(15) |
| Stamps | 31.04(16) | 56.52(15) | 81.96(9) | 76.90(13) | 81.92(10) | 82.21(8) | 87.87(5) | 82.29(7) | 62.11(14) | 85.36(6) | 80.40(11) | 91.24(3) | 78.03(12) | 91.29(2) | 89.53(4) | **92.59(1)** |
| thyroid | 54.20(16) | 83.70(13) | 84.26(12) | 94.03(2) | 92.38(4) | 91.44(6) | 90.64(7) | 90.64(8) | 61.71(15) | 88.48(9) | 78.82(14) | **94.73(1)** | 87.41(11) | 87.44(10) | 93.85(3) | 92.00(5) |
| vertebral | 10.84(16) | 39.90(14) | 43.61(11) | 42.27(13) | 42.30(12) | 55.50(8) | 84.67(2) | 49.93(9) | 49.61(10) | 19.76(15) | 73.51(6) | 67.18(7) | 76.71(5) | 82.82(4) | 83.51(3) | **85.53(1)** |
| vowels | 26.01(15) | 44.38(14) | 83.42(4) | 85.04(3) | **87.29(1)** | 77.38(6) | 69.89(8) | 66.13(10) | 61.97(12) | 2.16(16) | 77.68(5) | 85.21(2) | 53.71(13) | 69.04(9) | 64.99(11) | 75.03(7) |
| Waveform | 4.45(16) | 28.20(7) | 19.63(13) | 21.64(11) | 21.48(12) | 23.31(10) | 33.21(3) | 17.84(15) | **36.90(1)** | 19.28(14) | 24.56(9) | 28.70(6) | 25.72(8) | 30.35(4) | 29.80(5) | 35.17(2) |
| WBC | 42.75(16) | 64.57(14) | 80.40(10) | 86.30(3) | 87.76(2) | 75.07(11) | 80.64(9) | 83.80(5) | 57.40(15) | 69.26(13) | 70.07(12) | 84.64(4) | 83.51(7) | 83.30(8) | 83.61(6) | **90.25(1)** |
| WDBC | 59.41(16) | 75.75(14) | **100.00(1)** | **100.00(1)** | **100.00(1)** | **100.00(1)** | 92.74(10) | 91.33(12) | 65.47(15) | 97.73(7) | 84.18(13) | 99.22(6) | 96.07(9) | 97.37(8) | 91.84(11) | **100.00(1)** |
| Wilt | 4.04(16) | 67.79(7) | 8.32(13) | 8.11(15) | 8.40(12) | 37.53(8) | 77.68(2) | 37.08(9) | 32.53(10) | 8.15(14) | 69.38(5) | 10.65(11) | 68.56(6) | 69.07(5) | 69.19(4) | **79.52(1)** |
| wine | 16.12(16) | 73.82(14) | **100.00(1)** | **100.00(1)** | **100.00(1)** | **100.00(1)** | 97.04(10) | 44.06(15) | 94.98(8) | 98.39(9) | **100.00(1)** | 90.01(13) | 96.55(11) | 96.43(12) | 99.94(7) | 99.94(7) |
| WPBC | 22.73(16) | 56.46(12) | 54.11(13) | 59.75(10) | 59.76(9) | 57.91(11) | 62.82(3) | 43.33(15) | 53.72(14) | 64.61(8) | 83.56(2) | 77.41(7) | 79.13(6) | 82.34(5) | 82.43(4) | **85.03(1)** |
| yeast | 33.12(16) | 45.38(13) | 36.21(15) | 47.55(10) | 48.22(9) | 51.89(5) | 52.37(3) | 45.70(12) | 39.58(14) | 45.90(11) | 52.05(4) | 52.98(2) | 49.27(8) | 50.75(7) | 50.83(6) | **58.35(1)** |
| CIFAR10 | 9.28(15) | 19.13(11) | 30.37(2) | **31.07(1)** | 30.10(3) | 26.75(8) | 29.24(5) | 5.56(16) | 21.44(9) | 20.27(10) | 15.78(12) | 14.25(13) | 11.45(14) | 27.61(7) | 27.64(6) | 29.62(4) |
| FashionMNIST | 21.17(15) | 62.87(9) | 77.05(3) | **79.97(1)** | 77.72(2) | 70.89(8) | 73.09(5) | 13.46(16) | 50.32(11) | 56.68(10) | 45.81(14) | 49.73(12) | 46.53(13) | 72.12(7) | 73.06(6) | 75.42(4) |
| MNIST-C | 24.86(15) | 72.18(10) | 41.72(14) | **84.70(1)** | 84.19(2) | 80.97(3) | 80.04(4) | 24.71(16) | 65.85(12) | 75.69(8) | 53.85(13) | 69.89(11) | 74.13(9) | 78.47(7) | 79.89(5) | 79.46(6) |
| MVTec-AD | 57.98(14) | 73.33(8) | 55.46(15) | 77.29(5) | 74.61(7) | 72.21(10) | 81.28(2) | 52.75(16) | 61.10(13) | 71.39(11) | 66.66(12) | 73.22(9) | 77.21(6) | 80.00(4) | 81.28(3) | **82.36(1)** |
| SVHN | 7.93(15) | 16.07(12) | 24.51(3) | **27.06(1)** | 26.68(2) | 22.50(4) | 19.72(7) | 5.19(16) | 20.64(6) | 18.74(10) | 16.64(11) | 12.76(13) | 9.35(14) | 20.28(8) | 20.80(5) | 20.51(7) |
| Agnews | 7.28(16) | 52.97(6) | 53.35(5) | 54.60(4) | 55.92(3) | 57.65(2) | 45.38(9) | 8.56(15) | 46.99(8) | **64.87(1)** | 42.92(11) | 38.17(13) | 11.38(14) | 40.86(12) | 44.30(10) | 47.01(7) |
| Amazon | 6.04(15) | 16.95(12) | 27.75(4) | 30.39(2) | 29.65(3) | 25.79(5) | 22.95(7) | 5.83(16) | 20.66(9) | **37.26(1)** | 17.21(11) | 13.80(13) | 6.90(14) | 19.19(10) | 22.68(8) | 23.89(6) |
| Imdb | 4.83(16) | 18.16(12) | 21.82(10) | 23.88(3) | 23.47(5) | 27.31(2) | 22.36(9) | 5.82(15) | 23.31(6) | **33.11(1)** | 23.81(4) | 13.84(13) | 6.63(14) | 20.19(11) | 22.47(8) | 23.06(7) |
| Yelp | 9.08(14) | 23.77(13) | 42.56(2) | 41.01(4) | 42.21(3) | 39.34(5) | 35.10(6) | 5.86(16) | 31.82(9) | **53.80(1)** | 29.04(11) | 27.98(12) | 8.84(15) | 30.68(10) | 33.74(8) | 34.80(7) |
| 20news | 7.28(14) | 18.37(11) | 7.20(15) | 28.30(3) | 30.92(2) | 26.57(4) | 22.22(6) | 5.26(16) | 20.84(9) | **34.04(1)** | 21.36(8) | 14.13(13) | 15.73(12) | 21.60(7) | 23.32(5) | 20.01(10) |

Table D14: AUCROC of 16 label-informed algorithms on 57 benchmark datasets, with labeled anomaly ratio $\gamma_l = 50\%$. We show the performance rank in parenthesis (lower the better), and mark the best performing method(s) in **bold**.

| Datasets | GANomaly | DeepSAD | REPEN | DevNet | PReNet | FEAWAD | XGBOD | NB | SVM | MLP | ResNet | FTTransformer | RF | LGB | XGB | CatB |
|---|---|---|---|---|---|---|---|---|---|---|---|---|---|---|---|---|
| ALOI | 54.52(11) | 64.14(6) | 53.63(12) | 52.88(14) | 52.95(13) | 57.95(7) | **79.27(1)** | 52.70(15) | 52.07(16) | 57.46(8) | 56.16(9) | 54.72(10) | 78.90(2) | 70.12(5) | 70.28(4) | 70.28(3) |
| annthyroid | 76.85(16) | 95.54(9) | 83.31(11) | 82.81(13) | 82.58(14) | 91.43(10) | 99.21(5) | 83.01(12) | 82.03(15) | 96.62(8) | 98.13(7) | 98.97(6) | 99.09(3) | **99.38(1)** | 99.37(2) | 99.27(4) |
| backdoor | 86.48(15) | 98.24(6) | 89.60(14) | 98.11(7) | 94.07(12) | 98.48(5) | 97.27(9) | 85.15(16) | 97.21(10) | 97.69(8) | 92.79(13) | 96.56(11) | **99.69(1)** | 99.23(4) | 99.31(3) | 99.64(2) |
| breastw | 94.15(16) | 98.44(13) | 98.84(12) | 99.67(2) | 99.59(4) | 98.97(11) | 99.17(9) | **99.75(1)** | 98.32(14) | 99.62(3) | 95.52(15) | 99.10(10) | 99.19(8) | 99.42(5) | 99.40(6) | 99.38(7) |
| campaign | 58.36(15) | 78.11(13) | 57.94(16) | 86.53(8) | 87.42(7) | 77.21(14) | **92.93(1)** | 81.14(11) | 78.45(12) | 82.95(10) | 84.07(9) | 89.47(6) | 91.25(5) | 92.44(3) | 91.77(4) | 92.86(2) |
| cardio | 85.52(15) | 96.64(12) | 98.68(6) | 98.75(4) | 98.62(7) | 96.56(13) | **99.38(1)** | 94.69(14) | 97.54(10) | 96.88(11) | N/A(N/A) | 98.70(5) | 99.31(2) | 98.52(8) | 98.28(9) | 99.18(3) |
| Cardiocography | 58.27(16) | 94.69(11) | 95.80(6) | 95.00(10) | 95.22(9) | 94.30(12) | 96.00(4) | 93.07(14) | 94.29(13) | 95.79(7) | 89.09(15) | 95.52(8) | 96.23(3) | 96.42(2) | 95.95(5) | **96.53(1)** |
| celeba | 69.88(15) | 92.06(10) | 57.64(16) | 95.71(3) | 95.34(6) | 93.11(9) | 95.81(2) | 89.48(11) | 89.06(13) | 95.44(4) | 82.83(14) | 95.42(5) | 89.42(12) | **95.91(1)** | 93.73(8) | 95.04(7) |
| census | 58.46(16) | 82.01(12) | 69.47(15) | 91.36(4) | 89.89(7) | 84.60(11) | 92.51(2) | 73.31(13) | 86.39(10) | 86.63(9) | 70.73(14) | 86.68(8) | 90.87(6) | 91.72(3) | 91.29(5) | **92.95(1)** |
| cover | 42.11(16) | 99.95(3) | **99.95(1)** | 99.93(7) | 99.94(6) | 99.88(8) | 99.64(13) | 99.81(12) | 98.99(14) | 99.86(9) | 97.07(15) | 99.94(4) | 99.95(2) | 99.83(10) | 99.82(11) | 99.94(5) |
| donors | 57.68(16) | **100.00(1)** | 82.81(15) | **100.00(1)** | 99.94(10) | 99.99(5) | 99.98(8) | 99.60(14) | 99.91(11) | **100.00(1)** | 99.77(13) | 99.88(12) | 99.98(7) | 99.96(9) | 99.98(6) | 99.98(6) |
| fault | 66.32(16) | 79.36(7) | 76.32(11) | 75.60(12) | 77.47(9) | 75.53(13) | 81.29(4) | 69.99(15) | 76.87(10) | 80.56(5) | 74.97(14) | 78.64(8) | 83.03(2) | 81.77(3) | 80.15(6) | **84.08(1)** |
| fraud | 90.84(6) | 91.56(4) | 92.22(3) | 91.24(5) | **95.11(1)** | 82.43(13) | 89.93(7) | 88.59(8) | 78.12(14) | 86.39(12) | 73.00(15) | 87.20(11) | 88.12(9) | 41.66(16) | 87.38(10) | 93.64(2) |
| glass | 68.48(16) | 95.63(10) | 90.11(13) | 89.00(14) | 90.35(12) | 99.61(6) | 99.66(5) | 94.38(11) | 98.52(9) | 85.43(15) | 99.83(2) | 99.67(4) | 99.73(3) | 99.34(8) | 99.41(7) | **99.87(1)** |
| Hepatitis | 66.13(16) | 99.67(11) | 99.74(10) | 99.23(12) | 98.83(13) | 99.96(8) | 99.94(9) | 97.18(14) | 94.81(15) | **100.00(1)** | **100.00(1)** | **100.00(1)** | **100.00(1)** | **100.00(1)** | **100.00(1)** | **100.00(1)** |
| http | 99.77(11) | **100.00(1)** | 99.98(8) | **100.00(1)** | **100.00(1)** | 99.93(10) | 98.33(12) | 81.73(14) | 90.24(10) | 0.39(16) | **100.00(1)** | **100.00(1)** | **100.00(1)** | **100.00(1)** | **100.00(1)** | 99.97(9) |
| InternetAds | 68.93(15) | 90.13(9) | 94.33(2) | 92.95(5) | 92.47(7) | 89.31(12) | 92.35(8) | 76.24(14) | 89.42(11) | 89.53(10) | 80.04(13) | N/A(N/A) | 93.57(3) | 92.93(6) | 93.48(4) | **95.37(1)** |
| Ionosphere | 92.84(11) | 96.83(7) | 97.55(6) | 91.77(14) | 92.28(12) | 92.26(13) | 98.93(2) | 89.73(15) | 94.26(10) | 96.68(8) | 89.12(16) | 96.17(9) | 97.86(5) | **99.04(1)** | 98.80(3) | 98.77(4) |
| landsat | 55.06(16) | 94.52(4) | 60.57(15) | 80.19(12) | 79.86(13) | 87.52(11) | 94.60(3) | 71.73(14) | 90.24(10) | 92.93(8) | 91.23(9) | 93.96(7) | 95.09(2) | 94.47(5) | 94.45(6) | **95.46(1)** |
| letter | 70.16(16) | 83.75(10) | 88.74(6) | 84.17(9) | 83.32(11) | 82.56(13) | 91.91(2) | 70.70(15) | 81.80(14) | 82.59(12) | 84.90(8) | 89.13(5) | 90.57(4) | 91.58(3) | 87.49(7) | **94.25(1)** |
| Lymphography | 98.18(14) | 99.88(13) | **100.00(1)** | **100.00(1)** | **100.00(1)** | **100.00(1)** | **100.00(1)** | 99.92(12) | 96.82(15) | **100.00(1)** | 93.94(16) | **100.00(1)** | **100.00(1)** | **100.00(1)** | **100.00(1)** | **100.00(1)** |
| magic.gamma | 56.32(16) | 90.24(4) | 82.46(14) | 82.94(13) | 83.19(12) | 84.89(10) | 90.13(6) | 77.43(15) | 83.81(11) | 90.41(3) | 85.29(9) | 88.45(7) | 90.68(2) | 90.18(5) | 88.42(8) | **91.07(1)** |
| mammography | 77.28(15) | 94.42(2) | 93.17(11) | 93.28(10) | 93.29(8) | 93.95(5) | 94.12(3) | 91.32(12) | 70.70(16) | 93.28(9) | 93.62(6) | 93.92(4) | 93.47(7) | 87.29(14) | 91.21(13) | **94.48(1)** |
| mnist | 67.77(16) | 99.14(7) | 99.36(4) | 99.01(10) | 99.13(8) | 97.21(13) | 99.42(3) | 90.02(15) | 98.01(12) | 98.42(11) | 95.66(14) | 99.01(9) | **99.56(1)** | 99.31(5) | 99.19(6) | 99.51(2) |
| musk | 99.29(16) | **100.00(1)** | **100.00(1)** | **100.00(1)** | **100.00(1)** | **100.00(1)** | **100.00(1)** | 99.43(14) | 99.41(15) | **100.00(1)** | 99.98(13) | **100.00(1)** | **100.00(1)** | **100.00(1)** | **100.00(1)** | **100.00(1)** |
| optdigits | 46.90(16) | 99.97(6) | 99.99(3) | **99.99(1)** | 99.99(2) | 99.99(4) | 99.99(4) | 98.05(14) | 99.80(13) | 99.98(5) | 97.59(15) | 99.91(12) | 99.91(11) | 99.91(10) | 99.92(9) | 99.92(8) |
| PageBlocks | 78.46(16) | 97.21(8) | 94.08(12) | 88.09(15) | 90.53(14) | 96.22(9) | 98.31(4) | 91.83(13) | 94.17(11) | 95.63(10) | 97.49(7) | 97.61(6) | 98.29(5) | 98.39(2) | 98.34(3) | **98.87(1)** |
| pendigits | 59.36(16) | 99.99(2) | 99.88(8) | 99.76(13) | 99.81(11) | 99.87(9) | 99.93(6) | 99.69(14) | 99.77(12) | 99.83(10) | 98.70(15) | 99.97(3) | 99.99(3) | 99.95(5) | 99.91(7) | **100.00(1)** |
| Pima | 63.36(16) | 80.63(11) | 75.75(15) | 82.93(9) | 82.73(10) | 80.50(12) | 88.62(5) | 83.54(7) | 77.38(14) | 87.97(8) | 80.45(13) | 84.25(6) | **89.73(4)** | 89.63(2) | 88.73(4) | 89.13(3) |
| satellite | 75.12(16) | 95.28(7) | 81.26(14) | 85.38(12) | 84.85(13) | 86.32(11) | 95.55(3) | 78.88(15) | 91.43(8) | 90.66(10) | 91.13(9) | 95.39(6) | **96.25(1)** | 95.42(4) | 95.40(5) | 96.19(2) |
| satimage-2 | 96.83(13) | 99.28(3) | 99.45(2) | 99.06(4) | 99.05(5) | 98.31(7) | **99.45(1)** | 98.00(10) | 97.38(12) | 98.26(8) | 97.43(11) | 98.16(9) | 96.78(14) | 95.09(16) | 96.16(15) | 98.99(6) |
| shuttle | 65.45(16) | 98.97(6) | 98.91(7) | 97.57(14) | 97.67(12) | 98.42(9) | **99.98(1)** | 97.37(15) | 98.38(10) | 97.63(13) | 98.86(8) | 98.28(11) | 99.52(5) | 99.85(3) | 99.65(4) | 99.92(2) |
| skin | 52.08(16) | 99.92(3) | 89.02(15) | 95.70(12) | 95.35(13) | 98.77(11) | 99.88(7) | 93.94(14) | 99.73(10) | 99.88(6) | 99.93(2) | 99.90(4) | 99.77(9) | 99.89(5) | 99.87(8) | **99.94(1)** |
| smtp | 56.11(14) | 99.22(4) | 92.43(6) | 87.46(7) | 84.95(10) | 84.24(11) | 98.52(5) | 50.48(15) | 66.96(13) | 86.42(9) | 83.30(12) | 87.30(8) | **100.00(1)** | 43.94(16) | 99.72(3) | 99.99(2) |
| SpamBase | 57.07(16) | 94.37(8) | 91.77(14) | 92.05(12) | 94.09(9) | 93.64(11) | 97.38(2) | 83.88(15) | 91.82(13) | 97.05(4) | 93.71(10) | 95.20(7) | 95.74(6) | 97.15(3) | 96.55(5) | **97.69(1)** |
| speech | 47.72(16) | 59.94(14) | 73.34(7) | 80.36(2) | **81.56(1)** | 75.98(6) | 67.42(11) | 70.55(9) | 76.05(3) | 79.76(3) | 78.56(5) | 72.88(8) | 79.25(4) | 56.81(15) | 66.99(12) | 60.34(13) |
| Stamps | 74.37(16) | 98.03(14) | 98.84(11) | 98.41(13) | 99.00(9) | 98.85(10) | 99.61(4) | 99.07(8) | 97.30(15) | 99.12(7) | 98.47(12) | 99.65(3) | 99.32(6) | 99.67(2) | 99.56(5) | **99.72(1)** |
| thyroid | 92.97(16) | 99.04(13) | 99.76(5) | 99.76(6) | 99.76(7) | 99.76(8) | **99.90(1)** | 99.62(10) | 95.95(15) | 99.57(11) | 97.46(14) | 99.70(9) | 99.22(12) | 99.80(4) | 99.85(3) | 99.86(2) |
| vertebral | 39.88(16) | 88.40(10) | 79.82(14) | 81.09(13) | 82.54(12) | 92.65(8) | **98.71(1)** | 87.03(11) | 91.13(9) | 74.28(15) | 96.42(6) | 96.29(7) | 97.79(5) | 97.87(4) | 98.22(3) | 98.59(2) |
| vowels | 78.97(16) | 93.21(14) | 98.89(9) | 98.94(3) | 98.13(7) | 97.53(11) | 98.14(6) | 94.70(13) | 92.80(15) | **99.60(1)** | 96.27(12) | 98.85(5) | 97.60(10) | 99.43(2) | 97.60(10) | 99.13(3) |
| Waveform | 53.02(16) | 85.40(14) | 91.74(4) | 91.02(7) | 91.47(5) | 86.57(13) | 91.05(6) | 86.98(12) | 91.97(2) | **94.73(1)** | 91.22(5) | 95.22(15) | 90.30(9) | 87.11(11) | 90.44(8) | 89.69(10) | 91.77(3) |
| WBC | 95.21(13) | 98.38(10) | 99.22(6) | 99.43(4) | 99.14(7) | 92.44(15) | **99.90(1)** | 98.95(9) | 97.00(11) | 96.50(12) | 91.37(16) | 99.33(5) | 99.68(3) | 93.46(14) | 99.06(8) | 99.82(2) |
| WDBC | 97.76(15) | 99.72(12) | **100.00(1)** | **100.00(1)** | **100.00(1)** | **100.00(1)** | 99.29(13) | 98.75(14) | **100.00(1)** | **100.00(1)** | 99.31(16) | **100.00(1)** | 99.99(9) | 99.80(11) | 99.89(10) | **100.00(1)** |
| Wilt | 36.44(16) | 98.38(3) | 61.62(15) | 68.09(13) | 68.95(12) | 90.10(10) | 98.06(5) | 82.18(11) | 96.56(8) | 66.93(14) | 98.62(2) | 90.90(9) | 97.03(7) | 97.79(6) | **98.86(1)** | 98.86(1) |
| wine | 70.50(16) | 99.89(14) | **100.00(1)** | **100.00(1)** | **100.00(1)** | **100.00(1)** | **100.00(1)** | 99.60(14) | 98.73(15) | **100.00(1)** | 99.91(11) | **100.00(1)** | **100.00(1)** | **100.00(1)** | **100.00(1)** | **100.00(1)** |
| WPBC | 49.03(16) | 89.94(8) | 77.94(14) | 82.20(12) | 82.45(11) | 85.95(10) | **95.75(1)** | 70.04(15) | 81.80(13) | 86.21(9) | 93.25(6) | 94.19(5) | 92.37(7) | 94.79(4) | 95.72(2) | 95.44(3) |
| yeast | 48.87(15) | 67.62(12) | 45.20(16) | 66.34(11) | 68.52(10) | 68.58(9) | 71.08(4) | 65.68(13) | 55.86(14) | 69.74(8) | 69.76(7) | 71.13(12) | 69.51(13) | 85.17(3) | 84.27(6) | **75.62(1)** |
| CIFAR10 | 61.15(15) | 77.08(11) | 84.28(5) | **85.35(1)** | 82.31(7) | 79.64(9) | 85.06(4) | 60.77(16) | 80.99(8) | 79.19(10) | 66.86(14) | 71.13(12) | 69.51(13) | 85.17(3) | 84.27(6) | 85.33(2) |
| FashionMNIST | 80.16(15) | 94.68(9) | 96.20(2) | **96.57(1)** | 95.51(7) | 94.97(8) | 96.16(4) | 68.84(16) | 92.20(10) | 91.24(11) | 83.77(14) | 90.66(12) | 89.53(13) | 96.18(3) | 95.93(6) | 95.94(5) |
| MNIST-C | 72.91(16) | 94.30(10) | 82.81(14) | **95.81(1)** | 95.37(2) | 94.82(8) | 95.27(5) | 73.69(15) | 95.04(7) | 94.74(9) | 88.73(13) | 92.76(12) | 93.78(11) | 95.36(3) | 95.28(4) | 95.18(6) |
| MVTec-AD | 74.83(14) | 92.35(6) | 73.53(15) | 91.73(7) | 90.18(9) | 89.65(10) | 95.25(3) | 71.08(16) | 88.37(12) | 89.26(11) | 87.46(13) | 90.98(8) | 94.51(5) | 94.87(4) | 95.45(2) | **95.73(1)** |
| SVHN | 57.85(15) | 74.64(10) | 80.99(2) | **82.12(1)** | 80.34(3) | 77.30(9) | 79.31(6) | 51.66(16) | 77.99(8) | 73.79(11) | 69.37(12) | 68.41(13) | 65.37(14) | 79.61(5) | 78.75(7) | 79.92(4) |
| Agnews | 59.67(16) | 88.54(12) | 92.29(3) | 92.15(5) | 92.19(4) | 90.06(10) | 92.00(8) | 66.20(15) | 92.11(6) | **95.04(1)** | 89.53(11) | 87.88(13) | 70.95(14) | 91.90(9) | 92.03(7) | 92.60(2) |
| Amazon | 57.40(15) | 78.71(11) | 87.47(4) | 88.33(2) | 87.66(3) | 85.73(8) | 85.94(7) | 51.06(16) | 83.14(10) | **90.60(1)** | 77.20(13) | 78.22(12) | 62.23(14) | 84.04(9) | 86.14(6) | 86.47(5) |
| Imdb | 49.26(16) | 80.25(12) | 81.63(10) | 84.00(2) | 83.27(4) | 82.06(8) | 82.73(7) | 59.97(14) | 82.90(6) | **88.78(1)** | 81.33(11) | 79.21(13) | 59.70(15) | 81.93(9) | 83.19(5) | 83.93(3) |
| Yelp | 68.41(14) | 83.38(13) | 92.12(2) | 92.10(3) | 90.53(5) | 88.92(10) | 90.19(7) | 59.38(16) | 91.37(4) | **94.62(1)** | 86.64(11) | 85.83(12) | 66.45(15) | 89.71(9) | 89.95(8) | 90.23(6) |
| 20news | 57.80(15) | 76.18(10) | 59.88(14) | **83.71(1)** | 83.51(2) | 77.76(8) | 76.80(9) | 53.35(16) | 78.98(7) | 80.42(3) | 70.22(12) | 69.90(13) | 72.08(11) | 79.04(6) | 79.33(5) | 80.38(4) |

Table D15: AUCPR of 16 label-informed algorithms on 57 benchmark datasets, with labeled anomaly ratio $\gamma_l = 50\%$. We show the performance rank in parenthesis (lower the better), and mark the best performing method(s) in **bold**.

| Datasets | GANomaly | DeepSAD | REPEN | DevNet | PReNet | FEAWAD | XGBOD | NB | SVM | MLP | ResNet | FTTransformer | RF | LGB | XGB | CatB |
|---|---|---|---|---|---|---|---|---|---|---|---|---|---|---|---|---|
| ALOI | 3.80(16) | 7.79(6) | 4.68(11) | 4.29(13) | 4.29(14) | 5.80(10) | 14.42(3) | 4.04(15) | 6.38(7) | 5.97(9) | 6.00(8) | 4.58(12) | **19.73(1)** | 12.29(4) | 12.26(5) | 15.21(2) |
| annthyroid | 36.48(16) | 69.96(9) | 43.17(15) | 45.48(12) | 45.07(14) | 56.75(11) | 85.86(5) | 45.09(13) | 58.00(10) | 71.27(8) | 77.50(7) | 83.77(6) | 89.09(3) | **89.70(1)** | 89.60(2) | 86.92(4) |
| backdoor | 32.07(14) | 85.28(11) | 29.70(15) | 90.44(7) | 89.20(8) | 90.83(5) | 72.87(13) | 17.69(16) | 90.70(6) | 85.44(10) | 86.98(9) | 83.55(12) | **97.24(1)** | 92.06(4) | 93.08(3) | 96.60(2) |
| breastw | 88.43(16) | 96.18(13) | 97.24(12) | 99.33(2) | 99.12(4) | 98.43(8) | 97.60(11) | **99.52(1)** | 96.17(14) | 99.20(3) | 94.77(15) | 97.73(10) | 98.08(9) | 98.61(6) | 98.58(7) | 98.67(5) |
| campaign | 17.93(15) | 37.50(13) | 15.35(16) | 46.79(8) | 48.68(6) | 33.08(14) | **58.11(1)** | 38.49(12) | 38.90(11) | 42.72(9) | 40.12(10) | 47.50(7) | 53.96(3) | 53.83(4) | 52.12(5) | 55.79(2) |
| cardio | 44.50(15) | 86.44(12) | 93.47(2) | 91.40(7) | 91.50(6) | 87.30(11) | **93.69(1)** | 79.41(14) | 82.90(13) | 87.82(10) | N/A(N/A) | 91.72(5) | 93.46(3) | 87.83(9) | 89.77(8) | 93.07(4) |
| Cardiocography | 37.08(16) | 84.10(11) | 87.98(2) | 84.13(10) | 84.69(8) | 80.77(13) | 86.08(6) | 78.74(14) | 84.36(9) | 87.26(4) | 76.85(15) | 82.05(12) | 87.86(3) | 87.24(5) | 85.69(7) | **88.35(1)** |
| celeba | 8.61(15) | 29.10(8) | 3.31(16) | **34.64(1)** | 33.25(2) | 25.65(10) | 29.11(7) | 11.63(14) | 24.76(11) | 29.23(6) | 22.64(13) | 26.59(9) | 23.68(12) | 31.97(5) | 29.55(5) | 31.90(4) |
| census | 7.60(16) | 33.86(10) | 10.24(15) | 45.46(7) | 46.42(6) | 31.67(12) | **54.61(1)** | 12.22(14) | 37.34(8) | 36.76(9) | 27.78(13) | 33.71(11) | 48.97(4) | 49.10(3) | 47.54(5) | 53.74(2) |
| cover | 0.87(16) | **96.90(1)** | 96.33(3) | 96.60(2) | 95.84(4) | 94.01(6) | 87.82(12) | 89.71(10) | 81.19(14) | 91.62(9) | 83.88(13) | 92.57(7) | 94.82(5) | 80.36(15) | 89.69(11) | 91.82(8) |
| donors | 9.59(16) | **100.00(1)** | 17.09(15) | 100.00(2) | 98.94(12) | 99.45(9) | 99.53(8) | 88.90(14) | 98.69(13) | 100.00(2) | 99.66(7) | 100.00(2) | 99.33(11) | 99.72(6) | 99.35(10) | 99.73(5) |
| fault | 52.78(16) | 64.59(8) | 63.11(13) | 61.39(14) | 65.97(7) | 63.58(12) | 69.73(3) | 53.82(15) | 63.97(10) | 67.93(5) | 63.79(11) | 64.53(9) | 70.60(2) | 69.36(4) | 66.37(6) | **73.33(1)** |
| fraud | 42.42(12) | 44.58(10) | 42.54(11) | 57.41(3) | 58.06(2) | 54.79(5) | 55.51(4) | 22.65(15) | 30.67(14) | 48.17(9) | 39.66(13) | 53.37(7) | 53.74(6) | 0.15(16) | 51.59(8) | **61.97(1)** |
| glass | 16.25(16) | 46.49(10) | 33.61(12) | 26.60(15) | 28.31(13) | 95.30(5) | 94.82(6) | 44.49(11) | 92.24(9) | 27.92(14) | 96.78(2) | 96.67(3) | 96.44(4) | 92.99(8) | 93.48(7) | **97.47(1)** |
| Hepatitis | 31.56(16) | 98.42(11) | 99.00(10) | 93.79(12) | 91.56(13) | 99.79(8) | 99.69(9) | 83.73(14) | 83.17(15) | **100.00(1)** | **100.00(1)** | **100.00(1)** | **100.00(1)** | **100.00(1)** | **100.00(1)** | **100.00(1)** |
| http | 67.59(13) | **100.00(1)** | 93.99(11) | **100.00(1)** | **100.00(1)** | 96.62(10) | 66.78(14) | 96.68(8) | 66.78(14) | 0.37(16) | **100.00(1)** | **100.00(1)** | 0.38(15) | **100.00(1)** | **100.00(1)** | 94.12(10) |
| InternetAds | 49.41(14) | 80.95(9) | 87.79(2) | 86.41(3) | 86.22(4) | 78.45(11) | 85.50(6) | 42.15(15) | 83.85(7) | 74.60(12) | 69.61(13) | 78.89(10) | 83.73(8) | 85.87(5) | 85.87(5) | **91.04(1)** |
| Ionosphere | 90.45(15) | 95.80(8) | 96.72(6) | 91.65(13) | 91.95(12) | 92.40(11) | **98.80(1)** | 35.16(16) | 92.78(10) | 95.84(7) | 90.52(14) | 94.82(9) | 97.32(5) | 98.55(2) | 98.39(3) | 98.13(4) |
| landsat | 23.29(16) | 32.56(11) | 41.88(14) | 56.10(12) | 55.34(13) | 62.72(11) | 84.47(3) | 29.70(15) | 77.68(9) | 79.45(8) | 75.47(10) | 80.63(7) | 86.35(2) | 83.95(4) | 83.06(6) | **86.69(1)** |
| letter | 16.32(15) | 32.56(11) | 50.34(5) | 28.06(14) | 31.43(12) | 29.85(13) | 49.99(6) | 14.41(16) | 39.87(9) | 37.50(10) | 53.57(3) | 53.61(2) | 47.32(7) | 53.49(4) | 45.60(8) | **58.35(1)** |
| Lymphography | 74.88(16) | 98.72(12) | **100.00(1)** | **100.00(1)** | **100.00(1)** | **100.00(1)** | **100.00(1)** | 98.61(13) | 76.18(15) | **100.00(1)** | 94.16(14) | **100.00(1)** | **100.00(1)** | **100.00(1)** | **100.00(1)** | **100.00(1)** |
| magic.gamma | 42.51(16) | 85.16(5) | 76.64(10) | 72.86(14) | 74.75(13) | 74.90(12) | 85.48(4) | 69.93(15) | 79.92(9) | 86.86(2) | 75.90(11) | 80.67(8) | 85.77(3) | 84.85(6) | 81.78(7) | **86.87(1)** |
| mammography | 17.32(16) | 64.05(6) | 61.81(10) | 61.86(9) | 60.95(11) | 63.40(6) | 63.28(7) | 46.83(14) | 42.42(15) | 60.93(12) | 67.29(3) | 69.52(2) | 66.85(4) | 59.51(13) | 62.49(8) | **69.65(1)** |
| mnist | 16.32(16) | 94.42(5) | 95.50(3) | 90.42(11) | 91.40(9) | 81.82(14) | 95.11(4) | 50.06(15) | 88.99(12) | 92.22(8) | 83.97(13) | 91.37(10) | 96.22(2) | 94.15(6) | 93.77(7) | **96.31(1)** |
| musk | 89.43(16) | **100.00(4)** | 99.80(2) | **99.82(1)** | 99.79(3) | 99.64(4) | 99.29(9) | 66.18(15) | 96.78(13) | 99.53(5) | 89.41(14) | 98.77(7) | 98.33(8) | 98.16(11) | 98.02(12) | 98.28(10) |
| optdigits | 3.35(16) | 99.26(6) | 99.80(2) | **99.82(1)** | 99.79(3) | 99.64(4) | 99.29(9) | 66.18(15) | 96.78(13) | 99.53(5) | 89.41(14) | 98.77(7) | 98.33(8) | 98.16(11) | 98.02(12) | 98.28(10) |
| PageBlocks | 47.07(16) | 85.33(8) | 77.01(12) | 65.91(14) | 68.90(13) | 81.16(11) | 85.86(6) | 55.61(15) | 81.90(9) | 81.41(10) | 86.05(5) | 85.76(7) | 88.30(2) | 88.05(3) | 87.54(4) | **89.57(1)** |
| pendigits | 3.95(16) | 99.62(2) | 97.64(8) | 92.11(15) | 94.20(12) | 95.69(10) | 98.76(5) | 94.10(13) | 95.05(11) | 97.09(9) | 93.89(14) | 98.98(4) | 99.61(3) | 98.75(6) | 98.44(7) | **99.90(1)** |
| Pima | 46.44(16) | 63.68(14) | 58.93(15) | 70.59(11) | 70.65(10) | 69.24(12) | 83.00(4) | 71.56(9) | 65.62(13) | 72.61(6) | 72.61(7) | 71.62(8) | **84.52(1)** | 84.01(2) | 82.90(5) | 83.37(3) |
| satellite | 64.49(16) | 90.64(6) | 78.93(13) | 83.13(10) | 82.98(12) | 73.22(15) | 92.24(3) | 75.54(14) | 88.30(8) | 86.77(9) | 83.03(11) | 89.55(7) | **93.39(1)** | 91.35(4) | 90.83(5) | 93.30(2) |
| satimage-2 | 46.44(16) | **95.70(1)** | 93.59(3) | 92.70(6) | 92.06(9) | 93.36(4) | 90.73(11) | 92.42(8) | 90.37(13) | 90.31(14) | 85.55(15) | 94.64(2) | 92.59(7) | 91.02(10) | 90.40(12) | 93.15(5) |
| shuttle | 39.31(16) | 98.14(6) | 97.33(8) | 86.38(11) | 96.56(11) | 96.93(10) | **99.79(1)** | 95.86(15) | 96.38(13) | 96.17(14) | 97.74(7) | 96.84(9) | 99.22(4) | 99.29(2) | 99.22(4) | 99.59(2) |
| skin | 22.13(16) | 99.37(6) | 48.91(15) | 70.10(13) | 67.81(14) | 89.22(11) | **99.63(1)** | 86.56(12) | 98.66(10) | 99.34(7) | 99.46(3) | 99.20(8) | 99.02(9) | 99.39(5) | 99.40(4) | 99.60(2) |
| smtp | 21.46(13) | 58.81(5) | 50.05(8) | 66.70(3) | 66.69(4) | 17.13(14) | 50.25(7) | 0.06(15) | 50.01(10) | 50.03(9) | 44.46(11) | 33.96(12) | **100.00(1)** | 0.04(16) | 50.83(6) | 83.33(2) |
| SpamBase | 43.43(16) | 90.36(10) | 87.16(13) | 86.90(14) | 90.31(11) | 90.81(8) | 96.06(3) | 68.15(15) | 88.95(12) | 95.32(4) | 90.79(9) | 91.83(7) | 94.76(5) | 95.69(3) | 94.75(5) | **96.68(1)** |
| speech | 1.62(16) | 5.05(15) | 28.47(2) | 17.48(6) | 17.72(5) | 14.20(9) | 8.31(11) | 16.74(7) | **30.20(1)** | 17.87(4) | 25.25(3) | 16.61(8) | 6.86(12) | 12.87(10) | 5.81(13) | 5.48(14) |
| Stamps | 32.40(16) | 75.43(15) | 85.74(10) | 76.84(14) | 83.82(12) | 84.79(11) | 96.59(3) | 88.88(9) | 77.98(13) | 89.24(8) | 90.13(7) | 96.58(4) | 94.72(6) | 96.78(2) | 95.40(5) | **97.32(1)** |
| thyroid | 54.41(16) | 88.09(13) | 92.67(8) | 93.47(6) | 92.87(7) | 92.66(9) | **96.13(1)** | 90.49(12) | 75.40(15) | 90.97(11) | 65.42(10) | 91.30(10) | 93.64(4) | 93.48(5) | 94.22(3) | 95.04(2) |
| vertebral | 11.11(16) | 59.43(10) | 48.61(12) | 45.64(14) | 47.55(13) | 68.44(9) | 94.90(4) | 51.93(11) | 69.01(8) | 29.56(15) | 92.49(6) | 83.80(7) | 94.77(5) | **95.83(1)** | 95.17(3) | 95.67(2) |
| vowels | 26.48(16) | 65.76(15) | 88.90(5) | 90.64(3) | **92.03(1)** | 77.14(9) | 72.46(12) | 70.96(13) | 87.04(6) | 69.94(14) | 82.61(8) | 91.83(2) | 76.77(10) | 84.74(7) | 76.41(11) | 89.15(4) |
| Waveform | 4.48(16) | 34.85(10) | 20.18(12) | 18.23(14) | 19.39(13) | 22.08(11) | 40.77(4) | 14.57(15) | **57.61(1)** | 42.15(3) | 36.74(8) | 36.86(7) | 35.78(9) | 37.47(6) | 37.65(5) | 42.67(2) |
| WBC | 44.19(16) | 75.58(13) | 90.60(8) | 91.52(7) | 87.99(9) | 72.50(15) | **96.67(1)** | 87.71(10) | 75.58(14) | 81.24(12) | 83.81(11) | 92.75(4) | 94.83(3) | 92.01(6) | 92.15(5) | 96.59(2) |
| WDBC | 60.49(16) | 93.82(13) | **100.00(1)** | **100.00(1)** | **100.00(1)** | **100.00(1)** | 94.07(12) | **100.00(1)** | 83.41(15) | **100.00(1)** | 88.11(14) | **100.00(1)** | 99.54(9) | 97.63(10) | 97.10(11) | **100.00(1)** |
| Wilt | 3.93(16) | 80.95(7) | 7.44(15) | 8.15(14) | 8.41(12) | 45.54(10) | 84.14(3) | 39.55(11) | 81.37(6) | 8.31(13) | 83.54(4) | 61.97(9) | 85.74(2) | 78.64(8) | 81.66(5) | **87.16(1)** |
| wine | 18.91(16) | 98.82(14) | **100.00(1)** | **100.00(1)** | **100.00(1)** | **100.00(1)** | 94.07(12) | **100.00(1)** | 92.62(15) | **100.00(1)** | **100.00(1)** | **100.00(1)** | **100.00(1)** | **100.00(1)** | **100.00(1)** | **100.00(1)** |
| WPBC | 23.41(16) | 76.66(8) | 61.31(14) | 62.91(13) | 63.83(12) | 73.90(10) | 93.52(3) | 48.36(15) | 71.93(11) | 74.68(9) | 90.46(5) | 89.62(7) | 90.45(6) | 92.69(4) | 93.58(2) | **93.95(1)** |
| yeast | 33.33(16) | 48.18(12) | 34.34(15) | 48.71(11) | 49.38(10) | 51.40(8) | 56.01(4) | 45.44(13) | 40.07(14) | 50.76(9) | 51.57(7) | 57.82(2) | 57.11(3) | 54.62(6) | 55.49(5) | **61.24(1)** |
| CIFAR10 | 9.27(15) | 26.70(11) | **37.92(1)** | 35.42(2) | 34.62(5) | 28.25(9) | 35.17(4) | 7.41(16) | 28.54(8) | 26.97(10) | 23.86(12) | 19.38(13) | 14.98(14) | 34.03(7) | 34.29(6) | 35.23(3) |
| FashionMNIST | 21.70(16) | 73.62(8) | 81.48(3) | 81.65(2) | 80.38(5) | 71.13(10) | 79.70(7) | 22.68(15) | 72.33(9) | 67.38(11) | 63.52(12) | 59.75(14) | 61.88(13) | 80.50(4) | 80.09(6) | **81.92(1)** |
| MNIST-C | 19.01(16) | 82.75(9) | 42.17(14) | **86.63(1)** | 86.55(2) | 82.38(10) | 83.19(7) | 26.52(15) | 83.53(6) | 83.15(8) | 72.60(13) | 76.25(12) | 79.56(11) | 83.74(5) | 84.14(4) | 84.27(3) |
| MVTec-AD | 58.20(14) | 86.43(6) | 55.61(15) | 83.77(8) | 81.43(12) | 83.13(9) | 91.24(4) | 48.81(16) | 77.94(13) | 82.96(10) | 82.16(11) | 84.39(7) | 90.59(5) | 91.38(3) | 91.91(2) | **92.90(1)** |
| SVHN | 7.95(15) | 22.28(12) | 30.55(3) | 31.55(2) | **32.45(1)** | 25.91(9) | 26.62(8) | 5.22(16) | 26.37(6) | 25.11(10) | 24.03(11) | 16.38(13) | 12.82(14) | 28.27(4) | 27.02(6) | 28.07(5) |
| Agnews | 6.58(16) | 60.20(5) | 61.87(2) | 56.52(9) | 59.48(6) | 60.71(4) | 54.63(12) | 7.75(15) | 61.84(3) | **72.01(1)** | 58.21(8) | 47.78(13) | 18.29(14) | 56.47(11) | 56.49(10) | 59.21(7) |
| Amazon | 6.12(15) | 28.98(7) | 32.04(2) | 31.11(3) | 30.51(4) | 28.61(8) | 27.90(10) | 5.11(16) | 29.65(5) | **39.21(1)** | 23.66(12) | 19.34(13) | 7.79(14) | 26.09(11) | 29.00(6) | 28.00(9) |
| Imdb | 4.84(16) | 32.95(2) | 25.92(10) | 26.55(6) | 26.38(7) | 24.38(11) | 28.73(5) | 7.27(15) | 23.73(12) | **39.64(1)** | 30.03(3) | 22.33(13) | 7.31(14) | 26.00(9) | 26.24(8) | 28.96(4) |
| Yelp | 9.19(15) | 45.29(6) | 49.50(2) | 47.93(5) | 49.14(3) | 39.24(12) | 42.28(8) | 6.50(16) | 48.98(4) | **59.67(1)** | 40.32(11) | 34.63(13) | 11.37(14) | 40.41(10) | 40.66(9) | 44.61(7) |
| 20news | 7.33(15) | 28.66(8) | 7.73(14) | 36.72(3) | 37.02(2) | 30.81(5) | 26.29(10) | 5.34(16) | 32.63(4) | **41.92(1)** | 28.94(7) | 25.87(11) | 19.54(13) | 25.55(12) | 30.33(6) | 28.04(9) |

Table D16: AUCROC of 16 label-informed algorithms on 57 benchmark datasets, with labeled anomaly ratio $\gamma_l = 75\%$. We show the performance rank in parenthesis (lower the better), and mark the best performing method(s) in **bold**.

| Datasets | GANomaly | DeepSAD | REPEN | DevNet | PReNet | FEAWAD | XGBOD | NB | SVM | MLP | ResNet | FTTransformer | RF | LGB | XGB | CatB |
|---|---|---|---|---|---|---|---|---|---|---|---|---|---|---|---|---|
| ALOI | 56.22(11) | 66.49(6) | 51.94(15) | 52.85(14) | 50.82(16) | 63.28(7) | **80.73(1)** | 53.69(12) | 58.62(9) | 59.71(8) | 57.35(10) | 53.21(13) | 79.83(2) | 74.51(3) | 73.38(5) | 73.46(4) |
| annthyroid | 81.51(16) | 97.38(10) | 83.51(13) | 82.15(15) | 83.14(14) | 93.21(11) | 99.38(5) | 84.48(12) | 98.10(8) | 97.96(9) | 98.61(7) | 99.10(6) | 99.46(2) | 99.44(3) | **99.47(1)** | 99.42(4) |
| backdoor | 86.89(16) | 98.69(6) | 89.62(15) | 97.91(9) | 94.93(13) | 98.56(8) | 97.52(10) | 93.90(14) | 98.56(7) | 98.88(5) | 96.33(12) | 96.96(11) | **99.95(1)** | 99.73(2) | 99.60(4) | 99.64(3) |
| breastw | 95.04(16) | 99.22(14) | 99.46(12) | 99.73(5) | 99.74(4) | 98.76(15) | 99.68(10) | 99.69(9) | 99.55(11) | 99.75(3) | 99.39(13) | 99.70(8) | **99.79(1)** | 99.70(7) | 99.71(6) | 99.78(2) |
| campaign | 66.24(15) | 81.12(13) | 58.00(16) | 87.18(9) | 88.34(7) | 76.61(14) | **93.54(1)** | 81.89(12) | 84.32(11) | 86.75(10) | 89.83(6) | 90.05(6) | 91.84(5) | 93.29(3) | 92.47(4) | 93.52(2) |
| cardio | 87.07(15) | 98.39(11) | 99.27(7) | 98.80(9) | 98.56(10) | 97.84(12) | **99.57(1)** | 94.77(14) | 99.25(8) | 97.39(13) | N/A(N/A) | 99.42(4) | 99.48(2) | 99.35(5) | 99.34(6) | 99.47(3) |
| Cardiocography | 60.68(16) | 96.59(8) | 96.10(10) | 95.11(13) | 95.25(12) | 94.09(15) | 97.60(4) | 94.31(14) | 96.57(9) | 96.93(7) | 95.84(11) | 97.48(5) | 97.96(2) | 97.79(3) | 97.45(6) | **98.13(1)** |
| celeba | 63.72(15) | 93.78(10) | 57.64(16) | 95.82(3) | 95.78(4) | 93.98(9) | **96.28(1)** | 89.66(13) | 91.94(11) | 95.47(6) | 86.43(14) | 95.43(7) | 90.59(12) | 96.18(2) | 95.15(8) | 95.74(5) |
| census | 63.95(16) | 85.99(12) | 69.68(15) | 92.33(5) | 91.29(7) | 87.55(10) | 93.16(2) | 73.51(14) | 88.61(9) | 89.06(8) | 79.60(13) | 87.17(11) | 92.22(6) | 92.85(3) | 92.58(4) | **93.55(1)** |
| cover | 42.57(16) | **99.98(1)** | 99.95(7) | 99.95(6) | 99.96(6) | 99.80(13) | 99.72(15) | 99.89(12) | 99.97(2) | 99.95(9) | 99.96(5) | 99.94(11) | 99.96(4) | 99.95(10) | 99.78(14) | 99.97(3) |
| donors | 56.37(16) | **100.00(1)** | 82.83(15) | **100.00(1)** | 99.95(12) | **100.00(1)** | 100.00(9) | 99.60(14) | **100.00(1)** | **100.00(1)** | 99.83(13) | **100.00(1)** | 99.99(10) | 100.00(7) | 99.99(11) | 100.00(8) |
| fault | 67.43(16) | 81.36(8) | 77.04(12) | 75.54(14) | 78.07(11) | 76.71(13) | 83.10(5) | 69.77(15) | 80.51(9) | 82.96(6) | 78.39(10) | 82.07(7) | 86.70(2) | 85.46(3) | 84.68(4) | **86.93(1)** |
| fraud | 91.03(8) | 92.66(6) | 90.93(9) | 92.36(7) | 92.95(5) | 89.44(10) | 96.68(3) | 89.19(11) | 82.81(15) | 83.67(13) | 82.89(14) | 93.45(4) | 88.05(12) | 69.55(16) | **97.91(1)** | 97.81(2) |
| glass | 68.70(16) | 97.73(10) | 88.86(14) | 88.60(15) | 90.29(13) | 99.65(8) | 99.98(5) | 93.74(11) | 99.14(9) | 90.53(12) | 99.96(7) | 99.99(4) | **100.00(1)** | **100.00(1)** | 99.97(6) | **100.00(1)** |
| Hepatitis | 69.27(16) | 99.92(9) | 99.78(11) | 98.86(14) | 99.36(12) | 99.14(13) | 99.93(8) | 97.10(15) | 99.85(10) | **100.00(1)** | **100.00(1)** | **100.00(1)** | **100.00(1)** | **100.00(1)** | **100.00(1)** | **100.00(1)** |
| http | 99.80(12) | **100.00(1)** | 99.98(9) | **100.00(1)** | 99.73(13) | 99.93(11) | 98.33(13) | **100.00(1)** | 66.82(16) | **100.00(1)** | **100.00(1)** | **100.00(1)** | 100.00(8) | 78.47(15) | **100.00(1)** | 99.97(10) |
| InternetAds | 69.50(15) | 93.90(11) | 96.22(2) | **96.23(1)** | 95.92(4) | 93.34(12) | 94.86(7) | 81.88(14) | 94.14(9) | 94.09(10) | 90.88(13) | N/A(N/A) | 94.89(6) | 94.87(8) | 95.14(5) | 95.99(3) |
| Ionosphere | 93.59(15) | 98.34(7) | 98.27(8) | 94.44(13) | 94.08(14) | 97.60(11) | **99.24(1)** | 91.40(16) | 97.80(10) | 98.21(9) | 97.36(12) | 98.76(6) | 99.15(4) | 98.99(5) | 99.21(2) | 99.21(3) |
| landsat | 57.20(16) | 95.27(6) | 63.11(15) | 80.53(12) | 80.14(13) | 86.70(11) | 95.39(5) | 71.73(14) | 93.07(10) | 94.11(8) | 93.96(7) | 94.86(7) | 96.03(3) | 96.16(2) | 95.89(4) | **96.42(1)** |
| letter | 70.42(16) | 86.83(11) | 88.48(8) | 87.56(10) | 86.46(12) | 84.29(14) | 93.35(5) | 75.45(15) | 85.08(13) | 87.60(9) | 91.66(6) | 90.85(7) | 95.39(2) | 93.39(4) | 93.86(3) | **96.49(1)** |
| Lymphography | 98.31(16) | **100.00(1)** | **100.00(1)** | **100.00(1)** | **100.00(1)** | **100.00(1)** | **100.00(1)** | 99.71(15) | **100.00(1)** | **100.00(1)** | **100.00(1)** | **100.00(1)** | **100.00(1)** | **100.00(1)** | **100.00(1)** | **100.00(1)** |
| magic.gamma | 58.56(16) | 90.86(6) | 83.36(12) | 83.14(14) | 83.19(13) | 84.17(11) | 90.97(4) | 77.08(15) | 90.47(8) | 90.88(5) | 88.91(10) | 89.92(9) | 91.88(2) | 91.64(3) | 90.60(7) | **92.25(1)** |
| mammography | 75.53(16) | 95.71(2) | 93.01(9) | 93.10(8) | 92.81(12) | 95.62(4) | **95.71(1)** | 92.82(11) | 81.77(15) | 93.34(7) | 92.92(10) | 95.22(5) | 93.46(6) | 90.41(14) | 92.67(13) | 95.67(3) |
| mnist | 68.69(16) | 99.48(7) | 99.31(10) | 98.99(12) | 99.32(8) | 97.65(14) | 99.53(6) | 94.97(15) | 99.56(4) | 99.32(9) | 97.75(13) | 99.04(11) | 99.72(2) | **99.73(1)** | 99.55(5) | 99.71(3) |
| musk | **100.00(1)** | **100.00(1)** | **100.00(1)** | **100.00(1)** | **100.00(1)** | **100.00(1)** | **100.00(1)** | **100.00(1)** | **100.00(1)** | **100.00(1)** | **100.00(1)** | **100.00(1)** | **100.00(1)** | **100.00(1)** | **100.00(1)** | **100.00(1)** |
| optdigits | 47.52(16) | 99.99(2) | 99.98(6) | 99.99(4) | 99.99(3) | 99.98(10) | 99.99(9) | 94.53(15) | 99.98(5) | 99.10(14) | 99.76(13) | 99.98(8) | 99.98(7) | 99.97(12) | **99.99(1)** | 99.97(11) |
| PageBlocks | 79.45(16) | 98.08(7) | 95.25(12) | 89.15(15) | 90.10(14) | 95.55(11) | 98.73(4) | 92.29(13) | 96.88(9) | 96.49(10) | 97.61(8) | 98.44(6) | 98.78(3) | 98.82(2) | 98.62(5) | **99.05(1)** |
| pendigits | 62.56(16) | 99.99(3) | **99.99(1)** | 99.79(14) | 99.82(13) | 99.91(11) | 99.97(9) | 99.62(15) | 100.00(1) | 99.95(7) | 99.90(12) | 99.97(6) | 99.99(2) | 99.99(5) | 99.98(8) | 99.99(4) |
| Pima | 65.20(16) | 83.81(10) | 75.97(15) | 83.25(13) | 83.38(12) | 82.79(14) | 90.47(5) | 83.47(11) | 86.61(7) | 84.51(9) | 85.94(8) | 87.07(6) | **93.84(1)** | 92.00(3) | 91.30(4) | 92.02(2) |
| satellite | 77.11(16) | 96.22(5) | 81.55(14) | 84.27(13) | 84.99(12) | 86.32(11) | 96.03(6) | 78.66(15) | 94.43(9) | 91.78(10) | 94.60(8) | 95.88(7) | **97.08(1)** | 96.59(3) | 96.56(4) | 96.98(2) |
| satimage-2 | 97.70(14) | 99.09(7) | 99.40(3) | 99.40(4) | 99.24(5) | 98.39(10) | **99.74(1)** | 98.20(11) | 98.14(12) | 97.97(13) | 98.85(8) | 97.11(16) | 97.57(15) | 98.78(9) | 99.10(6) | 99.70(2) |
| shuttle | 78.80(16) | 99.43(6) | 98.60(9) | 97.57(14) | 97.59(13) | 98.20(11) | 99.99(5) | 97.74(12) | 98.89(7) | 97.56(15) | 98.75(8) | 98.55(10) | 100.00(2) | **100.00(1)** | 99.99(3) | 99.99(4) |
| skin | 52.94(16) | 99.92(8) | 89.30(15) | 95.77(12) | 95.28(13) | 98.77(11) | **99.97(1)** | 93.90(14) | 99.96(7) | 99.96(6) | 99.90(9) | 99.89(10) | 99.93(7) | 99.94(4) | 99.93(6) | 99.96(2) |
| smtp | 55.30(16) | 99.23(5) | 91.98(8) | 85.48(11) | 78.88(13) | 86.42(9) | 98.76(6) | 59.01(14) | 84.13(12) | 86.42(10) | **100.00(1)** | 95.83(7) | 99.99(2) | 56.21(15) | 99.70(4) | 99.99(3) |
| SpamBase | 59.25(16) | 95.94(10) | 93.20(13) | 92.03(14) | 94.01(12) | 94.35(11) | 97.88(3) | 84.31(15) | 96.61(7) | 97.33(5) | 96.05(9) | 96.35(8) | 97.28(6) | 97.95(2) | 97.58(4) | **98.14(1)** |
| speech | 47.76(16) | 63.41(14) | 79.23(6) | 84.01(2) | **85.32(1)** | 75.16(10) | 77.09(9) | 74.49(11) | 82.30(3) | 81.29(4) | 73.85(12) | 77.27(8) | 60.60(15) | 69.39(13) | 79.68(5) | 77.38(7) |
| Stamps | 76.83(16) | 98.69(13) | 98.70(12) | 98.42(14) | 99.08(10) | 98.34(15) | 99.88(2) | 99.08(11) | 99.20(9) | 99.21(8) | 99.67(7) | 99.75(6) | 99.86(4) | 99.88(3) | 99.85(5) | **99.93(1)** |
| thyroid | 93.27(16) | 99.55(16) | 99.80(9) | 99.82(7) | 99.77(11) | 99.80(8) | **99.95(1)** | 99.65(13) | 99.77(10) | 99.72(12) | 99.58(14) | 99.86(6) | 99.89(4) | 99.89(5) | 99.93(2) | 99.92(3) |
| vertebral | 40.75(16) | 92.02(10) | 81.76(14) | 81.74(15) | 82.32(13) | 93.53(8) | 99.14(3) | 86.78(11) | 92.61(9) | 84.63(12) | 97.89(7) | 98.29(6) | 99.25(2) | 98.82(5) | 98.82(4) | **99.35(1)** |
| vowels | 80.17(16) | 95.53(14) | 99.20(6) | 98.82(9) | 98.87(7) | 98.41(11) | 98.59(10) | 97.48(13) | 99.72(2) | 95.52(15) | 98.84(8) | **99.88(1)** | 99.57(4) | 99.22(5) | 99.63(3) | 99.70(3) |
| Waveform | 52.90(16) | 90.76(13) | 92.28(8) | 92.43(7) | 93.48(4) | 90.76(12) | 93.30(5) | 90.38(14) | 94.22(2) | **96.39(1)** | 83.91(15) | 91.53(10) | 90.98(11) | 93.12(6) | 92.01(9) | 93.70(3) |
| WBC | 95.93(16) | 98.93(13) | 99.18(10) | 99.73(4) | 99.10(11) | 99.38(8) | 99.69(5) | 98.91(14) | 98.97(12) | 98.85(15) | 99.91(2) | 99.63(6) | **99.93(1)** | 99.40(7) | 99.31(9) | 99.92(3) |
| WDBC | 97.85(16) | 99.99(11) | **100.00(1)** | **100.00(1)** | **100.00(1)** | **100.00(1)** | 99.52(15) | **100.00(1)** | **100.00(1)** | **100.00(1)** | **100.00(1)** | **100.00(1)** | 99.99(12) | 99.95(14) | 99.96(13) | **100.00(1)** |
| Wilt | 46.01(16) | 98.80(5) | 59.80(15) | 68.15(14) | 69.10(13) | 93.06(10) | 98.56(7) | 82.67(11) | 98.88(3) | 82.43(12) | 99.01(2) | 98.71(6) | 98.56(8) | 98.29(9) | 98.81(4) | **99.02(1)** |
| wine | 74.38(16) | 99.98(15) | **100.00(1)** | **100.00(1)** | **100.00(1)** | **100.00(1)** | **100.00(1)** | **100.00(1)** | **100.00(1)** | **100.00(1)** | **100.00(1)** | **100.00(1)** | **100.00(1)** | **100.00(1)** | **100.00(1)** | **100.00(1)** |
| WPBC | 49.93(16) | 95.22(8) | 80.56(14) | 84.11(12) | 84.08(13) | 91.45(10) | 98.83(2) | 71.44(15) | 91.87(9) | 90.37(11) | 96.87(7) | 97.27(6) | **99.33(1)** | 98.68(3) | 98.37(5) | 98.59(4) |
| yeast | 49.05(15) | 70.94(9) | 86.44(16) | 69.04(12) | 68.99(13) | 70.32(11) | 74.72(5) | 67.26(14) | 71.09(8) | 70.58(10) | 74.42(13) | 74.91(12) | 73.82(14) | 86.27(3) | 85.56(6) | **86.81(1)** |
| CIFAR10 | 61.85(16) | 80.15(11) | 86.13(4) | 86.60(2) | 84.02(8) | 81.32(10) | 85.87(5) | 66.03(15) | 84.43(7) | 81.41(9) | 74.42(13) | 74.91(12) | 73.82(14) | 86.27(3) | 85.56(6) | **86.81(1)** |
| FashionMNIST | 80.23(15) | 96.00(7) | 96.64(4) | **96.81(1)** | 95.94(8) | 95.39(9) | 96.59(5) | 72.55(16) | 94.54(10) | 93.72(11) | 91.80(14) | 93.53(12) | 92.88(13) | 96.77(2) | 96.71(3) | 96.36(6) |
| MNIST-C | 77.60(16) | 95.53(6) | 92.88(14) | 95.86(2) | 95.32(8) | 94.66(10) | 95.61(5) | 75.87(15) | **96.04(1)** | 95.25(9) | 94.13(12) | 93.45(13) | 94.54(11) | 95.70(4) | 95.82(3) | 95.42(7) |
| MVTec-AD | 74.57(15) | 96.60(6) | 73.90(16) | 93.84(11) | 92.35(13) | 94.07(9) | 97.67(5) | 77.46(14) | 93.22(12) | 94.06(10) | 94.82(8) | 95.42(7) | 97.83(3) | 97.76(4) | 97.87(2) | **98.27(1)** |
| SVHN | 57.63(15) | 77.98(10) | 82.41(2) | **83.44(1)** | 82.00(3) | 79.20(9) | 81.23(8) | 53.15(16) | 81.30(7) | 77.03(11) | 75.33(12) | 70.31(13) | 68.77(14) | 81.54(6) | 81.63(5) | 81.88(4) |
| Agnews | 57.92(16) | 83.75(11) | 87.46(6) | 88.74(2) | 87.51(5) | 84.50(10) | 86.26(9) | 52.28(16) | 88.63(3) | **91.58(1)** | 83.46(12) | 81.56(13) | 63.96(14) | 87.15(7) | 86.37(8) | 87.84(4) |
| Amazon | 57.55(15) | 83.75(11) | 87.46(6) | 88.74(2) | 87.51(5) | 84.50(10) | 86.26(9) | 52.28(16) | 88.63(3) | **91.58(1)** | 83.46(12) | 81.56(13) | 63.96(14) | 87.15(7) | 86.37(8) | 87.84(4) |
| Imdb | 49.90(16) | 81.70(12) | 83.50(8) | 83.37(10) | 82.85(11) | 83.96(6) | 83.56(7) | 61.03(15) | 86.83(2) | **89.39(1)** | 83.50(9) | 80.53(13) | 64.42(14) | 84.38(4) | 84.11(5) | 85.43(3) |
| Yelp | 67.99(15) | 89.63(11) | 93.15(5) | 92.48(4) | 90.78(10) | 91.84(6) | 91.74(7) | 59.81(16) | 94.49(2) | **95.24(1)** | 89.00(13) | 89.07(12) | 70.94(14) | 91.63(8) | 91.61(9) | 92.41(5) |
| 20news | 58.25(15) | 84.84(5) | 59.11(14) | 86.33(3) | 85.63(4) | 80.03(11) | 83.95(6) | 56.25(16) | 86.45(2) | **86.55(1)** | 81.84(10) | 76.91(13) | 79.56(12) | 83.32(8) | 82.75(9) | 83.42(7) |

Table D17: AUCPR of 16 label-informed algorithms on 57 benchmark datasets, with labeled anomaly ratio $\gamma_l = 75\%$. We show the performance rank in parenthesis (lower the better), and mark the best performing method(s) in **bold**.

| Datasets | GANomaly | DeepSAD | REPEN | DevNet | PReNet | FEAWAD | XGBOD | NB | SVM | MLP | ResNet | FTTransformer | RF | LGB | XGB | CatB |
|---|---|---|---|---|---|---|---|---|---|---|---|---|---|---|---|---|
| ALOI | 3.90(16) | 8.77(7) | 4.49(12) | 4.19(14) | 4.17(15) | 5.82(10) | 16.99(5) | 4.25(13) | 9.13(6) | 7.17(8) | 6.56(9) | 5.05(11) | **24.07(1)** | 18.92(3) | 18.05(4) | 19.77(2) |
| annthyroid | 40.97(16) | 73.75(10) | 42.93(15) | 44.64(14) | 45.04(13) | 59.76(11) | 88.78(5) | 47.44(12) | 79.51(8) | 77.15(9) | 80.98(7) | 86.84(6) | **90.61(1)** | 89.80(4) | 90.59(2) | 90.44(3) |
| backdoor | 32.90(15) | 92.07(7) | 29.77(16) | 90.62(9) | 88.94(12) | 92.35(6) | 75.37(14) | 81.09(13) | 94.44(5) | 91.79(8) | 89.33(11) | 89.82(10) | **98.71(1)** | 95.25(4) | 95.46(3) | 97.00(2) |
| breastw | 89.92(16) | 98.37(15) | 98.79(13) | 99.47(5) | 99.48(4) | 98.80(12) | 99.29(8) | 99.43(6) | 98.95(11) | 99.53(3) | 98.60(14) | 99.28(9) | **99.57(1)** | 99.26(10) | 99.37(7) | 99.56(2) |
| campaign | 28.98(15) | 41.70(12) | 15.38(16) | 48.78(8) | 49.98(7) | 34.27(14) | **60.85(1)** | 38.41(13) | 46.84(9) | 46.58(10) | 45.04(11) | 50.01(6) | 57.47(4) | 58.70(3) | 54.62(5) | 58.93(2) |
| cardio | 49.13(15) | 91.99(9) | 95.64(3) | 91.91(10) | 91.91(10) | 91.06(13) | **96.20(1)** | 80.81(14) | 95.72(2) | 90.49(6) | 91.16(12) | 79.33(14) | 95.36(4) | 94.32(8) | 94.37(7) | 93.56(2) |
| Cardiocography | 40.53(16) | 88.95(9) | 88.93(10) | 84.29(13) | 85.47(12) | 79.88(15) | 92.45(3) | 80.81(14) | 89.52(7) | 90.49(6) | 87.42(11) | 89.39(8) | **93.58(1)** | 92.04(4) | 91.23(5) | 93.36(2) |
| celeba | 4.98(15) | 33.76(5) | 3.31(16) | 34.13(3) | 33.27(6) | 29.04(12) | **35.81(1)** | 11.20(14) | 29.05(11) | 29.80(9) | 23.20(13) | 30.74(7) | 29.12(10) | 34.23(2) | 30.63(8) | 34.07(4) |
| census | 8.76(16) | 42.39(9) | 10.32(15) | 46.79(7) | 48.61(6) | 36.75(12) | 57.11(2) | 11.94(14) | 45.59(8) | 41.23(10) | 36.85(11) | 35.01(13) | 53.87(5) | 56.36(3) | 54.24(4) | **58.57(1)** |
| cover | 0.89(16) | **98.29(1)** | 96.36(8) | 97.34(4) | 96.83(6) | 93.67(13) | 93.06(14) | 91.91(15) | 97.82(2) | 96.01(9) | 97.35(3) | 94.79(11) | 97.03(5) | 94.82(10) | 94.76(12) | 96.79(7) |
| donors | 8.39(16) | **100.00(1)** | 17.09(15) | **100.00(1)** | 98.90(13) | **100.00(1)** | 99.93(8) | 88.90(14) | **100.00(1)** | **100.00(1)** | 99.83(11) | **100.00(1)** | 99.89(10) | 99.94(7) | 99.73(12) | 99.93(9) |
| fault | 54.25(16) | 69.03(8) | 66.75(11) | 62.40(14) | 67.29(10) | 63.61(13) | 72.71(5) | 55.44(15) | 70.49(7) | 72.64(6) | 66.47(12) | 68.94(9) | 78.01(2) | 76.37(3) | 75.54(4) | **78.59(1)** |
| fraud | 42.52(12) | 46.78(9) | 38.28(14) | 59.11(3) | 36.84(16) | 55.47(6) | 45.45(10) | 21.45(15) | 43.98(11) | 51.83(8) | 39.79(13) | 58.72(5) | **62.83(1)** | 18.16(16) | 51.98(7) | 60.82(2) |
| glass | 16.40(16) | 57.61(10) | 32.30(13) | 26.43(15) | 30.01(14) | 92.66(8) | 99.36(6) | 43.85(11) | 83.26(9) | 39.82(12) | 99.32(7) | 99.70(4) | **100.00(1)** | **100.00(1)** | 99.52(5) | **100.00(1)** |
| Hepatitis | 37.56(16) | 99.62(8) | 98.96(11) | 93.59(14) | 96.74(12) | 94.23(13) | 99.59(9) | 82.09(15) | 99.34(10) | **100.00(1)** | **100.00(1)** | **100.00(1)** | **100.00(1)** | **100.00(1)** | **100.00(1)** | **100.00(1)** |
| http | 73.48(14) | **100.00(1)** | 93.99(12) | **100.00(1)** | **100.00(1)** | 90.00(13) | 96.68(9) | 96.68(9) | 66.82(15) | **100.00(1)** | **100.00(1)** | **100.00(1)** | 99.44(8) | 63.50(16) | **100.00(1)** | 94.12(11) |
| InternetAds | 51.82(14) | 88.69(8) | 91.96(6) | 92.19(2) | 91.55(4) | 85.33(10) | 89.37(7) | 47.20(15) | 91.43(5) | 84.36(12) | 84.81(11) | N/A(N/A) | 84.05(13) | 87.54(9) | 90.22(6) | **92.58(1)** |
| Ionosphere | 91.39(15) | 98.06(6) | 97.97(9) | 94.01(13) | 93.95(14) | 98.02(7) | **99.41(1)** | 88.57(16) | 97.58(11) | 97.67(10) | 97.07(12) | 98.02(8) | 99.07(3) | 98.98(4) | 99.11(2) | 98.91(5) |
| landsat | 24.48(16) | 85.55(6) | 42.47(14) | 57.03(13) | 57.85(12) | 60.29(11) | 87.66(4) | 29.25(15) | 82.06(9) | 82.33(8) | 78.22(10) | 84.05(7) | 89.20(2) | 88.50(3) | 87.56(5) | **89.57(1)** |
| letter | 16.62(16) | 37.95(11) | 48.59(9) | 31.84(13) | 33.81(12) | 28.78(14) | 62.10(7) | 18.44(15) | 50.94(8) | 48.15(10) | 68.24(2) | 65.14(4) | 52.29(3) | 62.82(5) | 62.56(6) | **74.19(1)** |
| Lymphography | 76.38(16) | **100.00(1)** | **100.00(1)** | **100.00(1)** | 100.00(14) | 88.07(15) | **100.00(1)** | 89.61(15) | **100.00(1)** | **100.00(1)** | **100.00(1)** | **100.00(1)** | **100.00(1)** | **100.00(1)** | **100.00(1)** | **100.00(1)** |
| magic.gamma | 44.43(16) | 86.61(7) | 77.37(11) | 73.85(14) | 75.38(13) | 77.37(12) | 87.18(5) | 69.45(15) | 87.08(6) | 87.58(4) | 82.94(10) | 83.75(9) | 88.34(2) | 87.71(3) | 85.72(8) | **88.99(1)** |
| mammography | 17.29(16) | 66.18(7) | 62.40(10) | 62.24(11) | 61.19(12) | 58.87(13) | 66.55(6) | 48.60(15) | 56.34(14) | 63.16(9) | 68.39(4) | 71.68(2) | 69.46(3) | 65.80(8) | 66.92(5) | **74.01(1)** |
| mnist | 16.50(16) | 96.41(6) | 95.20(9) | 90.45(13) | 92.99(10) | 78.75(14) | 95.88(7) | 68.97(15) | 97.39(4) | 95.39(8) | 91.97(11) | 91.11(12) | 97.61(2) | 97.48(3) | 96.69(5) | **97.76(1)** |
| musk | **100.00(1)** | 99.72(2) | 99.69(5) | 99.63(4) | 99.67(3) | 99.37(10) | 99.38(8) | 29.78(15) | **100.00(1)** | 99.59(6) | 97.47(14) | 98.57(13) | 99.38(9) | **100.00(1)** | **100.00(1)** | 99.13(11) |
| optdigits | 3.23(16) | 99.72(2) | 99.69(5) | 99.63(4) | 99.37(10) | 99.38(8) | 99.38(8) | 29.78(15) | **100.00(1)** | 99.59(6) | 97.47(14) | 98.57(13) | 99.38(9) | 99.45(7) | 99.07(12) | 99.13(11) |
| PageBlocks | 47.44(16) | 87.97(8) | 79.11(11) | 67.37(14) | 69.38(13) | 74.23(12) | 88.50(6) | 55.98(15) | 85.16(9) | 84.10(10) | 89.25(5) | 89.84(3) | 90.23(2) | 89.47(4) | 88.25(7) | **91.76(1)** |
| pendigits | 5.40(16) | 99.70(3) | 97.55(9) | 93.20(15) | 94.63(13) | 96.28(12) | 99.01(5) | 94.11(14) | **99.89(1)** | 98.72(8) | 97.23(11) | 97.49(10) | 99.72(2) | 98.91(7) | 99.01(6) | 99.40(4) |
| Pima | 47.65(16) | 69.97(14) | 61.29(15) | 70.81(12) | 70.85(11) | 73.46(10) | 83.70(5) | 70.78(13) | 77.73(8) | 75.04(9) | 78.85(7) | 78.85(6) | **90.24(1)** | 88.13(2) | 87.26(4) | 88.08(3) |
| satellite | 67.69(16) | 92.78(6) | 80.24(13) | 82.61(12) | 83.15(11) | 80.19(14) | 83.42(5) | 75.71(15) | 91.64(7) | 87.99(10) | 88.68(9) | 89.60(8) | **95.05(1)** | 94.01(3) | 93.45(4) | 94.90(2) |
| satimage-2 | 50.10(16) | **96.19(1)** | 93.81(7) | 93.58(8) | 91.96(11) | 93.19(10) | 95.49(2) | 91.50(12) | 93.24(9) | 89.32(15) | 91.40(13) | 89.70(14) | 94.61(4) | 94.85(3) | 94.44(6) | 94.55(5) |
| shuttle | 58.59(16) | 98.78(6) | 97.11(10) | 96.47(13) | 96.43(13) | 96.73(11) | 99.64(5) | 96.21(14) | 97.69(8) | 96.18(15) | 98.19(7) | 99.22(2) | 99.68(4) | 99.66(5) | **99.70(1)** | 99.75(3) |
| skin | 22.52(16) | 99.56(6) | 49.51(15) | 70.60(13) | 67.45(14) | 89.78(11) | **99.86(1)** | 85.56(12) | 99.33(8) | 99.82(2) | 99.15(10) | 99.22(9) | 99.68(4) | 99.66(5) | 99.50(7) | 99.78(3) |
| smtp | 20.87(15) | 67.14(4) | 66.71(5) | 66.69(6) | 66.68(8) | 36.14(12) | 50.30(10) | 16.71(16) | 66.69(7) | 50.03(11) | **100.00(1)** | 34.50(13) | 83.33(2) | 33.37(14) | 50.79(9) | 83.33(2) |
| SpamBase | 45.24(16) | 92.82(10) | 88.57(13) | 86.87(14) | 90.32(12) | 91.44(11) | 94.78(5) | 68.56(15) | 95.47(6) | 95.25(7) | 93.36(8) | 92.99(9) | 96.63(4) | 97.03(2) | 96.56(3) | **97.49(1)** |
| speech | 1.63(16) | 5.82(14) | 35.94(2) | 18.27(7) | 17.74(9) | 20.91(5) | 13.09(13) | 18.11(8) | **42.05(1)** | 24.00(4) | 32.06(3) | 17.56(10) | 5.46(15) | 13.23(12) | 18.90(6) | 15.15(11) |
| Stamps | 34.54(16) | 80.17(14) | 85.33(11) | 77.21(15) | 84.96(12) | 82.52(13) | **99.21(2)** | 88.76(10) | 94.68(8) | 90.90(9) | 96.32(7) | 97.40(6) | 96.64(4) | 96.39(5) | 95.28(5) | 97.16(2) |
| thyroid | 55.50(16) | 99.73(4) | 92.90(9) | 94.01(7) | 92.34(10) | 93.56(8) | **98.29(1)** | 90.46(15) | 92.22(11) | 92.01(12) | 91.82(13) | 96.07(4) | 96.59(3) | 95.28(5) | 95.15(6) | 97.16(2) |
| vertebral | 11.30(16) | 69.15(9) | 50.63(13) | 46.16(15) | 46.54(14) | 68.11(10) | 88.11(4) | 51.25(12) | 71.68(8) | 52.58(11) | 94.93(6) | 92.60(7) | 98.42(2) | 98.18(3) | 97.72(5) | **98.71(1)** |
| vowels | 28.43(16) | 77.70(13) | 92.72(5) | 89.35(8) | 91.77(7) | 86.37(11) | 88.18(10) | 69.26(15) | 96.83(2) | 72.65(14) | 94.98(3) | **97.38(1)** | 82.28(12) | 92.44(6) | 88.19(9) | 94.09(4) |
| Waveform | 4.47(16) | 47.70(6) | 19.72(15) | 19.83(14) | 24.14(11) | 23.36(12) | 51.91(3) | 22.56(13) | **66.56(1)** | 53.82(2) | 40.45(10) | 46.09(8) | 43.72(9) | 50.61(5) | 46.28(7) | 51.39(4) |
| WBC | 48.77(16) | 88.97(12) | 89.93(11) | 94.77(9) | 84.18(15) | 95.14(7) | 95.80(5) | 88.35(13) | 94.55(10) | 84.55(14) | 98.10(2) | 95.90(4) | **98.46(1)** | 95.19(6) | 95.00(8) | 97.04(3) |
| WDBC | 61.66(16) | 99.63(11) | **100.00(1)** | **100.00(1)** | **100.00(1)** | **100.00(1)** | 96.99(15) | **100.00(1)** | **100.00(1)** | **100.00(1)** | **100.00(1)** | **100.00(1)** | 99.54(12) | 98.61(14) | 99.07(13) | **100.00(1)** |
| Wilt | 4.81(16) | 84.42(9) | 7.11(15) | 8.19(14) | 8.49(13) | 55.31(10) | 88.49(3) | 40.55(11) | 88.02(5) | 24.46(12) | 87.91(6) | **96.92(1)** | 89.87(2) | 85.82(8) | 86.32(7) | 88.17(4) |
| wine | 24.84(16) | 99.85(15) | **100.00(1)** | **100.00(1)** | **100.00(1)** | **100.00(1)** | **100.00(1)** | **100.00(1)** | **100.00(1)** | **100.00(1)** | **100.00(1)** | **100.00(1)** | **100.00(1)** | **100.00(1)** | **100.00(1)** | **100.00(1)** |
| WPBC | 23.91(16) | 86.09(9) | 64.58(14) | 69.31(13) | 70.04(12) | 84.92(10) | 97.86(5) | 48.03(15) | 87.85(8) | 81.52(11) | 96.25(7) | 97.43(6) | 98.11(2) | **98.76(1)** | 98.04(3) | 98.04(4) |
| yeast | 33.54(16) | 51.46(11) | 36.34(15) | 49.24(12) | 48.99(13) | 54.83(7) | 60.76(5) | 46.99(14) | 52.52(9) | 51.54(10) | 55.07(6) | 54.38(8) | 63.63(2) | 61.78(3) | 61.63(4) | **65.46(1)** |
| CIFAR10 | 9.42(16) | 31.80(11) | **41.15(1)** | 38.42(4) | 36.93(8) | 33.51(9) | 38.04(7) | 9.52(15) | 38.16(6) | 32.83(10) | 30.76(12) | 24.34(13) | 19.97(14) | 39.16(3) | 38.24(5) | 39.37(2) |
| FashionMNIST | 22.73(16) | 81.04(9) | 84.13(3) | 82.96(6) | 82.05(7) | 69.36(14) | 83.07(5) | 28.08(15) | 81.15(8) | 76.82(10) | 76.37(11) | 70.18(13) | 72.64(12) | 84.54(2) | 83.87(4) | **84.83(1)** |
| MNIST-C | 17.18(16) | 86.53(5) | 42.39(14) | 86.58(4) | 87.02(2) | 83.09(11) | 84.95(10) | 28.38(15) | **88.68(1)** | 86.65(3) | 85.02(9) | 80.84(13) | 82.07(12) | 86.77(7) | 86.34(6) | 85.82(8) |
| MVTec-AD | 58.22(14) | 93.06(6) | 56.17(16) | 87.71(11) | 85.56(13) | 88.56(10) | 95.54(5) | 56.50(15) | 86.81(12) | 90.24(9) | 92.72(7) | 91.40(8) | 96.14(2) | 96.08(4) | 96.14(3) | **96.80(1)** |
| SVHN | 7.89(15) | 28.91(11) | 34.00(3) | 34.17(2) | **34.87(1)** | 27.04(12) | 31.61(8) | 5.39(16) | 33.62(4) | 30.09(9) | 29.51(10) | 20.43(13) | 15.43(14) | 33.54(5) | 32.96(6) | 32.53(7) |
| Agnews | 6.23(16) | 70.02(4) | 64.27(6) | 55.82(12) | 58.75(11) | 61.56(9) | 59.97(10) | 8.77(15) | 71.93(2) | **74.72(1)** | 70.14(3) | 53.18(13) | 22.80(14) | 62.20(7) | 61.82(8) | 64.27(5) |
| Amazon | 6.15(15) | 35.79(3) | 32.58(4) | 32.03(5) | 31.44(7) | 26.90(12) | 28.22(11) | 5.24(16) | 41.67(2) | **43.32(1)** | 31.75(6) | 25.48(13) | 8.50(14) | 30.58(9) | 29.92(10) | 30.85(8) |
| Imdb | 5.00(16) | 34.28(4) | 29.92(8) | 27.08(11) | 26.50(12) | 28.58(10) | 28.68(9) | 7.75(15) | 38.36(2) | **44.09(1)** | 35.61(3) | 21.04(13) | 8.67(14) | 30.36(7) | 31.13(6) | 31.46(5) |
| Yelp | 9.02(15) | 56.25(3) | 53.63(4) | 49.62(7) | 50.44(5) | 46.06(11) | 46.38(9) | 6.52(16) | **64.11(1)** | 63.68(2) | 44.52(12) | 41.14(13) | 13.27(14) | 47.41(8) | 50.00(6) | 50.22(6) |
| 20news | 7.38(14) | 44.78(4) | 7.35(15) | 40.42(7) | 43.69(5) | 42.82(6) | 39.74(8) | 6.08(16) | 50.47(2) | **54.33(1)** | 45.62(3) | 35.25(11) | 31.17(13) | 33.29(12) | 39.34(9) | 38.83(10) |

Table D18: AUCROC of 16 label-informed algorithms on 57 benchmark datasets, with labeled anomaly ratio $\gamma_l = 100\%$. We show the performance rank in parenthesis (lower the better), and mark the best performing method(s) in **bold**.

| Datasets | GANomaly | DeepSAD | REPEN | DevNet | PReNet | FEAWAD | XGBOD | NB | SVM | MLP | ResNet | FTTransformer | RF | LGB | XGB | CatB |
|---|---|---|---|---|---|---|---|---|---|---|---|---|---|---|---|---|
| ALOI | 57.04(12) | 68.95(6) | 52.22(16) | 53.84(14) | 53.32(15) | 65.82(7) | 82.38(2) | 54.35(13) | 62.88(9) | 64.54(8) | 60.40(10) | 57.47(11) | **82.85(1)** | 79.37(3) | 76.55(5) | 78.54(4) |
| annthyroid | 82.58(16) | 98.17(10) | 84.54(13) | 82.70(15) | 82.81(14) | 89.96(12) | 99.57(3) | 91.67(11) | 98.72(8) | 98.62(9) | 98.98(7) | 99.13(6) | **99.60(1)** | 99.56(4) | 99.57(2) | 99.55(5) |
| backdoor | 88.57(16) | 99.54(5) | 89.65(15) | 98.32(12) | 95.39(14) | 98.73(9) | 99.20(7) | 97.53(13) | 98.62(10) | 99.69(4) | 98.94(8) | 98.50(11) | **99.97(1)** | 99.47(6) | 99.87(3) | 99.97(2) |
| breastw | 97.28(16) | 99.82(9) | 99.01(14) | 99.72(11) | 99.77(10) | 99.10(13) | 99.90(5) | 98.87(15) | 99.66(12) | 99.82(8) | **99.95(1)** | 99.86(7) | 99.92(4) | 99.92(3) | 99.87(6) | 99.93(2) |
| campaign | 57.15(16) | 84.72(12) | 58.03(15) | 87.35(11) | 88.73(9) | 79.81(14) | 93.74(3) | 82.61(13) | 87.84(10) | 89.10(8) | 89.30(7) | 91.27(6) | 92.79(5) | 93.79(2) | 93.35(4) | **93.93(1)** |
| cardio | 87.96(15) | 99.20(10) | 99.42(9) | 98.96(11) | 98.87(12) | 98.21(13) | 99.81(2) | 97.64(14) | 99.71(4) | 99.44(8) | N/A(N/A) | **99.82(1)** | 99.73(3) | 99.64(5) | 99.49(7) | 99.61(6) |
| Cardiotocography | 64.62(16) | 96.95(9) | 96.09(11) | 94.82(13) | 95.38(12) | 94.44(14) | 98.14(6) | 93.75(15) | 96.87(10) | 97.21(8) | 98.01(7) | 98.20(5) | 98.33(3) | 98.54(2) | 98.28(4) | **98.57(1)** |
| celeba | 64.22(15) | 95.09(9) | 57.68(16) | 96.12(4) | 95.88(6) | 94.48(10) | **96.61(1)** | 90.51(14) | 94.36(11) | 96.03(5) | 91.21(13) | 95.45(8) | 92.89(12) | 96.50(2) | 95.70(7) | 96.18(3) |
| census | 63.13(16) | 87.32(12) | 69.76(15) | 92.54(6) | 91.64(7) | 87.75(11) | 93.80(2) | 71.46(14) | 91.23(8) | 90.24(9) | 82.49(13) | 88.76(10) | 92.81(5) | 93.56(3) | 93.16(4) | **94.05(1)** |
| cover | 46.16(16) | 99.97(7) | 99.95(11) | 99.96(9) | 99.96(10) | 99.91(13) | 99.89(15) | 99.90(14) | 99.98(5) | 99.97(8) | 99.99(2) | **99.99(1)** | 99.98(6) | 99.98(4) | 99.99(12) | 99.98(3) |
| donors | 50.22(16) | **100.00(1)** | 82.80(15) | **100.00(1)** | 99.95(13) | **100.00(1)** | 100.00(10) | 99.60(14) | **100.00(1)** | **100.00(1)** | **100.00(1)** | **100.00(1)** | 100.00(9) | **100.00(1)** | 100.00(12) | **100.00(1)** |
| fault | 68.47(16) | 83.57(8) | 78.55(12) | 77.29(13) | 79.21(11) | 76.51(14) | 85.29(6) | 70.44(15) | 82.56(10) | 84.49(7) | 83.31(9) | 87.41(5) | 89.22(3) | 89.64(2) | 88.65(4) | **89.80(1)** |
| fraud | 90.90(12) | 92.87(5) | 90.95(11) | 91.19(10) | 92.20(7) | 94.66(3) | 95.74(2) | 88.64(13) | 91.31(9) | 84.01(14) | 80.31(15) | 92.75(6) | 92.09(8) | 32.09(16) | **96.41(1)** | 94.46(4) |
| glass | 68.90(16) | 98.51(10) | 89.45(14) | 88.78(15) | 90.44(13) | **100.00(1)** | 95.64(2) | 93.70(11) | 99.47(9) | 92.67(12) | 99.73(8) | **100.00(1)** | **100.00(1)** | **100.00(1)** | **100.00(1)** | **100.00(1)** |
| Hepatitis | 75.90(16) | **100.00(1)** | 99.72(12) | 99.09(14) | 99.45(13) | 99.93(10) | 99.85(11) | 97.41(15) | **100.00(1)** | **100.00(1)** | **100.00(1)** | **100.00(1)** | **100.00(1)** | **100.00(1)** | **100.00(1)** | **100.00(1)** |
| http | 99.80(15) | 96.50(7) | 96.49(8) | 97.11(3) | 97.16(2) | 94.14(13) | 95.75(9) | 85.96(14) | 95.28(11) | **97.63(1)** | 97.00(5) | N/A(N/A) | 97.04(4) | 95.14(12) | 95.37(10) | 96.91(6) |
| InternetAds | 70.14(15) | 99.25(8) | 98.28(11) | 94.76(14) | 94.88(13) | 98.12(12) | **100.00(1)** | 93.96(15) | 98.92(10) | 99.07(9) | 99.98(6) | N/A(N/A) | 100.00(3) | 99.98(5) | 100.00(2) | 99.94(4) |
| Ionosphere | 93.90(16) | 99.25(8) | 98.29(11) | 94.76(14) | 94.88(13) | 98.12(12) | **100.00(1)** | 93.96(15) | 98.92(10) | 99.07(9) | 99.98(6) | 99.56(7) | 100.00(3) | 99.98(5) | 100.00(2) | 99.94(4) |
| landsat | 57.87(16) | 95.62(8) | 60.98(15) | 80.41(12) | 80.31(13) | 86.86(11) | 95.89(7) | 71.88(14) | 94.77(9) | 94.52(10) | 96.35(5) | 96.27(6) | 96.62(4) | 96.78(3) | 96.79(2) | **96.88(1)** |
| letter | 70.88(16) | 87.01(12) | 90.09(9) | 87.70(11) | 86.17(13) | 84.35(14) | 94.93(4) | 75.85(15) | 90.82(8) | 89.59(10) | 94.80(5) | 91.41(7) | 96.64(2) | 96.29(3) | 94.02(6) | **97.30(1)** |
| Lymphography | 98.40(16) | **100.00(1)** | **100.00(1)** | **100.00(1)** | **100.00(1)** | **100.00(1)** | **100.00(1)** | 99.59(15) | **100.00(1)** | **100.00(1)** | **100.00(1)** | **100.00(1)** | **100.00(1)** | **100.00(1)** | **100.00(1)** | **100.00(1)** |
| magic.gamma | 56.75(16) | 91.45(8) | 83.17(14) | 83.36(12) | 83.31(13) | 86.11(11) | 91.68(6) | 75.84(15) | 90.60(10) | 91.17(9) | 91.60(7) | 91.95(5) | 92.49(3) | 92.68(2) | 92.37(4) | **93.15(1)** |
| mammography | 76.95(16) | 95.75(5) | 92.88(14) | 93.14(12) | 93.24(11) | 95.84(2) | 95.64(4) | 92.96(13) | 87.43(15) | 95.17(5) | 93.85(8) | 95.06(6) | 94.72(7) | 93.49(10) | 93.63(9) | **95.95(1)** |
| mnist | 72.30(16) | 99.68(8) | 99.29(11) | 98.99(13) | 99.25(12) | 98.08(14) | 99.73(6) | 95.35(15) | 99.69(7) | 99.75(5) | 99.49(10) | 99.59(9) | 99.81(4) | 99.82(2) | 99.81(3) | **99.83(1)** |
| musk | **100.00(1)** | **100.00(1)** | **100.00(1)** | **100.00(1)** | **100.00(1)** | **100.00(1)** | **100.00(1)** | **100.00(1)** | **100.00(1)** | **100.00(1)** | **100.00(1)** | **100.00(1)** | **100.00(1)** | **100.00(1)** | **100.00(1)** | **100.00(1)** |
| optdigits | 51.61(16) | **100.00(1)** | 99.98(12) | 99.98(10) | 99.99(9) | 99.97(13) | 99.99(9) | 94.43(15) | **100.00(1)** | 99.97(14) | **100.00(1)** | 100.00(3) | 99.98(11) | 100.00(4) | 99.99(7) | 100.00(6) |
| PageBlocks | 79.86(16) | 98.51(8) | 95.66(12) | 89.24(15) | 90.83(14) | 96.00(11) | 99.01(5) | 93.31(13) | 96.88(10) | 97.29(9) | 98.53(7) | 98.72(6) | 99.02(4) | 99.25(2) | 99.13(3) | **99.31(1)** |
| pendigits | 57.49(16) | 99.99(3) | 99.88(10) | 99.79(12) | 99.77(14) | 99.84(11) | 99.97(7) | 99.61(15) | **100.00(1)** | 99.94(8) | 99.99(5) | 99.98(6) | 99.98(4) | 99.93(9) | 99.96(6) | 100.00(2) |
| Pima | 66.35(16) | 85.52(10) | 77.22(15) | 78.37(14) | 83.44(13) | 83.13(14) | 92.44(5) | 83.66(12) | 87.12(8) | 85.59(9) | 86.26(7) | 90.29(6) | **96.12(1)** | 93.96(3) | 93.32(4) | 94.17(2) |
| satellite | 80.79(15) | 96.74(7) | 81.89(14) | 83.58(13) | 83.96(12) | 88.24(11) | 94.72(8) | 78.28(16) | 95.53(9) | 94.38(10) | 96.87(6) | 97.32(5) | 97.41(3) | 97.41(4) | **97.51(1)** | 97.49(2) |
| satimage-2 | 98.57(11) | 99.68(2) | 99.53(3) | 99.50(4) | 99.47(5) | 98.19(15) | 99.22(8) | 98.50(12) | 99.17(8) | 97.94(16) | 99.22(7) | 99.82(10) | 99.33(14) | 98.42(13) | 98.87(9) | **99.76(1)** |
| shuttle | 96.34(16) | 99.52(7) | 98.58(11) | 97.57(15) | 97.58(14) | 97.82(13) | 100.00(3) | 99.13(9) | 99.02(10) | 98.20(12) | 99.71(6) | 99.13(8) | 100.00(2) | 100.00(5) | 100.00(4) | **100.00(1)** |
| skin | 51.02(16) | 99.84(10) | 88.75(15) | 96.19(12) | 95.34(13) | 98.71(11) | 99.99(3) | 93.93(14) | 99.95(9) | 99.96(8) | 99.98(5) | 99.97(6) | **99.99(1)** | 99.98(3) | 99.96(7) | 99.98(4) |
| smtp | 55.30(16) | 92.23(5) | 91.98(8) | 85.48(10) | 78.88(13) | 85.30(11) | 98.76(6) | 59.01(14) | 84.13(12) | 86.42(9) | **100.00(1)** | 95.83(7) | 99.99(2) | 56.21(15) | 99.70(4) | 99.99(2) |
| SpamBase | 62.20(16) | 97.20(9) | 94.74(12) | 92.29(14) | 94.11(13) | 96.37(11) | 98.18(4) | 86.42(15) | 97.12(10) | 97.88(7) | 97.68(8) | 97.92(6) | 98.01(5) | 98.01(5) | 98.43(3) | **98.51(1)** |
| speech | 47.79(16) | 65.84(14) | 81.01(5) | 85.35(2) | **85.67(1)** | 76.97(9) | 79.04(7) | 75.88(11) | 84.43(3) | 81.33(4) | 73.54(13) | 76.33(10) | 63.62(15) | 74.82(12) | 80.90(6) | 78.78(8) |
| Stamps | 78.11(16) | 99.05(12) | 98.97(13) | 98.54(15) | 99.09(11) | 99.31(9) | 99.91(5) | 98.97(14) | 99.65(8) | 99.23(10) | 99.96(4) | 99.88(7) | 99.99(3) | **100.00(1)** | 99.89(6) | **100.00(1)** |
| thyroid | 93.19(16) | 99.85(8) | 99.85(8) | 99.81(9) | 99.79(10) | 99.76(13) | 99.96(3) | 99.69(14) | 99.78(12) | 99.78(11) | 99.87(7) | 99.93(6) | 99.97(2) | 99.93(5) | 99.96(4) | **99.97(1)** |
| vertebral | 43.13(16) | 93.38(8) | 82.43(13) | 81.08(15) | 82.18(14) | 93.04(9) | **99.64(1)** | 86.48(12) | 91.50(10) | 86.88(11) | 99.20(5) | 99.01(7) | 99.43(3) | 99.02(6) | 99.34(4) | 99.43(2) |
| vowels | 80.74(16) | 98.07(13) | 99.45(7) | 99.03(9) | 98.92(10) | 99.29(8) | 98.39(12) | 97.86(14) | **99.98(1)** | 96.29(15) | 99.97(2) | 99.92(3) | 98.72(11) | 99.68(5) | 99.65(6) | 99.80(4) |
| Waveform | 53.00(16) | 93.02(13) | 93.01(14) | 93.40(10) | 93.97(5) | 93.15(12) | 93.75(7) | 91.21(14) | 95.36(2) | **96.84(1)** | 81.72(15) | 93.85(6) | 93.53(9) | 94.81(4) | 93.62(8) | 94.86(3) |
| WBC | 96.14(16) | 98.92(12) | 99.17(11) | 99.56(10) | 98.88(14) | 99.71(9) | **100.00(1)** | 98.55(15) | **100.00(1)** | 98.92(13) | **100.00(1)** | **100.00(1)** | **100.00(1)** | **100.00(1)** | **100.00(1)** | **100.00(1)** |
| WDBC | 98.00(16) | **100.00(1)** | **100.00(1)** | **100.00(1)** | **100.00(1)** | **100.00(1)** | **100.00(1)** | **100.00(1)** | **100.00(1)** | **100.00(1)** | **100.00(1)** | **100.00(1)** | **100.00(1)** | **100.00(1)** | **100.00(1)** | **100.00(1)** |
| Wilt | 46.78(16) | 98.90(6) | 58.15(15) | 68.16(14) | 68.68(13) | 84.69(11) | 99.14(2) | 83.03(12) | **99.18(1)** | 86.66(10) | 99.14(3) | 98.86(7) | 98.49(9) | 98.83(8) | 99.07(4) | 99.00(5) |
| wine | 80.09(16) | **100.00(1)** | **100.00(1)** | **100.00(1)** | **100.00(1)** | **100.00(1)** | **100.00(1)** | **100.00(1)** | **100.00(1)** | **100.00(1)** | **100.00(1)** | **100.00(1)** | **100.00(1)** | **100.00(1)** | **100.00(1)** | **100.00(1)** |
| WPBC | 51.56(16) | 96.79(8) | 81.07(14) | 85.62(12) | 84.19(13) | 94.44(10) | 99.93(5) | 72.04(15) | 95.34(9) | 92.14(11) | 99.47(7) | 99.88(6) | **100.00(1)** | **100.00(1)** | 99.96(4) | 99.98(3) |
| yeast | 49.03(15) | 72.41(8) | 46.92(16) | 86.92(5) | 87.40(2) | 85.33(8) | 82.92(10) | 67.53(14) | 71.70(9) | 71.15(10) | 73.88(7) | 75.64(6) | 78.36(2) | 77.69(3) | 77.11(4) | **79.41(1)** |
| CIFAR10 | 62.48(16) | 82.71(11) | 86.92(5) | 87.40(2) | 85.33(8) | 82.92(10) | 87.37(3) | 69.33(15) | 86.12(7) | 83.03(9) | 78.88(12) | 77.50(13) | 75.75(14) | 87.32(4) | 86.61(6) | **87.63(1)** |
| FashionMNIST | 80.64(15) | 96.64(7) | 96.89(4) | 96.92(3) | 96.45(8) | 96.03(9) | 96.81(5) | 77.49(16) | 94.73(12) | 95.15(11) | 95.39(10) | 93.51(13) | 93.22(14) | **96.99(1)** | 96.99(2) | 96.66(6) |
| MNIST-C | 72.81(16) | 96.11(3) | 83.01(14) | 95.98(6) | 95.66(10) | 95.19(11) | 96.05(5) | 77.29(15) | **96.30(1)** | 95.95(8) | 95.96(7) | 94.23(13) | 94.91(12) | 96.10(4) | 96.19(2) | 95.67(9) |
| MVTec-AD | 73.44(16) | 98.69(6) | 74.12(15) | 94.52(11) | 93.20(13) | 95.74(10) | 98.95(5) | 83.69(14) | 94.36(12) | 97.02(9) | 98.55(7) | 97.42(8) | **99.28(1)** | 99.18(3) | 98.97(4) | 99.20(2) |
| SVHN | 58.26(15) | 80.02(9) | 83.13(3) | **84.16(1)** | 83.40(2) | 78.62(12) | 81.98(8) | 54.97(16) | 82.57(7) | 78.98(11) | 79.20(10) | 72.36(13) | 70.04(14) | 82.71(5) | 82.72(4) | 82.59(6) |
| Agnews | 57.95(16) | 94.00(5) | 93.30(8) | 92.37(11) | 92.30(12) | 92.39(10) | 93.10(9) | 68.87(15) | 94.56(3) | **96.49(1)** | 95.24(2) | 91.87(13) | 80.80(14) | 93.87(7) | 93.98(6) | 94.04(4) |
| Amazon | 56.66(15) | 83.14(12) | 88.70(4) | 89.03(3) | 88.25(6) | 86.41(10) | 87.50(9) | 52.06(16) | 91.50(2) | **92.11(1)** | 85.47(11) | 82.30(13) | 66.75(14) | 87.99(7) | 87.62(8) | 88.41(5) |
| Imdb | 50.05(16) | 81.07(12) | 82.83(10) | 83.79(9) | 82.49(11) | 84.52(8) | 84.66(7) | 66.93(15) | 88.40(2) | **89.88(1)** | 86.51(3) | 80.52(13) | 67.50(14) | 85.54(5) | 84.99(6) | 86.03(4) |
| Yelp | 68.90(15) | 91.06(11) | 93.29(3) | 92.18(6) | 92.05(9) | 91.19(10) | 92.18(7) | 60.09(16) | 95.29(2) | **95.65(1)** | 90.60(12) | 89.17(13) | 76.25(14) | 92.65(5) | 92.12(8) | 92.77(4) |
| 20news | 58.18(15) | 85.39(8) | 59.28(14) | 86.67(3) | 86.50(4) | 83.50(11) | 86.18(5) | 56.21(16) | **88.36(1)** | 86.86(2) | 83.72(10) | 77.86(13) | 82.04(12) | 85.44(7) | 86.04(6) | 85.34(9) |

Table D19: AUCPR of 16 label-informed algorithms on 57 benchmark datasets, with labeled anomaly ratio $\gamma_l = 100\%$. We show the performance rank in parenthesis (lower the better), and mark the best performing method(s) in **bold**.

| Datasets | GANomaly | DeepSAD | REPEN | DevNet | PReNet | FEAWAD | XGBOD | NB | SVM | MLP | ResNet | FTTransformer | RF | LGB | XGB | CatB |
|---|---|---|---|---|---|---|---|---|---|---|---|---|---|---|---|---|
| ALOI | 3.92(16) | 9.31(7) | 4.57(12) | 4.30(13) | 4.19(15) | 6.02(10) | 18.12(5) | 4.21(14) | 11.46(6) | 9.06(8) | 7.68(9) | 5.01(11) | **31.35(1)** | 26.05(2) | 23.15(4) | 25.44(3) |
| annthyroid | 45.77(13) | 78.04(10) | 45.07(15) | 45.35(14) | 44.64(16) | 53.95(12) | 93.23(2) | 60.64(11) | 80.97(8) | 79.61(9) | 85.09(7) | 86.50(6) | **93.25(1)** | 92.97(3) | 92.44(4) | 92.36(5) |
| backdoor | 39.08(15) | 96.79(5) | 29.75(16) | 91.06(11) | 89.83(12) | 91.65(10) | 85.58(13) | 84.92(14) | 95.00(8) | 95.84(7) | 96.30(6) | 92.72(9) | **99.33(1)** | 98.22(3) | 97.87(4) | 99.32(2) |
| breastw | 94.78(16) | 99.66(9) | 97.83(14) | 99.46(11) | 99.56(10) | 98.91(13) | 99.81(5) | 96.36(15) | 99.22(12) | 99.67(8) | **99.90(1)** | 99.71(7) | 99.83(4) | 99.85(3) | 99.74(6) | 99.85(2) |
| campaign | 20.82(15) | 46.45(12) | 15.39(16) | 49.78(11) | 50.98(9) | 37.76(14) | **61.45(1)** | 38.67(13) | 51.95(7) | 51.60(8) | 50.33(10) | 54.05(6) | 60.31(3) | 59.83(4) | 58.63(5) | 60.96(2) |
| cardio | 53.07(15) | 95.14(10) | 96.27(9) | 92.91(13) | 93.03(12) | 94.60(11) | 98.46(2) | 81.01(14) | 97.85(3) | 96.40(8) | N/A(N/A) | **98.65(1)** | 97.79(4) | 97.41(5) | 93.56(6) | 95.38(1) |
| Cardiotocography | 44.34(16) | 90.49(9) | 88.97(11) | 82.95(14) | 85.97(12) | 83.16(13) | 94.35(5) | 79.80(15) | 89.57(10) | 91.69(6) | 93.19(7) | 94.66(4) | 95.12(3) | 95.21(2) | 93.50(6) | **95.38(1)** |
| celeba | 6.93(15) | 37.04(2) | 3.31(16) | 35.50(4) | 34.49(6) | 26.01(12) | **37.10(1)** | 11.63(14) | 32.21(9) | 31.10(11) | 24.68(13) | 33.37(7) | 32.08(10) | 34.64(5) | 33.02(8) | 36.36(3) |
| census | 8.46(16) | 48.55(9) | 10.35(15) | 48.11(10) | 49.12(8) | 33.77(13) | 60.70(4) | 10.97(14) | 53.14(6) | 49.79(7) | 42.95(12) | 45.35(11) | 57.07(5) | 61.81(2) | 60.95(3) | **63.13(1)** |
| cover | 0.92(16) | 98.18(6) | 96.54(13) | 97.44(9) | 97.25(12) | 94.12(14) | 97.63(8) | 92.12(15) | 98.27(5) | 97.36(11) | 98.75(2) | **98.95(1)** | 98.18(7) | 98.37(4) | 97.41(10) | 98.45(3) |
| donors | 6.87(16) | **100.00(1)** | 17.07(15) | **100.00(1)** | 98.80(13) | 100.00(11) | 100.00(10) | 88.90(14) | **100.00(1)** | **100.00(1)** | **100.00(1)** | **100.00(1)** | 100.00(9) | **100.00(1)** | 99.96(12) | **100.00(1)** |
| fault | 55.63(16) | 73.77(10) | 69.59(11) | 64.79(13) | 68.92(12) | 64.77(14) | 77.52(6) | 57.44(15) | 74.86(9) | 75.47(8) | 76.60(7) | 79.65(5) | 82.69(3) | 83.41(2) | 81.64(4) | **83.70(1)** |
| fraud | 42.84(13) | 51.77(8) | 38.40(14) | 59.09(2) | 58.72(3) | 57.48(5) | 45.83(12) | 21.73(15) | 51.19(9) | 54.51(7) | 48.85(10) | 54.99(6) | **62.77(1)** | 15.52(16) | 47.34(11) | 58.67(4) |
| glass | 16.49(16) | 68.14(10) | 32.12(13) | 26.89(15) | 30.84(14) | **100.00(1)** | 99.36(7) | 81.98(12) | 84.02(9) | 42.64(11) | 93.38(8) | **100.00(1)** | **100.00(1)** | **100.00(1)** | **100.00(1)** | **100.00(1)** |
| Hepatitis | 43.83(16) | **100.00(1)** | 98.76(12) | 95.28(14) | 97.23(13) | 99.60(10) | 99.22(11) | 82.15(15) | **100.00(1)** | **100.00(1)** | **100.00(1)** | **100.00(1)** | **100.00(1)** | **100.00(1)** | **100.00(1)** | **100.00(1)** |
| http | 73.48(16) | **100.00(1)** | 93.99(14) | **100.00(1)** | **100.00(1)** | 83.33(15) | **100.00(1)** | 96.68(13) | **100.00(1)** | **100.00(1)** | **100.00(1)** | N/A(N/A) | **100.00(1)** | **100.00(1)** | **100.00(1)** | **100.00(1)** |
| InternetAds | 54.44(14) | 93.88(2) | 92.42(9) | 92.53(7) | 93.31(5) | 88.96(13) | 90.96(11) | 50.63(15) | 93.07(6) | 93.74(3) | **94.09(1)** | N/A(N/A) | 92.48(8) | 90.87(12) | 91.44(10) | 93.62(4) |
| Ionosphere | 91.77(16) | 99.02(6) | 97.85(12) | 94.06(14) | 94.22(13) | 98.37(11) | **99.99(1)** | 92.82(15) | 98.81(10) | 98.81(9) | 99.96(6) | 99.45(7) | 99.99(3) | 99.97(5) | 99.99(2) | 99.98(4) |
| landsat | 25.71(16) | 86.53(8) | 43.41(14) | 56.64(13) | 58.10(12) | 62.70(11) | 88.70(7) | 29.25(15) | 85.30(9) | 83.89(10) | 89.05(6) | 89.53(5) | 90.25(4) | 90.78(3) | **90.82(1)** | 90.80(2) |
| letter | 16.59(16) | 43.58(11) | 54.56(10) | 32.80(13) | 35.94(12) | 30.17(14) | 71.78(4) | 19.50(15) | 61.32(8) | 56.30(9) | 74.77(2) | 74.62(3) | 65.08(7) | 73.44(3) | 71.06(5) | **79.14(1)** |
| Lymphography | 78.35(16) | **100.00(1)** | **100.00(1)** | **100.00(1)** | 100.00(14) | 85.01(15) | **100.00(1)** | **100.00(1)** | **100.00(1)** | **100.00(1)** | **100.00(1)** | **100.00(1)** | **100.00(1)** | **100.00(1)** | **100.00(1)** | **100.00(1)** |
| magic.gamma | 52.20(16) | 88.07(9) | 77.59(12) | 74.86(14) | 75.47(13) | 80.79(11) | 88.68(7) | 68.16(15) | 87.51(10) | 88.10(8) | 88.77(6) | 89.02(5) | 89.37(4) | 90.06(2) | 89.68(3) | **90.68(1)** |
| mammography | 18.41(16) | 66.99(8) | 61.99(12) | 61.73(13) | 61.56(14) | 62.72(11) | 70.55(6) | 49.19(15) | 63.43(10) | 65.35(9) | 70.20(7) | 71.21(5) | 73.71(2) | 72.49(3) | 72.21(4) | **76.89(1)** |
| mnist | 18.25(16) | 97.45(10) | 95.07(11) | 90.62(13) | 92.54(12) | 85.35(14) | 97.79(8) | 68.06(15) | 98.12(7) | 98.55(4) | 98.16(6) | 97.70(9) | 98.36(5) | 98.70(2) | 98.68(3) | **98.71(1)** |
| musk | 100.00(8) | **100.00(1)** | 99.57(11) | 99.61(10) | 100.00(8) | **100.00(1)** | **100.00(1)** | 21.13(15) | **100.00(1)** | 99.43(14) | 100.00(7) | **100.00(1)** | 99.92(3) | 99.50(12) | 99.89(4) | 99.86(5) |
| optdigits | 3.70(16) | 99.86(6) | 99.57(11) | 99.61(10) | 99.70(8) | 99.44(13) | 99.70(9) | 21.13(15) | **100.00(1)** | 99.43(14) | 100.00(4) | 99.92(3) | 99.50(12) | 99.89(4) | 99.89(4) | 99.86(5) |
| PageBlocks | 52.66(16) | 90.21(8) | 80.32(11) | 67.63(14) | 70.57(13) | 75.78(12) | 92.16(5) | 60.98(15) | 84.80(10) | 86.23(9) | 91.69(7) | 92.34(4) | 93.01(2) | 92.85(3) | 91.75(6) | **93.54(1)** |
| pendigits | 4.94(16) | 99.52(5) | 97.37(11) | 93.67(13) | 93.45(14) | 92.54(15) | 99.49(6) | 93.77(12) | **99.89(1)** | 98.54(10) | 99.70(2) | 98.75(9) | 99.56(4) | 99.26(7) | 99.16(8) | 99.58(3) |
| Pima | 48.61(16) | 73.47(10) | 63.60(15) | 71.68(13) | 72.62(11) | 72.46(12) | 88.71(5) | 70.58(14) | 78.43(8) | 76.53(9) | 84.89(6) | 84.51(7) | **93.79(1)** | 89.96(3) | 89.51(4) | 91.05(2) |
| satellite | 75.97(15) | 94.20(8) | 81.46(14) | 82.14(13) | 82.58(12) | 85.60(11) | 94.59(7) | 75.87(16) | 92.89(9) | 91.15(10) | 94.97(6) | 95.49(5) | 95.59(4) | 95.76(3) | **95.86(1)** | 95.82(2) |
| satimage-2 | 53.98(16) | 96.56(2) | 93.98(10) | 93.34(11) | 93.15(12) | 92.52(13) | 96.49(3) | 91.61(14) | 95.62(5) | 90.87(15) | 94.79(8) | 94.20(9) | 95.42(6) | **98.71(1)** | 95.09(7) | 96.43(4) |
| shuttle | 82.87(16) | 98.98(7) | 97.19(10) | 96.49(13) | 96.39(14) | 96.32(15) | 99.99(3) | 97.19(11) | 97.76(9) | 96.64(12) | 99.38(8) | 98.03(8) | **99.97(1)** | 99.94(3) | 99.79(?) | **100.00(1)** |
| skin | 22.52(16) | 99.18(10) | 48.47(15) | 73.57(13) | 67.63(14) | 89.96(11) | 99.62(5) | 84.26(12) | 99.76(8) | 99.64(9) | 99.90(2) | 99.82(6) | **99.97(1)** | 99.94(3) | 99.92(4) | 99.92(4) |
| smtp | 20.87(15) | 67.14(4) | 66.71(5) | 66.69(6) | 66.68(7) | 66.69(7) | 50.30(11) | 16.71(16) | 66.69(8) | 50.03(12) | **100.00(1)** | 34.50(13) | 83.33(3) | 33.37(14) | 50.79(10) | 83.33(2) |
| SpamBase | 47.72(16) | 96.07(9) | 91.59(12) | 87.57(14) | 90.59(13) | 94.51(11) | 97.45(5) | 71.73(15) | 96.99(6) | 96.69(8) | 96.82(7) | 97.06(?) | 97.71(?) | 97.59(4) | 97.69(3) | **97.90(1)** |
| speech | 1.63(16) | 6.91(15) | 39.96(2) | 20.85(8) | 19.82(11) | 13.53(13) | 16.39(12) | 21.23(7) | **44.86(1)** | 24.70(4) | 35.80(3) | 23.71(5) | 7.25(14) | 20.10(10) | 20.52(9) | 22.34(6) |
| Stamps | 36.99(16) | 87.44(12) | 86.72(13) | 78.69(15) | 84.96(14) | 89.77(10) | 99.18(5) | 87.65(11) | 97.34(8) | 90.63(9) | 99.64(4) | 98.73(7) | 99.88(3) | **100.00(1)** | 98.16(6) | 98.68(1) |
| thyroid | 55.72(16) | 91.59(14) | 94.40(8) | 93.83(9) | 93.33(10) | 92.30(12) | 98.17(3) | 89.07(15) | 92.07(13) | 93.22(10) | 95.54(4) | 94.99(7) | 98.52(2) | 97.41(5) | 98.16(4) | **98.68(1)** |
| vertebral | 11.71(16) | 72.70(9) | 51.14(13) | 45.26(15) | 45.62(14) | 67.65(10) | **99.31(1)** | 51.19(12) | 72.89(8) | 58.07(11) | 98.06(6) | 97.52(7) | 99.21(4) | 98.70(5) | 99.24(3) | 99.26(2) |
| vowels | 28.96(16) | 84.53(13) | 92.75(7) | 91.67(9) | 91.72(8) | 91.67(10) | 85.22(12) | 69.26(15) | **99.53(1)** | 77.44(14) | 99.42(2) | 97.99(3) | 87.46(11) | 97.57(5) | 97.55(6) | 96.14(4) |
| Waveform | 4.55(16) | 52.55(8) | 22.34(13) | 21.50(14) | 24.61(12) | 32.51(11) | 54.97(6) | 20.41(15) | **69.28(1)** | 61.47(2) | 50.48(10) | 56.22(5) | 51.00(9) | 56.86(4) | 53.89(7) | 58.13(3) |
| WBC | 50.43(16) | 85.45(13) | 88.65(11) | 92.76(10) | 82.24(14) | 95.00(9) | **100.00(1)** | 74.60(15) | **100.00(1)** | 85.59(12) | **100.00(1)** | **100.00(1)** | **100.00(1)** | **100.00(1)** | **100.00(1)** | **100.00(1)** |
| WDBC | 64.49(16) | **100.00(1)** | **100.00(1)** | **100.00(1)** | **100.00(1)** | **100.00(1)** | **100.00(1)** | **100.00(1)** | **100.00(1)** | **100.00(1)** | **100.00(1)** | **100.00(1)** | **100.00(1)** | **100.00(1)** | **100.00(1)** | **100.00(1)** |
| Wilt | 4.93(16) | 88.15(3) | 6.70(15) | 8.18(14) | 8.36(13) | 37.94(11) | 93.01(4) | 40.92(10) | 88.15(8) | 34.94(12) | **94.53(1)** | 90.59(7) | 91.83(5) | 90.86(6) | 93.30(3) | 93.30(3) |
| wine | 34.98(16) | **100.00(1)** | **100.00(1)** | **100.00(1)** | **100.00(1)** | **100.00(1)** | **100.00(1)** | **100.00(1)** | **100.00(1)** | **100.00(1)** | **100.00(1)** | **100.00(1)** | **100.00(1)** | **100.00(1)** | **100.00(1)** | **100.00(1)** |
| WPBC | 24.72(16) | 90.22(9) | 67.49(14) | 72.16(12) | 68.99(13) | 86.38(10) | 99.84(5) | 48.38(15) | 92.87(8) | 85.23(11) | 99.13(7) | 99.78(6) | **100.00(1)** | **100.00(1)** | 99.89(4) | 99.95(3) |
| yeast | 33.52(16) | 53.48(9) | 36.10(15) | 47.62(12) | 47.84(11) | 47.60(13) | 60.66(5) | 47.07(14) | 54.05(8) | 51.83(10) | 56.39(7) | 59.75(6) | 65.43(2) | 61.99(4) | 63.04(3) | **66.16(1)** |
| CIFAR10 | 9.58(16) | 38.61(9) | **44.06(1)** | 39.52(7) | 39.39(8) | 31.04(12) | 40.60(6) | 10.77(15) | 42.50(2) | 38.00(11) | 38.22(10) | 28.13(13) | 23.98(14) | 42.03(4) | 41.35(5) | 42.28(3) |
| FashionMNIST | 23.88(16) | 86.78(3) | 86.09(6) | 84.07(9) | 83.05(10) | 76.64(14) | 85.42(7) | 29.05(15) | 81.81(11) | 85.08(8) | 86.29(4) | 81.79(12) | 76.87(13) | **87.11(1)** | 86.19(5) | 86.19(5) |
| MNIST-C | 17.38(16) | 90.03(2) | 42.69(14) | 87.01(9) | 87.54(7) | 84.75(12) | 86.01(10) | 31.43(15) | 89.79(3) | 89.35(4) | **90.41(1)** | 84.76(11) | 83.11(13) | 87.60(6) | 87.94(5) | 87.03(8) |
| MVTec-AD | 57.05(15) | 96.41(7) | 56.48(16) | 88.80(11) | 87.18(13) | 90.87(10) | 97.92(5) | 66.75(14) | 88.38(12) | 95.26(8) | 97.89(6) | 95.13(9) | 98.55(3) | **98.67(1)** | 98.33(4) | 98.62(2) |
| SVHN | 8.06(15) | 31.84(11) | 35.70(4) | 36.07(2) | **37.10(1)** | 24.98(12) | 32.38(9) | 5.60(16) | 35.97(3) | 34.03(9) | 35.12(5) | 24.87(13) | 17.52(14) | 34.16(7) | 34.14(8) | 34.45(6) |
| Agnews | 6.34(16) | 75.56(9) | 65.97(6) | 56.15(13) | 57.30(12) | 62.46(10) | 61.84(11) | 8.82(15) | 72.66(4) | **78.37(1)** | 76.51(2) | 63.03(9) | 29.80(14) | 64.65(8) | 64.83(7) | 66.27(5) |
| Amazon | 6.08(16) | 35.31(4) | 35.54(13) | 32.76(6) | 32.65(8) | 31.13(11) | 30.13(12) | 5.21(16) | **48.05(1)** | 45.78(2) | 38.71(3) | 23.17(13) | 9.82(14) | 32.54(7) | 32.39(5) | 32.89(5) |
| Imdb | 5.05(16) | 35.68(4) | 29.74(9) | 26.93(12) | 27.52(11) | 31.85(7) | 28.69(10) | 9.34(15) | **48.29(1)** | 46.38(2) | 40.71(3) | 23.94(13) | 11.26(14) | 32.30(6) | 30.83(8) | 32.89(5) |
| Yelp | 9.25(15) | 59.64(4) | 54.23(5) | 48.89(9) | 49.49(8) | 42.50(13) | 48.22(11) | 6.53(16) | **66.00(1)** | 65.54(2) | 54.82(4) | 42.96(12) | 16.24(14) | 50.98(7) | 50.58(6) | 51.38(6) |
| 20news | 7.40(14) | 50.10(4) | 7.30(15) | 39.63(10) | 43.64(7) | 44.22(6) | 45.64(5) | 6.23(16) | 51.84(3) | **57.40(1)** | 53.65(2) | 36.44(12) | 33.60(13) | 39.28(11) | 43.31(8) | 42.66(9) |