# OpenReview forum: "ADBench: Anomaly Detection Benchmark"
_NeurIPS.cc/2022/Track/Datasets_and_Benchmarks — NeurIPS 2022 Datasets and Benchmarks _

### Official Review · Reviewer_U8uU · 2022-07-07
**The conclusion of each comparison is interesting and helpful for future research.**

**Rating:** 6
**Confidence:** 4

**Strengths:**

The paper is well written and easy to follow.

The comparison is comprehensive and covers a range of datasets

The paper analyzes the performance of models from each type of supervision (such as supervised, unsupervised, and semi-supervised).

The types of the anomaly are also taken into account during evaluation (such as local, global, dependency, and clustered).

The robustness analysis is interesting and well-thought.

**Weaknesses:**

My first concern is on the additional CV and NLP datasets that are firstly adopted and included in the ADBench. However, this part is not presented in the paper and remains unclear. While authors claim that AD benchmarks for CV and NLP are naturally different from tabular AD, it would be helpful how to adapt CV/NLP data to tabular data.

It is also interesting to discuss the difference between the anomaly, the saliency [1], and the camouflage [2]. What makes an object anomaly, salient, or camouflaged? Do the shallow or deep models learn such desired outlier/features?


From Table 1 the reviewer would expect to see that this paper conducts an extensive benchmark (coverage), analyze the difference between real-world and synthetic datasets for anomaly detection (data source), the difference in computational cost with respect to the model type (algorithm type), and the comparison angle. The current paper addresses well the expectation of the coverage and the comparison angles. However, several comparisons are missed. It would be interesting to include:

1.	What is the domain gap between the real-world and synthetic datasets? How do AD models perform when they are trained on a real-world/synthetic dataset and tested on another synthetic/real-world dataset?

2.	What is the inference time of both Shallow and DL models? The paper presents the running and the convergence time of the models. However, the inference speed should be also taken into account.

From Figures 4 and 6, we can conclude that shallow methods such as XGB, LGB, and CATB perform better than deep methods such as ResNet and FTTransformer. This is a strange conclusion since firstly deep NN has already shown better performance in other domains (CV/NLP). Secondly, ResNet and FTTransformer are more recent methods compared to XGB, LGB, and CATB. Thirdly, in the original publication [3], the performances of ResNet and FTTransformer are better than XGB and CATB. The author should clarify this issue. Maybe it is due to the evaluation metric. But from this regard, this benchmark is not complete and the conclusion from this paper is biased since more meaningful metrics should be taken into account.


It would be also interesting to include the recent ICLR papers such as [4] in comparison.

[1 ]Salient Objects in Clutter: Bringing Salient Object Detection to the Foreground (ECCV 18)
[2] Concealed Object Detection (TPAMI 22)
[3] Revisiting Deep Learning Models for Tabular Data (NIPS21)
[4] Anomaly Detection for Tabular Data with Internal Contrastive Learning (ICLR22)

**Additional Feedback:**

Please refer to weakness to address the concern. The authors are also encouraged to go through the whole paper to correct the typos

**Clarity:**

While the paper is overall well written, there are many typos in the current version. The reviewer lists several examples as follows, while there exist A LOT of others:

majority are designed -> the majority

we focuses -> we focus

requirements in industry -> requirements in the industry

unsupervised algorithms is statistically -> unsupervised algorithms are statistically

best unsupervised methods -> the best unsupervised methods

are therefore limited to detect - > are therefore limited to detecting

improving detection performance, and leverage - > improving detection performance and leveraging

covers both shallow and deep learning (DL) algorithms, and consider -> and considers

latest unsupervised - > the latest

in natural language processing -> in the

as great addition to the community –> as a great

more complex representation -> more complex representations

there are a group -> there is

varying level of -> varying levels of

anomalies in the training set is known -> are known

anomalies at interest -> of interest

generation of normal samples are -> is

probability density function of generated anomalies are -> is

observe how do AD algorithms respond to it - > observe how AD algorithms respond to it

discuss more generalized impact -> discuss the more generalized impact

make prediction on -> make a prediction on

as testing set -> as the testing set

**Correctness:**

The reviewer is not certain about the evaluation protocol which seems to result in an abnormal conclusion. In this paper, the AUCROC and AUCPR are used to analyze the performance. From this metric, early models achieve better performance than recent deep methods. However, if the metrics become ‘CA AD HE JA HI AL EP YE CO YA MI’, it can be seen from [1] that deep models perform better. The authors should address this concern.



**Documentation:**

According to the official Github, several codes are released for the reproducibility check.


**Ethics:**

No special concern

**Relation To Prior Work:**

The difference compared to prior works is trivial and significant.

**Summary And Contributions:**

This paper provides a detailed and thorough benchmarking of tabular AD. They conduct studies on 30 detection algorithms’ performance among 55 benchmark datasets. The paper pays extra attention to the type of supervision, type of anomaly, and algorithm robustness. The conclusion of each comparison is interesting and helpful for future research.

---

> ### Author Response · Authors · 2022-08-11
> **Response to Reviewer U8uU (paper revision Aug 11th) P1**
>
> **Q1**. My first concern is on the additional CV and NLP datasets that are firstly adopted and included in the ADBench. However, this part is not presented in the paper and remains unclear. While authors claim that AD benchmarks for CV and NLP are naturally different from tabular AD, it would be helpful how to adapt CV/NLP data to tabular data.
>
> **R1**. Great point! We now expand Section 3.2 and Appx. B.2 to detail how these CV and NLP datasets are created and provide the generation script on [GitHub repo](https://github.com/Minqi824/ADBench/tree/main/datasets).
> In a nutshell, we follow popular AD literature DeepSAD [1], ADIB [2], and DATE [3] to create these new AD datasets by using deep models to extract representations from CV and NLP datasets. Including these (originally) non-tabular datasets helps to see whether tabular AD methods can work on CV/NLP data after necessary preprocessing.
>
> Meanwhile, we provide the download link to the raw version of the newly added datasets in [ReadMe](https://github.com/Minqi824/ADBench/tree/main/datasets), as well as the second version of CV/MLP datasets by using different pre-trained models (i.e., ViT [4] and RoBERTa [5]; see Appx. B.2 for details) to extract the features from these CV and NLP datasets. Although we still analyze our results based on the original version using Bert and ResNet18 for NLP and CV datasets respectively. we encourage readers to check the other version. In the later iteration of ADBench, we will also try to understand the impact of backbone choice on the downstream tasks.
>
> [1] Ruff, Lukas, et al. "Deep semi-supervised anomaly detection." ICLR, 2019.
>
> [2] Deecke, Lucas, et al. "Transfer-based semantic anomaly detection." ICML, 2021.
>
> [3] Manolache, Andrei, Florin Brad, and Elena Burceanu. "DATE: Detecting Anomalies in Text via Self-Supervision of Transformers." NAACL, 2021.
>
> [4] Dosovitskiy, Alexey, et al. "An Image is Worth 16x16 Words: Transformers for Image Recognition at Scale." ICLR, 2020.
>
> [5] Liu, Yinhan, et al. "Roberta: A robustly optimized bert pretraining approach." arXiv (2019).
>
> ----
>
> **Q2**. It is also interesting to discuss the difference between the anomaly, the saliency [1], and the camouflage [2]. What makes an object anomalous, salient, or camouflaged? Do the shallow or deep models learn such desired outlier/features?
>
> **R2**. We appreciate that the reviewer mentioned these interesting points. We have enriched our related work (Section 2.3) to cover more relevant topics, including out-of-distribution and novelty detection [1,2], and salient [3,4] and camouflaged object detection [5,6]. Due to the scope of ADBench, we will leave a broader coverage of these relevant topics for future work.
>
> [1] A unified survey on anomaly, novelty, open-set, and out-of-distribution detection: Solutions and future challenges. arXiv, 2021.
>
> [2] OpenOOD: Benchmarking Generalized Out-of-Distribution Detection. openreview, under submission, 2022
>
> [3] Salient Objects in Clutter: Bringing Salient Object Detection to the Foreground. ECCV, 2018.
>
> [4] Re-thinking co-salient object detection. TPAMI, 2021.
>
> [5] Camouflaged object detection. CVPR, 2020.
>
> [6] Concealed Object Detection. TPAMI, 2022.
>
> ----
>
> **Q3**. From Table 1 the reviewer would expect to see that this paper conducts an extensive benchmark (coverage), analyze the difference between real-world and synthetic datasets for anomaly detection (data source), the difference in computational cost with respect to the model type (algorithm type), and the comparison angle. The current paper addresses well the expectation of the coverage and the comparison angles. However, several comparisons are missed. It would be interesting to include:
>
> **Q3.1**. What is the domain gap between the real-world and synthetic datasets? How do AD models perform when they are trained on a real-world/synthetic dataset and tested on another synthetic/real-world dataset?
>
> **R3.1**. This sounds like an interesting angle! We could envision the scenario of training an AD model on synthetic anomalies to use in real-world applications with a similar anomaly composition. We are not aware of works in this category yet, and add this interesting idea to the Future research direction 3 in Section 4.3.
>
>
> **Q3.2**. What is the inference time of both Shallow and DL models? The paper presents the running and convergence time of the models. However, the inference speed should also be taken into account.
>
> **R3.2**. This is a great point! We have added the runtime in Fig. 4d and the inference time in Appx. Fig. D6. We also provide a brief discussion of runtime in Section 4.2, which identifies some fast algorithms and mentions that computational cost should also be considered.

---

> > ### Author Response · Authors · 2022-08-11
> > **Response to Reviewer U8uU (paper revision Aug 11th) P2**
> >
> > **Q4**. From Figures 4 and 6, we can conclude that shallow methods such as XGB, LGB, and CATB perform better than deep methods such as ResNet and FTTransformer. This is a strange conclusion since firstly deep NN has already shown better performance in other domains (CV/NLP). Secondly, ResNet and FTTransformer are more recent methods compared to XGB, LGB, and CATB. Thirdly, in the original publication [3], the performances of ResNet and FTTransformer are better than XGB and CATB. The author should clarify this issue. Maybe it is due to the evaluation metric. But from this regard, this benchmark is not complete and the conclusion from this paper is biased since more meaningful metrics should be taken into account.
> >
> > The reviewer is not certain about the evaluation protocol which seems to result in an abnormal conclusion. In this paper, the AUCROC and AUCPR are used to analyze the performance. From this metric, early models achieve better performance than recent deep methods. However, if the metrics become ‘CA AD HE JA HI AL EP YE CO YA MI’, it can be seen from [1] that deep models perform better. The authors should address this concern.
> >
> > **R4**. We appreciate this important question! First, a few recent studies find that tree-based ensemble methods outperform the latest deep models for tabular data [1,2]. Indeed, tree ensembles have been found to work well in tabular AD [3]. Even in recent time-series AD benchmarks, ensemble trees outperform deep models in many cases, e.g., Fig. 7a of [4] where iForest is better than LSTM/CNN and Fig. 5 of [5] where gradient boosted tree regressors outperform GAN and LSTM. Thus, it is not surprising to see some deep models are inferior to ensemble trees in ADBench.
> >
> > Regarding the reason, we assume ensemble trees are good at handling imbalanced AD datasets via aggregation, and node-splitting of trees is also suited for tabular datasets that lack intrinsic structures. We have updated Section 4.2 to include some of the above works and pointed out this interesting direction for future research.
> >
> > [1] Borisov, Vadim, et al. "Deep neural networks and tabular data: A survey." arXiv, 2021.
> >
> > [2] Grinsztajn, Léo, Edouard Oyallon, and Gaël Varoquaux. "Why do tree-based models still outperform deep learning on tabular data?." arXiv, 2022.
> >
> > [3] Vargaftik, Shay, et al. "Rade: Resource-efficient supervised anomaly detection using decision tree-based ensemble methods." Machine Learning, 2020.
> >
> > [4] Paparrizos, John, et al. "TSB-UAD: an end-to-end benchmark suite for univariate time-series anomaly detection." VLDB, 2022.
> >
> > [5] Lai, Kwei-Herng, et al. "Revisiting time series outlier detection: Definitions and benchmarks." NeurIPS. 2021.
> >
> > ----
> >
> > **Q5**. It would also be interesting to include the recent ICLR papers such as [4] in comparison.
> > [4] Anomaly Detection for Tabular Data with Internal Contrastive Learning (ICLR22)
> >
> > **R5**. Thanks for mentioning this recent work. We are looking into it and considering including it in a later version (in our long-term plan). For now, we add it to our Future direction 1 in Section 4.2 and Section 5.
> >
> > ----
> >
> > **Q6**. While the paper is overall well written, there are many typos in the current version. The reviewer lists several examples as follows, while there exist A LOT of others.
> >
> >
> > **R6**. We appreciate the corrections. We have fixed them with another round of proofreading. We are so grateful for your diligence and devotion in reviewing ADBench. It would not be in its current form without your help.

---

### Official Review · Reviewer_8D8R · 2022-07-23
**Comprehensive AD benchmark with new comparison angles**

**Rating:** 7
**Confidence:** 4

**Strengths:**

1. This paper provides a comprehensive AD benchmark containing many domains' datasets.
2. This paper provides multiple angles on benchmark anomaly detection algorithms: for example, comparing semi-supervised and supervised anomaly detection is fascinating; studying different types of anomalies can be helpful for AD algorithm analysis and model selection.


**Weaknesses:**

The comparison of shallow and deep anomaly detection approaches can be improved. For example, this paper assumes the training data contains a certain amount of anomalies; however, some deep anomaly detection models (e.g., DeepSVDD, AE) believe the training data is anomaly-free so that they can learn the normal representations. Thus the current comparison results may not be fair.

**Additional Feedback:**

To extend the scope of current ADBench, I would suggest to add some datasets that are relevant to explainable AD [1] and Fair AD [2,3].

[1] Pang, Guansong, and Charu Aggarwal. "Toward explainable deep anomaly detection." Proceedings of the 27th ACM SIGKDD Conference on Knowledge Discovery & Data Mining. 2021.

[2] Zhang, Hongjing, and Ian Davidson. "Towards fair deep anomaly detection." Proceedings of the 2021 ACM Conference on Fairness, Accountability, and Transparency. 2021.

[3] Shekhar, Shubhranshu, Neil Shah, and Leman Akoglu. "Fairod: Fairness-aware outlier detection." Proceedings of the 2021 AAAI/ACM Conference on AI, Ethics, and Society. 2021.



**Clarity:**

The paper reads well and the text is fluent. However, the figure 3 is not clear to me. some local anomalies look similar to the global anomalies.

**Correctness:**

The claims are correct and sound. However, the evaluation methods for benchmark results could be improved (for example, the comparison of shallow and deep approaches may not be fair, the DeepSVDD assumes the training data is anomaly-free, but that's not the case for this paper's experiments.)

**Documentation:**

The datasets and code are made available in a GitHub project. Sufficient details are provided for others to replicable the experiments reported in the paper.

**Ethics:**

Not that I can see.

**Relation To Prior Work:**

The related work section clearly discussed the difference between the proposed work and previous works.

**Summary And Contributions:**

Although there are many existing AD benchmarks, this paper proposes a comprehensive AD benchmark containing 55 datasets from different domains and covers both deep and shallow algorithms. Besides the comprehensiveness, this paper studies AD from different angles like types of supervision and anomalies which are highly beneficial for AD research. Moreover, all the datasets and source code are made public for reproduction.

---

> ### Author Response · Authors · 2022-08-11
> **Response to Reviewer 8D8R (paper revision Aug 11th)**
>
> **Q1**. The comparison of shallow and deep anomaly detection approaches can be improved. For example, this paper assumes the training data contains a certain amount of anomalies; however, some deep anomaly detection models (e.g., DeepSVDD, AE) believe the training data is anomaly-free so that they can learn the normal representations. Thus the current comparison results may not be fair.
> The claims are correct and sound. However, the evaluation methods for benchmark results could be improved (for example, the comparison of shallow and deep approaches may not be fair, the DeepSVDD assumes the training data is anomaly-free, but that's not the case for this paper's experiments.)
>
> **R1**. We appreciate the thought of handling DeepSVDD and other methods that assume anomaly-free samples during training with special care. Currently, in our unsupervised setting, we assume no label information is available during training, and using anomaly-free samples here may leak label information. For a fair comparison, we do not use anomaly-free training samples for unsupervised methods. Differently, for these semi-supervised methods that assume anomaly-free training samples (e.g., GANomaly), we remove the labeled anomalies during the training phase.
>
> Also, DeepSVDD paper (Section 3.1) mentions that “we assume *most of the training data $D_n$* is normal, which is often the case in one-class classification tasks”. Thus, we believe the current setting of ADBench follows DeepSVDD’s data assumption. We are happy to discuss this further.
>
> ----
>
> **Q2**. The paper reads well and the text is fluent. However, figure 3 is not clear to me. some local anomalies look similar to global anomalies.
>
> **R2**. We assume the visual similarity of a small percentage of local and global anomalies in Fig. 3 is caused by compressing 18-dimension Lymphography to 2 dimensions by the non-linear t-SNE. However, the significant algorithm performance difference for local (Fig. 5a) and global (Fig. 5b) anomalies show that the injected anomalies are largely different. For instance, the best performing method for local anomalies, local outlier factor (LOF), only ranks the 9-th in the global anomaly setting, justifying the major difference between different types of anomalies.
>
> We provide another visualization for this in Appx. Fig. B2, using Ionosphere dataset. In this case, the injected local and global anomalies look more distinct.
>
> ----
>
> **Q3**. To extend the scope of the current ADBench, I would suggest adding some datasets that are relevant to explainable AD [1] and Fair AD [2,3].
>
> **R3**. Thanks for mentioning these interesting works. We have highlighted these potential data sources in our future direction (Section 5), including AD papers on explainability and interpretability [1, 6] and fairness-aware AD [2,3,4,5] as you suggested.
>
> Again, we appreciate all these great thoughts on considering the anomaly-free training setting and expanding the scope of datasets. As a long-term open-source project, we will gradually include these points in iterations.
>
> [1] Pang, Guansong, and Charu Aggarwal. "Toward explainable deep anomaly detection." KDD. 2021.
>
> [2] Zhang, Hongjing, and Ian Davidson. "Towards fair deep anomaly detection." FAccT. 2021.
>
> [3] Shekhar, Shubhranshu, Neil Shah, and Leman Akoglu. "Fairod: Fairness-aware outlier detection." AIES, 2021.
>
> [4] Davidson, Ian, and Selvan Suntiha Ravi. "A framework for determining the fairness of outlier detection." ECAI, 2020.
>
> [5] Song, Hanyu, Peizhao Li, and Hongfu Liu. "Deep Clustering based Fair Outlier Detection." KDD, 2021.
>
> [6] Xu, Hongzuo, Yijie Wang, Songlei Jian, Zhenyu Huang, Yongjun Wang, Ning Liu, and Fei Li. "Beyond outlier detection: Outlier interpretation by attention-guided triplet deviation network.". WWW, 2021.

---

### Official Review · Reviewer_sH17 · 2022-07-24
**comprehensive benchmark with new dataset contribution**

**Rating:** 7
**Confidence:** 3

**Strengths:**

1. the ADBench proposed in the paper includes the most number of datasets(55) and algorithms(30) compared to previous benchmarks.
2. it evaluates the algorithms with most various angles such as level of supervision, type of anomaly and dataset condition.
3. the evaluations are well performed and explained, e.g all evaluates are done under different percentage of labeled anomalies. The author explains the evaluation results and insights which also provide directions for future research directions.

**Weaknesses:**

1. The paper depends on 1 single method to extract the features for each of the newly added datasets that may cause the feature themselves to be biased towards certain methods.
2. The paper briefly mentions using a generative model to generate anomaly, however, no more details are given in the main text or appendix. This seems critical for readers to learn about the 7 newly introduced datasets.
3. The author provides average rank across multiple datasets in the main section and attaches the full results in the appendix, but it will be helpful if the author could give some insights into the results, e.g highlight the best performance on a certain dataset. The full table is a little bit hard to interpret now.
4. Since the paper created 7 new datasets, maybe it's better to separate the dataset from the codebase for easier access
5. There seem to be a misuse of terms, e.g "real-world", in some places, the author refers to the newly added dataset as synthetic datasets and in some other places, the author refers to them as "real-world" datasets.

**Additional Feedback:**

N/A

**Clarity:**

Yes. The paper is well written. It summarizes its contributions at the very beginning and then splits the main text into 3 focuses which correspond to the 3 angles the author proposes. The paper is easy to read and the tables and figures are well explained.

**Correctness:**

# Benchmark

The benchmark evaluation in ADBench looks correct.

# New datasets

The author introduces how the new datasets are generated without giving much details. We don't know if the dataset is biased or not.

**Documentation:**

## ADBench
The benchmarks proposed in this paper are fully open sourced on GitHub with licensing. Therefore it's easily accessible.

## Dataset
The datasets introduced in this paper are hosted in the same place with the code under the same licensing. However, there seems to be a lack of documentation of how the data are generated and best practices to use the data.

**Ethics:**

The newly introduced datasets seem to be developed from existing public datasets. However, these datasets are not referenced in the paper as indicated by original dataset publishers.

**Relation To Prior Work:**

Yes. The author adds a table to compare the work with previous works and outlines the key differences.

**Summary And Contributions:**

# Submission summary and contribution
The paper proposes a comprehensive benchmark for evaluating the performance of anomaly detection(AD) methods. Plus, it contributes 7 new datasets developed from popular datasets.

The paper proposes to evaluate AD methods in the following 3 angles
1. the extent of supervision such as unsupervised, semi-supervised and fully supervised
2. the type of anomalies such as local, global, dependency and clustered.
3. the performance of models with noisy data

All experiments are focused on the 3 angles with varying percentage of labeled anomaly samples. The paper presents the results of 30 methods which reveal their true performances and a few key conclusions are drawn from the results such as no unsupervised algorithms can statistically outperform. Based on the observations, the author proposes a few future directions for the research community to further explore.

---

> ### Author Response · Authors · 2022-08-11
> **Response to Reviewer sH17 (paper revision Aug 11th) P1**
>
> **Q1**. The paper depends on 1 single method to extract the features for each of the newly added datasets that may cause the feature themselves to be biased towards certain methods.
>
> **R1**. Great point! We follow popular DeepSAD [1], ADIB [2], and DATE [3]’s setting to create these new datasets by using deep models to extract representations from CV and NLP datasets. Following your comment, we now provide another version of CV/NLP datasets using ViT [4] and RoBERTa [5] as the backbone (see Appx. B.2 for details) to extract the features and release both versions at [datasets](https://github.com/Minqi824/ADBench/tree/main/datasets).
>
> Although the paper results are based on the original version using Bert and ResNet18 for NLP and CV datasets, we encourage readers to check the other version. In the later iteration of ADBench, we will also try to understand the impact of backbone choice on the downstream tasks.
>
> [1] Ruff, Lukas, et al. "Deep semi-supervised anomaly detection." ICLR, 2019.
>
> [2] Deecke, Lucas, et al. "Transfer-based semantic anomaly detection." ICML, 2021.
>
> [3] Manolache, Andrei, Florin Brad, and Elena Burceanu. "DATE: Detecting Anomalies in Text via Self-Supervision of Transformers." NAACL, 2021.
>
> [4] Dosovitskiy, Alexey, et al. "An Image is Worth 16x16 Words: Transformers for Image Recognition at Scale." ICLR, 2020.
>
> [5] Liu, Yinhan, et al. "Roberta: A robustly optimized bert pretraining approach." arXiv (2019).
>
> ----
>
> **Q2**. The paper briefly mentions using a generative model to generate anomalies, however, no more details are given in the main text or appendix. This seems critical for readers to learn about the 7 newly introduced datasets. The author introduces how the new datasets are generated without giving many details. We don't know if the dataset is biased or not.
>
> **R2**. We appreciate the questions on the 10 datasets (originally 7) introduced in ADBench from CV and NLP domains. First, we want to clarify we do not generate synthetic anomalies for these datasets, but repurpose these multi-classification datasets for AD following the procedures in [1,2,3].  Simply put, we set one of the classes as normal and downsample the remaining classes to 5% of the total instances as anomalies by default, and report the average results over all the respective classes. Here are some examples. For Amazon and Imdb datasets, we regard the original negative class as the anomaly class. For Yelp dataset, we regard the reviews of 0 and 1 stars as the anomaly class, and the reviews of 3 and 4 stars as the normal class. More details can be found in Appx. B.2. Again, this process does not involve any generative models.
>
> For a separate purpose, we use the generative model (i.e., GMM) to inject specific types of anomalies (e.g., local, global, etc.) in Section 3.3.2. This is a separate process from the CV/NLP dataset adaptation discussed above. With these different types of synthetic anomalies, we can better understand detector performance regarding anomaly types. We follow the latest anomaly generation introduced in [3], which has been widely used in recent AD works including a KDD’22 paper [5]. We also plan to use more advanced techniques for anomaly generation, and we will closely follow the work in this direction and update ADBench accordingly. Again, we confirm that GMM is not used in creating the 10 CV and NLP datasets.
>
> [1] Emmott, Andrew, Shubhomoy Das, Thomas Dietterich, Alan Fern, and Weng-Keen Wong. "A meta-analysis of the anomaly detection problem." arXiv, 2015.
>
> [2] Ruff, Lukas, et al. "Deep one-class classification." ICML, 2018.
>
> [3] Ruff, Lukas, Robert A. Vandermeulen, Nico Görnitz, Alexander Binder, Emmanuel Müller, Klaus-Robert Müller, and Marius Kloft. "Deep semi-supervised anomaly detection. ICLR, 2019.
>
> [4] Steinbuss, Georg, and Klemens Böhm. "Benchmarking unsupervised outlier detection with realistic synthetic data." TKDD, 2021.
>
> [5] Zhang, Sean, Varun Ursekar, and Leman Akoglu. “Sparx: Distributed Outlier Detection at Scale”. KDD 2022 (Accepted, to appear).
>
> ----
>
> **Q3**. The author provides average rank across multiple datasets in the main section and attaches the full results in the appendix, but it will be helpful if the author could give some insights into the results, e.g highlight the best performance on a certain dataset. The full table is a little bit hard to interpret now.
>
> **R3**.Thank you so much for the suggestion. We now highlight the best model (as well as rank) per dataset in Appendix tables D4-D19.
>
> ----
>
> **Q4**. Since the paper created 7 new datasets, maybe it's better to separate the dataset from the codebase for easier access.
>
> **R4**. Following your advice, we make these datasets separate folders and provide the download link to the raw version of them in [ReadMe](https://github.com/Minqi824/ADBench/tree/main/datasets). Additionally, we add references to all the datasets in Appx. Table 2 so that readers can find more descriptions of the datasets.

---

> > ### Author Response · Authors · 2022-08-11
> > **Response to Reviewer sH17 (paper revision Aug 11th) P2**
> >
> > **Q5**. There seem to be a misuse of terms, e.g "real-world", in some places, the author refers to the newly added dataset as synthetic datasets and in some other places, the author refers to them as "real-world" datasets.
> >
> > **R5**. Great catch! We have made corresponding changes in Sections 3.2, 3.3.2, 4, and the Appendix.
> >
> > ----
> >
> > **Q6**. Dataset documentation and reference to the original datasets
> > The datasets introduced in this paper are hosted in the same place with the code under the same licensing. However, there seems to be a lack of documentation of how the data are generated and best practices for using the data.
> >
> > **R6**. As mentioned in Q4, we make these datasets a separate folder and provide the download link to the raw version of them in [dataset](https://github.com/Minqi824/ADBench/tree/main/datasets), along with a ReadME file. Additionally, we add references to all the datasets in Appx. Table B1 so that readers can find more descriptions of the datasets.
> >
> > In addition to the more detailed description of the CV/NLP datasets generation in Section 3.2 and Appx. B.2, our [dataset](https://github.com/Minqi824/ADBench/tree/main/datasets) now provides Google Colab scripts for generating these representations, along with other technical details.
> >
> > ----
> >
> > **Q7**. The newly introduced datasets seem to be developed from existing public datasets. However, these datasets are not referenced in the paper as indicated by the original dataset publishers.
> >
> > **R7**. We add the references to all the datasets in Appx. Table 2, Column 7. We sincerely thank your suggestions for making ADBench more accessible and usable :)

---

> > > ### Comment · Reviewer_sH17 · 2022-08-26
> > > **Response**
> > >
> > > Thank you for the updates. I think the paper and the accompanied codebase/datasets are valuable to the community. So I will still recommend "accept" for the paper.

---

### Official Review · Reviewer_xNAm · 2022-07-25
**ADBench: Anomaly Detection Benchmark**

**Rating:** 6
**Confidence:** 3
**Clarity:** The paper is clearly written and easy…

**Strengths:**

1. ADBench has a large algorithm collection with unsupervised, semi-supervised, and full-supervised anomaly detection algorithms. It includes the latest deep learning and ensemble learning methods.
2. ADBench covers a wide range of anomaly detection application domains, including healthcare, finance, natural language processing, computer vision, etc.
3. ADBench studies multiple types of synthetic anomalies in Figure 3, which can help us better understand the impact of different anomaly types.
4. ADBench additionally considers three noisy and corruption settings to better evaluate the algorithms. It highlights the importance of developing robust and noise-resilient anomaly detection algorithms.


**Weaknesses:**

Compared to previous anomaly detection benchmarks, the proposed ADBench takes more datasets into account, but more do not necessarily mean better. One question that the authors forget to justify is that, whether these 55 datasets are really anomaly detection datasets?

Indeed, some of these datasets are not anomaly detection datasets from my perspective. In the definition, an anomaly is a data object that deviates significantly from the majority of the objects[1], and thus usually accounts for only a small fraction of the dataset. However,  in 7 out of 55 datasets used by ADBench, anomalies make up about half of the dataset. For example, 75% of samples are labeled as anomalies in the Parkinson dataset, which is unreasonable.

[1] Han et al. Data Mining: Concepts and Techniques. 2011.


**Additional Feedback:**

1. More justifications can be provided on whether all 55 datasets in ADBench are really designed for anomaly detection.
2. Besides AUC-ROC and AUC-PR, some other metrics can be used (e.g., various F1 scores, the top-K accuracy) for a more comprehensive comparison.
3. Figures 4 and 5 are too small to distinguish the performance of the compared algorithms.

**Correctness:**

Yes. The experiments are appropriately designed and performed. The model comparison is comprehensive and fair. However, I have some concerns about the datasets. Please refer to Weaknesses for more details.


**Documentation:**

Yes. The proposed ADBench, including datasets and code, is available on Github.


**Ethics:**

Not applicable.

**Relation To Prior Work:**

This paper clearly discusses the differences with existing literature in section 2 and Table 1. In particular, it considers more ensemble learning and deep learning methods, incorporates both real-world and synthetic datasets, and provides diverse comparison angles.
Some additional References:
[1] Lavin A, Ahmad S. Evaluating real-time anomaly detection algorithms--the Numenta anomaly benchmark[C]//2015 IEEE 14th international conference on machine learning and applications (ICMLA). IEEE, 2015: 38-44.
[2] Akcay S, Ameln D, Vaidya A, et al. Anomalib: A Deep Learning Library for Anomaly Detection[J]. arXiv preprint arXiv:2202.08341, 2022.


**Summary And Contributions:**

This paper presents ADBench, a comprehensive tabular anomaly detection benchmark with 30 algorithms and 55 datasets. They focus on three comparison perspectives, including the availability of supervision, the anomaly types, and the algorithm robustness under noise and data corruption. They conduct extensive experiments to show the value of algorithm selection, label supervision, and prior knowledge.

---

> ### Author Response · Authors · 2022-08-11
> **Response to Reviewer xNAm (paper revision Aug 11th)**
>
> **Q1**. Compared to previous anomaly detection benchmarks, the proposed ADBench takes more datasets into account, but more do not necessarily mean better. One question that the authors forget to justify is that, whether these 55 datasets are really anomaly detection datasets?
> Indeed, some of these datasets are not anomaly detection datasets from my perspective. In the definition, an anomaly is a data object that deviates significantly from the majority of the objects[1], and thus usually accounts for only a small fraction of the dataset. However, in 7 out of 55 datasets used by ADBench, anomalies make up about half of the dataset. For example, 75% of samples are labeled as anomalies in the Parkinson dataset, which is unreasonable.
> More justifications can be provided on whether all 55 datasets in ADBench are really designed for anomaly detection.
>
> **R1**. Great point! As discussed in Section 2.2, we originally curated the list of datasets from the ODDS Library, DAMI Repository, and Anomaly Detection Meta-Analysis Benchmarks, which are all widely used in AD research. It is unclear what is the reasonable threshold for anomaly ratios in AD tasks, while we agree the bottom line is the number of anomalies should be fewer than normal samples.
>
> With your suggestion, we have removed seven datasets from ADBench with anomaly ratio >=40% and added 9 new datasets from [1,2,3,4]. See Appx. Table B1 for an updated list of datasets (we crossed 7 datasets, and highlighted 9 added datasets in yellow during the revision). After this change, all 57 datasets in ADBench have a below 40% anomaly ratio, and 40 datasets have below 10% of anomalies. We provide a histogram of the anomaly ratio distribution in Appx. Fig. B1 (the median=0.05). We appreciate this opportunity to make ADBench better.
>
> We also want to thank you again for pointing out this important question which has been long ignored in AD benchmarks. It will also be helpful to analyze the algorithm performance under varying levels of anomaly ratios (not the labeled ratio but the intrinsic anomaly ratio). We have listed this as one of the future directions in Section 5.
>
> [1] Pang, Guansong, et al. "Deep learning for anomaly detection: A review." CSUR, 2021.
>
> [2] Mu, Norman, and Justin Gilmer. "Mnist-c: A robustness benchmark for computer vision." arXiv, 2019.
>
> [3] Bergmann, Paul, et al. "MVTec AD--A comprehensive real-world dataset for unsupervised anomaly detection." CVPR, 2019.
>
> [4] Lang, Ken. "Newsweeder: Learning to filter netnews." In Machine Learning Proceedings 1995, pp. 331-339. Morgan Kaufmann, 1995.
>
> ----
>
> **Q2**. This paper clearly discusses the differences with existing literature in section 2 and Table 1. In particular, it considers more ensemble learning and deep learning methods, incorporates both real-world and synthetic datasets, and provides diverse comparison angles. Some additional References:
> - [1] Lavin, Alexander, and Subutai Ahmad. "Evaluating real-time anomaly detection algorithms--the Numenta anomaly benchmark.". ICMLA, 2015.
> - [2] Akcay, Samet, Dick Ameln, Ashwin Vaidya, Barath Lakshmanan, Nilesh Ahuja, and Utku Genc. "Anomalib: A Deep Learning Library for Anomaly Detection." arXiv, 2022.
>
> **R2**. Thanks for pointing them out! We have included these nice benchmarks for time-series and image AD in Section 2.2 as related work.
>
> ----
>
> **Q3**. Besides AUC-ROC and AUC-PR, some other metrics can be used (e.g., various F1 scores, the top-K accuracy) for a more comprehensive comparison.
>
> **R3**. We appreciate the thought of including more metrics. For now, we report both AUC-ROC and AUC-PR, which should be sufficient for most cases. In the later iteration, we will add the results on F1 and Precision@rank k (where k is the number of actual anomalies) to the Appendix. As a long-term open-source project, we would be happy to provide as much useful information as possible.
>
> ----
>
> **Q4**. Figures 4 and 5 are too small to distinguish the performance of the compared algorithms.
>
> **R4**. Agreed! Now we have remade and/or resized Fig. 4 for better readability. Due to the space limitation, we have slightly improved Fig. 5 and verified its readability by printing it out on paper. After the rebuttal period, we will cut out all the current revision remarks for space and make Fig. 5 in two rows.

---

### Official Review · Reviewer_89C5 · 2022-07-26
**A nice benchmark and analysis for Anomaly Detection**

**Rating:** 8
**Confidence:** 4

**Strengths:**


* The authors have proposed a large collection of data for benchmarking Anomaly Detection algorithms. They have performed a comprehensive analysis of existing methods and provided various insights regarding the robustness and inductive biases of these methods. The addition of synthetic data which can be used to test the inductive assumptions and types of anomalies that can be detected can be a very powerful "model debugging and development" tool.

* The paper provides several insights on existing AD methods and reports their performance when various amounts of supervision is used. The paper contains discussions about the limitations of existing AD models.

* The code and the dataset are publicly available. The paper is well written and easy to follow.

* I believe that these kinds of _unified_ benchmarks are very important for Anomaly Detection since most works are using different experimental setups, which makes the methods harder to compare and can lead to reproducibility issues. This might be a very useful resource for the AD community.


**Weaknesses:**


* You have not included datasets tailored for anomaly detection in your benchmark, such as MVTec-AD [[1]] or MNIST-C [[2]]. Do you have a reason for omitting them?

* I understand that the newly added datasets (CIFAR10, fMNIST, SVHN, AGNews, Amazon, IMDB, and Yelp) are present in the form of representations extracted from neural networks (BERT and ResNet18). This could limit the overall AD performance of the models, since the data is already projected to a different space by the feature extraction backbone, leading to anomalies that could be clustered together with the normal data. Moreover, less general models that use the data structural information as an inductive bias for learning will suffer from these benchmark design decisions. For example, [[3]], [[4]] and [[5]] are methods specifically designed for one-class classification in text, with [[3]] and [[5]] achieving much better performance on the AG News dataset when compared to your best results (83.1, 90.0 vs. 68.97 ROCAUC). This might be due to the different train/validation/test splits used but might also indicate that end-to-end training is desirable on these datasets. Please consider adding an option for downloading the "raw" dataset in your library.

* Some scores reported on your synthetic data are very high (for example 99.91 ROCAUC for the KNN on Global Anomalies). I still believe that the datasets are useful for testing the inductive biases when designing a model, but I don't see any value in using them when computing the overall performance metrics of new methods. It's not clear to me if you've also used them when computing the mean performance in places such as Figure 4.

[1]: https://openaccess.thecvf.com/content_CVPR_2019/papers/Bergmann_MVTec_AD_--_A_Comprehensive_Real-World_Dataset_for_Unsupervised_Anomaly_CVPR_2019_paper.pdf
[2]: https://arxiv.org/pdf/1906.02337.pdf
[3]: https://aclanthology.org/P19-1398/
[4]: https://aclanthology.org/2021.eacl-main.296/
[5]: https://aclanthology.org/2021.naacl-main.25/

**Additional Feedback:**

It would be nice to consider expanding this work with different modalities (such as videos, for example ShanghaiTech, or graphs). I'm looking forward to seeing how this project evolves!

**Clarity:**


The paper is generally well written but contains some confusing parts.

* I don't understand the claim that "none of the benchmarked unsupervised algorithms is statistically better than others" (line 35, figure 4 caption), it seems that the biggest performance gap is between DeepSVDD (51.07 ROCAUC) and CBLOF (70.20 ROCAUC), 19.13 points in ROCAUC seems like a major performance gap?

* In figure 4 you have a cross symbol for both subfigures (a) and (b), but the explanation for the symbol is just under (b). Please consider moving it in the figure caption.

* In the figure 4 captions some text seems to be missing: "(a) shows that no unsupervised algorithm can statistically outperform." - outperform what? Please expand.

* In the figure 4 captions you mention that the scores present in the figure are the "AD model performance on 55 real-world datasets.", I suppose that the reported numbers are the average score over all the datasets. Please be explicit in the captions and consider adding the standard deviations over the datasets.

* You mentioned multiple times that the collection of data represents "55 real-world datasets", please expand. What makes them "real-world"? For example, your collection of data contains MNIST and Fashion MNIST - it could be argued that they can be used in "real-world" applications and were collected from the real world, but they are curated and are usually used for benchmarking algorithms.

* In Figure 7 you show the performance change under noisy and corrupted data. I would argue that the performance difference on the OY axis is not the clearest way of expressing the degradation: for example, you could get the impression that DeepSVDD is robust to noise since its performance doesn't change a lot when using corrupted data, in reality, DeepSVDD can't lose a lot of performance since it's already close to random in the uncorrupted scenario (51.07 ROCAUC). Please consider plotting the average ROCAUC score on the OY axis (as you did in Figure 6).

* You mentioned that your work is focusing on Tabular AD (line 26), but you include datasets such as CIFAR10, SVHN, AGNews and so on, which are not tabular. Please consider rephrasing or expanding the explanation.

* Regarding Sec. 4.4, "Unsupervised methods are more susceptible to duplicated anomalies": does "anomalies are duplicated by 6 times" mean that you now have the _all_ the anomalous samples appear multiple (6) times in the test data, or that you sampled _some_ anomalies and made them appear more frequently? Please consider rephrasing and expanding the explanation.

* Line 38: "(...) we observe that *the* best unsupervised methods (...)"

**Correctness:**


* The datasets used are relevant to Anomaly Detection

* The experimental setup and the metrics used are sound for AD.

* The benchmarked methods are well known and the authors performed multiple experiment runs to obtain the mean performance.

**Documentation:**

The code and data are made publicly available on Github under a BSD-2 license.

**Ethics:**

No ethical concerns.

**Relation To Prior Work:**

The paper cites and discusses a lot of related and prior work.

**Summary And Contributions:**

* The authors propose a comprehensive Anomaly Detection benchmark, which consists of 55 datasets.

* A total of 93,654 experiments were performed on the proposed data, with both classical and deep learning models.

* The benchmark is constructed taking into account various degrees of supervision, inductive assumptions, and data modalities.

* The paper contains several insights regarding different models' performances and proposes several future directions for research in Anomaly Detection.

---

> ### Author Response · Authors · 2022-08-11
> **Response to Reviewer 89C5 (paper revision Aug 11th) P1**
>
> **Q1**. You have not included datasets tailored for anomaly detection in your benchmark, such as MVTec-AD [1] or MNIST-C [2]. Do you have a reason for omitting them?
>
> **R1**. Thanks for mentioning this. We have included these two datasets in the revision, along with adding 6 tabular AD datasets from [3] and another NLP dataset from [4]. Please see Appx. Table B1 for the complete list of 57 datasets.
>
> [1] Bergmann, Paul, et al. "MVTec AD--A comprehensive real-world dataset for unsupervised anomaly detection." CVPR, 2019.
>
> [2] Mu, Norman, and Justin Gilmer. "Mnist-c: A robustness benchmark for computer vision." arXiv, 2019.
>
> [3] Pang, Guansong, et al. "Deep learning for anomaly detection: A review." CSUR, 2021.
>
> [4] Lang, Ken. "Newsweeder: Learning to filter netnews." In Machine Learning Proceedings 1995, pp. 331-339. Morgan Kaufmann, 1995.
>
> ----
>
> **Q2**.  Please consider adding an option for downloading the "raw" dataset in your library.
> I understand that the newly added datasets (CIFAR10, fMNIST, SVHN, AGNews, Amazon, IMDB, and Yelp) are present in the form of representations extracted from neural networks (BERT and ResNet18). This could limit the overall AD performance of the models, since the data is already projected to a different space by the feature extraction backbone, leading to anomalies that could be clustered together with the normal data. Moreover, less general models that use the data structural information as an inductive bias for learning will suffer from these benchmark design decisions. For example, [3], [4] and [5] are methods specifically designed for one-class classification in text, with [3] and [5] achieving much better performance on the AG News dataset when compared to your best results (83.1, 90.0 vs. 68.97 ROCAUC). This might be due to the different train/validation/test splits used but might also indicate that end-to-end training is desirable on these datasets.
>
> **R2**. Great point! We follow popular AD literature DeepSAD [1], ADIB [2], and DATE [3] to create these new AD datasets by using deep models to extract representations from CV and NLP datasets. Following your advice, we now provide the download link to the raw version of the newly added datasets in [ReadMe](https://github.com/Minqi824/ADBench/tree/main/datasets). Additionally, we add references to all the datasets in Appx. Table B1 so that readers can find more descriptions of the datasets. Meanwhile, we also provide another version (let us call it V2) of datasets using ViT [4] and RoBERTa [5] as the backbone (see Appx. B.2 for details) to extract the features from these CV and NLP datasets, and release both versions at [datasets](https://github.com/Minqi824/ADBench/tree/main/datasets).
>
> We still analyze our results based on the original version using ResNet18 and Bert for CV and NLP datasets, while we encourage readers to check the V2. In the later iteration of ADBench, we will try to understand the impact of backbone choice on the downstream AD tasks.
>
> [1] Ruff, Lukas, et al. "Deep semi-supervised anomaly detection." ICLR, 2019.
>
> [2] Deecke, Lucas, et al. "Transfer-based semantic anomaly detection." ICML, 2021.
>
> [3] Manolache, Andrei, Florin Brad, and Elena Burceanu. "DATE: Detecting Anomalies in Text via Self-Supervision of Transformers." NAACL, 2021.
>
> [4] Dosovitskiy, Alexey, et al. "An Image is Worth 16x16 Words: Transformers for Image Recognition at Scale." ICLR, 2020.
>
> [5] Liu, Yinhan, et al. "Roberta: A robustly optimized bert pretraining approach." arXiv (2019).
>
> ----
>
> **Q3**. Some scores reported on your synthetic data are very high (for example 99.91 ROCAUC for the KNN on Global Anomalies). I still believe that the datasets are useful for testing the inductive biases when designing a model, but I don't see any value in using them when computing the overall performance metrics of new methods. It's not clear to me if you've also used them when computing the mean performance in places such as Figure 4.
>
> **R3**. Rest assured. We *do not* use these synthetic datasets in comparing different algorithms in Sections 4.2 and 4.4. They are only used in Section 4.3 (and Appx. D.2) in analyzing the anomaly types; only Fig. 5 and Fig. 6 is based on these synthetic datasets. We also want to explain the reason for seeing high scores on these synthetic datasets. These simple synthetic datasets only contain a single type of anomalies, while the benchmark datasets are the aggregation of multiple (unknown) types of anomalies which is more challenging.

---

> > ### Author Response · Authors · 2022-08-11
> > **Response to Reviewer 89C5 (paper revision Aug 11th) P2**
> >
> > **Q4**. I don't understand the claim that "none of the benchmarked unsupervised algorithms is statistically better than others" (line 35, figure 4 caption), it seems that the biggest performance gap is between DeepSVDD (51.07 ROCAUC) and CBLOF (70.20 ROCAUC), 19.13 points in ROCAUC seems like a major performance gap?
> >
> > **R4**. In ADBench, we use critical-difference plots to show group-wise performance differences among AD algorithms; the technique is used in some recent studies [1,2] that involve comparing a group of methods on a collection of datasets. To this end, the result only shows group-wise conclusions that the top 1 method is not significantly different from the rest of the group. For instance, the horizontal line crossing the top1-5 AD methods means that none of them is statistically different from the rest 4. So the key takeaway is if we get a dataset with limited knowledge, using the top 1 method in Fig. 4 will not be statistically better than the top 2, 3 methods, etc.
> >
> > You are absolutely right that the top 1 method (CBLOF) is statistically better than the bottom 1 (DeepSVDD) in a pairwise test (one-sided Wilcoxon Rank test; p=9.06e-10). Indeed, Fig. 4 suggests almost all the unsupervised methods are statistically better than DeepSVDD.
> >
> > [1] Ismail Fawaz, Hassan, et al. "Deep learning for time series classification: a review." Data mining and knowledge discovery, 2019.
> >
> > [2] Shekhar, Shubhranshu, and Leman Akoglu. "Incorporating privileged information to unsupervised anomaly detection." ECML/PKDD, 2018.
> >
> > ----
> >
> > **Q5**. Clarity of Figure 4:
> > - In figure 4 you have a cross symbol for both subfigures (a) and (b), but the explanation for the symbol is just under (b). Please consider moving it in the figure caption.average
> > - In the figure 4 captions some text seems to be missing: "(a) shows that no unsupervised algorithm can statistically outperform." - outperform what? Please expand.
> > - In the figure 4 captions you mention that the scores present in the figure are the "AD model performance on 55 real-world datasets.", I suppose that the reported numbers are the average score over all the datasets. Please be explicit in the captions and consider adding the standard deviations over the datasets.
> >
> > **R5**. Thank you so much for these suggestions. We did cut Fig. 4 caption aggressively to save space. In this revision, we have improved Fig. 4 and provided additional information like the performance distribution and train time by boxplots.
> >
> > ----
> >
> > **Q6**. You mentioned multiple times that the collection of data represents "55 real-world datasets", please expand. What makes them "real-world"? For example, your collection of data contains MNIST and Fashion MNIST - it could be argued that they can be used in "real-world" applications and were collected from the real world, but they are curated and are usually used for benchmarking algorithms.
> >
> > **R6**. We have made corresponding changes in Sections 3.2, 3.3.2, 4, and the Appendix.
> >
> > ----
> >
> > **Q7**. In Figure 7 you show the performance change under noisy and corrupted data. I would argue that the performance difference on the OY axis is not the clearest way of expressing the degradation: for example, you could get the impression that DeepSVDD is robust to noise since its performance doesn't change a lot when using corrupted data, in reality, DeepSVDD can't lose a lot of performance since it's already close to random in the uncorrupted scenario (51.07 ROCAUC). Please consider plotting the average ROCAUC score on the OY axis (as you did in Figure 6).
> >
> > **R7**. That will be a great addition! There might be two ways to demonstrate the degradation: (i) the current Fig. 7 shows the relative performance change and (ii) Appx. Fig. D9 shows the absolute performance (as you suggested). We now provide both of them with a holistic view.

---

> > > ### Author Response · Authors · 2022-08-11
> > > **Response to Reviewer 89C5 (paper revision Aug 11th) P3**
> > >
> > > **Q8**. You mentioned that your work is focusing on Tabular AD (line 26), but you include datasets such as CIFAR10, SVHN, AGNews and so on, which are not tabular. Please consider rephrasing or expanding the explanation.
> > >
> > > **R8**. Thanks for this question. As we explain in Q2, existing tubular AD papers, e.g., DeepSAD [1], ADIB [2], DATE [3] use deep models to extract representations from CV and NLP datasets for detection. One reason is that some shallow models, such as OCSVM, cannot directly run on (large, high-dimensional) CV datasets. Also, it is interesting to see whether tabular AD methods can work on CV/NLP data representations, which carry values in real-world deployment. Moreover, the extracted representations often lead to better downstream detection results [2]. Thus, we extract features from CV and NLP datasets by deep models to create “tabular’’ versions of them. Although not perfect, this may provide insights into shallow methods’ performance on (originally infeasible) CV and NLP datasets. We have updated Section 3.2 and Appx. B.2 with more details.
> > >
> > > [1] Ruff, Lukas, et al. "Deep semi-supervised anomaly detection." ICLR, 2019.
> > >
> > > [2] Deecke, Lucas, et al. "Transfer-based semantic anomaly detection." ICML, 2021.
> > >
> > > [3] Manolache, Andrei, Florin Brad, and Elena Burceanu. "DATE: Detecting Anomalies in Text via Self-Supervision of Transformers." NAACL, 2021.
> > >
> > > ----
> > >
> > > **Q9**. Regarding Sec. 4.4, "Unsupervised methods are more susceptible to duplicated anomalies": does "anomalies are duplicated by 6 times" mean that you now have the all the anomalous samples appear multiple (6) times in the test data, or that you sampled some anomalies and made them appear more frequently? Please consider rephrasing and expanding the explanation.
> > >
> > > **R9**. We motivate the setting by application settings that the same anomalies may appear multiple times. This may be caused by recording errors or just like recurring frauds in credit card transactions. To simulate this, we first split the data into train and test set, then duplicate the anomalies (both features and labels) up to 6 times in both sets, and observe how AD algorithms' performance changes. We have updated Section 3.3.3 to describe the process.
> > >
> > > ----
> > >
> > > **Q10**. Line 38: "(...) we observe that the best unsupervised methods (...)"
> > >
> > > **R10**. Good catch. We fix this along with another round of proofreading.
> > >
> > > ----
> > >
> > > **Q11**. It would be nice to consider expanding this work with different modalities (such as videos, for example ShanghaiTech, or graphs). I'm looking forward to seeing how this project evolves!
> > >
> > > **R11**. Thanks for mentioning this, and we have listed this as one of the future directions in Section 5. Indeed, we already have a series of works focusing on different data modalities, e.g., time-series AD [1] and graph AD [2]. One interesting direction is building multimodal AD datasets and benchmarks, which can be important for the AD community.
> > >
> > > We also want to take this opportunity to thank you for these detailed comments on paper clarification and future directions. ADBench has been significantly improved by these great thoughts :)
> > >
> > > [1] Lai, Kwei-Herng, Daochen Zha, Junjie Xu, Yue Zhao, Guanchu Wang, and Xia Hu. "Revisiting time series outlier detection: Definitions and benchmarks." NeurIPS, 2021.
> > >
> > > [2] Liu, Kay, Yingtong Dou, Yue Zhao, Xueying Ding, Xiyang Hu, Ruitong Zhang, Kaize Ding et al. "Benchmarking Node Outlier Detection on Graphs." arXiv preprint arXiv:2206.10071, 2022.

---

> > > > ### Comment · Reviewer_89C5 · 2022-08-23
> > > > **Response to the authors**
> > > >
> > > > Many thanks to the authors for promptly responding to my review and addressing my raised concerns!
> > > >
> > > > I still strongly believe that the paper should be accepted and I will keep my initial rating.

---

### Official Review · Reviewer_Ycib · 2022-07-28

**Rating:** 7
**Confidence:** 2

**Strengths:**

- Large, well-designed benchmark for tabular anomaly detection
- Lots of datasets and methods
- Explores interesting settings, e.g., supervision type, corrupted data, different types of anomalies
- Adds new evaluation settings using embeddings from DNNs on various text and image datasets
- Good GitHub repository. Everything seems easily available.

**Weaknesses:**

- At the end of the day, it doesn't feel like there is much novelty beyond the increased scale and the findings that come with it. This is OK, but still something to keep in mind.
- Some issues with related work and correctness (see below)

**Additional Feedback:**

It would be good to bold the best-performing methods in the tables in the appendix (or indicate visually that there are multiple indistinguishable methods).

**Clarity:**

The paper is well-written, although there is superfluous self-praise in some parts of the paper, e.g., "We implement all the semi-supervised methods and release them along with ADBench—we consider them as great addition to the community." The latter part is for the community to judge and doesn't need to be said explicitly.

**Correctness:**

It's unclear whether the anomaly generation method outlined in lines 157-159 is faithful to the original dataset. It fits a GMM to the data in order to generate anomalies in a controlled manner, but this might remove interesting structure in the original dataset.

**Documentation:**

There is sufficient documentation

**Ethics:**

No concerns

**Relation To Prior Work:**

The paper is marketed as comprehensive, but it completely ignores the large body of work on out-of-distribution detection. I understand that the OOD detection community is separate from the tabular anomaly detection community, but the two areas are related, so it would be good to at least mention OOD detection methods (e.g., MSP, Energy-based, Mahalanobis).

**Summary And Contributions:**

This paper introduces ADBench, a tabular anomaly detection benchmark. The problem setting is that one is given a dataset and asked to identify the outliers in the dataset. This paper compares a wide variety of methods (30 methods total, supervised, semi-supervised, and unsupervised) on a wide variety of settings (55 datasets, different kinds of anomalies, corrupted data). There is not much novelty in the datasets or methods, but there are some interesting findings afforded by the large evaluation (e.g., none of the unsupervised methods are better than the others).

---------------------
Updating my score to a 7 after author response; I'm not an expert in this area, so my confidence will remain at a 2

---

> ### Author Response · Authors · 2022-08-11
> **Response to Reviewer Ycib (paper revision Aug 11th)**
>
> **Q1**. At the end of the day, it doesn't feel like there is much novelty beyond the increased scale and the findings that come with it. This is OK, but still something to keep in mind.
>
> **R1**. Thanks for sharing this thought. As you and all other reviewers find, the key contributions of ADBench (as a benchmark) are its wide coverage of algorithms and datasets, and the insights that can only be unlocked via such large-scale analysis, e.g., the necessity of model selection in unsupervised AD. We hope that ADBench may be helpful for future AD research.
>
> ----
>
> **Q2**. Some issues with related work and correctness (see below)
> The paper is marketed as comprehensive, but it completely ignores the large body of work on out-of-distribution detection. I understand that the OOD detection community is separate from the tabular anomaly detection community, but the two areas are related, so it would be good to at least mention OOD detection methods (e.g., MSP, Energy-based, Mahalanobis).
>
> **R2**. Great point! Following your suggestion, we enrich the literature on OOD, novelty detection, open-set recognition, and salient and camouflage detection (as mentioned by Reviewer U8uU) in Section 2.3.
>
> ----
>
> **Q3**. It's unclear whether the anomaly generation method outlined in lines 157-159 is faithful to the original dataset. It fits a GMM to the data in order to generate anomalies in a controlled manner, but this might remove an interesting structure in the original dataset.
>
>
> **R3**. We follow the latest anomaly generation method introduced in [1], which has been used in recent AD works including a KDD’22 paper [2] where it takes the GMM approach to simulate different types of anomalies (see Section 4.1.1 of [2]). One advantage of GMM is that it  “allows for the generation of *characterizable* outliers” [1], which may be unclear in more complex generative models like generative adversarial networks (GAN).
>
> We appreciate the advice of considering more advanced techniques for anomaly generation, and we will closely follow this direction and update ADBench accordingly.
>
> [1] Steinbuss, Georg, and Klemens Böhm. "Benchmarking unsupervised outlier detection with realistic synthetic data." TKDD, 2021.
>
> [2] Zhang, Sean, Varun Ursekar, and Leman Akoglu. “Sparx: Distributed Outlier Detection at Scale”. KDD 2022 (Accepted, to appear).
>
> ----
>
> **Q4**. The paper is well-written, although there is superfluous self-praise in some parts of the paper, e.g., "We implement all the semi-supervised methods and release them along with ADBench—we consider them as a great addition to the community." The latter part is for the community to judge and doesn't need to be said explicitly.
>
> **R4**. We have removed this statement in Section 3.2, while still sincerely wishing the community to find these implementations helpful :)
>
> ----
>
> **Q5**. It would be good to bold the best-performing methods in the tables in the appendix (or indicate visually that there are multiple indistinguishable methods).
>
> **R5**. Good catch! We have highlighted all the best-performing methods in the Appendix tables (Appx. Table D4-D19). Again, thanks a lot for these useful comments to make ADBench more complete work!

---

> > ### Comment · Reviewer_Ycib · 2022-08-20
> > **The authors addressed my concerns**
> >
> > The author response and updated paper addresses all of my concerns. I now think the paper should be accepted and have updated my score to a 7.

---

### Author Response · Authors · 2022-08-11
**Summary of Our Responses (as of paper revision Aug 11th) and Long-term Plan**

**We sincerely thank all the reviewers for their encouraging and insightful comments. We have carefully read through them and provided corresponding responses individually.**

We upload **the revised paper and the supplementary material**, with changes highlighted in blue and callout boxes for specific reviewer(s)’ attention. We provide the single pdf with the main content+supplmentary material in the supplementary.

**The primary changes are summarized below**:

- **Related Work**: We enrich the related work in Section 2.3 to discuss relevant topics, e.g., OOD, novelty detection, open-set detection, and salient and camouflaged object detection.
- **More Datasets**: We refresh the dataset list by removing 7 datasets due to their high anomaly ratio (>=40%) and adding 9 additional datasets (highlighted in yellow in Appx. Table B1), including 3 CV/NLP datasets, and 6 tabular datasets from a recent AD survey. Consequently, this revision is based on 57 benchmark datasets (see Section 3.2 and Appx. B.2 for details), of which 10 are repurposed CV/NLP datasets released by us (marked in brown in Appx. Table B1). We have run the experiments with the enriched data corpus and updated the paper with new figures (Fig. 4-7 and Appx. Fig. B1-B2, D3-D11) and tables (Appx. Table B1-B3, and D4-D19). **The key takeaways and insights stay the same**.
- **Documentation and Accessibility**: We provide more details on datasets in Section 3.2 and Appx B.2, e.g., how to adapt CV and NLP datasets for tabular AD and its merits. Both the generated datasets and the link to raw datasets are now available at [data](https://github.com/Minqi824/ADBench/tree/main/datasets) with more documentation.
- **Figures and Tables**: We improve the readability of selected figures and tables by increasing their sizes, adding more descriptions, and highlighting key information. For the tables in the Appendix, we highlight the best method(s) per dataset.
- **Additional Analysis**: we add the boxplot of AUCROC and runtime in Fig. 4c and 4d, and provide the inference time plot in Appx. Fig. D5.
- **Future Directions**: we extend the future directions of ADBench in Section 5.


**Long-term plan**: We commit to maintaining and enriching ADBench in long run, as many of our open-source AD works (e.g., PyOD [1], SUOD [2], and TODS [3]). The next step is distributing ADBench via PyPI for easier dataset and script access, as well as new algorithm benchmarking. Also, we consider developing an AD leaderboard based on ADBench in the future.

[1] Zhao, Yue, Zain Nasrullah, and Zheng Li. "PyOD: A Python Toolbox for Scalable Outlier Detection." JMLR, 2019.

[2] Zhao, Yue, et al. "SUOD: Accelerating large-scale unsupervised heterogeneous outlier detection." MLSys, 2021.

[3] Lai, Kwei-Herng, Daochen Zha, Junjie Xu, Yue Zhao, Guanchu Wang, and Xia Hu. "Revisiting time series outlier detection: Definitions and benchmarks." NeurIPS Benchmark and Datasets. 2021.

---

> ### Author Response · Authors · 2022-08-20
> **Follow-up On Our Revision and Responses**
>
> We want to send this friendly reminder and are happy to address additional comments reviewers may have. Also, If our revision and responses address your questions, we want to kindly ask for your reconsideration of the score.
>
> Thank you so much for devoting time to making ADBench a better work. We are looking forward to any further discussions :)

---

> > ### Author Response · Authors · 2022-08-26
> > **Happy to Address Any Additional Questions**
> >
> > We appreciate all the initial comments and the acknowledgment of the revision and responses.
> >
> > Since the author-reviewer discussion deadline is coming, we want to kindly inquire if other reviewers have any additional thoughts for the revision and responses. We would like to take this valuable opportunity to make ADBench a better work :)

---

### Meta-Review · Area_Chair_sNJa · 2022-09-07

**Recommendation:** Accept
**Confidence:** 3

**Metareview:**

All reviewers agree on accepting the paper. The authors appear to have addressed the concerns of the reviewers.

---

### Decision · Program_Chairs · 2022-09-16

Accept